# Combinatorial Reinforcement Learning with Preference Feedback

## Abstract

In this paper, we consider combinatorial reinforcement learning with preference feedback, where a learning agent sequentially offers an action—an assortment of multiple items—to a user, whose preference feedback follows a multinomial logit (MNL) model. This framework allows us to model real-world scenarios, particularly those involving long-term user engagement, such as in recommender systems and online advertising. However, this framework faces two main challenges: 1) the unknown value of each item, unlike traditional MNL bandits (which only account for single-step preference feedback), and (2) the difficulty of ensuring optimism with tractable assortment selection in the combinatorial action space. In this paper, we assume a contextual MNL preference model, where mean utilities are linear, and the value of each item is approximated using general function approximation. We propose an algorithm, $\texttt{MNL-V}Q\texttt{L}$, that addresses these challenges, making it both computationally and statistically efficient. As a special case, for linear MDPs (with the MNL preference feedback), we establish the first regret lower bound in this framework and show that $\texttt{MNL-V}Q\texttt{L}$ achieves near-optimal regret. To the best of our knowledge, this is the first work to provide statistical guarantees in combinatorial RL with preference feedback.

## 1 Introduction

We first formally state the concept of *Combinatorial Reinforcement Learning* (RL), which we refer to as a class of RL problems where the action space is combinatorial, meaning that the agent selects a combination or subset of base actions from a set of possible base actions. Although some previous studies have addressed problems within this setting—particularly in deep RL (Sunehag et al., 2015; He et al., 2016; Swaminathan et al., 2017; Metz et al., 2017; Ryu et al., 2019; Ie et al., 2019; Delarue et al., 2020; McInerney et al., 2020; Vlassis et al., 2021; Chaudhari et al., 2024), with less emphasis on theoretical RL—to the best of our knowledge, it appears that no prior work has formally and theoretically defined the concept of combinatorial RL.[1] This framework is especially relevant for real-world applications such as recommender systems and online advertising, where multiple items (base actions) must be selected simultaneously, such as a set of products to recommend or advertisements to display. The challenge in combinatorial RL lies in the exponentially large action space and the need to efficiently optimize the agent's action selection while balancing exploration and exploitation (a challenge even for single action selection), while considering the long-term effects of these actions.

One of the most widely encountered settings in combinatorial RL is *preference feedback* over combinatorial actions, commonly seen in streaming services, online retail, and similar platforms. Despite the broad applicability of this setting, theoretical studies have predominantly focused on the multinomial logit (MNL) *bandit* model (Rusmevichientong et al., 2010; Sauré & Zeevi, 2013; Agrawal et al., 2017; 2019; Oh & Iyengar, 2019; 2021; Perivier & Goyal, 2022; Agrawal et al., 2023; Zhang & Sugiyama, 2024; Lee & Oh, 2024). The MNL bandit framework focuses on assortment (a set of items) selection by selecting subsets of items and receiving feedback on chosen items, modeled by the MNL model (McFadden, 1977). However, these studies take a *myopic* approach, optimizing for immediate rewards without considering the long-term impact on user behavior.

---

[1] Surprisingly, this is in contrast to the rich theoretical literature in combinatorial bandits (Chen et al., 2013; Kveton et al., 2015b;a; Combes et al., 2015), which extend the multi-armed bandit problem.

While MNL bandits have been widely studied, the myopic approach is limiting in many real-world scenarios. For example in recommender systems, incorporating the long-term impact of recommendations opens the door to balancing short-term engagement with long-term user satisfaction. For instance, recommending *junk* product or content might lead to high immediate reward but it can decrease user satisfaction over time due to fatigue. This trade-off between immediate and long-term outcomes is not captured by traditional MNL bandit approaches. See Appendix A for a more details.

On the empirical side, several studies have explored long-term user engagement in recommendation systems, particularly using deep RL (Swaminathan et al., 2017; Ie et al., 2019; McInerney et al., 2020; Vlassis et al., 2021; Chaudhari et al., 2024). However, there is a significant gap in the theoretical understanding of combinatorial RL with preference feedback, particularly within the RL framework. To the best of our knowledge, no theoretical work has yet explored this important problem setting.

In this paper, we aim to address this gap by rigorously studying the problem of combinatorial RL with preference feedback. Our goal is to develop a provably efficient algorithm that maximizes long-term user engagement by incorporating state transitions (e.g., historical behavior) in decision-making. The key challenges in this framework are: (1) the unknown long-term value of each item due to the stochastic nature of rewards and transitions, (2) the difficulty of selecting an assortment that ensures optimism while considering tractable assortment optimization in the combinatorial action space.

To tackle theses challenges, we first decompose the $Q$-function of an assortment into two components: the preference model and the item values, inspired by Ie et al. (2019). Since the long-term value of each item is unknown (unlike in MNL bandits), we compute optimistic item values and select an assortment based on those values. To ensure optimism (Lemma D.9), we propose a novel method to estimate the *optimistic preference model* by carefully alternating between *optimistic* and *pessimistic* utilities (for the preference model) based on the optimistic item values (Equation 7 and 8). This method requires a more sophisticated proof analysis to address both cases (Lemma D.2, D.5, and D.7). Finally, once the optimistic item values and *optimistic* (or *pessimistic*) utilities are established, we avoid the need for naive combinatorial enumeration when selecting the assortment (Equation 9) by reformulating the optimization problem as a linear programming (LP), inspired by Davis et al. (2013).

In this paper, We consider the contextual MNL preference model with linear mean utilities (Agrawal & Goyal, 2013; Cheung & Simchi-Levi, 2017; Agrawal et al., 2019; Oh & Iyengar, 2019; 2021; Amani & Thrampoulidis, 2021; Perivier & Goyal, 2022; Agrawal et al., 2023; Zhang & Sugiyama, 2024; Lee & Oh, 2024) and use general function approximation to estimate item values (Jiang et al., 2017; Wang et al., 2020; Jin et al., 2021; Du et al., 2021; Foster et al., 2021; Agrawal et al., 2023; Zhao et al., 2023). Inspired by Agarwal et al. (2023), we use function approximation in the $Q$-type setting (Jin et al., 2021), employing point-wise optimism for estimating item values. The algorithm then constructs optimistic choice probabilities based on optimistic or pessimistic MNL utilities to ensure sufficient exploration and guarantee optimism. Our main contributions are summarized as:

- We propose a computationally efficient algorithm, `MNL-VQL`, that achieves a regret upper bound of $\tilde{\mathcal{O}}\big(dH\sqrt{K} + \sqrt{\dim(\mathcal{F})KH\log|\mathcal{F}|}\big)$, where $H$ is the horizon length, $K$ is the total number of episodes, $d$ is the feature dimension of the MNL preference model, and $\dim(\mathcal{F})$ is the generalized Eluder dimension (see Definition 3) of the function class $\mathcal{F}$, under the parameterized contextual MNL preference model and general function approximation for item values (Theorem 1). To the best of our knowledge, this is the first theoretical regret guarantee in combinatorial RL with preference feedback.

- For the special case of linear MDPs, `MNL-VQL` obtains a regret upper bound of $\tilde{\mathcal{O}}(dH\sqrt{K} + d^{\mathrm{lin}}\sqrt{HK})$, where $d^{\mathrm{lin}}$ is the feature dimension of the linear MDPs (Theorem 2). Furthermore, we establish a regret lower bound of $\Omega(d\sqrt{HK} + d^{\mathrm{lin}}\sqrt{HK})$, demonstrating the near-optimality of our algorithm for linear MDPs (Theorem 3). To the best of our knowledge, this is the first regret lower bound in linear MDPs with preference feedback.

## 2 RELATED WORK

**MNL bandits.** The MNL bandits were initially studied in Rusmevichientong et al. (2010), followed by a line of improvements (Filippi et al., 2010; Rusmevichientong et al., 2010; Agrawal et al., 2017; Oh & Iyengar, 2019; Faury et al., 2020; Abeille et al., 2021; Faury et al., 2022; Oh & Iyengar, 2021;

Perivier & Goyal, 2022; Agrawal et al., 2023; Lee & Oh, 2024). In MNL bandits, the goal is to offer an assortment that maximizes the expected rewards, which are adaptively learned based on user preference feedback from the offered assortment. However, there are no state transitions, and it is assumed that the value of each item is known, with the value of the outside option fixed at zero. In contrast, our study extends this by not only estimating the MNL model but also the item values.

**Combinatorial RL with preference feedback.** Recently, several studies have demonstrated the empirical success of combinatorial RL with preference feedback (Swaminathan et al., 2017; Ie et al., 2019; McInerney et al., 2020; Vlassis et al., 2021; Chaudhari et al., 2024), where a set of items is offered to a user, and (relative) choice feedback along with a reward is received, leading to a transition to the next state. However, theoretical results quantifying the benefits of such methods are still few and far between. A closely related work is cascading RL (Du et al., 2024), which also involves selecting a set of items. However, in cascading RL, items are offered to the user one by one, and the user decides only whether to choose the currently offered item. As a result, this framework does not capture relative preference feedback across multiple items. Furthermore, in cascading RL, the probability of choosing each item is independent of the others, which is not the case in our framework.

Another related line of work is preference-based RL (PbRL) (Akrour et al., 2012; Wirth et al., 2017; Christiano et al., 2017; Ouyang et al., 2022; Saha et al., 2023; Zhu et al., 2023; Zhan et al., 2023), where the policy is optimized based on relative, rather than absolute, preference feedback. However, our framework differs from PbRL in that our goal is not to offer just a single item, but to offer multiple items—a combinatorial (base) action—at each timestep.

**Provably efficient RL with nonlinear function approximation.** RL under nonlinear function approximation has gained attention (Jiang et al., 2017; Wang et al., 2020; Jin et al., 2021; Du et al., 2021; Foster et al., 2021; Agarwal et al., 2023; Zhao et al., 2023) for modeling complex function spaces like neural networks. Among these, Agarwal et al. (2023); Zhao et al. (2023) achieved the best-known regret guarantees under general function approximation by introducing the concept of generalized Eluder dimension to handle weighted regression. Inspired by their work, we estimate the value of items (referred to as *item-level Q-value*) using general function approximation in this paper.

## 3 PROBLEM SETTING

### 3.1 NOTATIONS

For an integer $n > 0$, we denote $[n] := \{1, 2, \ldots, n\}$. For a positive semi-definite matrix $\Lambda \in \mathbb{R}^{d \times d}$, the norm $\| \cdot \|_\Lambda$ on $\mathbb{R}^d$ is defined as $\|x\|_\Lambda^2 := x^\top \Lambda x$. Let $| \cdot |$ denote the cardinality of a set.

### 3.2 COMBINATORIAL MDPs WITH PREFERENCE FEEDBACK

In this paper, we consider a time-inhomogeneous episodic and *combinatorial Markov decision processes (MDPs) with preference feedback*, $\mathcal{M}(\mathcal{S}, \mathcal{I}, \mathcal{A}, M, \{\mathcal{P}_h\}_{h=1}^H, \{\mathbb{P}_h\}_{h=1}^H, \{r_h\}_{h=1}^H, H)$. Here, $\mathcal{S}$ is the set of states. Each state $s \in \mathcal{S}$ reflects the user's status, capturing both relatively static user features (e.g., demographics, interests) and relevant user history or past behavior (e.g., past recommendations, items purchased or clicked). $\mathcal{I} := \{a_1, \ldots, a_N, a_0\}$ is the ground set of items (base actions), where $a_1, \ldots, a_N$ are items and $a_0$ refers to the "outside option", meaning the user has chosen none of the items from the offered set of items (referred to as an "assortment" throughout the paper). It is included in every assortment *by default*. $\mathcal{A}$ is the set of candidate assortments that always include the outside option $a_0$, contain at least one item (other than $a_0$), and have at most $M$ items (including $a_0$), i.e., $\mathcal{A} = \{A \subseteq \mathcal{I} : a_0 \in A, 1 \leqslant |A \backslash \{a_0\}| \leqslant M\}$, where $A$ is an assortment. For any $(s, A) \in \mathcal{S} \times \mathcal{A}$, we denote $\mathcal{P}_h(a|s, A)$ as the probability of the user choosing on item $a \in A$ (including the outside option $a_0$). Furthermore, we let $\mathbb{P}_h : \mathcal{S} \times \mathcal{I} \to \Delta_\mathcal{S}$ and $r_h : \mathcal{S} \times \mathcal{I} \times \mathcal{S} \to \mathbb{R}$ characterize the transition kernel and instantaneous reward, respectively, at a given horizon $h \in [H]$. Throughout this paper, we assume that $\sum_{h=1}^H r_h(s_h, a_h, s_{h+1}) \in [0, 1]$ for all possible sequence $(s_1, a_1, \ldots, s_H, a_H, s_{H+1})$. $H \in \mathbb{Z}_+$ is the length of each episode. A policy $\pi : \mathcal{S} \to \mathcal{A}$ is a mapping from the state space to the assortment space. Since the optimal policy is non-stationary in an episodic MDP, we use $\pi$ to refer to the $H$-tuple $\{\pi_h\}_{h=1}^H$.

In each episode $k \in [K]$, an initial state $s_1^k$ is picked *arbitrarily* (e.g., a user arrives at the system). The agent then follows a policy $\pi^k$ starting from $s_1^k$. At each step $h \in [H]$, the agent observes the

current state $s_h^k$ (e.g., historical behaviors of the user) and offers an assortment $A_h^k = \pi_h^k(s_h^k)$. The user's preference feedback $a_h^k \in A_h^k$ is then observed, which is drawn based on the choice probability $\mathcal{P}_h(\cdot|s_h^k, A_h^k)$. Next, the system transitions to the next state $s_{h+1}^k \sim \mathbb{P}_h(\cdot|s_h^k, a_h^k)$ and receives a reward $r_h(s_h^k, a_h^k, s_{h+1}^k)$. After $H$ steps, the episode terminates, and the agent proceeds to the next.

For any policy $\pi = \{\pi_h\}_{h=1}^H$, we define the value function of policy $\pi$, denoted as $V_h^\pi : \mathcal{S} \to \mathbb{R}$, as the expected sum of rewards under the policy $\pi$ until the end of the episode when starting from $s_h = s$, i.e., $V_h^\pi(s) := \mathbb{E}\left[\sum_{h'=h}^H r_{h'}(s_{h'}, a_{h'}, s_{h'+1}) \,\middle|\, s_h = s\right]$. Similarly, we define the action-value function of policy $\pi$, $Q_h^\pi : \mathcal{S} \times \mathcal{A} \to \mathbb{R}$, as the expected sum of rewards under policy $\pi$, starting from step $h$ until the end of the episode after taking action $A$ in state $s$; that is, $Q_h^\pi(s, A) := \mathbb{E}\left[\sum_{h'=h}^H r_{h'}(s_{h'}, a_{h'}, s_{h'+1}) \,\middle|\, s_h = s, A_h = A\right]$. Furthermore, we define the *item-level Q-value function* (also called the $\overline{Q}$-*value*) $\overline{Q}_h^\pi(s, a) := \sum_{s'} \mathbb{P}_h(s'|s, a)(r_h(s, a, s') + V_{h+1}^\pi(s'))$. Then, the Bellman equation for assortment RL is denoted as follows:

$$Q_h^\pi(s, A) = \sum_{a \in A} \mathcal{P}_h(a|s, A) \left(\sum_{s' \in \mathcal{S}} \mathbb{P}_h(s'|s, a)\left(r_h(s, a, s') + V_{h+1}^\pi(s')\right)\right) = \sum_{a \in A} \mathcal{P}_h(a|s, A)\overline{Q}_h^\pi(s, a).$$

There exists an optimal policy $\pi^\star$, which gives the optimal value function for all states (Puterman, 2014), i.e., $V_h^{\pi^\star}(s) = \sup_\pi V_h^\pi(s)$ for all $(s, h) \in \mathcal{S} \times [H]$. For notational simplicity, we abbreviate $V^{\pi^\star}$ as $V^\star$. Similarly, we define the optimal Q-value function as $Q^\star$ and the *item-level optimal Q-value function* as $\overline{Q}^\star$. Then, $Q^\star$ satisfies the following Bellman optimality equation:

$$Q_h^\star(s, A) = \sum_{a \in A} \mathcal{P}_h(a|s, A) \left(\sum_{s' \in \mathcal{S}} \mathbb{P}_h(s'|s, a)\left(r_h(s, a, s') + V_{h+1}^\star(s')\right)\right) = \sum_{a \in A} \mathcal{P}_h(a|s, A)\overline{Q}_h^\star(s, a).$$

For any $V : \mathcal{S} \to \mathbb{R}$ and $h \in [H]$, we define the *item-level Bellman operator* of $V$ as $\mathcal{T}_h V : \mathcal{S} \times \mathcal{I} \to \mathbb{R}$, such that for all $(s, a) \in \mathcal{S} \times \mathcal{I}$, $\mathcal{T}_h V(s, a) := \mathbb{E}_{s' \sim \mathbb{P}_h(\cdot|s, a)}\left[r_h(s, a, s') + V(s') \mid s, a\right]$. The definition of value functions ensures that they satisfy the equation $\overline{Q}_h^\star(s, a) = \mathcal{T}_h V_{h+1}^\star(s, a)$. We also define the *second moment item-level Bellman operator* of $V$ as $\mathcal{T}_h^2 V : \mathcal{S} \times \mathcal{I} \to \mathbb{R}$ such that for all $(s, a) \in \mathcal{S} \times \mathcal{I}$, $\mathcal{T}_h^2 V(s, a) := \mathbb{E}_{s' \sim \mathbb{P}_h(\cdot|s, a)}[(r_h(s, a, s') + V(s'))^2 \mid s, a]$.

The agent's objective is to find a policy that maximizes its expected cumulative reward over $K$ episodes. Equivalently, our goal is to minimize the cumulative regret over $K$ episodes, defined as **Regret**$(\mathcal{M}, K) := \sum_{k=1}^K V_1^\star(s_1^k) - V_1^{\pi_k}(s_1^k)$.

## 3.3 MULTINOMIAL LOGIT PREFERENCE MODEL

In this paper, we make a structural assumption about the MDP $\mathcal{M}$, where the user's choice probability $\{\mathcal{P}_h\}_{h=1}^H$ follows multinomial logit (MNL) model (McFadden, 1977) parameterized by $\{\theta_h^\star\}_{h=1}^H$. We denote $\mathcal{P}_h(\cdot|s, A; \theta_h^\star)$ as equivalent to $\mathcal{P}_h(\cdot|s, A)$, explicitly showing the dependence on the parameter $\theta_h^\star$. Throughout the paper, we use $\mathcal{P}_h(\cdot|s, A)$ and $\mathcal{P}_h(\cdot|s, A; \theta_h^\star)$ interchangeably.

**Assumption 1** (MNL preference model). *Let there exist a known feature map $\phi : \mathcal{S} \times \mathcal{I} \to \mathbb{R}^d$ and an unknown $\theta_h^\star \in \Theta$ for all $h \in [H]$, where $\Theta = \{\theta \in \mathbb{R}^d : \|\theta\|_2 \leqslant B\}$. Then, for any $(s, A, H) \in \mathcal{S} \times \mathcal{A} \times [H]$, the probability of choosing any item $a \in A$ is defined as:*

$$\mathcal{P}_h(a|s, A) = \mathcal{P}_h(a|s, A; \theta_h^\star) := \frac{\exp\left(\phi(s, a)^\top \theta_h^\star\right)}{\sum_{a' \in A} \exp\left(\phi(s, a')^\top \theta_h^\star\right)}.$$

*Also, we assume that $\|\phi(s, a)\|_2 \leqslant 1$ for all $(s, a) \in \mathcal{S} \times \mathcal{I}$ and $B = \tilde{\mathcal{O}}(1)$.*

Here, without loss of generality, we assume that $\phi(s, a_0) = 0$ for all $s \in \mathcal{S}$ [2], which implies that $\exp(\phi(s, a_0)^\top \theta_h^\star) = 1$. Thus, the preference model can be equivalently expressed as $\mathcal{P}_h(a|s, A; \theta_h^\star) = \exp\left(\phi(s, a)^\top \theta_h^\star\right) / \left(1 + \sum_{a' \in A \setminus \{a_0\}} \exp\left(\phi(s, a')^\top \theta_h^\star\right)\right)$.

---

[2] By subtracting $\phi(s, a_0)$ from each $\phi(s, a)$, where $a \in \mathcal{I}$, and defining $\phi'(s, a) := \phi(s, a) - \phi(s, a_0)$, we can ensure that $\phi'(s, a_0) = 0$. This implies that $\exp(\phi'(s, a_0)^\top \theta_h^\star) = 1$. This assumption is commonly made in contextual MNL bandits (Oh & Iyengar, 2019; 2021; Perivier & Goyal, 2022; Agrawal et al., 2023; Zhang & Sugiyama, 2024; Lee & Oh, 2024).

Following the previous MNL bandit literature (Chen et al., 2020; Oh & Iyengar, 2021; Perivier & Goyal, 2022; Zhang & Sugiyama, 2024; Lee & Oh, 2024), we also introduce the following constant:

**Definition 1** (Problem-dependent constant). *We define the problem-dependent constant $\kappa$ as*

$$\kappa := \min_{A \in \mathcal{A}, a \in A \setminus \{a_0\}, \boldsymbol{\theta} \in \Theta, h \in [H]} \mathcal{P}_h(a|s, A, \boldsymbol{\theta}) \mathcal{P}_h(a_0|s, A, \boldsymbol{\theta}) > 0.$$

A small $\kappa$ indicates a larger deviation from the linear model. Note that this value can be extremely large, so it is crucial to avoid any dependency on $\kappa$ in the main term of our regret bound.

### 3.4 GENERALIZED FUNCTION APPROXIMATION FOR ITEM-LEVEL $Q$-FUNCTION

We estimate the item-level $Q$-functions (referred to as $\overline{Q}$-values) using general function approximation. Specifically, the agent is given a function class $\mathcal{F} := \{\mathcal{F}_h\}_{h=1}^H$, where each set $\mathcal{F}_h$ is composed of functions $f_h : \mathcal{S} \times \mathcal{I} \to [0, L]$. We assume $L = \mathcal{O}(1)$ throughout the paper. Since no reward is collected in the $(H+1)^{\text{th}}$ steps, we set $f_{H+1} = 0$. We denote $\mathcal{N}$ as the maximal size of function class, i.e., $\mathcal{N} = \max_{h \in [H]} |\mathcal{F}_h|$. We consider a class of episodic MDPs such that the value functions satisfy the completeness and realizability assumptions under a function class $\mathcal{F}$.

**Assumption 2** (Completeness & Realizability). *For each $h \in [H]$ and any $V : \mathcal{S} \to [0, 1]$, we assume that $\overline{Q}_h^\star \in \mathcal{F}_h$, and there exists $f_h, f_h' \in \mathcal{F}_h$ such that*

$$f_h(s, a) = \mathcal{T}_h V(s, a), \quad \text{and} \quad f_h'(s, a) = \mathcal{T}_h^2 V(s, a), \quad \forall (s, a) \in \mathcal{S} \times \mathcal{I}.$$

**Remark 1.** *The completeness and realizability assumptions are standard in RL with general function approximation (Wang et al., 2021; Jin et al., 2021; Agarwal et al., 2023; Zhao et al., 2023). Our assumption is the same as those in Agarwal et al. (2023); Zhao et al. (2023), but stronger than those in (Wang et al., 2021; Jin et al., 2021), especially, in terms of the second moment completeness. However, this assumption is essential for using point-wise exploration bonuses and achieving a tighter regret bound. Additionally, it naturally holds for both tabular and linear MDPs.*

Since our results include the regret bound under the linear MDPs as a special case (in Section 5.2), we formally define the linear MDP as follows:

**Definition 2** (Linear MDPs, Yang & Wang 2019; Jin et al. 2020). *An MDP $\mathcal{M}$ is a linear MDP if we have a known feature mapping $\psi : \mathcal{S} \times \mathcal{I} \to \mathbb{R}^{d^{lin}}$, and there exist $d^{lin}$ unknown (signed) measures $\boldsymbol{\mu}_h^\star = (\mu_h^{(1)}, \ldots, \mu_h^{(d^{lin})})$ over $\mathcal{S}$ and unknown vector $\mathbf{w}_h^\star \in \mathbb{R}^{d^{lin}}$, such that for any $(s, a) \in \mathcal{S} \times \mathcal{I}$, we have $\mathbb{P}_h(\cdot|s, a) = \langle \psi(s, a), \boldsymbol{\mu}_h^\star(\cdot) \rangle$ and $r_h(s, a) = \langle \psi(s, a), \mathbf{w}_h^\star \rangle$. We assume that $\sup_{(s,a) \in \mathcal{S} \times \mathcal{I}} \|\psi(s, a)\|_2 \leqslant 1$, $\max\{\|\sum_{s \in \mathcal{S}} |\boldsymbol{\mu}_h^\star(s)|\|_2, \|\mathbf{w}_h^\star\|_2\} \leqslant \sqrt{d^{lin}}$ for all $h \in [H]$.*

We also assume that $\sum_{h=1}^H r_h \in [0, 1]$. Proposition 2.3 of Jin et al. (2020) shows that linear MDPs satisfy Assumption 2 under the linear function class $\mathcal{F}_h^{\text{lin}}$ defined as follows:

$$\mathcal{F}_h^{\text{lin}} := \left\{ \langle \psi(\cdot, \cdot), \boldsymbol{\omega}_h \rangle : \boldsymbol{\omega}_h \in \mathbb{R}^{d^{\text{lin}}}, \|\boldsymbol{\omega}_h\|_2 \leqslant 2\sqrt{d^{\text{lin}}} \right\}, \quad \text{for any } h \in [H]. \tag{1}$$

To capture the complexity of exploration in the MDP, we define the *generalized Eluder dimension*, which is a weighted regression version of the original definition by Russo & Van Roy (2013).

**Definition 3** (Generalized Eluder dimension, Agarwal et al. 2023; Zhao et al. 2023). *Let $\rho > 0$, a sequence of state-item pairs $\mathbf{Z} = \{z^k\}_{k=1}^K$, where $z^k = (s^k, a^k)$, and a sequence of positive numbers $\boldsymbol{\sigma} = \{\sigma^k\}_{k=1}^K$. The generalized Eluder dimension of a function class $\mathcal{F} : \mathcal{S} \times \mathcal{I} \to [0, L]$ with respect to $\rho$ is defined as $\dim_{\nu,K}(\mathcal{F}) := \sum_{\mathbf{Z}, \boldsymbol{\sigma}:|Z|=K, \sigma \geqslant \nu} \dim(\mathcal{F}, \mathbf{Z}, \boldsymbol{\sigma})$, where*

$$\dim(\mathcal{F}, \mathbf{Z}, \boldsymbol{\sigma}) := \sum_{k=1}^K \min \left( 1, \frac{1}{(\sigma^k)^2} D_{\mathcal{F}}^2 \left( z^k; \{z^\tau\}_{\tau=1}^{k-1}, \{\sigma^\tau\}_{\tau=1}^{k-1} \right) \right),$$

$$D_{\mathcal{F}}^2 \left( z; \{z^\tau\}_{\tau=1}^{k-1}, \{\sigma^\tau\}_{\tau=1}^{k-1} \right) := \sup_{f_1, f_2, \in \mathcal{F}} \frac{(f_1(z) - f_2(z))^2}{\sum_{\tau=1}^{k-1} \frac{1}{(\sigma^\tau)^2} (f_1(z^\tau) - f_2(z^\tau))^2 + \rho}.$$

*We write $d_\nu := \frac{1}{H} \sum_{h=1}^H \dim_{\nu,K}(\mathcal{F}_h)$ when function class $\{\mathcal{F}_h\}_{h=1}^H$ is clear form context.*

According to Theorem 4.6 of Zhao et al. (2023), the generalized Eluder dimension is upper bounded by the standard Eluder dimension (Russo & Van Roy, 2013) up to logarithmic terms.

---

**Algorithm 1** `MNL-VQL`, **MNL** Preference Model with **V**ariance-weighted Item-level **Q-L**earning

---

1: **Inputs:** parameter space $\Theta$, function class $\{\mathcal{F}_h\}_{h=1}^H$, consistent bonus oracle $\mathcal{B}$.
2: **Parameters:** $\{\alpha_h^k, \beta_{h,1}^k, \beta_{h,2}^k, \bar{\beta}_h^k\}_{(k,h)\in[H]\times[K]}$, $\{u_k\}_{k=1}^K$, $\rho$, bonus error $\epsilon_b$, $\nu$, $\delta$.
3: **Initialize:** confidence interval $\mathcal{C}_h^1 = \Theta$, dataset $\mathcal{D}_h^0 = \varnothing$ for all $h \in [H]$.
4: Generate $\{\mathcal{D}_h^1\}_{h=1}^H$ from initial state $s_1^1$ by random policy and set $\sigma_h^1 = \bar{\sigma}_h^1 = 2$ for all $h \in [H]$.
5: **for** episode $k = 2, \cdots, K$ **do**
6:     **for** horizon $h = H, H-1, \ldots, 1$ **do**
        // CONSTRUCT CI FOR MNL PREFERENCE MODEL
7:         Update $\hat{\boldsymbol{\theta}}_h^k$ by Equation 2 and the confidence interval $\mathcal{C}_h^k$ by Equation 3.
        // CONSTRUCT CI FOR OPTIMISTIC $\overline{Q}$-VALUES
8:         $\hat{f}_{h,1}^k \in \arg\min_{f_h \in \mathcal{F}_h} \sum_{\tau=1}^{k-1} \frac{1}{(\bar{\sigma}_h^\tau)^2} \left( f_h(s_h^\tau, a_h^\tau) - r_h^\tau - V_{h+1,1}^k(s_{h+1}^\tau) \right)^2$.
9:         $b_{h,1}^k \leftarrow \mathcal{B}\left( \{\bar{\sigma}_h^\tau\}_{\tau=1}^{k-1}, \mathcal{D}_h^{k-1}, \mathcal{F}_h, \hat{f}_{h,1}^k, \beta_{h,1}^k, \rho, \epsilon_b \right)$ (see Definition 4).
10:        Update $f_{h,1}^k(\cdot, \cdot) \leftarrow \min\left\{ \hat{f}_{h,1}^k(\cdot, \cdot) + b_{h,1}^k(\cdot, \cdot), 1 \right\}$.
        // CONSTRUCT CI FOR OVERLY OPTIMISTIC, PESSIMISTIC $\overline{Q}$-VALUES
11:        $\hat{f}_{h,j}^k \in \arg\min_{f_h \in \mathcal{F}_h} \sum_{\tau=1}^{k-1} \left( f_h(s_h^\tau, a_h^\tau) - r_h^\tau - V_{h+1,j}^k(s_{h+1}^\tau) \right)^2, j = \pm 2$.
12:        $b_{h,2}^k \leftarrow \mathcal{B}\left( \{\mathbf{1}^\tau\}_{\tau=1}^{k-1}, \mathcal{D}_h^{k-1}, \mathcal{F}_h, \hat{f}_{h,2}^k, \beta_{h,2}^k, \rho, \epsilon_b \right)$ (see Definition 4).
13:        Update $f_{h,2}^k(\cdot, \cdot) \leftarrow \min\left\{ \hat{f}_{h,2}^k(\cdot, \cdot) + 2b_{h,1}^k(\cdot, \cdot) + b_{h,2}^k(\cdot, \cdot), 1 \right\}$.
14:        Update $f_{h,-2}^k(\cdot, \cdot) \leftarrow \max\left\{ \hat{f}_{h,-2}^k(\cdot, \cdot) - b_{h,2}^k(\cdot, \cdot), 0 \right\}$.
        // CONSTRUCT CI FOR VARIANCE ESTIMATOR
15:        $\hat{g}_h^k \in \arg\min_{g_h \in \mathcal{F}_h} \sum_{\tau=1}^{k-1} \left( g_h(s_h^\tau, a_h^\tau) - \left( r_h^\tau + V_{h+1,1}^k(s_{h+1}^\tau) \right)^2 \right)^2$.
        // UPDATE VALUES
16:        Update $\widetilde{\mathcal{P}}_{h,j}^k(\cdot|\cdot, \cdot)$ by Equation 8.
17:        Update $Q_{h,j}^k(\cdot, A) \leftarrow \sum_{a \in A} \widetilde{\mathcal{P}}_{h,j}^k(a|\cdot, A) f_{h,j}^k(\cdot, a), j = 1, \pm 2$.
18:        Update $V_{h,j}^k(\cdot) \leftarrow \max_{A \in \mathcal{A}} Q_{h,j}^k(\cdot, A), j = 1, \pm 2$.
19:     **end for**
20:     Receive initial state $s_1^k$.
    // ROLLOUT POLICY
21:     **for** $h = 1, 2, \ldots, H$ **do**
22:         Offer $A_h^k$ by Equation 10 and receive $a_h^k, r_h^k$, and $s_{h+1}^k$.
23:         Update $\mathcal{D}_h^k \leftarrow \mathcal{D}_h^{k-1} \cup \{s_h^k, a_h^k, r_h^k, s_{h+1}^k\}$.
24:         Update $\sigma_h^k$ and $\bar{\sigma}_h^k$ by Equation 5 .
25:     **end for**
26: **end for**

---

# 4 ALGORITHM

In this section, we introduce an algorithm, which, to the best of our knowledge, is the first to provide statistical guarantees in combinatorial RL with preference feedback while maintaining computational tractability. **Step 1** involves online parameter estimation for the MNL preference model, proposed by (Zhang & Sugiyama, 2024; Lee & Oh, 2024). **Steps 2, 3, and 4** implement variance-weighted regression to tighten the regret bound, as outlined in (Agarwal et al., 2023). **Step 5**, which is our main contribution, ensures optimism in a computationally efficient manner, even with uncertainty in item-level $Q$-values. **Step 6**, inspired by (Agarwal et al., 2023), introduces an exploration step that employs a modified threshold to account for estimation errors from the MNL preference model.

**Step 1. Loss function for MNL preference model and online parameter estimation (Line 7).**
At episode $k$ and horizon $h$, given the user's choice feedback $c_h^k \in A_h^k$, the response for each item $a_{i_m} \in A_h^k$ is defined as $y_h^k(a_{i_m}) := \mathbb{1}(c_h^k = a_{i_m}) \in \{0, 1\}$. Therefore, the response variable $\mathbf{y}_h^k := (y_h^k(a_0), y_h^k(a_{i_1}), \ldots y_h^k(a_{i_l}))$, where $l \leqslant M - 1$, is sampled from a multinomial distribution:

$\mathbf{y}_h^k \sim \mathrm{MNL}\{1, \mathcal{P}_h(a_0|s_h^k, A_h^k; \boldsymbol{\theta}_h^\star), \ldots, \mathcal{P}_h(a_l|s_h^k, A_h^k; \boldsymbol{\theta}_h^\star)\}$, where the parameter $1$ indicates that $\mathbf{y}_h^k$ is a single-trial sample, i.e., $y_h^k(a_0) + \sum_{m=1}^l y_h^k(a_{i_m}) = 1$. Then, for any $(k, h) \in [K] \times [H]$, the multinomial logistic loss function is defined as:

$$\ell_h^k(\boldsymbol{\theta}) := - \sum_{a \in A_h^k} y_h^k(a) \log \mathcal{P}_h(a|s_h^k, A_h^k; \boldsymbol{\theta}).$$

Inspired by Zhang & Sugiyama (2024); Lee & Oh (2024), for all $(k, h) \in [K] \times [H]$, we use the online mirror descent algorithm to estimate the true parameter $\boldsymbol{\theta}_h^\star$ as follows:

$$\boldsymbol{\theta}_h^{k+1} \in \underset{\boldsymbol{\theta} \in \Theta}{\arg\min} \langle \nabla \ell_h^k(\boldsymbol{\theta}_h^k), \boldsymbol{\theta} \rangle + \frac{1}{2\eta} \|\boldsymbol{\theta} - \boldsymbol{\theta}_h^k\|_{\tilde{\mathbf{H}}_h^k}^2, \quad \text{where } \Theta = \left\{ \boldsymbol{\theta} \in \mathbb{R}^d : \|\boldsymbol{\theta}\|_2 \leqslant B \right\}, \quad (2)$$

where $\eta = \mathcal{O}(\log M)$ is the step-size, $\tilde{\mathbf{H}}_h^k := \mathbf{H}_h^k + \eta \nabla^2 \ell_h^k(\boldsymbol{\theta}_h^k)$, and $\mathbf{H}_h^k := \lambda \mathbf{I}_d + \sum_{\tau=1}^{k-1} \nabla^2 \ell_h^\tau(\boldsymbol{\theta}_h^{\tau+1})$.

**Remark 2.** *This online estimator is efficient in both computation and storage. Using the standard online mirror descent formulation (Orabona, 2019), Equation 2 can be solved efficiently with a computational cost of only $\mathcal{O}(Md^3)$, which is completely independent of $k$ (Mhammedi et al., 2019; Zhang & Sugiyama, 2024; Lee & Oh, 2024). In terms of storage, the estimator does not need to retain all historical data, as both $\tilde{\mathbf{H}}_h^k$ and $\mathbf{H}_h^k$ can be updated incrementally, requiring only $\mathcal{O}(d^2)$ storage.*

Now, we define the confidence interval as follows:

$$\mathcal{C}_h^k := \left\{ \boldsymbol{\theta} \in \Theta : \|\boldsymbol{\theta} - \boldsymbol{\theta}_h^k\|_{\mathbf{H}_h^k} \leqslant \alpha_h^k \right\}, \quad \text{where } \alpha_h^k = \tilde{\mathcal{O}}(\sqrt{d}). \quad (3)$$

Then, with high probability, we have $\boldsymbol{\theta}_h^\star \in \mathcal{C}_h^k$ for all $(k, h) \in [K] \times [H]$ (refer Corollary D.1).

**Step 2. Weighted regression and optimistic estimation for item-level $Q$-function (Line 8-10).**
Using the past dataset, we solve the following (weighted) regression problem to fit $\mathcal{T}_h V_{h+1}^k$:

$$\hat{f}_{h,1}^k \in \underset{f_h \in \mathcal{F}_h}{\arg\min} \sum_{\tau=1}^{k-1} \frac{1}{(\bar{\sigma}_h^\tau)^2} \left( f_h(s_h^\tau, a_h^\tau) - r_h^\tau - V_{h+1,1}^k(s_{h+1}^\tau) \right)^2, \quad (4)$$

where $(\bar{\sigma}_h^k)^2$ is a variance upper bound, i.e., $(\bar{\sigma}_h^k)^2 \geqslant \mathbb{V}[r_h + V_{h+1,1}^k(s_{h+1})|s_h^k, a_h^k]$, which will be specified later. Given $\hat{f}_{h,1}^k$, we define the version space $\mathcal{F}_h^k$ as follows:

$$\mathcal{F}_{h,1}^k := \left\{ f_h \in \mathcal{F}_h : \sum_{\tau=1}^{k-1} \frac{1}{(\bar{\sigma}_h^\tau)^2} \left( f_h(s_h^\tau, a_h^\tau) - \hat{f}_{h,1}^k(s_h^\tau, a_h^\tau) \right)^2 \leqslant \left( \beta_{h,1}^k \right)^2 \right\},$$

where $\beta_{h,1}^k = \tilde{\mathcal{O}}(\sqrt{\log \mathcal{N}})$. We can ensure that $\mathcal{T}_h V_{h+1}^k \in \mathcal{F}_h^k$ with high probability (Proposition D.2). Then, an *optimistic $\overline{Q}$-value* estimate at horizon $h$ is defined as $f_{h,1}^k := \hat{f}_{h,1}^k + b_h^k$, where $b_h^k$ is the optimistic bonus. The bonus is calculated as $b_h^k(s, a) = \max_{f_h \in \mathcal{F}_h} f_h(s, a) - \min_{f_h \in \mathcal{F}_h} f_h(s, a)$. In general, this uncertainty bonus has a high complexity, as the maximizing and minimizing functions can differ arbitrarily for each $(s, a) \in \mathcal{S} \times \mathcal{I}$. To address this, we introduce a low-complexity bonus oracle that approximately dominates the value obtained from the point-wise maximization over $\mathcal{F}_h$.

**Definition 4** (Oracle $\mathcal{B}$ for bonus function $b_h^k$, Agarwal et al. 2023). *For any $(h, k) \in [H] \times [K]$, sequence of $\{\bar{\sigma}_h^\tau\}_{\tau=1}^{k-1}$ and dataset $\mathcal{D}_h^{k-1} = \{(s_h^\tau, a_h^\tau, r_h^\tau, s_{h+1}^\tau)\}_{\tau=1}^{k-1}$, function class $\mathcal{F}_h$ with $\hat{f}_h \in \mathcal{F}_h$, $\beta_h, \rho > 0$, error parameter $\epsilon_b \geqslant 0$, the bonus oracle $\mathcal{B}(\{\bar{\sigma}_h^\tau\}_{\tau=1}^{k-1}, \mathcal{D}_h^{k-1}, \mathcal{F}_h, \hat{f}_h, \beta_h, \rho, \epsilon_b)$ outputs a bonus function $b_h(\cdot)$ such that, for any $z_h = (s_h, a_h) \in \mathcal{S} \times \mathcal{I}$, we have*

- *$b_h : \mathcal{S} \times \mathcal{I} \to \mathbb{R}_+$ belongs to a bonus function class $\mathcal{W}$ and denote $\mathcal{N}_b = |\mathcal{W}|$.*

- *$b_h(z_h) \geqslant \max \left\{ |f_h(z_h) - \hat{f}_h(z_h)|, f_h \in \mathcal{F}_h : \sum_{\tau=1}^{k-1} \frac{1}{(\bar{\sigma}_h^\tau)^2} \left( f_h(z_h^\tau) - \hat{f}_h^k(z_h^\tau) \right)^2 \leqslant (\beta_h)^2 \right\}.$*

- *$b_h(z_h) \leqslant C \cdot \left( \mathcal{D}_{\mathcal{F}_h} \left( z_h; \{z_h^\tau\}_{\tau=1}^{k-1}, \{\bar{\sigma}_h^\tau\}_{\tau=1}^{k-1} \right) \cdot \sqrt{(\beta_h)^2 + \rho} + \epsilon_b \cdot \beta_h \right)$ with $0 < C < \infty.$*

*Further we say the oracle $\mathcal{B}$ is consistent if for any $k < k'$ with consistent $\{\bar{\sigma}_h^\tau\}_{\tau=1}^{k-1} \subseteq \{\bar{\sigma}_h^\tau\}_{\tau=1}^{k'-1}$, $\mathcal{D}_h^{k-1} \subseteq \mathcal{D}_h^{k'-1}$, $\beta_h^k$ non-decreasing in $k$ for each $h \in [H]$ and $\hat{f}_h^k$ as defined in Equation 4, it holds that $\mathcal{B}(\{\bar{\sigma}_h^\tau\}_{\tau=1}^{k-1}, \mathcal{D}_h^{k-1}, \mathcal{F}_h, \hat{f}_h^k, \beta_h^k, \rho, \epsilon_b) \geqslant \mathcal{B}(\{\bar{\sigma}_h^\tau\}_{\tau=1}^{k'-1}, \mathcal{D}_h^{k'-1}, \mathcal{F}_h, \hat{f}_h^{k'}, \beta_h^{k'}, \rho, \epsilon_b)$ element-wise.*

With the oracle $\mathcal{B}$, we can efficiently calculate the optimistic $\overline{Q}$-value estimate $f_h^k$ with an error of $\epsilon_b$.

**Remark 3.** *Following Kong et al. (2021); Wang et al. (2020); Agarwal et al. (2023), we use the online sensitivity sub-sampling method for efficient implementation (refer Appendix B).*

**Step 3. Overly optimistic and pessimistic estimation for item-level $Q$-function (Line 11-14).** For a sharp analysis of the convergence of the optimistic estimate $f_{h,1}^k$, we define an *overly optimistic $\overline{Q}$-value* estimate $f_{h,2}^k$, as well as an *overly pessimistic $\overline{Q}$-value* estimate $f_{h,-2}^k$. Similarly to $f_{h,1}^k$, they are calculated by solving an *unweighted* regression problem (Line 11), and by adding (or subtracting) a bonus function, which is the output of the bonus oracle $\mathcal{B}$ (Line 13-14).

**Step 4. Variance estimation (Line 15 and 24).** To calculate $\bar{\sigma}_h^k$ introduced in Equation 4, we first estimate the second moment by solving the *unweighted* regression problem:

$$\hat{g}_h^k \in \arg\min_{g_h \in \mathcal{F}_h} \sum_{\tau=1}^{k-1} \left( g_h(s_h^\tau, a_h^\tau) - \left( r_h^\tau + V_{h+1,1}^k(s_{h+1}^\tau) \right)^2 \right)^2.$$

Then, denoting $z_h^k = (s_h^k, a_h^k)$ for simplicity, we calculate the estimated variance as follows (this is an informal description; for the precise formulation, see Equation D.9 and Equation D.10 in Appendix):

$$\left( \sigma_h^k \right)^2 = \min \left\{ 4, \hat{g}_h^k(z_h^k) - \left( \hat{f}_{h,-2}^k(z_h^k) \right)^2 + D_{\mathcal{F}_h}(z_h^k; \{z_h^\tau\}_{\tau=1}^{k-1}, \{\mathbf{1}^\tau\}_{\tau=1}^{k-1}) \cdot \mathcal{O}\left( \sqrt{\log \mathcal{N}\mathcal{N}_b} \right) \right\}$$

$$\bar{\sigma}_h^k = \max \left\{ \sigma_h^k, \nu, \mathcal{O}(\log \mathcal{N}\mathcal{N}_b) \cdot \sqrt{\max \left\{ f_{h,2}^k(z_h^k) - f_{h,-2}^k(z_h^k), D_{\mathcal{F}_h}(z_h^k; \{z_h^\tau\}_{\tau=1}^{k-1}, \{\bar{\sigma}_h^\tau\}_{\tau=1}^{k-1}) \right\}} \right\},$$
(5)

where $\nu$ serves as a lower bound on the variance estimate to ensure the stability of the algorithm.

**Step 5. Efficient optimistic $Q$-value estimation based on unknown item values (Line 16-18).** In this step, we address our main challenge: selecting an optimistic assortment based on the optimistic $\overline{Q}$-values, which incorporate uncertainty, while ensuring computational tractability.

To introduce optimism and encourage exploration, we need to solve the following optimization problem using the optimistic (or pessimistic) estimates of the $\overline{Q}$-values, specifically $f_{h,j}^k$ for $j = 1, \pm 2$:

$$A_{h,j}^k \in \arg\max_{A \in \mathcal{A}} \max_{\boldsymbol{\theta} \in \mathcal{C}_h^k} \sum_{a \in A} \mathcal{P}_h^k(a|s_h^k, A; \boldsymbol{\theta}) f_{h,j}^k(s_h^k, a), \quad \text{where } \mathcal{C}_h^k \text{ is defined in Equation 3.} \quad (6)$$

One of naive approach to solving the optimization problem in Equation 6 is to add bonus terms to $\sum_{a \in A} \mathcal{P}_h^k(a|s_h^k, A; \boldsymbol{\theta}_h^k) f_{h,j}^k(s_h^k, a)$ for each assortment $A$, and then enumerating all $A \in \mathcal{A}$ to find the maximum. However, this approach results in an exponential computational cost of $\mathcal{O}(|\mathcal{I}|^M)$.

To avoid this exponential computational cost, inspired by Oh & Iyengar (2019; 2021), we use optimistic MNL utilities instead of directly adding bonus terms to $\sum_{a \in A} \mathcal{P}_h^k(a|s_h^k, A; \boldsymbol{\theta}_h^k) f_{h,j}^k(s_h^k, a)$. However, unlike MNL bandits (Rusmevichientong et al., 2010; Sauré & Zeevi, 2013; Agrawal et al., 2017; 2019; Oh & Iyengar, 2019; 2021), simply using optimistic utilities does not always guarantee optimism. Specifically, increasing the MNL utilities to their optimistic values (e.g., UCB) reduces the probability of choosing the outside option $a_0$, by definition. In MNL bandits, the item values are known, and the value of the outside option $a_0$ is fixed at zero. Therefore, increasing the MNL utilities (using the optimistic utilities) lowers the probability of choosing the outside option, which in turn increases the expected value of the item values.

However, in our setting, using the optimistic utilities can decrease the expected value of $f_{h,j}^k$. To explain why: even if the true value of the outside option, $\overline{Q}_h^\star(s, a_0)$, is the lowest, its estimated value, $f_{h,j}^k(s, a_0)$, can be the highest—i.e., $f_{h,j}^k(s, a_0) > f_{h,j}^k(s, a)$ for all $a \in \mathcal{I} \backslash \{a_0\}$—due to uncertainty. In this case, increasing the MNL utilities results in a decrease in the expected value of $f_{h,j}^k$. This challenge arises from the unknown item values, $\overline{Q}_h^\star$, which is one of the main difficulties we face in combinatorial RL with preference feedback.

Hence, since we must rely on the estimates $f_{h,j}^k$ instead of the true values $\overline{Q}_h^\star$, a more refined approach is required to use the utility based on $f_{h,j}^k$. Given the confidence interval in Equation 3, we define the

*optimistic utility* $\widetilde{v}_h^k(s, a)$ and the *pessimistic utility* $\breve{v}_h^k(s, a)$ as:

$$\widetilde{v}_h^k(s,a) := \phi(s,a)^\top \boldsymbol{\theta}_h^k + \alpha_h^k \|\phi(s,a)\|_{\left(\mathbf{H}_h^k\right)^{-1}}, \quad \breve{v}_h^k(s,a) := \phi(s,a)^\top \boldsymbol{\theta}_h^k - \alpha_h^k \|\phi(s,a)\|_{\left(\mathbf{H}_h^k\right)^{-1}}, \quad (7)$$

We then use the optimistic utility only when $f_{h,j}^k(s, a_0)$ is not the highest estimate to calculate the *optimistic choice* probabilities $\widetilde{\mathcal{P}}_{h,j}^k$:

$$\widetilde{\mathcal{P}}_{h,j}^k(a|s, A) := \begin{cases} \dfrac{\exp\left(\widetilde{v}_h^k(s,a)\right)}{\sum_{a' \in A} \exp\left(\widetilde{v}_h^k(s,a')\right)}, & \text{if } \exists a \in \mathcal{I} \backslash \{a_0\} \text{ s.t. } f_{h,j}^k(s,a) \geqslant f_{h,j}^k(s,a_0) \\ \dfrac{\exp\left(\breve{v}_h^k(s,a)\right)}{\sum_{a' \in A} \exp\left(\breve{v}_h^k(s,a')\right)}, & \text{otherwise.} \end{cases} \quad (8)$$

Equipped with $\widetilde{\mathcal{P}}_{h,j}^k$, we select the assortments $A_{h,j}^k$ for each $j = 1, \pm 2$ as follows:

$$A_{h,j}^k \in \underset{A \in \mathcal{A}}{\arg\max}\, Q_{h,j}^k(s_h^k, A), \quad \text{where } Q_{h,j}^k(s_h^k, A) = \sum_{a \in A} \widetilde{\mathcal{P}}_{h,j}^k(a|s_h^k, A) f_{h,j}^k(s_h^k, a). \quad (9)$$

Here, we refer to $Q_{h,j}^k$ as the *optimistic Q-values*. This construction can induce sufficient exploration and guarantee optimism (Lemma D.9). Furthermore, by using the optimistic (or pessimistic) utilities for each item, instead of calculating bonus terms for each $A \in \mathcal{A}$, the optimization problem in Equation 9 can be solved efficiently (Davis et al., 2013).

**Remark 4.** *The optimization problem in Equation 9 can be transformed into a linear programming (LP), thus making it solvable in polynomial time with respect to $|\mathcal{I}|$ (see Appendix C).*

**Step 6. Exploration policy (Line 22).** We then offer the assortment $A_h^k$ to the user as follows:

$$A_h^k = \begin{cases} A_{h,1}^k & \text{if } f_{h',1}^k(s_{h'}^k, a_{h'}) \geqslant f_{h',2}^k(s_{h'}^k, a_{h'}) - u_k, \quad \forall a_{h'} \in A_{h',1}^k, \forall h' \leqslant h, \\ A_{h,2}^k & \text{otherwise,} \end{cases} \quad (10)$$

where $u_k$ is a carefully chosen threshold (see Table D.2 for the exact value). When the optimistic sequence $f_{h,1}^k$ and the overly optimistic sequence $f_{h,2}^k$ diverge beyond a certain threshold, we offer the assortment $A_{h,2}^k$, which is selected based on $f_{h,2}^k$. This approach ensures that by occasionally using $f_{h,2}^k$, the variance upper bound $\bar{\sigma}_h^k$, estimated from $f_{h,2}^k$, does not become overly pessimistic.

## 5 MAIN RESULTS

### 5.1 NONLINEAR FUNCTION APPROXIMATION FOR ITEM-LEVEL $Q$-FUNCTION

**Theorem 1** (Regret upper bound of MNL-VQL, proof in Section D). *Suppose Assumptions 1 and 2 hold. We assume that we have the generalized Eluder dimension $\dim_{\nu, K}(\mathcal{F}_h)$, for $h \in [H]$, as defined in Definition 3 with $\rho = 1$, and access to a consistent bonus oracle $\mathcal{B}$ satisfying Definition 4 with $\epsilon_b = \mathcal{O}(1/KH)$. Let $d_\nu = \frac{1}{H} \sum_{h=1}^H \dim_{\nu, K}(\mathcal{F}_h)$ with $\nu = \sqrt{1/KH}$, and set $u_k = \mathcal{O}(\sqrt{\log \mathcal{N}} \cdot (\log \mathcal{N} \mathcal{N}_b \cdot H^{5/2} \sqrt{d_\nu} + dH^{5/2} \sqrt{\log \mathcal{N} \mathcal{N}_b})/\sqrt{K})$. Then, for any $\delta < 1/(H^2 + 14)$, with probability at least $1 - \delta$, the regret of MNL-VQL is upper-bounded by:*

$$\mathbf{Regret}(\mathcal{M}, K) = \tilde{\mathcal{O}}\bigg( \underbrace{dH\sqrt{K} + \frac{1}{\kappa}d^2 H^2}_{\text{regret from MNL model}} + \underbrace{\sqrt{d_\nu H K \log \mathcal{N}} + d_\nu H^5 \log \mathcal{N} \log^2(\mathcal{N} \mathcal{N}_b)}_{\text{regret from general function approximation of } \overline{Q}} \bigg),$$

*where $d$ is the feature dimension of the MNL preference model, $\mathcal{N}$ is the maximum size of the function class, i.e., $\mathcal{N} = \max_{h \in [H]} |\mathcal{F}_h|$, and $\mathcal{N}_b$ it the size of the bonus function class, i.e., $\mathcal{N}_b = |\mathcal{W}|$.*

**Discussion of Theorem 1.** The first two terms arise from the regret of the MNL preference model, while the other two terms come from the regret associated with the general function approximation for item-level $Q$-values. When $H = 1$, reducing our setting to MNL bandits (though not exactly the traditional MNL bandits, as we consider a more general case where item values are unknown and the value of the outside option can be non-zero), the first two terms of our regret simplify to

$\tilde{\mathcal{O}}(d\sqrt{K} + \frac{1}{\kappa}d^2)$. This matches the known minimax optimal regret established by Lee & Oh (2024). Note that we avoid the detrimental dependence on $\kappa$ in our leading term. The other two terms of our regret, incurred from estimating item-level $Q$-values using general function approximation, are identical to the regret proposed by Agarwal et al. (2023) are only slightly worse in the lower-order terms compared to Zhao et al. (2023). Since we haven't focused on optimizing the lower-order terms, we believe they could be easily improved with a more careful analysis, as done by Zhao et al. (2023).

With respect to computational cost, by using the online sensitivity sub-sampling method (Algorithm B.1), we can efficiently implement the bonus oracle $\mathcal{B}$ (see Remark 3), with $\log |\mathcal{W}| = \log \mathcal{N}_b = \tilde{\mathcal{O}}\left(\max_{h \in [H]} \dim_{\nu,K}(\mathcal{F}_h) \cdot \log \mathcal{N} \log |\mathcal{S} \times \mathcal{I}|\right)$. Furthermore, we can avoid the exponential computational cost required to solve the optimization in Equation 9 (see Remark 4). As a result, our algorithm is both computationally tractable and statistically efficient.

### 5.2 LINEAR FUNCTION APPROXIMATION FOR ITEM-LEVEL $Q$-FUNCTION

As a special case, we also consider linear MDPs (Definition 2) with preference feedback. To clearly indicate the dependency on parameters, we denote the linear MDPs as $\mathcal{M}_{\boldsymbol{\theta}^\star, \boldsymbol{\mu}^\star, \mathbf{w}^\star}$, where $\boldsymbol{\theta}^\star = \{\boldsymbol{\theta}_h^\star\}_{h=1}^H, \boldsymbol{\mu}^\star = \{\boldsymbol{\mu}_h^\star\}_{h=1}^H$, and $\mathbf{w}^\star = \{\mathbf{w}_h^\star\}_{h=1}^H$. Note that the bonus oracle can be easily implemented using the standard elliptical bonus, which satisfies all the necessary properties (refer Appendix E).

**Theorem 2** (Informal, Regret upper bound for linear MDPs, complete version in Section E). *In linear MDPs, under the same conditions as Theorem 1, with appropriately chosen parameters, the cumulative regret of* `MNL-VQL` *is upper-bounded with probability at least $1 - \delta$ as:*

$$\textbf{Regret}\left(\mathcal{M}_{\boldsymbol{\theta}^\star, \boldsymbol{\mu}^\star, \mathbf{w}^\star}, K\right) = \tilde{\mathcal{O}}\bigg( \underbrace{dH\sqrt{K} + \frac{1}{\kappa}d^2H^2}_{\text{regret from MNL model}} + \underbrace{d^{lin}\sqrt{HK} + (d^{lin})^6 H^5}_{\text{regret from linear MDPs}} \bigg).$$

We also establish a regret lower bound for linear MDPs by constructing a novel multi-layered MDP (see Figure E.1) with a combinatorial action space and preference feedback.

**Theorem 3** (Regret lower bound for linear MDPs, proof in Section F). *Suppose that $d \geqslant 2$, $d^{lin} \geqslant 6$, $H \geqslant 3$, and $K \geqslant \max\{C \cdot (d^{lin} - 5)^2 H(H + 1)^2, C' \cdot (d - 1)^4(1 + H)/H\}$ for some constant $C, C' > 0$. Then, for any algorithm, there exists an episodic linear MDP $\mathcal{M}_{\boldsymbol{\theta}, \boldsymbol{\mu}, \mathbf{w}}$ with MNL preference feedback such that the worst-case expected regret is lower bounded as follows:*

$$\sup_{\boldsymbol{\theta}, \boldsymbol{\mu}, \mathbf{w}} \mathbb{E}_{\boldsymbol{\theta}, \boldsymbol{\mu}, \mathbf{w}}\left[\textbf{Regret}\left(\mathcal{M}_{\boldsymbol{\theta}, \boldsymbol{\mu}, \mathbf{w}}, K\right)\right] = \Omega\left(d\sqrt{HK} + d^{lin}\sqrt{HK}\right).$$

**Discussion of Theorem 2 and 3.** If $K \geqslant \tilde{\mathcal{O}}\left(d^2 H^2/\kappa^2 + (d^{lin})^{10} H^9\right)$, the regret upper bound for linear MDPs scales as $\tilde{\mathcal{O}}(dH\sqrt{K} + d^{lin}\sqrt{HK})$. Comparing this to the lower bound, there is only a gap of a factor of $\sqrt{H}$ in the regret term from the MNL model (the first term in each regret). Therefore, if $d^{lin} \geqslant d\sqrt{H}$, our algorithm achieves the minimax optimal regret (up to logarithmic factors) of $\tilde{\mathcal{O}}(d^{lin}\sqrt{HK})$. Note that if we rescale the rewards to be $1/H$ in the lower bounds of Zhou et al. (2021a) to align with our setting, their regret bound becomes the same as the second term of our regret bound, $\Omega(d^{lin}\sqrt{HK})$, which is related to linear function approximation. To the best of our knowledge, this is the first regret lower bound in linear MDPs with preference feedback.

**Numerical Experiment.** We empirically evaluate the performance of our algorithm in linear MDPs (with preference feedback), showing that it outperforms baseline algorithms while remaining computationally efficient even with a large number of items $|\mathcal{I}|$. Due to space constraints, the results are provided in Appendix G.

## 6 CONCLUSION

In this work, we study combinatorial RL with preference feedback, extending MNL bandit problems to account for the influence of user states and state transitions in applications like recommendation systems. Assuming an MNL preference model with linear utilities and using general function approximation for item values, we propose an efficient algorithm, `MNL-VQL`, which, to the best of our knowledge, provides the first statistical guarantee. As a special case, in linear MDPs, `MNL-VQL` achieves near-optimal regret, compared to our established lower bound.

# 7 REPRODUCIBILITY STATEMENT

We provide all the assumptions necessary to derive our theoretical results in Section 3, and the complete proofs of our main results and claims are included in the Appendix.

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

# Appendix

## Table of Contents

## A ILLUSTRATIVE EXPLANATION FOR COMBINATORIAL RL WITH PREFERENCE FEEDBACK

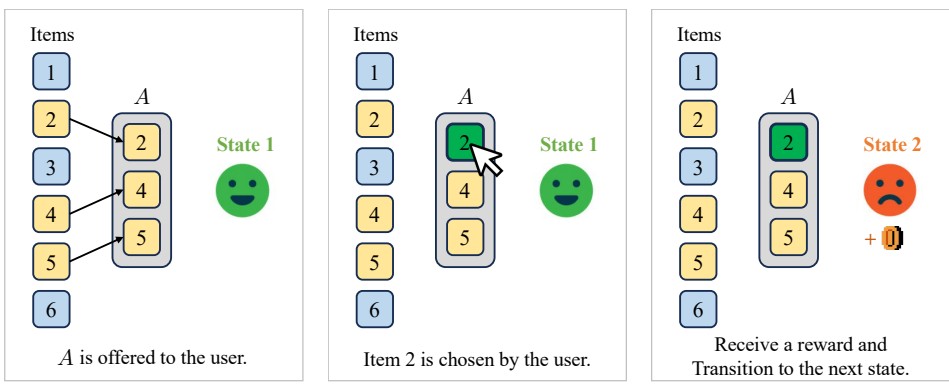

Figure A.1: Illustration of combinatorial RL with preference feedback.

In this section, we provide additional explanation of our framework, combinatorial RL with preference feedback, for better clarity. In this framework, at each episode, as a user arrives at the system (starting in the initial state, e.g., high loyalty), a learning agent selects an assortment $A$ (a set of items) and

offers it to the user (the first figure in Figure A.1). The user then chooses an item from the assortment $A$ (the second figure in Figure A.1). The agent receives a reward, along with preference (or choice) feedback, and transitions to the next state (e.g., lower loyalty) (the last figure in Figure A.1). This process repeats until the episode concludes.

The key advantage of this framework is its ability to capture the long-term value of choosing an item by considering state transitions and avoiding myopic decisions. For instance, in Figure A.1, a user may choose a *junk* item that provides a high immediate reward. However, repeatedly recommending such items can lead to user fatigue, resulting in a transition to a state of lower satisfaction or loyalty to the system, ultimately leading to a lower cumulative reward.

We compare our framework with other related works.

**vs MNL bandits.** Our framework can be considered as a multi-step extension of MNL bandits (Rusmevichientong et al., 2010; Sauré & Zeevi, 2013; Agrawal et al., 2017; 2019; Oh & Iyengar, 2019; 2021; Perivier & Goyal, 2022; Agrawal et al., 2023; Zhang & Sugiyama, 2024; Lee & Oh, 2024). In MNL bandits, there are no state transitions; thus, in Figure E.1, the user exits the system immediately after receiving a reward.

Another important difference is that, in MNL bandits, the value (reward) of choosing an item is assumed to be known, and the value of choosing the outside option $a_0$ is always assumed to be zero. In contrast, in our framework, the value of choosing an item is unknown due to the stochastic nature of rewards and transition probabilities. Additionally, we allow the value of choosing the outside option $a_0$ to be non-zero.

**vs Cascading RL.** In cascading RL (Du et al., 2024), the agent also selects a set of items, and state transitions are taken into account when making decisions. However, these items are offered to the user one by one, and the user decides whether to choose the currently offered item.

For example, in Figure E.1, only the first item, item 2, is shown to the user. If the user chooses item 2, the agent receives a reward and transitions to the next state. If the user does not choose item 2, the next item, item 4, is shown. If the user does not choose any item, the agent considers this as choosing the outside option $a_0$ and transitions to the next state.

Cascading RL fundamentally differs from our framework because the user does not compare multiple items at once, so it does not involve relative preference feedback. Another key distinction is that in cascading RL, the probability of choosing each item is independent of the others in the chosen set of items In contrast, in our MNL preference model, the choice probability of an item is influenced by the other items in the assortment.

**vs PbRL.** In preference-based RL (PbRL) (Akrour et al., 2012; Wirth et al., 2017; Christiano et al., 2017; Ouyang et al., 2022; Saha et al., 2023; Zhu et al., 2023; Zhan et al., 2023), the agent learns not from explicit numerical rewards, but through preferences as feedback. The user is presented with two (or sometimes multiple) items and chooses a preferred one.

In our framework, if we treat the reward signal generated by the user's choice as a preference signal instead of a numerical reward, we can learn the policy based on user preferences, similar to PbRL, without relying on explicit rewards. However, our framework differs fundamentally from PbRL because our goal is not just to offer a single item, but multiple items—a combinatorial (base) action—at each timestep.

# B IMPLEMENTING BONUS ORACLE $\mathcal{B}$ USING ONLINE-SUBSAMPLING

The guarantees of Algorithm 1 rely on a consistent bonus oracle, $\mathcal{B}$, that satisfies Definition 4. To implement this oracle, we use the online sensitivity sub-sampling approach described by Agarwal et al. (2023), which builds on the original sensitivity sub-sampling method proposed by Kong et al. (2021) and Wang et al. (2020).

For completeness, we include the sub-sampling procedure in Algorithm B.1 and show its guarantees in Proposition B.2. Let $z = (s, a) \in \mathcal{S} \times \mathcal{I}$. We first define the weighted data set $\mathcal{Z}$, where each

element is $(z, \bar{\sigma}(z))$, and introduce the *weighted sensitivity score* as follows:

$$\text{sensitivity}_{\mathcal{Z},\mathcal{F},\gamma,\nu}(z) = \min \left\{ \sup_{f,f' \in \mathcal{F}} \frac{\frac{1}{\bar{\sigma}^2(z)} (f(z) - f'(z))^2}{\min \left\{ \sum_{z' \in \mathcal{Z}} \frac{1}{\bar{\sigma}^2(z')} (f(z') - f'(z'))^2 , \frac{K(H+1)^2}{\nu^2} \right\} + \gamma^2}, 1 \right\}.$$

Now we introduce the sub-sampling procedure.

---

**Algorithm B.1** Online Sensitivity Sub-sampling with Weights

---

1: **Inputs:** function class $\mathcal{F}$, current sub-sampled dataset $\hat{\mathcal{Z}} \subseteq \mathcal{S} \times \mathcal{I}$, new state-action pair $s, a$,
   parameter $\gamma$, threshold $\nu > 0$, failure probability $\delta$.
2: **Parameter:** $1 \leqslant C < \infty$
3: Let $p_z$ be the smallest real number such that

   $$1/p_z \text{ is an integer and } p_z \geqslant \min \left\{ 1, C \cdot \text{sensitivity}_{\hat{\mathcal{Z}},\mathcal{F},\gamma,\nu}(z) \cdot \log(K\mathcal{N}/\delta) \right\}.$$

4: Independently add $1/p_z$ copies of $(z, \bar{\sigma}(z))$ into $\hat{\mathcal{Z}}$ with probability $p_z$.
5: **Return:** $\hat{\mathcal{Z}}$.

---

For the weighted dataset $\mathcal{Z}_h^{k-1} = \{(s_h^\tau, a_h^\tau), \bar{\sigma}_h^\tau\}_{\tau=1}^{k-1}$, we defined $\|f\|^2_{\mathcal{Z}_h^{k-1}} = \sum_{z \in \mathcal{Z}_h^{k-1}} \frac{1}{\bar{\sigma}^2(z)} f^2(z)$,
i.e., weighted sum of $\ell_2$-norm square. We denote $\hat{\mathcal{Z}}_h^{k-1}$ as the dataset sub-sampled from $\mathcal{Z}_h^{k-1}$. At
every $(k, h) \in [K] \times [H]$, we call Algorithm B.1 with the current sub-sampled dataset $\hat{\mathcal{Z}}_h^{k-1}$ and the
new state action-pair $z_h^k = (s_h^k, a_h^k)$ to generate the next sub-sampled dataset $\hat{\mathcal{Z}}_h^k$.

The following proposition shows that the distance of any two functions measured by the historical
dataset $\mathcal{Z}_h^{k-1}$ is well approximated by the subsamping dataset $\hat{\mathcal{Z}}_h^{k-1}$. Additionally, it shows that both
the number of distinct elements in $|\hat{\mathcal{Z}}_h^{k-1}|$ and its the total size (counting repetitions) do not scale
with $\text{poly}(\mathcal{S})$.

**Proposition B.1** (Guarantees of online sensitivity sub-sampling, Proposition 13 of Agarwal et al.
2023). *Let $z = (s, a) \in \mathcal{S} \times \mathcal{I}$. When $\bar{\sigma}(z) \geqslant \nu$ for any $z$, then with probability at least $1 - \delta$, it
holds that*

$$\sup_{f_1,f_1:\|f_1-f_2\|^2_{\mathcal{Z}_h^k} \leqslant \gamma^2} |f_1(z) - f_2(z)| \leqslant \sup_{f_1,f_1:\|f_1-f_2\|^2_{\hat{\mathcal{Z}}_h^k} \leqslant 10^2\gamma^2} |f_1(z) - f_2(z)|$$

$$\leqslant \sup_{f_1,f_1:\|f_1-f_2\|^2_{\mathcal{Z}_h^k} \leqslant 10^4\gamma^2} |f_1(z) - f_2(z)|.$$

*Further, for any $(k, h) \in [K] \times [H]$, the number of distinct elements in sub-sampled dataset $\hat{\mathcal{Z}}_h^k$
is always bounded by $\mathcal{O}\left( \log \frac{K\mathcal{N}}{\delta} \cdot \max_{h \in [H]} \dim_{\nu,K}(\mathcal{F}_h) \right)$ and the total size of $\hat{\mathcal{Z}}_h^k$ is bounded by
$\mathcal{O}(K^3/\delta)$.*

We can assert that the predictive differences between the functions are preserved up to constant
factors, while requiring significantly less data. Then, the size of the bonus class $\mathcal{W}$ in Definition 4 is
bounded as follows:

**Proposition B.2** (Implementing $\mathcal{B}$ using online-subsampling, Corollary 14 of Agarwal et al. 2023).
*There exists an algorithm (see Algorithm B.1) such that, with probability at least $1 - \delta$, implements a
consistent bonus oracle $\mathcal{B}$ with $\epsilon_b = 0$ for all $(k, h) \in [K] \times [H]$, where*

$$\log |\mathcal{W}| \leqslant \mathcal{O}\left( \max_{h \in [H]} \dim_{\nu,K}(\mathcal{F}_h) \cdot \log \frac{K\mathcal{N}}{\delta} \log \frac{K|\mathcal{S} \times \mathcal{I}|}{\delta} \right).$$

## C  EFFICIENT COMBINATORIAL OPTIMIZATION

In this section, we explain how to solve the combinatorial optimization problem in Equation 9,
following the method outlined in Davis et al. (2013); Ie et al. (2019).

To find an assortment $A \in \mathcal{A}$ that maximizes the optimistic $Q$-value, a fundamental step in $Q$-learning and crucial for inducing exploration, we must solve the following combinatorial optimization problem:

$$\max_{A \in \mathcal{A}} \sum_{a \in A} \widetilde{\mathcal{P}}_{h,j}^k(a|s_h^k, A) f_{h,j}^k(s_h^k, a), \tag{C.1}$$

where $\widetilde{\mathcal{P}}_{h,j}^k$ is the optimistic choice probability as defined in Equation 8 (also in Equation D.17), and $f_{h,j}^k$ is the $\overline{Q}$-value estimate (item-level $Q$-values) as defined in Equation D.15.

Fix $(k, h, s, j) \in [K] \times [H] \times \mathcal{S} \times \{1, 2, -2\}$. For simplicity, we will abbreviate these indices. Accordingly, we denote $w_a = \exp\left(\widetilde{v}_h^k(s_h^k, a)\right)$ or $w_a = \exp\left(\breve{v}_h^k(s_h^k, a)\right)$, depending on the value of $j$. Additionally, let $\tilde{f}_a = f_{h,j}^k(s_h^k, a)$ for simplicity.

We can then express the optimization problem in Equation C.1 in terms of $w$ as fractional mixed-integer program (MIP), with binary variables $x_a \in \{0, 1\}$ for each item $a \in \mathcal{I}\backslash\{a_0\}$, indicating whether $a$ is included in the assortment $A$:

$$\max \frac{w_{a_0}\tilde{f}_{a_0} + \sum_{a \in \mathcal{I}\backslash\{a_0\}} x_a w_a \tilde{f}_a}{w_{a_0} + \sum_{a' \in \mathcal{I}\backslash\{a_0\}} x_{a'} w_{a'}} \tag{C.2}$$

$$\text{s.t.} \sum_{a \in \mathcal{I}\backslash\{a_0\}} x_a = M - 1;$$

$$x_a \in \{0, 1\}, \ \forall a \in \mathcal{I}\backslash\{a_0\}.$$

By Chen & Hausman (2000), the binary indicator in the MIP can be relaxed, resulting in the following fractional linear program (LP):

$$\max \frac{w_{a_0}\tilde{f}_{a_0} + \sum_{a \in \mathcal{I}\backslash\{a_0\}} x_a w_a \tilde{f}_a}{w_{a_0} + \sum_{a' \in \mathcal{I}\backslash\{a_0\}} x_{a'} w_{a'}} \tag{C.3}$$

$$\text{s.t.} \sum_{a \in \mathcal{I}\backslash\{a_0\}} x_a = M - 1;$$

$$0 \leqslant x_a \leqslant 1, \ \forall a \in \mathcal{I}\backslash\{a_0\}.$$

Since this relaxed problem is a fractional linear program (LP), using the Charnes-Cooper method (Cooper et al., 1962), it can be transformed into a (non-fractional) LP. To achieve this, we introduce additional variables:

$$t = \frac{1}{w_{a_0} + \sum_{a' \in \mathcal{I}\backslash\{a_0\}} x_{a'} w_{a'}}, \quad y_a = \frac{x_a}{w_{a_0} + \sum_{a' \in \mathcal{I}\backslash\{a_0\}} x_{a'} w_{a'}}.$$

Then, we can obtain the following LP:

$$\max \sum_{a \in \mathcal{I}\backslash\{a_0\}} \tilde{f}_a w_a y_a + \tilde{f}_{a_0} w_{a_0} t \tag{C.4}$$

$$\text{s.t.} \sum_{a \in \mathcal{I}\backslash\{a_0\}} w_a y_a + w_{a_0} t = 1;$$

$$\sum_{a \in \mathcal{I}\backslash\{a_0\}} y_a \leqslant (M - 1)t;$$

$$t \geqslant 0, \quad 0 \leqslant x_a \leqslant 1, \ \forall a \in \mathcal{I}\backslash\{a_0\}.$$

The optimal solution $(y_{a_1}^\star, \ldots, y_{a_N}^\star, t^\star)$ to this LP in Equation C.4 provides the optimal $x_i$ values for the fractional LP in Equation C.3 by setting $x_a = y_a^\star/t^\star$. This, in ture, determines the optimal assortment in the original fractional MIP (Equation C.2) by including any item where $y_a^\star > 0$. Thus, the optimization problem is proven to be solvable in polynomial time.

Table D.1: Summary of notations

| Notation | Meaning | Remark |
|---|---|---|
| $\mathcal{S}, \mathcal{A}, \mathcal{I}$ | state space, action (assortment) space, item set | |
| $k, h$ | $k \in [K]$ episode, $h \in [H+1]$ horizon | |
| $r_h^k, s_h^k, A_h^k, a_h^k$ | reward, state, action and item at $k, h$ | |
| $r_h, s_h, A_h, a_h$ | random reward, state, action and item $h$ | |
| $z$ | shorthand for state-item pair $(s, a)$ | |
| $\overline{Q} : \mathcal{S} \times \mathcal{I} \to \mathbb{R}$ | item-level $Q$-value function | |
| $\mathcal{T}_h, \mathcal{T}_h^2$ | Bellman operator and second-moment operator | |
| $\mathcal{D}_h^{k-1}$ | data set $\{(s_h^\tau, a_h^\tau, r_h^\tau, s_{h+1}^\tau)\}_{\tau=1}^{k-1}$ | |
| $\mathcal{F}_h$ | function class for horizon $h \in [H]$ | Ass. 2 |
| $\mathcal{F}_h^{\mathrm{lin}}$ | linear function class for horizon $h \in [H]$ | Eqn. 1 |
| $\mathcal{F}_h^{\mathrm{lin}}(\epsilon_c)$ | $\epsilon_c$-cover of linear function class $\mathcal{F}_h^{\mathrm{lin}}$ | |
| $\mathcal{W}$ | bonus function class defined for bonus oracle $\mathcal{B}$ | Def. 4 |
| $\epsilon_b$ | error parameter for bonus oracle | |
| $\mathcal{N}$ | maximal size of function class, i.e., $\max_{h \in [H]} \|\mathcal{F}_h\|$ | |
| $\mathcal{N}_b$ | size of bonus function class $\|\mathcal{W}\|$ | |
| $D_{\mathcal{F}}^2\left(z; \{z^\tau\}_{\tau=1}^{k-1}, \{\sigma^\tau\}_{\tau=1}^{k-1}\right)$ | $:= \sup_{f_1, f_2, \in \mathcal{F}} \frac{(f_1(z) - f_2(z))^2}{\sum_{\tau=1}^{k-1} \frac{1}{(\sigma^\tau)^2}(f_1(z^\tau) - f_2(z^\tau))^2 + \rho}$ | $\rho$ param. |
| $\dim_{\nu, K}(\mathcal{F})$ | generalized Eluder dimension defined in Definition 3 | $\nu$ param. |
| $d_\nu$ | $:= \frac{1}{H} \sum_{h=1}^{H} \dim_{\nu, K}(\mathcal{F}_h)$ (Definition 3) | $\nu$ param. |
| $\ell_h^k(\boldsymbol{\theta})$ | $-\sum_{a \in A_h^k} y_h^k(a) \log \mathcal{P}_h(a\|s_h^k, A_h^k; \boldsymbol{\theta})$, loss for MNL model at $k, h$ | Eqn. D.1 |
| $\mathbf{H}_h^k, \tilde{\mathbf{H}}_h^k$ | $= \lambda \mathbf{I}_d + \sum_{\tau=1}^{k-1} \nabla^2 \ell_h^\tau(\boldsymbol{\theta}^{\tau+1}), = \mathbf{H}_h^k + \eta \nabla^2 \ell_h^k(\boldsymbol{\theta}_h^k)$, respectively | Eqn. D.3 |
| $\mathcal{C}_h^k$ | confidence interval for MNL model at $k, h$ | Eqn. D.5 |
| $f_{h,1}^k$ | optimistic $\overline{Q}$ at $k, h$ | |
| $\hat{f}_{h,1}^k$ | solution of fitting weighted regression at $k, h$ | Eqn. D.7 |
| $\mathcal{F}_{h,1}^k$ | version space of optimistic $\overline{Q}$ at $k, h$ | Eqn. D.8 |
| $f_{h,\pm 2}^k$ | overly optimistic (pessimistic) $\overline{Q}$ at $k, h$ | |
| $\hat{f}_{h,\pm 2}^k$ | solution of fitting unweighted regression at $k, h$ | Eqn. D.11 |
| $\mathcal{F}_{h,\pm 2}^k$ | version space of overly optimistic (pessimistic) $\overline{Q}$ at $k, h$ | Eqn. D.12 |
| $\hat{g}_h^k$ | solution of fitting second-moment regression at $k, h$ | Eqn. D.13 |
| $\mathcal{G}_h^k$ | version space of second-moment estimates at $k, h$ | Eqn. D.14 |
| $\mathcal{E}^\theta$ | event that $\{\boldsymbol{\theta}_h^\star \in \mathcal{C}_h^k$ for all $k \geqslant 1$ and all $h \in [H]\}$ | |
| $\mathcal{E}_h^k$ | event that $\{\mathcal{T}_h V_{h+1,j}^k \in \mathcal{F}_{h,j}^k$ for $j = 1, \pm 2$ and $\mathcal{T}_h^2 V_{h+1,1}^k \in \mathcal{G}_h^k\}$ | Ass. 2 |
| $\mathcal{E}_{\leqslant k}$ | joint event that $\bigcap_{\tau=1}^{k} \bigcap_{h=1}^{H} \mathcal{E}_h^\tau$ | |
| $\tilde{v}_h^k(s_h^k, a), \breve{v}_h^k(s_h^k, a)$ | optimistic (pessimistic) utility defined in Equation 7 | |
| $\tilde{\mathcal{P}}_h^k(a\|s, A)$ | optimistic choice probability defined in Equation 8 | |
| $Q_{h,j}^k(s, A)$ | $:= \sum_{a \in A} \tilde{\mathcal{P}}_{h,j}^k(a\|s, A) f_{h,j}^k(s, a)$ for $j = 1, \pm 2$ | |
| $V_{h,j}^k(s)$ | $:= \max_{A \in \mathcal{A}} Q_{h,j}^k(s, A)$ for $j = 1, \pm 2$ | |
| $A_{h,j}^k$ | $\in \arg\max_{A \in \mathcal{A}} \sum_{a \in A} \tilde{\mathcal{P}}_{h,j}^k(a\|s_h^k, A) f_{h,j}^k(s_h^k, a)$ for $j = 1, \pm 2$ | |
| $A_h^k$ | chosen assortment at $k, h$ by assortment selection rule in Equation 10 | |
| $Q_h^k(s, A), V_h^k(s)$ | *realized* optimistic values determined by Equation D.18 | Eqn. 10 |
| $h_k$ | random $h$ when first taking action $A_{h,2}^k$ at $k$, i.e., $A_h^k = A_{h,2}^k$ | Eqn. 10 |
| $\mathcal{K}_o, \mathcal{K}_{oo}$ | disjoint subsets of $[K]$ when $h_k = H + 1$ or $h_k \in [H]$ | Eqn. 10 |
| $b_{h,j}^k$ | bonus term obtained in Line 9 and 12 using $\mathcal{B}$ | Def. 4 |
| $\mathbb{E}_{\mathbb{P}}[\cdot\|s_h^k, a_h^k], \mathbb{V}_{\mathbb{P}}[\cdot\|s_h^k, a_h^k]$ | $\mathbb{E}_{s_{h+1} \sim \mathbb{P}_h(\cdot\|s_h^k, a_h^k)}[\cdot\|s_h^k, a_h^k], \mathbb{V}_{s_{h+1} \sim \mathbb{P}_h(\cdot\|s_h^k, a_h^k)}[\cdot\|s_h^k, a_h^k]$ | |
| $\mathbb{E}_{\mathcal{P}}[\cdot\|s_h^k, A_h^k]$ | $\mathbb{E}_{a_h \sim \mathcal{P}_h(\cdot\|s_h^k, A_h^k)}[\cdot\|s_h^k, A_h^k]$ | |

Table D.2: Summary of parameter choices

| Notation | Choice | Remark |
|---|---|---|
| $\delta$ | $\delta \in (0, 1/(H^2 + 14))$ | |
| $\delta_h^k$ | $\delta/((K+1)(H+1))$ | |
| $\eta$ | $\frac{1}{2}\log(M+1) + B + 1$, step-size parameter for OMD | Eqn. D.2 |
| $\lambda$ | $84\sqrt{2}d\eta$, regularization parameter | |
| $\alpha_h^k$ | $\mathcal{O}(\sqrt{d}\log k \log M)$, confidence radius of $\mathcal{C}_h^k$ | Eqn. D.6 |
| $\epsilon_c$ | error due to taking covering of function class | |
| $\nu$ | $\sqrt{1/KH}$ | Def. 3 |
| $\rho$ | $1$ | Def. 3 |
| $o(\delta)$ | $\sqrt{\log \frac{\mathcal{N}^2(2\log(4LK/\nu)+2)(\log(8L/\nu^2)+2)}{\delta}}$ | |
| $\iota(\delta)$ | $3\sqrt{\log \frac{\mathcal{N}\mathcal{N}_b(2\log(4LK/\nu)+2)(\log(8L/\nu^2)+2)}{\delta}}$ | |
| $\beta_{h,1}^k$ | $\sqrt{\left(6\sqrt{\rho} + 156\right)\cdot\log\frac{\mathcal{N}^2(K+1)(H+1)(2\log\frac{4LK}{\nu}+2)(\log\frac{8L}{\nu^2}+2)}{\delta}}$, confidence radius of $\mathcal{F}_{h,1}^k$ | Eqn. D.8 |
| $\iota'(\delta)$ | $\sqrt{2\log\frac{\mathcal{N}\mathcal{N}_b(2\log(18LK)+2)(\log(18L)+2)}{\delta}}$ | |
| $\beta_{h,2}^k$ | $\sqrt{2(24L+21)(\iota'(\delta_h^k))^2}$, confidence radius of $\mathcal{F}_{h,\pm2}^k$ | Eqn. D.12 |
| $\iota''(\delta)$ | $\sqrt{2\log\frac{\mathcal{N}\mathcal{N}_b(2\log(32LK)+2)(\log(32L)+2)}{\delta}}$ | |
| $\bar{\beta}_h^k$ | $\sqrt{8(11L+9)(\iota''(\delta_h^k))^2}$, confidence radius of $\mathcal{G}_h^k$ | Eqn. D.14 |
| $(\sigma_h^k)^2$ | $\min\left\{4, \ \hat{g}_h^k(z_h^k) - \left(\hat{f}_{h,-2}^k(z_h^k)\right)^2 \right.$ | Eqn. 5 |
| | $\left. + D_{\mathcal{F}_h}^2\left(z_h^k; \{z_h^\tau\}_{\tau=1}^{k-1}, \{\mathbf{1}^\tau\}_{\tau=1}^{k-1}\right)\cdot\left(\sqrt{\left(\bar{\beta}_h^k\right)^2 + \rho} + 2L\sqrt{\left(\beta_{h,2}^k\right)^2 + \rho}\right)\right\}$ | |
| $\bar{\sigma}_h^k$ | $\max\left\{\sigma_h^k, \nu, \sqrt{2}\iota(\delta_h^k)\sqrt{f_{h,2}^k(z_h^k) - f_{h,-2}^k(z_h^k)},\right.$ | |
| | $\left. 2\left(\sqrt{o(\delta_h^k)} + \iota(\delta_h^k)\right)\cdot\sqrt{D_{\mathcal{F}_h}\left(z_h^k; \{z_h^\tau\}_{\tau=1}^{k-1}, \{\bar{\sigma}_h^\tau\}_{\tau=1}^{k-1}\right)}\right\}$ | Eqn. D.7 |
| $u_k$ | $C\cdot\left(\sqrt{\log\frac{\mathcal{N}KH}{\nu\delta}}\cdot\left(\log\frac{\mathcal{N}\mathcal{N}_bKH}{\nu\delta}\cdot H^{5/2}\sqrt{d_\nu} + \sqrt{k}H\epsilon_b\right)\right.$ | |
| | $\left. + dH^{5/2}\log K\log M\sqrt{\log\frac{\mathcal{N}\mathcal{N}_bKH}{\nu\delta}}\right)/\sqrt{k}$ for $C > 0$ | Eqn. 10 |

## D  PROOF OF THEOREM 1

### D.1  NOTATIONS AND PRELIMINARIES

In this subsection, for easy reference, we introduce notations and definitions used throughout the proof. The key notations are summarized in Table D.1, and the specific parameter choices are listed in Table D.2.

**Online parameter update and confidence interval for MNL preference model.** We define the multinomial logistic loss function at $(k,h) \in [K] \times [H]$ as follows:

$$\ell_h^k(\boldsymbol{\theta}) := -\sum_{a \in A_h^k} y_h^k(a)\log\mathcal{P}_h(a|s_h^k, A_h^k; \boldsymbol{\theta}). \tag{D.1}$$

To achieve constant-time parameter estimation, we use the online mirror descent algorithm to estimate the true parameter $\boldsymbol{\theta}_h^\star$:

$$\boldsymbol{\theta}_h^{k+1} \in \underset{\boldsymbol{\theta} \in \Theta}{\arg\min} \langle \nabla\ell_h^k(\boldsymbol{\theta}_h^k), \boldsymbol{\theta} \rangle + \frac{1}{2\eta}\|\boldsymbol{\theta} - \boldsymbol{\theta}_h^k\|_{\tilde{\mathbf{H}}_h^k}^2, \quad \text{where } \Theta = \left\{ \boldsymbol{\theta} \in \mathbb{R}^d : \|\boldsymbol{\theta}\|_2 \leqslant B \right\}, \quad \text{(D.2)}$$

where $\eta = \frac{1}{2}\log(M+1) + B + 1$ is the step-size parameter, and the related matrices are defined as:

$$\mathbf{H}_h^k := \lambda\mathbf{I}_d + \sum_{\tau=1}^{k-1} \nabla^2\ell_h^\tau(\boldsymbol{\theta}_h^{\tau+1}),$$

$$\tilde{\mathbf{H}}_h^k := \mathbf{H}_h^k + \eta\nabla^2\ell_h^k(\boldsymbol{\theta}_h^k), \tag{D.3}$$

where

$$\nabla^2\ell_h^k(\boldsymbol{\theta}) = \sum_{a \in A_h^k} \mathcal{P}_h(a|s_h^k, A_h^k; \boldsymbol{\theta})\phi(s_h^k, a)\phi(s_h^k, a)^\top$$

$$- \sum_{a \in A_h^k} \sum_{a' \in A_h^k} \mathcal{P}_h(a|s_h^k, A_h^k; \boldsymbol{\theta})\mathcal{P}_h(a'|s_h^k, A_h^k; \boldsymbol{\theta})\phi(s_h^k, a)\phi(s_h^k, a')^\top.$$

By a standard online mirror descent formulation (Orabona, 2019), Equation D.2 can be solved using a single projected gradient step through the following equivalent formula:

$$\bar{\boldsymbol{\theta}}_h^{k+1} = \boldsymbol{\theta}_h^k - \eta\left(\tilde{H}_h^k\right)^{-1}\nabla\ell_h^k(\boldsymbol{\theta}_h^k), \quad \text{and} \quad \boldsymbol{\theta}_h^{k+1} \in \underset{\boldsymbol{\theta} \in \Theta}{\arg\min} \|\boldsymbol{\theta} - \bar{\boldsymbol{\theta}}_h^{k+1}\|_{\tilde{H}_h^k}, \tag{D.4}$$

which enjoys a computational cost of only $\mathcal{O}(Md^3)$, completely independent of $k$ (Mhammedi et al., 2019; Zhang & Sugiyama, 2024; Lee & Oh, 2024).

We define the confidence interval at $(k, h) \in [K] \times [H]$ as follows:

$$\mathcal{C}_h^k := \left\{ \boldsymbol{\theta} \in \Theta : \left\|\boldsymbol{\theta} - \boldsymbol{\theta}_h^k\right\|_{\mathbf{H}_h^k} \leqslant \alpha_h^k \right\}, \tag{D.5}$$

where the radius of the confidence interval $\mathcal{C}_h^k$ is as follows:

$$\alpha_h^k = \left[ 2\eta\left( (3\log(1 + (M+1)k) + B + 2)\left( \frac{17}{16}\lambda + 2\sqrt{\lambda}\log\left( \frac{2\sqrt{1+2k}}{\delta} \right) \right) \right.\right.$$

$$\left.\left. + 16\left( \log\left( \frac{2\sqrt{1+2k}}{\delta} \right) \right)^2 \right) + \frac{7\sqrt{6}}{6}d\eta\log\left( 1 + \frac{k+1}{2\lambda} \right) + 2 \right) + \lambda B^2 \right]^{1/2} \tag{D.6}$$

Then, we define the optimistic and pessimistic utility as follows:

$$\widetilde{v}_h^k(s, a) := \phi(s, a)^\top\boldsymbol{\theta}_h^k + \alpha_h^k\|\phi(s, a)\|_{(\mathbf{H}_h^k)^{-1}}, \quad \check{v}_h^k(s, a) := \phi(s, a)^\top\boldsymbol{\theta}_h^k - \alpha_h^k\|\phi(s, a)\|_{(\mathbf{H}_h^k)^{-1}},$$

**Regression and confidence intervals for item-level functions.** In this paper, we define $\mathcal{N} := \max_{h \in [H]} |\mathcal{F}_h|$ as the maximum size of the function classes $\mathcal{F}_1, \ldots, \mathcal{F}_H$ and $\mathcal{N}_b = |\mathcal{W}|$ as the size of the bonus function class $\mathcal{W}$.

For all $(k, h) \in [K] \times [H]$, the weighted regression problem for fitting the optimistic item-level $Q$-functions, $\overline{Q}$, along with the version space of these functions, is defined as:

$$\hat{f}_{h,1}^k \in \underset{f_h \in \mathcal{F}_h}{\arg\min} \sum_{\tau=1}^{k-1} \frac{1}{(\bar{\sigma}_h^\tau)^2}\left( f_h(s_h^\tau, a_h^\tau) - r_h^\tau - V_{h+1,1}^k(s_{h+1}^\tau) \right)^2, \tag{D.7}$$

$$\mathcal{F}_{h,1}^k := \left\{ f_h \in \mathcal{F}_h : \sum_{\tau=1}^{k-1} \frac{1}{(\bar{\sigma}_h^\tau)^2}\left( f_h(s_h^\tau, a_h^\tau) - \hat{f}_{h,1}^k(s_h^\tau, a_h^\tau) \right)^2 \leqslant \left(\beta_{h,1}^k\right)^2 \right\}. \tag{D.8}$$

Let $z_h^k = (s_h^k, a_h^k)$. The parameters are as follows (for $k \geqslant 2$):

$$
\left(\sigma_h^k\right)^2 := \min \left\{ 4, \; \hat{g}_h^k(z_h^k) - \left(\hat{f}_{h,-2}^k(z_h^k)\right)^2 \right.
$$

$$
\left. + D_{\mathcal{F}_h}\left(z_h^k; \{z_h^\tau\}_{\tau=1}^{k-1}, \{\mathbf{1}^\tau\}_{\tau=1}^{k-1}\right) \cdot \left(\sqrt{\left(\bar{\beta}_h^k\right)^2 + \rho} + \sqrt{\left(\beta_{h,2}^k\right)^2 + \rho}\right) \right\}, \quad \text{(D.9)}
$$

$$
\bar{\sigma}_h^k := \max \left\{ \sigma_h^k, \nu, \sqrt{2}\iota(\delta_h^k)\sqrt{f_{h,2}^k(z_h^k) - f_{h,-2}^k(z_h^k)}, \right.
$$

$$
\left. 2\left(\sqrt{o(\delta_h^k)} + \iota(\delta_h^k)\right) \cdot \sqrt{D_{\mathcal{F}_h}\left(z_h^k; \{z_h^\tau\}_{\tau=1}^{k-1}, \{\bar{\sigma}_h^\tau\}_{\tau=1}^{k-1}\right)} \right\}, \quad \text{(D.10)}
$$

$$
\beta_{h,1}^k := \sqrt{(6\sqrt{\rho} + 156) \cdot \log \frac{\mathcal{N}^2(K+1)(H+1)(2\log\frac{4LK}{\nu} + 2)(\log\frac{8L}{\nu^2} + 2)}{\delta}},
$$

$$
o(\delta_h^k) := \sqrt{\log \frac{\mathcal{N}^2(2\log(4LK/\nu) + 2)(\log(8L/\nu^2) + 2)}{\delta_h^k}}, \quad \delta_h^k := \frac{\delta}{(K+1)(H+1)},
$$

$$
\iota(\delta_h^k) := 3\sqrt{\log \frac{\mathcal{N}\mathcal{N}_b(2\log(4LK/\nu) + 2)(\log(8L/\nu^2) + 2)}{\delta_h^k}}.
$$

For all $(k, h) \in [K] \times [H]$, the unweighted regression for fitting overly optimistic and overly pessimistic item-level $Q$-functions, along with the version space of them, is defined as follows:

$$
\hat{f}_{h,\pm 2}^k \in \arg\min_{f_h \in \mathcal{F}_h} \sum_{\tau=1}^{k-1} \left(f_h(s_h^\tau, a_h^\tau) - r_h^\tau - V_{h+1,\pm 2}^k(s_{h+1}^\tau)\right)^2, \quad \text{(D.11)}
$$

$$
\mathcal{F}_{h,\pm 2}^k := \left\{ f_h \in \mathcal{F}_h : \sum_{\tau=1}^{k-1} \left(f_h(s_h^\tau, a_h^\tau) - \hat{f}_{h,\pm 2}^k(s_h^\tau, a_h^\tau)\right)^2 \leqslant \left(\beta_{h,2}^k\right)^2 \right\}. \quad \text{(D.12)}
$$

We choose the parameters as follows:

$$
\beta_{h,2}^k := \sqrt{2(24L + 21)(\iota'(\delta_h^k))^2},
$$

$$
\iota'(\delta_h^k) := \sqrt{2\log \frac{\mathcal{N}\mathcal{N}_b(2\log(18LK) + 2)(\log(18L) + 2)}{\delta_h^k}}, \quad \delta_h^k := \frac{\delta}{(K+1)(H+1)}.
$$

For all $(k, h) \in [K] \times [H]$, the unweighted regression for fitting second-moment function values to item-level $Q$-functions, and their version space, is as follows:

$$
\hat{g}_h^k \in \arg\min_{g_h \in \mathcal{F}_h} \sum_{\tau=1}^{k-1} \left(g_h(s_h^\tau, a_h^\tau) - \left(r_h^\tau + V_{h+1,1}^k(s_{h+1}^\tau)\right)^2\right)^2, \quad \text{(D.13)}
$$

$$
\mathcal{G}_h^k := \left\{ g_h \in \mathcal{F}_h : \sum_{\tau=1}^{k-1} \left(g_h(s_h^\tau, a_h^\tau) - \hat{g}_h^k(s_h^\tau, a_h^\tau)\right)^2 \leqslant \left(\bar{\beta}_h^k\right)^2 \right\}. \quad \text{(D.14)}
$$

We choose the parameters as follows:

$$
\bar{\beta}_h^k := \sqrt{8(11L + 9)(\iota''(\delta_h^k))^2},
$$

$$
\iota''(\delta_h^k) := \sqrt{2\log \frac{\mathcal{N}\mathcal{N}_b(2\log(32LK) + 2)(\log(32L) + 2)}{\delta}}, \quad \delta_h^k := \frac{\delta}{(K+1)(H+1)}.
$$

Given the center of the constructed confidence intervals, $\hat{f}_{h,j}^k$ for $j = 1, \pm 2$, we define the optimistic, overly optimistic, and overly pessimistic $\overline{Q}$-values as follows:

$$f_{h,1}^k(\cdot, \cdot) := \min\left\{\hat{f}_{h,1}^k(\cdot, \cdot) + b_{h,1}^k(\cdot, \cdot), 1\right\},$$

$$f_{h,2}^k(\cdot, \cdot) := \min\left\{\hat{f}_{h,2}^k(\cdot, \cdot) + 2b_{h,1}^k(\cdot, \cdot) + b_{h,2}^k(\cdot, \cdot), 1\right\},$$

$$f_{h,-2}^k(\cdot, \cdot) := \max\left\{\hat{f}_{h,-2}^k(\cdot, \cdot) - b_{h,2}^k(\cdot, \cdot), 0\right\}. \tag{D.15}$$

**Good events.** We define the following good events:

$$\mathcal{E}^\theta := \left\{\forall k \geqslant 1, \forall h \in [H] : \boldsymbol{\theta}_h^\star \in \mathcal{C}_h^k\right\}, \tag{D.16}$$

$$\mathcal{E}_{\leqslant K} := \bigcap_{k=1}^K \bigcap_{h=1}^H \mathcal{E}_h^k,$$

$$\mathcal{E}_h^k := \mathcal{E}_{h,1}^k \bigcap \mathcal{E}_{h,2}^k \bigcap \mathcal{E}_{h,-2}^k \bigcap \bar{\mathcal{E}}_h^k,$$

where $\mathcal{E}_{h,j}^k := \left\{\mathcal{T}_h V_{h+1,1}^k \in \mathcal{F}_{h,j}^k\right\}$ for $j = 1, \pm 2$, and $\bar{\mathcal{E}}_h^k := \left\{\mathcal{T}_h^2 V_{h+1,1}^k \in \mathcal{G}_h^k\right\}$.

**Optimistic $Q$-values.** For $(k, h, s, A) \in [K] \times [H] \times \mathcal{S} \times \mathcal{A}$ and for $j = 1, \pm 2$, we define the optimistic choice probability as follows:

$$\widetilde{\mathcal{P}}_{h,j}^k(a|s, A) := \begin{cases} \dfrac{\exp\left(\widetilde{v}_h^k(s, a)\right)}{\sum_{a' \in A} \exp\left(\widetilde{v}_h^k(s, a')\right)}, & \text{if } \exists a \in \mathcal{I} \backslash \{a_0\} \text{ s.t. } f_{h,j}^k(s, a) \geqslant f_{h,j}^k(s, a_0) \\ \dfrac{\exp\left(\breve{v}_h^k(s, a)\right)}{\sum_{a' \in A} \exp\left(\breve{v}_h^k(s, a')\right)}, & \text{otherwise,} \end{cases} \tag{D.17}$$

where

$$\widetilde{v}_h^k(s, a) := \phi(s, a)^\top \boldsymbol{\theta}_h^k + \alpha_h^k \|\phi(s, a)\|_{(\mathbf{H}_h^k)^{-1}}, \quad \breve{v}_h^k(s, a) := \phi(s, a)^\top \boldsymbol{\theta}_h^k - \alpha_h^k \|\phi(s, a)\|_{(\mathbf{H}_h^k)^{-1}}.$$

Next, we define the optimistic $Q$-values for $j = 1, \pm 2$, each constructed using $f_{h,j}^k$ and $\widetilde{\mathcal{P}}_{h,j}^k$:

$$Q_{h,j}^k(s, A) = \sum_{a \in A} \widetilde{\mathcal{P}}_{h,j}^k(a|s, A) f_{h,j}^k(s, a), \quad V_{h,j}^k(s) = \max_{A \in \mathcal{A}} Q_{h,j}^k(s, A).$$

For convenience, we also define the *realized* optimistic value functions at $(k, h) \in [K] \times [H]$ as follows:

$$Q_h^k(s, A) := \begin{cases} Q_{h,1}^k(s, A) & \text{if } A_h^k = A_{h,1}^k, \\ Q_{h,2}^k(s, A) & \text{otherwise,} \end{cases} \quad V_h^k(s) = \max_{A \in \mathcal{A}} Q_h^k(s, A), \tag{D.18}$$

where $A_h^k$ is the assortment offered to the user by the assortment selection rule in Equation 10 (or equivalently in Equation D.19). Therefore, we write $\pi_h^k(s_h^k) = \arg\max_{A \in \mathcal{A}} Q_h^k(s_h^k, A)$.

**Design of exploration policy.** At each episode $k$, the agent collects data using both $A_{h,1}^k$ and $A_{h,2}^k$, where $A_{h,j}^k \in \arg\max_{A \in \mathcal{A}} \sum_{a \in A} \widetilde{\mathcal{P}}_{h,j}^k(a|s_h^k, A) f_{h,j}^k(s_h^k, a)$ for $j = 1, 2$. Given a sequence of pre-specified $\{u_k\}_{k=1}^K$, at episode $k$, the agent select an assortment based on the following rule:

$$A_h^k = \begin{cases} A_{h,1}^k & \text{if } f_{h',1}^k(s_{h'}^k, a_{h'}) \geqslant f_{h',2}^k(s_{h'}^k, a_{h'}) - u_k, \quad \forall a_{h'} \in A_{h',1}^k, \forall h' \leqslant h. \\ A_{h,2}^k & \text{otherwise.} \end{cases} \tag{D.19}$$

We denote $h_k \in [H + 1]$ as the (random) horizon at which the agent first begins offering the assortment $A_{h,2}^k$. More formally, for $h \leqslant h_k$, the assortment offered is $A_h^k = A_{h,1}^k$, and for $h \geqslant h_k$, the assortment offered is $A_h^k = A_{h,2}^k$. We then divide the set of episodes $[K]$ into two disjoint subsets: $\mathcal{K}_{\mathrm{o}}$ and $\mathcal{K}_{\mathrm{oo}}$, such that

$$\mathcal{K}_{\mathrm{o}} := \{k \in [K] : h_k = H + 1\}, \quad \text{and} \quad \mathcal{K}_{\mathrm{oo}} := \{k \in [K] : h_k \leqslant H\}.$$

Later in the proof, we separately bound the regret for each case.

**Other notations.** Throughout the proof, we use $z = (s, a)$, $z_h = (s_h, a_h)$ and $z_h^k = (s_h^k, a_h^k)$ interchangeably. We sometimes use $\mathcal{P}_h(\cdot|s, A; \boldsymbol{\theta}_h^\star)$ instead of $\mathcal{P}_h(\cdot|s, A)$ to explicitly indicate the dependence on the parameter $\boldsymbol{\theta}_h^\star$. For simplicity, we denote $\mathbb{E}_{\mathbb{P}}[\cdot|s_h^k, a_h^k] = \mathbb{E}_{s_{h+1} \sim \mathbb{P}_h(\cdot|s_h^k, a_h^k)}[\cdot|s_h^k, a_h^k]$, $\mathbb{V}_{\mathbb{P}}[\cdot|s_h^k, a_h^k] = \mathbb{V}_{s_{h+1} \sim \mathbb{P}_h(\cdot|s_h^k, a_h^k)}[\cdot|s_h^k, a_h^k]$, and $\mathbb{E}_{\mathcal{P}}[\cdot|s_h^k, A_h^k] = \mathbb{E}_{a_h \sim \mathcal{P}_h(\cdot|s_h^k, A_h^k)}[\cdot|s_h^k, A_h^k]$.

## D.2 CONFIDENCE INTERVALS AND GOOD EVENTS

In this subsection, we show that, given the construction of confidence intervals $\mathcal{C}_h^k$ and $\mathcal{F}_{h,j}^k$ for $j = 1, \pm 2$, the good events $\mathcal{E}^\theta$ and $\mathcal{E}_{\leqslant K}$ occurs with high probability.

**Proposition D.1** (Online parameter confidence interval, Lemma 1 of Lee & Oh 2024). *Let $\delta \in (0, 1)$. Under Assumption 1, for the confidence interval defined in Equation 3 with*

$$
\alpha_h^k = \Bigg[ 2\eta \Bigg( (3\log(1 + (M+1)k) + B + 2)\Big( \frac{17}{16}\lambda + 2\sqrt{\lambda} \log\Big( \frac{2\sqrt{1+2k}}{\delta} \Big) \Big)
$$

$$
+ 16\Big( \log\Big( \frac{2\sqrt{1+2k}}{\delta} \Big) \Big)^2 \Big) + \frac{7\sqrt{6}}{6} d\eta \log\Big( 1 + \frac{k+1}{2\lambda} \Big) + 2 \Bigg) + \lambda B^2 \Bigg]^{1/2}
$$

$$
= \mathcal{O}(\sqrt{d} \log k \log M),
$$

$\eta = \frac{1}{2}\log(M+1) + B + 1$ *and* $\lambda = 84\sqrt{2}d\eta$, *and for any* $h \in [H]$, *we have*

$$
\Pr[\forall k \geqslant 1, \boldsymbol{\theta}_h^\star \in \mathcal{C}_h^k] \geqslant 1 - \delta.
$$

Now, we define the good event for the preference model $\mathcal{E}^\theta$ as follows:

$$
\mathcal{E}^\theta := \big\{ \forall k \geqslant 1, \forall h \in [H] : \boldsymbol{\theta}_h^\star \in \mathcal{C}_h^k \big\}.
$$

Then, by applying proposition D.1 and using a union bound over $h \in [H]$, we obtain the following corollary:

**Corollary D.1** (Good event for MNL preference model). *Under the same assumption and settings as in Proposition D.1, for $\delta \in (0, 1)$, with probability at least $1 - \delta$, the good event for the preference model $\mathcal{E}^\theta$ happens, i.e., $\boldsymbol{\theta}_h^\star \in \mathcal{C}_h^k$ for all $k \geqslant 1$ and all $h \in [H]$.*

The following proposition shows that $\mathcal{T}_h V_{h+1,j}^k$ for $j = 1, \pm 2$, and $\mathcal{T}_h^2 V_{h+1,1}^k$ lie within their respective confidence intervals.

**Proposition D.2** (Good event for general functions, Proposition 33 of Agarwal et al. 2023). *Suppose Algorithm 1 uses a consistent bonus oracle satisfying Definition 4. Let $\delta \in (0, 1/5)$. Then, with probability at least $1 - 5\delta$, the good event $\mathcal{E}_{\leqslant K} = \bigcap_{k=1}^K \bigcap_{h=1}^H \mathcal{E}_h^k$ happens, that is, $\mathcal{T}_h V_{h+1,1}^k \in \mathcal{F}_{h,1}^k$, $\mathcal{T}_h V_{h+1,\pm 2}^k \in \mathcal{F}_{h,\pm 2}^k$, and $\mathcal{T}_h^2 V_{h+1,1}^k \in \mathcal{G}_h^k$ for all $(k, h) \in [K] \times [H]$.*

## D.3 BOUND FOR MNL PREFERENCE MODEL

In this subsection, we provide proofs for several properties of the MNL preference model.

The following lemma presents both the optimistic and pessimistic utilities.

**Lemma D.1.** *For any $(k, h, s, a) \in [K] \times [H] \times \mathcal{S} \times \mathcal{I}$, let $\widetilde{v}_h^k(s, a) := \phi(s,a)^\top \boldsymbol{\theta}_h^k + \alpha_h^k \|\phi(s,a)\|_{(\mathbf{H}_h^k)^{-1}}$ and $\breve{v}_h^k(s,a) = \phi(s,a)^\top \boldsymbol{\theta}_h^k - \alpha_h^k \|\phi(s,a)\|_{(\mathbf{H}_h^k)^{-1}}$. Under the good event $\mathcal{E}^\theta$ defined in Equation D.16, it holds that*

$$
0 \leqslant \widetilde{v}_h^k(s,a) - \phi(s,a)^\top \boldsymbol{\theta}_h^\star \leqslant 2\alpha_h^k \|\phi(s,a)\|_{(\mathbf{H}_h^k)^{-1}},
$$

$$
\text{and } 0 \leqslant \phi(s,a)^\top \boldsymbol{\theta}_h^\star - \breve{v}_h^k(s,a) \leqslant 2\alpha_h^k \|\phi(s,a)\|_{(\mathbf{H}_h^k)^{-1}}.
$$

*Proof of Lemma D.1.* Conditioning on the good event $\mathcal{E}^\theta$ holds, we have

$$
\big| \phi(s,a)^\top \boldsymbol{\theta}_h^k - \phi(s,a)^\top \boldsymbol{\theta}_h^\star \big| \leqslant \|\phi(s,a)\|_{(\mathbf{H}_h^k)^{-1}} \big\| \boldsymbol{\theta}_h^k - \boldsymbol{\theta}_h^\star \big\|_{\mathbf{H}_h^k} \leqslant \alpha_h^k \|\phi(s,a)\|_{(\mathbf{H}_h^k)^{-1}},
$$

where the first inequality holds by Hölder's inequality, and the last inequality holds by Corollary D.1. Therefore, it follows that

$$
\widetilde{v}_h^k(s,a) - \phi(s,a)^\top \boldsymbol{\theta}_h^\star = \phi(s,a)^\top \boldsymbol{\theta}_h^k - \phi(s,a)^\top \boldsymbol{\theta}_h^\star + \alpha_h^k \|\phi(s,a)\|_{(\mathbf{H}_h^k)^{-1}}
$$

$$
\leqslant 2\alpha_h^k \|\phi(s,a)\|_{(\mathbf{H}_h^k)^{-1}}.
$$

Furthermore, since $\phi(s,a)^\top \boldsymbol{\theta}_h^k - \phi(s,a)^\top \boldsymbol{\theta}_h^\star \geq -\alpha_h^k \|\phi(s,a)\|_{(\mathbf{H}_h^k)^{-1}}$, we also get

$$\breve{v}_h^k(s,a) - \phi(s,a)^\top \boldsymbol{\theta}_h^\star = \phi(s,a)^\top \boldsymbol{\theta}_h^k - \phi(s,a)^\top \boldsymbol{\theta}_h^\star + \alpha_h^k \|\phi(s,a)\|_{(\mathbf{H}_h^k)^{-1}} \geq 0.$$

The second statement directly follows from the results mentioned above. $\qquad\square$

Lemma D.2 is useful for proving optimism (Lemma D.9) and bounding the approximation error of the optimistic $\overline{Q}$ (Lemma D.14).

**Lemma D.2.** *For all $(k,h,s,A) \in [K] \times [H] \times \mathcal{S} \times \mathcal{A}$ and any $j \in \{1,2\}$, under the good event $\mathcal{E}^\theta$ defined in Equation D.16, there exists a subset $\tilde{A} \subseteq A$ such that $\tilde{A} \in \mathcal{A}$ and*

$$\max\left\{\sum_{a\in A} \mathcal{P}_h(a|s,A) f_{h,j}^k(s,a), \sum_{a\in A} \widetilde{\mathcal{P}}_{h,j'}^k(a|s,A) f_{h,j}^k(s,a)\right\} \leq \sum_{a\in\tilde{A}} \widetilde{\mathcal{P}}_{h,j}^k(a|s,\tilde{A}) f_{h,j}^k(s,a),$$

*where $j' \neq j$.*

*Proof of Lemma D.2.* First, recall that, without loss of generality, we assumed that $\phi(s,a_0) = 0$ for all $s \in \mathcal{S}$. Therefore, the true preference model and the optimistic preference model can be written as

$$\mathcal{P}_h(a|s,A) = \frac{\exp\left(\phi(s,a)^\top \boldsymbol{\theta}_h^\star\right)}{1 + \sum_{a'\in A\backslash\{a_0\}} \exp\left(\phi(s,a')^\top \boldsymbol{\theta}_h^\star\right)},$$

$$\widetilde{\mathcal{P}}_{h,j}^k(a|s,A) = \begin{cases} \dfrac{\exp\left(\widetilde{v}_h^k(s,a)\right)}{1 + \sum_{a'\in A\backslash\{a_0\}} \exp\left(\widetilde{v}_h^k(s,a')\right)}, & \text{if } \exists a \in \mathcal{I}\backslash\{a_0\} \text{ s.t. } f_{h,j}^k(s,a) \geq f_{h,j}^k(s,a_0) \\ \dfrac{\exp\left(\breve{v}_h^k(s,a)\right)}{1 + \sum_{a'\in A\backslash\{a_0\}} \exp\left(\breve{v}_h^k(s,a')\right)}, & \text{otherwise.} \end{cases}$$

$$\text{(D.20)}$$

Fix $s \in \mathcal{S}$ and $A \in \mathcal{A}$. We now present the proof by considering two cases: (i) $f_{h,j}^k(s,a_0) > f_{h,j}^k(s,a)$ for all $a \in A$ and (ii) $\exists a \in A\backslash\{a_0\}$ such that $f_{h,j}^k(s,a_0) \leq f_{h,j}^k(s,a)$.

**Case (i)** $f_{h,j}^k(s,a_0) > f_{h,j}^k(s,a)$ for all $a \in A$.
We denote $\tilde{a} \in \arg\max_{a\in A\backslash\{a_0\}} f_{h,j}^k(s,a)$. Let $\tilde{A} = \{a_0, \tilde{a}\}$. Since $f_{h,j}^k(s,a_0) > f_{h,j}^k(s,a)$ for all $a \in A$ and $a_0$ is always included in $A$, removing any item $a \in A\backslash\{a_0\}$ from $A$ increases the expected value of $f_{h,j}^k$. Thus, we get

$$\sum_{a\in A} \mathcal{P}_h(a|s,A) f_{h,j}^k(s,a) \leq \sum_{a\in\tilde{A}} \mathcal{P}_h(a|s,\tilde{A}) f_{h,j}^k(s,a)$$

$$\text{and } \sum_{a\in A} \widetilde{\mathcal{P}}_{h,j'}^k(a|s,A) f_{h,j}^k(s,a) \leq \sum_{a\in\tilde{A}} \widetilde{\mathcal{P}}_{h,j'}^k(a|s,\tilde{A}) f_{h,j}^k(s,a). \qquad \text{(D.21)}$$

By the definition of $\widetilde{\mathcal{P}}_{h,j}^k$ in Equation 8, we use the pessimistic utility $\breve{v}_h^k(s,a)$ in this case. Since $\breve{v}_h^k(s,a) \leq \phi(s,a)^\top \boldsymbol{\theta}_h^\star$ by Lemma D.1, using this utility, $\breve{v}_h^k(s,a)$, reduces the probability of selecting $\tilde{a}$ (compared to the true choice probability $\mathcal{P}_h$). Moreover, we know that $f_h^k(s,a_0) \geq f_h^k(s,\tilde{a})$, we have

$$\sum_{a\in\tilde{A}} \mathcal{P}_h(a|s,\tilde{A}) f_{h,j}^k(s,a) \leq \sum_{a\in\tilde{A}} \widetilde{\mathcal{P}}_{h,j}^k(a|s,\tilde{A}) f_{h,j}^k(s,a). \qquad \text{(D.22)}$$

Furthermore, if $\widetilde{\mathcal{P}}_{h,j'}^k$ is constructed using the pessimistic utility $\breve{v}_h^k(s,a)$, then, $\widetilde{\mathcal{P}}_{h,j'}^k = \widetilde{\mathcal{P}}_{h,j}^k$. However, if $\widetilde{\mathcal{P}}_{h,j'}^k$ is constructed using the optimistic utility $\widetilde{v}_h^k(s,a)$, replacing $\widetilde{v}_h^k(s,a)$ with $\breve{v}_h^k(s,a)$ (which is equivalent to replacing $\widetilde{\mathcal{P}}_{h,j'}^k$ with $\widetilde{\mathcal{P}}_{h,j}^k$) decreases the probability of choosing $\tilde{a}$, meaning increase the expected value of $f_{h,j}^k$. Thus, we get

$$\sum_{a\in\tilde{A}} \widetilde{\mathcal{P}}_{h,j'}^k(a|s,\tilde{A}) f_{h,j}^k(s,a) \leq \sum_{a\in\tilde{A}} \widetilde{\mathcal{P}}_{h,j}^k(a|s,\tilde{A}) f_{h,j}^k(s,a). \qquad \text{(D.23)}$$

Combining Equation D.21, Equation D.22, and Equation D.23, we have

$$\max\left\{\sum_{a\in A}\mathcal{P}_h(a|s,A)f_{h,j}^k(s,a),\sum_{a\in A}\widetilde{\mathcal{P}}_{h,j'}^k(a|s,A)f_{h,j}^k(s,a)\right\}\leqslant\sum_{a\in\tilde{A}}\widetilde{\mathcal{P}}_{h,j}^k(a|s,\tilde{A})f_{h,j}^k(s,a).$$

**Case (ii)** $\exists a\in A\backslash\{a_0\}$ such that $f_{h,j}^k(s,a_0)\leqslant f_{h,j}^k(s,a)$.

Let $\tilde{A}=\left\{A'\subseteq A:f_{h,j}^k(s,a)\geqslant f_{h,j}^k(s,a_0),\forall a\in A'\right\}$. Note that $|\tilde{A}|\geqslant 2$ and $a_0\in\tilde{A}$ by definition of action space $\mathcal{A}$. By selecting $\tilde{A}$ instead of $A$, we exclude items with the small values of $f_{h,j}^k(s,a)$, thereby increasing the expected value of $f_{h,j}^k$, i.e.,

$$\sum_{a\in A}\mathcal{P}_h(a|s,A)f_{h,j}^k(s,a)\leqslant\sum_{a\in\tilde{A}}\mathcal{P}_h(a|s,\tilde{A})f_{h,j}^k(s,a)$$

$$\text{and }\sum_{a\in A}\widetilde{\mathcal{P}}_{h,j'}^k(a|s,A)f_{h,j}^k(s,a)\leqslant\sum_{a\in\tilde{A}}\widetilde{\mathcal{P}}_{h,j'}^k(a|s,\tilde{A})f_{h,j}^k(s,a).\tag{D.24}$$

By the definition of $\widetilde{\mathcal{P}}_{h,j}^k$ in Equation 8, we use the optimistic utility $\widetilde{v}_h^k(s,a)$ in this case. Since $\phi(s,a)^\top\boldsymbol{\theta}_h^\star\leqslant\widetilde{v}_h^k(s,a)$ by Lemma D.1, this choice of utility increases the probability of choosing item $a\neq\tilde{A}\backslash\{a_0\}$ compared to the true $\mathcal{P}_h$), implying that

$$\sum_{a\in\tilde{A}}\mathcal{P}_h(a|s,\tilde{A})f_{h,j}^k(s,a)\leqslant\sum_{a\in\tilde{A}}\widetilde{\mathcal{P}}_{h,j}^k(a|s,\tilde{A})f_{h,j}^k(s,a).\tag{D.25}$$

Moreover, if $\widetilde{\mathcal{P}}_{h,j'}^k$ is constructed using the pessimistic utility $\breve{v}_h^k(s,a)$, replacing $\breve{v}_h^k(s,a)$ with $\widetilde{v}_h^k(s,a)$ (which is equivalent to replacing $\widetilde{\mathcal{P}}_{h,j'}^k$ with $\widetilde{\mathcal{P}}_{h,j}^k$) increases the probability of choosing item $a\neq\tilde{A}\backslash\{a_0\}$. However, if $\widetilde{\mathcal{P}}_{h,j'}^k$ is constructed using the optimistic utility $\widetilde{v}_h^k(s,a)$, we have $\widetilde{\mathcal{P}}_{h,j}^k=\widetilde{\mathcal{P}}_{h,j'}^k$. To this end, we get

$$\sum_{a\in\tilde{A}}\widetilde{\mathcal{P}}_{h,j'}^k(a|s,\tilde{A})f_{h,j}^k(s,a)\leqslant\sum_{a\in\tilde{A}}\widetilde{\mathcal{P}}_{h,j}^k(a|s,\tilde{A})f_{h,j}^k(s,a).\tag{D.26}$$

Combining Equation D.24, Equation D.25, and Equation D.26, we have

$$\max\left\{\sum_{a\in A}\mathcal{P}_h(a|s,A)f_{h,j}^k(s,a),\sum_{a\in A}\widetilde{\mathcal{P}}_{h,j'}^k(a|s,A)f_{h,j}^k(s,a)\right\}\leqslant\sum_{a\in\tilde{A}}\widetilde{\mathcal{P}}_{h,j}^k(a|s,\tilde{A})f_{h,j}^k(s,a).$$

This concludes the proof of Lemma D.2. $\qquad\square$

Lemma D.3 is an elliptical potential lemma used for bounding the regret incurred from the MNL preference model (Lemma D.6 and D.7).

**Lemma D.3** (Elliptical potential lemma, Lemma E.2 and H.3 of Lee & Oh 2024). *Assume that $\lambda\geqslant 2$ and $\phi(s,a_0)=0$ for all $s\in\mathcal{S}$. For any $(k,h,a)\in[K]\times[H]\times\mathcal{I}$, we define $\widetilde{\phi}(s_h^k,a)=\phi(s_h^k,a)-\mathbb{E}_{a'\sim\mathcal{P}_h(\cdot|s_h^k,A_h^k;\boldsymbol{\theta}_h^{k+1})}[\phi(s_h^k,a)]$. Then, for $\mathbf{H}_h^k$ defined in Equation D.3, and for any $h\in[H]$, the following statements hold true:*

$$\sum_{\tau=1}^k\sum_{a\in A_h^\tau}\mathcal{P}_h\big(a|s_h^\tau,A_h^\tau;\boldsymbol{\theta}_h^{\tau+1}\big)\mathcal{P}_h\big(a_0|s_h^\tau,A_h^\tau;\boldsymbol{\theta}_h^{\tau+1}\big)\|\phi(s_h^\tau,a)\|_{(\mathbf{H}_h^\tau)^{-1}}^2\leqslant 2d\log\left(1+\frac{k}{d\lambda}\right),$$

$$\sum_{\tau=1}^k\sum_{a\in A_h^\tau}\mathcal{P}_h\big(a|s_h^\tau,A_h^\tau;\boldsymbol{\theta}_h^{\tau+1}\big)\|\widetilde{\phi}(s_h^\tau,a)\|_{(\mathbf{H}_h^\tau)^{-1}}^2\leqslant 2d\log\left(1+\frac{k}{d\lambda}\right),$$

$$\sum_{\tau=1}^k\max\left\{\max_{a\in A_h^\tau}\|\phi(s_h^\tau,a)\|_{(\mathbf{H}_h^\tau)^{-1}}^2,\max_{a\in A_h^\tau}\|\widetilde{\phi}(s_h^\tau,a)\|_{(\mathbf{H}_h^\tau)^{-1}}^2\right\}\leqslant\frac{2}{\kappa}d\log\left(1+\frac{k}{d\lambda}\right).$$

Lemma D.4 is used to derive the tight bound for the second-order regret term of the MNL preference model (LemmaD.7).

**Lemma D.4** (Lemma E.3 of Lee & Oh 2024). *Let $M \in \mathbf{Z}^+$. Define $R : \mathbb{R}^M \to \mathbb{R}$, such that for any $\boldsymbol{v} = (v_1, \ldots, v_M) \in \mathbb{R}^M$, $R(\boldsymbol{v}) = \sum_{m=1}^M \frac{\exp(v_m)}{1 + \sum_{l=1}^M \exp(v_l)}$. Let $p_m(\boldsymbol{v}) = \frac{\exp(v_m)}{1 + \sum_{l=1}^M \exp(v_l)}$. Then, for all $m \in [M]$, we have*

$$\left| \frac{\partial^2 R}{\partial m \partial n} \right| \leq \begin{cases} 3 p_m(\boldsymbol{v}) & \text{if } m = n, \\ 2 p_m(\boldsymbol{v}) p_n(\boldsymbol{v}) & \text{if } m \neq n. \end{cases}$$

Lemma D.5 is crucial for deriving the $\kappa$-improved bound for the MNL preference model (LemmaD.7), enabling the analysis.

**Lemma D.5** (Overly optimistic choice probability). *We define*

$$\tilde{\tilde{\mathcal{P}}}_{h,j}^k(a|s, A) = \begin{cases} \dfrac{\exp\left( \phi(s,a)^\top \boldsymbol{\theta}_h^\star + 2\alpha_h^k \|\phi(s,a)\|_{(\mathbf{H}_h^k)^{-1}} \right)}{\sum_{a' \in A} \exp\left( \phi(s,a')^\top \boldsymbol{\theta}_h^\star + 2\alpha_h^k \|\phi(s,a')\|_{(\mathbf{H}_h^k)^{-1}} \right)}, & \text{if } \exists a \in \mathcal{I} \backslash \{a_0\} \text{ s.t.} \\ & f_{h,j}^k(s,a) \geq f_{h,j}^k(s,a_0), \\[4mm] \dfrac{\exp\left( \phi(s,a)^\top \boldsymbol{\theta}_h^\star - 2\alpha_h^k \|\phi(s,a)\|_{(\mathbf{H}_h^k)^{-1}} \right)}{\sum_{a' \in A} \exp\left( \phi(s,a')^\top \boldsymbol{\theta}_h^\star - 2\alpha_h^k \|\phi(s,a')\|_{(\mathbf{H}_h^k)^{-1}} \right)}, & \text{otherwise.} \end{cases}$$

$$\tag{D.27}$$

*Let $A_{h,j}^k \in \arg\max_{A \in \mathcal{A}} \sum_{a \in A} \tilde{\mathcal{P}}_{h,j}(a|s_h^k, A) f_{h,j}^k(s_h^k, a)$, where $j \in \{1, 2\}$. Then, under the good event $\mathcal{E}^\theta$, for all $(k, h, j) \in [K] \times [H] \times \{1, 2\}$, we have*

$$\sum_{a \in A_{h,j}^k} \tilde{\mathcal{P}}_{h,j}^k(a|s_h^k, A_{h,j}^k) f_{h,j}^k(s_h^k, a) \leq \sum_{a \in A_{h,j}^k} \tilde{\tilde{\mathcal{P}}}_{h,j}^k(a|s_h^k, A_{h,j}^k) f_{h,j}^k(s_h^k, a).$$

*Proof of Lemma D.5.* Fix $(k, h, j) \in [K] \times [H] \times \{1, 2\}$. We consider the two cases: (i) $f_{h,j}^k(s_h^k, a_0) > f_{h,j}^k(s_h^k, a)$ for all $a \in \mathcal{I}$ and (ii) $\exists a \in \mathcal{I} \backslash \{a_0\}$ such that $f_{h,j}^k(s_h^k, a_0) \leq f_{h,j}^k(s_h^k, a)$.

**Case (i)** $f_{h,j}^k(s_h^k, a_0) > f_{h,j}^k(s_h^k, a)$ for all $a \in \mathcal{I}$.
Recall that, by the definition of $\tilde{\mathcal{P}}_{h,j}^k$ in Equation 8, we use the pessimistic utility $\breve{v}_h^k(s, a)$ to construct $\tilde{\mathcal{P}}_{h,j}^k$ in this case. Note that the outside option $a_0$ must be included in the assortment, i.e., $a_0 \in A_h^k$. Moreover, under the event $\mathcal{E}^\theta$, by Lemma D.1, we have

$$\breve{v}_h^k(s_h^k, a) \geq \phi(s_h^k, a)^\top \boldsymbol{\theta}_h^\star - 2\alpha_h^k \|\phi(s_h^k, a)\|_{(\mathbf{H}_h^k)^{-1}}.$$

Thus, since we assume, without loss of generality, that $\phi(s_h^k, a_0) = 0$ (refer Equation D.20), using $\tilde{\tilde{\mathcal{P}}}_{h,j}^k$ instead of $\tilde{\mathcal{P}}_{h,j}^k$ decreases the probability of choosing any item $a \in A_h^k \backslash \{a_0\}$. As a result, the expected value of $f_{h,j}^k$ increases, since $f_{h,j}^k(s, a_0) \geq f_{h,j}^k(s, a)$ for all $a \in A_h^k$. Formally, we have

$$\sum_{a \in A_{h,j}^k} \tilde{\mathcal{P}}_{h,j}^k(a|s_h^k, A_{h,j}^k) f_{h,j}^k(s_h^k, a) \leq \sum_{a \in A_{h,j}^k} \tilde{\tilde{\mathcal{P}}}_{h,j}^k(a|s_h^k, A_{h,j}^k) f_{h,j}^k(s_h^k, a).$$

**Case (ii)** $\exists a \in \mathcal{I} \backslash \{a_0\}$ such that $f_{h,j}^k(s_h^k, a_0) \leq f_{h,j}^k(s_h^k, a)$.
First, we show that for all $a \in A_h^k \backslash \{a_0\}$, we have $f_{h,j}^k(s_h^k, a) \geq \sum_{a \in A_{h,j}^k} \tilde{\mathcal{P}}_{h,j}^k(a|s_h^k, A_{h,j}^k) f_{h,j}^k(s_h^k, a)$. Suppose that there exists $a \in A_h^k \backslash \{a_0\}$ for which $f_{h,j}^k(s_h^k, a) < \sum_{a \in A_{h,j}^k} \tilde{\mathcal{P}}_{h,j}^k(a|s_h^k, A_{h,j}^k) f_{h,j}^k(s_h^k, a)$. Then, removing item $a$ from the assortment $A_h^k$ results in the increase in the expected value of $f_{h,j}^k$. Consequently, this contradicts the optimality of $A_h^k$. Hence, we get

$$f_{h,j}^k(s_h^k, a) \geq \sum_{a \in A_{h,j}^k} \tilde{\mathcal{P}}_{h,j}^k(a|s_h^k, A_{h,j}^k) f_{h,j}^k(s_h^k, a), \quad \forall a \in A_h^k \backslash \{a_0\}.$$

On the other hand, recall that, by the definition of $\tilde{\mathcal{P}}_{h,j}^k$ in Equation 8, we use the pessimistic utility $\tilde{v}_h^k(s, a)$ to construct $\tilde{\mathcal{P}}_{h,j}^k$ in this case. Furthermore, by Lemma D.1, we know that

$$\tilde{v}_h^k(s_h^k, a) \leq \phi(s_h^k, a)^\top \boldsymbol{\theta}_h^\star + 2\alpha_h^k \|\phi(s_h^k, a)\|_{(\mathbf{H}_h^k)^{-1}}.$$

If we increase $\widetilde{v}_h^k(s_h^k, a)$ to $\phi(s_h^k, a)^\top \boldsymbol{\theta}_h^\star + 2\alpha_h^k \|\phi(s_h^k, a)\|_{(\mathbf{H}_h^k)^{-1}}$ for all $a \in A_h^k \backslash \{a_0\}$, the probability of choosing the outside option decreases (because $\phi(s_h^k, a_0) = 0$). In other words, the sum of probabilities of choosing $a \in A_h^k \backslash \{a_0\}$ increases. Since $f_{h,j}^k(s_h^k, a) \geqslant \sum_{a \in A_{h,j}^k} \widetilde{\mathcal{P}}_{h,j}^k(a|s_h^k, A_{h,j}^k) f_{h,j}^k(s_h^k, a)$ for all $a \in A_h^k \backslash \{a_0\}$, the expected value of $f_{h,j}^k$ increases. Formally, we get

$$\sum_{a \in A_{h,j}^k} \widetilde{\mathcal{P}}_{h,j}^k(a|s_h^k, A_{h,j}^k) f_{h,j}^k(s_h^k, a) \leqslant \sum_{a \in A_{h,j}^k} \widetilde{\widetilde{\mathcal{P}}}_{h,j}^k(a|s_h^k, A_{h,j}^k) f_{h,j}^k(s_h^k, a).$$

This concludes the proof. $\qquad \square$

Lemma D.6 will be used to carefully bound the sum of $b_{h,1}^k$ (Lemma D.17). Note that the following MNL bandit regret improves upon the one proposed in (Oh & Iyengar, 2021) by a factor of $1/\sqrt{\kappa}$, which can be exponentially large.

**Lemma D.6** (Crude bound for MNL bandits). *For any $h \in [H]$, $j = \{1, 2, -2\}$ and subset $\mathcal{K} \in [K]$, under the good event $\mathcal{E}^\theta$ defined in Equation D.16, we have*

$$\sum_{k \in \mathcal{K}} \left| \sum_{a \in A_h^k} \left( \widetilde{\mathcal{P}}_{h,j}^k(a|s_h^k, A_h^k) - \mathcal{P}_h(a|s_h^k, A_h^k) \right) f_{h,j}^k(s_h^k, a) \right| \leqslant \mathcal{O}\left( \frac{1}{\sqrt{\kappa}} d\sqrt{|\mathcal{K}|} \cdot \log K \log M \right),$$

*where $M$ is the maximum size of the assortment.*

*Proof of Lemma D.6.* We denote $M_h^k$ as the size of the assortment at horizon $h$ in episode $k$, i.e., $M_h^k = |A_h^k|$. For any $j \in \{1, 2, -2\}$, we define a function $R_j : \mathbb{R}^{M_h^k} \to \mathbb{R}$ such that, for all $\boldsymbol{v} \in \mathbb{R}^{M_h^k}$, $R_j(\boldsymbol{v}) = \sum_{m=1}^{M_h^k} \frac{\exp(v_m) f_{h,j}^k(s_h^k, a_{i_m})}{1 + \sum_{l=1}^{M_h^k} \exp(v_l)}$.

For simplicity, we denote $v_{h,j}^k(s, a)$ as the utility, which can represent either the optimistic utility $\widetilde{v}_h^k(s, a)$ or the pessimistic utility $\breve{v}_h^k(s, a)$, as determined by Equation 8, depending on $f_{h,j}^k$. Let $\boldsymbol{v}_{h,j}^k(s_h^k) = \left( v_{h,j}^k(s_h^k, a) \right)_{a \in A_h^k} \in \mathbb{R}^{M_h^k}$ and $\boldsymbol{v}_h^\star(s_h^k) = \left( \phi(s_h^k, a)^\top \boldsymbol{\theta}_h^\star \right)_{a \in A_h^k} \in \mathbb{R}^{M_h^k}$. Then, by the mean value theorem, there exists a vector $\bar{\boldsymbol{v}}_{h,j}^k(s_h^k)$, which is a convex combination of $\boldsymbol{v}_{h,j}^k(s_h^k)$ and $\boldsymbol{v}_h^\star(s_h^k)$, such that

$$\sum_{k \in \mathcal{K}} \left| \sum_{a \in A_h^k} \left( \widetilde{\mathcal{P}}_{h,j}^k(a|s_h^k, A_h^k) - \mathcal{P}_h(a|s_h^k, A_h^k) \right) f_{h,j}^k(s_h^k, a) \right|$$
$$= \sum_{k \in \mathcal{K}} \left| R_j \left( \boldsymbol{v}_{h,j}^k(s_h^k) \right) - R_j \left( \boldsymbol{v}_h^\star(s_h^k) \right) \right|$$
$$= \sum_{k \in \mathcal{K}} \left| \nabla R_j \left( \bar{\boldsymbol{v}}_{h,j}^k(s_h^k) \right)^\top \left( \boldsymbol{v}_{h,j}^k(s_h^k) - \boldsymbol{v}_h^\star(s_h^k) \right) \right|.$$

Therefore, we get

$$
\sum_{k \in \mathcal{K}} \left| \nabla R_j \left( \bar{\boldsymbol{v}}_{h,j}^k(s_h^k) \right)^\top \left( \boldsymbol{v}_{h,j}^k(s_h^k) - \boldsymbol{v}_h^\star(s_h^k) \right) \right|
$$

$$
= \sum_{k \in \mathcal{K}} \left| \sum_{a \in A_h^k} \frac{\exp\left( \bar{v}_{h,j}^k(s_h^k, a) \right) f_{h,j}^k(s_h^k, a)}{\sum_{a'' \in A_h^k} \exp\left( \bar{v}_{h,j}^k(s_h^k, a'') \right)} \left( v_{h,j}^k(s_h^k, a) - \phi(s_h^k, a)^\top \boldsymbol{\theta}_h^\star \right) \right.
$$

$$
\left. - \sum_{a \in A_h^k} \sum_{a' \in A_h^k} \frac{\exp\left( \bar{v}_{h,j}^k(s_h^k, a) \right) f_{h,j}^k(s_h^k, a) \exp\left( \bar{v}_{h,j}^k(s_h^k, a') \right)}{\left( \sum_{a'' \in A_h^k} \exp\left( \bar{v}_{h,j}^k(s_h^k, a'') \right) \right)^2} \left( v_{h,j}^k(s_h^k, a') - \phi(s_h^k, a')^\top \boldsymbol{\theta}_h^\star \right) \right|
$$

$$
= \sum_{k \in \mathcal{K}} \left| \sum_{a \in A_h^k} \mathcal{P}_h\left( a | s_h^k, A_h^k; \bar{\boldsymbol{v}}_{h,j}^k(s_h^k) \right) \left( v_{h,j}^k(s_h^k, a) - \phi(s_h^k, a)^\top \boldsymbol{\theta}_h^\star \right) \right.
$$

$$
\left. \cdot \left( f_{h,j}^k(s_h^k, a) - \mathbb{E}_{a' \sim \mathcal{P}_h\left( \cdot | s_h^k, A_h^k; \bar{\boldsymbol{v}}_{h,j}^k(s_h^k) \right)} \left[ f_{h,j}^k(s_h^k, a') \right] \right) \right|
$$

$$
\leqslant 2 \sum_{k \in \mathcal{K}} \sum_{a \in A_h^k} \mathcal{P}_h\left( a | s_h^k, A_h^k; \bar{\boldsymbol{v}}_{h,j}^k(s_h^k) \right) \left| \left( v_{h,j}^k(s_h^k, a) - \phi(s_h^k, a) \right)^\top \boldsymbol{\theta}_h^\star \right|, \tag{D.28}
$$

where the inequality is from $f_{h,j}^k \leqslant 1$. Recall that $v_{h,j}^k(s_h^k, a)$ can be either $\widetilde{v}_h^k(s_h^k, a)$ or $\breve{v}_h^k(s_h^k, a)$. Then, by Lemma D.1, we have $\left| \left( v_{h,j}^k(s_h^k, a) - \phi(s_h^k, a)^\top \boldsymbol{\theta}_h^\star \right) \right| \leqslant 2\alpha_h^k \|\phi(s_h^k, a)\|_{(\mathbf{H}_h^k)^{-1}}$. Hence, we can further bound the right-hand side of Equation D.28.

$$
2 \sum_{k \in \mathcal{K}} \sum_{a \in A_h^k} \mathcal{P}_h\left( a | s_h^k, A_h^k; \bar{\boldsymbol{v}}_{h,j}^k(s_h^k) \right) \left| \left( v_{h,j}^k(s_h^k, a) - \phi(s_h^k, a) \right)^\top \boldsymbol{\theta}_h^\star \right|
$$

$$
\leqslant 4\alpha_h^K \sum_{k \in \mathcal{K}} \sum_{a \in A_h^k} \mathcal{P}_h\left( a | s_h^k, A_h^k; \bar{\boldsymbol{v}}_{h,j}^k(s_h^k) \right) \|\phi(s_h^k, a)\|_{(\mathbf{H}_h^k)^{-1}}
$$

$$
\leqslant 4\alpha_h^K \sqrt{\sum_{k \in \mathcal{K}} \sum_{a \in A_h^k} \mathcal{P}_h\left( a | s_h^k, A_h^k; \bar{\boldsymbol{v}}_{h,j}^k(s_h^k) \right)} \sqrt{\sum_{k \in \mathcal{K}} \sum_{a \in A_h^k} \mathcal{P}_h\left( a | s_h^k, A_h^k; \bar{\boldsymbol{v}}_{h,j}^k(s_h^k) \right) \|\phi(s_h^k, a)\|_{(\mathbf{H}_h^k)^{-1}}^2}
$$

$$
\leqslant 4\alpha_h^K \sqrt{|\mathcal{K}|} \cdot \sqrt{\sum_{k=1}^K \sum_{a \in A_h^k} \|\phi(s_h^k, a)\|_{(\mathbf{H}_h^k)^{-1}}^2}
$$

$$
= 4\alpha_h^K \sqrt{|\mathcal{K}|} \cdot \sqrt{\sum_{k=1}^K \sum_{a \in A_h^k} \frac{\mathcal{P}_h\left( a | s_h^k, A_h^k; \boldsymbol{\theta}_h^{k+1} \right) \mathcal{P}_h\left( a_0 | s_h^k, A_h^k; \boldsymbol{\theta}_h^{k+1} \right)}{\mathcal{P}_h\left( a | s_h^k, A_h^k; \boldsymbol{\theta}_h^{k+1} \right) \mathcal{P}_h\left( a_0 | s_h^k, A_h^k; \boldsymbol{\theta}_h^{k+1} \right)} \|\phi(s_h^k, a)\|_{(\mathbf{H}_h^k)^{-1}}^2}
$$

$$
\leqslant 4\alpha_h^K \sqrt{|\mathcal{K}|} \cdot \sqrt{\frac{1}{\kappa} \cdot \sum_{k=1}^K \sum_{a \in A_h^k} \mathcal{P}_h\left( a | s_h^k, A_h^k; \boldsymbol{\theta}_h^{k+1} \right) \mathcal{P}_h\left( a_0 | s_h^k, A_h^k; \boldsymbol{\theta}_h^{k+1} \right) \|\phi(s_h^k, a)\|_{(\mathbf{H}_h^k)^{-1}}^2}
$$

$$
\leqslant 4\alpha_h^K \sqrt{|\mathcal{K}|} \cdot \sqrt{\frac{1}{\kappa} \cdot 2d \log\left( 1 + \frac{K}{d\lambda} \right)}, \tag{D.29}
$$

where the first inequality holds because $\alpha_h^1 \leqslant \cdots \leqslant \alpha_h^K$, the second inequality follows from the Cauchy-Schwarz inequality, the second-to-the last inequality holds due to the definition of $\kappa$, and the last inequality holds by Lemma D.3.

Combining Equation D.28 and Equation D.29, and plugging in the value of $\alpha_h^k$, we derive that

$$
\sum_{k \in \mathcal{K}} \left| \sum_{a \in A_h^k} \left( \widetilde{\mathcal{P}}_{h,j}^k(a | s_h^k, A_h^k) - \mathcal{P}_h(a | s_h^k, A_h^k) \right) f_{h,j}^k(s_h^k, a) \right| = \mathcal{O}\left( \frac{1}{\sqrt{\kappa}} d\sqrt{|\mathcal{K}|} \cdot \log K \log M \right).
$$

$\square$

Lemma D.7 is crucial for obtaining a $\kappa$-independent regret in our leading term. While the proof is largely inspired by Lee & Oh (2024), extending their result to our setting is non-trivial because the unknown item values $f_{h,j}^k$ add complexity to the analysis.

**Lemma D.7** ($\kappa$-improved bound for MNL bandits). *For any $h \in [H]$ and subset $\mathcal{K} \in [K]$, let $J(k,h) : \mathcal{K} \times [H] \to \{1,2\}$ be the one-to-one function that maps from $\mathcal{K} \times [H]$ to the index set $\{1,2\}$ such that $A_h^k = A_{h,J(k,h)}^k \in \arg\max_{A \in \mathcal{A}} \sum_{a \in A} \widetilde{\mathcal{P}}_{h,J(k,h)}^k(a|s_h^k, A) f_{h,J(k,h)}^k(s_h^k, a)$. Then, under the good event $\mathcal{E}^\theta$ defined in Equation D.16, we have*

$$\sum_{k \in \mathcal{K}} \sum_{a \in A_h^k} \left( \widetilde{\mathcal{P}}_{h,J(k,h)}^k(a|s_h^k, A_h^k) - \mathcal{P}_h(a|s_h^k, A_h^k) \right) f_{h,J(k,h)}^k(s_h^k, a)$$

$$= \mathcal{O}\left( d\sqrt{|\mathcal{K}|} \cdot \log K \log M + \frac{1}{\kappa} d^2 (\log K \log M)^2 \right).$$

*Proof of Lemma D.7.* We begin by defining $\widetilde{\widetilde{\mathcal{P}}}_{h,j}^k(a|s, A)$ as given in D.27. Then, by Lemma D.5, we have

$$\sum_{k \in \mathcal{K}} \sum_{a \in A_h^k} \left( \widetilde{\mathcal{P}}_{h,J(k,h)}^k(a|s_h^k, A_h^k) - \mathcal{P}_h(a|s_h^k, A_h^k) \right) f_{h,J(k,h)}^k(s_h^k, a)$$

$$\leqslant \sum_{k \in \mathcal{K}} \sum_{a \in A_h^k} \left( \widetilde{\widetilde{\mathcal{P}}}_{h,J(k,h)}^k(a|s_h^k, A_h^k) - \mathcal{P}_h(a|s_h^k, A_h^k) \right) f_{h,J(k,h)}^k(s_h^k, a).$$

We denote $M_h^k$ as the size of the assortment at horizon $h$ in episode $k$, i.e., $M_h^k = |A_h^k|$. We define a function $\tilde{R} : \mathbb{R}^{M_h^k} \to \mathbb{R}$ such that for all $\boldsymbol{v} \in \mathbb{R}^{M_h^k}$, $\tilde{R}(\boldsymbol{v}) = \sum_{m=1}^{M_h^k} \frac{\exp(v_m) f_{h,J(k,h)}^k(s_h^k, a_{i_m})}{1 + \sum_{l=1}^{M_h^k} \exp(v_l)}$.

For any $(k,h) \in \mathcal{K} \times [H]$ and all $a \in \mathcal{I}$, we denote $v_h^k(s_h^k, a)$ as the utility, which can be either $\phi(s_h^k, a)^\top \boldsymbol{\theta}_h^\star + 2\alpha_h^k \|\phi(s_h^k, a)\|_{(\mathbf{H}_h^k)^{-1}}$ or $\phi(s_h^k, a)^\top \boldsymbol{\theta}_h^\star - 2\alpha_h^k \|\phi(s_h^k, a)\|_{(\mathbf{H}_h^k)^{-1}}$, determined deterministically based on the history up to $(k,h)$:

$$v_h^k(s_h^k, a) = \begin{cases} \phi(s_h^k, a)^\top \boldsymbol{\theta}_h^\star + 2\alpha_h^k \|\phi(s_h^k, a)\|_{(\mathbf{H}_h^k)^{-1}}, & \text{if } \exists a \in \mathcal{I}\backslash\{a_0\} \\ & \text{s.t. } f_{h,J(k,h)}^k(s_h^k, a) \geqslant f_{h,J(k,h)}^k(s_h^k, a_0) \\ \phi(s_h^k, a)^\top \boldsymbol{\theta}_h^\star - 2\alpha_h^k \|\phi(s_h^k, a)\|_{(\mathbf{H}_h^k)^{-1}}, & \text{if } \forall a \in \mathcal{I}\backslash\{a_0\} \\ & f_{h,J(k,h)}^k(s_h^k, a) < f_{h,J(k,h)}^k(s_h^k, a_0). \end{cases}$$

Let $\boldsymbol{v}_h^k(s_h^k) = \left( v_h^k(s_h^k, a) \right)_{a \in A_h^k} \in \mathbb{R}^{M_h^k}$ and $\boldsymbol{v}_h^\star(s_h^k) = \left( \phi(s_h^k, a)^\top \boldsymbol{\theta}_h^\star \right)_{a \in A_h^k} \in \mathbb{R}^{M_h^k}$. Thanks to exact second-order Taylor expansion, we obtain that

$$\sum_{k \in \mathcal{K}} \sum_{a \in A_h^k} \left( \widetilde{\mathcal{P}}_{h,J(k,h)}^k(a|s_h^k, A_h^k) - \mathcal{P}_h(a|s_h^k, A_h^k) \right) f_{h,J(k,h)}^k(s_h^k, a)$$

$$= \sum_{k \in \mathcal{K}} \tilde{R}(\boldsymbol{v}_h^k(s_h^k)) - \tilde{R}(\boldsymbol{v}_h^\star(s_h^k))$$

$$= \underbrace{\sum_{k \in \mathcal{K}} \nabla \tilde{R}(\boldsymbol{v}_h^\star(s_h^k))^\top \left( \boldsymbol{v}_h^k(s_h^k) - \boldsymbol{v}_h^\star(s_h^k) \right)}_{(A)}$$

$$+ \underbrace{\frac{1}{2} \sum_{k \in \mathcal{K}} \left( \boldsymbol{v}_h^k(s_h^k) - \boldsymbol{v}_h^\star(s_h^k) \right)^\top \nabla^2 \tilde{R}(\bar{\boldsymbol{v}}_h^k(s_h^k)) \left( \boldsymbol{v}_h^k(s_h^k) - \boldsymbol{v}_h^\star(s_h^k) \right)}_{(B)}, \qquad \text{(D.30)}$$

where $\bar{\boldsymbol{v}}_h^k(s_h^k) = \left( \bar{v}_h^k(s_h^k, a) \right)_{a \in A_h^k} \in \mathbb{R}^{M_h^k}$ is the convex combination of $\boldsymbol{v}_h^k(s_h^k)$ and $\boldsymbol{v}_h^\star(s_h^k)$.

We first bound the term (A) in Equation D.30.

$$
\sum_{k \in \mathcal{K}} \nabla \tilde{R}(\boldsymbol{v}_h^\star(s_h^k))^\top \left(\boldsymbol{v}_h^k(s_h^k) - \boldsymbol{v}_h^\star(s_h^k)\right)
$$

$$
= \sum_{k \in \mathcal{K}} \sum_{a \in A_h^k} \frac{\exp\left(\phi(s_h^k, a)^\top \boldsymbol{\theta}_h^\star\right) f_{h, J(k,h)}^k(s_h^k, a)}{\sum_{a'' \in A_h^k} \exp\left(\phi(s_h^k, a'')^\top \boldsymbol{\theta}_h^\star\right)} \left(v_h^k(s_h^k, a) - \phi(s_h^k, a)^\top \boldsymbol{\theta}_h^\star\right)
$$

$$
- \sum_{a \in A_h^k} \sum_{a' \in A_h^k} \frac{\exp\left(\phi(s_h^k, a)^\top \boldsymbol{\theta}_h^\star\right) f_{h, J(k,h)}^k(s_h^k, a) \exp\left(\phi(s_h^k, a')^\top \boldsymbol{\theta}_h^\star\right)}{\left(\sum_{a'' \in A_h^k} \exp\left(\phi(s_h^k, a'')^\top \boldsymbol{\theta}_h^\star\right)\right)^2} \left(v_h^k(s_h^k, a') - \phi(s_h^k, a')^\top \boldsymbol{\theta}_h^\star\right)
$$

$$
= \sum_{k \in \mathcal{K}} \sum_{a \in A_h^k} \mathcal{P}_h\left(a | s_h^k, A_h^k; \boldsymbol{\theta}_h^\star\right) f_{h, J(k,h)}^k(s_h^k, a)
$$

$$
\cdot \left( \left(v_h^k(s_h^k, a) - \phi(s_h^k, a)^\top \boldsymbol{\theta}_h^\star\right) - \sum_{a' \in A_h^k} \mathcal{P}_h\left(a' | s_h^k, A_h^k; \boldsymbol{\theta}_h^\star\right) \left(v_h^k(s_h^k, a') - \phi(s_h^k, a')^\top \boldsymbol{\theta}_h^\star\right) \right).
$$

$$
\text{(D.31)}
$$

We bound the right-hand side of Equation D.31 by examining two separate cases.

**Case (i)** For $(k, h) \in \mathcal{K} \times [H]$ such that $v_h^k(s_h^k, a) = \phi(s_h^k, a)^\top \boldsymbol{\theta}_h^\star + 2\alpha_h^k \|\phi(s_h^k, a)\|_{(\mathbf{H}_h^k)^{-1}}$.
In this case, by denoting $\mathbb{E}_{\boldsymbol{\theta}_h^\star}[\cdot] = \mathbb{E}_{a' \sim \mathcal{P}_h(\cdot | s_h^k, A_h^k; \boldsymbol{\theta}_h^\star)}[\cdot]$ for simplicity, we get

$$
\nabla \tilde{R}(\boldsymbol{v}_h^\star(s_h^k))^\top \left(\boldsymbol{v}_h^k(s_h^k) - \boldsymbol{v}_h^\star(s_h^k)\right)
$$

$$
= 2\alpha_h^k \sum_{a \in A_h^k} \mathcal{P}_h\left(a | s_h^k, A_h^k; \boldsymbol{\theta}_h^\star\right) f_{h, J(k,h)}^k(s_h^k, a) \left( \|\phi(s_h^k, a)\|_{(\mathbf{H}_h^k)^{-1}} - \mathbb{E}_{\boldsymbol{\theta}_h^\star}\left[ \|\phi(s_h^k, a')\|_{(\mathbf{H}_h^k)^{-1}} \right] \right)
$$

$$
\leqslant 2\alpha_h^k \sum_{a \in A_h^{k,+}} \mathcal{P}_h\left(a | s_h^k, A_h^k; \boldsymbol{\theta}_h^\star\right) \left( \|\phi(s_h^k, a)\|_{(\mathbf{H}_h^k)^{-1}} - \mathbb{E}_{\boldsymbol{\theta}_h^\star}\left[ \|\phi(s_h^k, a')\|_{(\mathbf{H}_h^k)^{-1}} \right] \right)
$$

$$
\leqslant 2\alpha_h^k \sum_{a \in A_h^{k,+}} \mathcal{P}_h\left(a | s_h^k, A_h^k; \boldsymbol{\theta}_h^\star\right) \left( \|\phi(s_h^k, a)\|_{(\mathbf{H}_h^k)^{-1}} - \left\|\mathbb{E}_{\boldsymbol{\theta}_h^\star}\left[\phi(s_h^k, a')\right]\right\|_{(\mathbf{H}_h^k)^{-1}} \right)
$$

$$
\leqslant 2\alpha_h^k \sum_{a \in A_h^{k,+}} \mathcal{P}_h\left(a | s_h^k, A_h^k; \boldsymbol{\theta}_h^\star\right) \left\|\phi(s_h^k, a) - \mathbb{E}_{\boldsymbol{\theta}_h^\star}\left[\phi(s_h^k, a')\right]\right\|_{(\mathbf{H}_h^k)^{-1}}
$$

$$
\leqslant 2\alpha_h^k \sum_{a \in A_h^k} \mathcal{P}_h\left(a | s_h^k, A_h^k; \boldsymbol{\theta}_h^\star\right) \left\|\phi(s_h^k, a) - \mathbb{E}_{\boldsymbol{\theta}_h^\star}\left[\phi(s_h^k, a')\right]\right\|_{(\mathbf{H}_h^k)^{-1}}, \qquad \text{(D.32)}
$$

where, in the first inequality, we define $A_h^{k,+} \subseteq A_h^k$ as the subset of items in $A_h^k$ such that the term $\|\phi(s_h^k, a)\|_{(\mathbf{H}_h^k)^{-1}} - \mathbb{E}_{\boldsymbol{\theta}_h^\star}\left[\|\phi(s_h^k, a')\|_{(\mathbf{H}_h^k)^{-1}}\right] \geqslant 0$ and $f_{h, J(k,h)}^k \in [0, 1]$, the second inequality holds due to Jensen's inequality, and the second-to-last inequality holds due to the fact that $\|\mathbf{a}\| = \|\mathbf{a} - \mathbf{b} + \mathbf{b}\| \leqslant \|\mathbf{a} - \mathbf{b}\| + \|\mathbf{b}\|$ for any vectors $\mathbf{a}, \mathbf{b} \in \mathbb{R}^d$.

For simplicity, we abbreviate the expectations as $\mathbb{E}_{\boldsymbol{\theta}_h^{k+1}}[\cdot] = \mathbb{E}_{a' \sim \mathcal{P}_h(\cdot | s_h^k, A_h^k; \boldsymbol{\theta}_h^{k+1})}[\cdot]$. Additionally, let $\bar{\phi}(s_h^k, a) = \phi(s_h^k, a) - \mathbb{E}_{\boldsymbol{\theta}_h^\star}\left[\phi(s_h^k, a')\right]$ and $\widetilde{\phi}(s_h^k, a) = \phi(s_h^k, a) - \mathbb{E}_{\boldsymbol{\theta}_h^{k+1}}\left[\phi(s_h^k, a')\right]$. Then, we

have

$$\sum_{a \in A_h^k} \mathcal{P}_h\left(a|s_h^k, A_h^k; \boldsymbol{\theta}_h^\star\right) \left\|\phi(s_h^k, a) - \mathbb{E}_{\boldsymbol{\theta}_h^\star}\left[\phi(s_h^k, a')\right]\right\|_{(\mathbf{H}_h^k)^{-1}}$$

$$= \sum_{a \in A_h^k} \mathcal{P}_h\left(a|s_h^k, A_h^k; \boldsymbol{\theta}_h^\star\right) \left\|\bar{\phi}(s_h^k, a)\right\|_{(\mathbf{H}_h^k)^{-1}}$$

$$\leqslant \sum_{a \in A_h^k} \mathcal{P}_h\left(a|s_h^k, A_h^k; \boldsymbol{\theta}_h^\star\right) \left\|\bar{\phi}(s_h^k, a) - \widetilde{\phi}(s_h^k, a)\right\|_{(\mathbf{H}_h^k)^{-1}}$$

$$+ \sum_{a \in A_h^k} \left(\mathcal{P}_h\left(a|s_h^k, A_h^k; \boldsymbol{\theta}_h^\star\right) - \mathcal{P}_h\left(a|s_h^k, A_h^k; \boldsymbol{\theta}_h^{k+1}\right)\right) \left\|\widetilde{\phi}(s_h^k, a)\right\|_{(\mathbf{H}_h^k)^{-1}}$$

$$+ \sum_{a \in A_h^k} \mathcal{P}_h\left(a|s_h^k, A_h^k; \boldsymbol{\theta}_h^{k+1}\right) \left\|\widetilde{\phi}(s_h^k, a)\right\|_{(\mathbf{H}_h^k)^{-1}}, \tag{D.33}$$

where the inequality holds by the triangle inequality. Now, we bound the terms on the right-hand side of Equation D.33 individually. For the first term, we get

$$\sum_{a \in A_h^k} \mathcal{P}_h\left(a|s_h^k, A_h^k; \boldsymbol{\theta}_h^\star\right) \left\|\bar{\phi}(s_h^k, a) - \widetilde{\phi}(s_h^k, a)\right\|_{(\mathbf{H}_h^k)^{-1}}$$

$$= \sum_{a \in A_h^k} \mathcal{P}_h\left(a|s_h^k, A_h^k; \boldsymbol{\theta}_h^\star\right) \left\|\mathbb{E}_{\boldsymbol{\theta}_h^{k+1}}\left[\phi(s_h^k, a')\right] - \mathbb{E}_{\boldsymbol{\theta}_h^\star}\left[\phi(s_h^k, a')\right]\right\|_{(\mathbf{H}_h^k)^{-1}}$$

$$= \sum_{a \in A_h^k} \mathcal{P}_h\left(a|s_h^k, A_h^k; \boldsymbol{\theta}_h^\star\right) \left\|\sum_{a' \in A_h^k} \left(\mathcal{P}_h\left(a'|s_h^k, A_h^k; \boldsymbol{\theta}_h^{k+1}\right) - \mathcal{P}_h\left(a'|s_h^k, A_h^k; \boldsymbol{\theta}_h^\star\right)\right) \phi(s_h^k, a')\right\|_{(\mathbf{H}_h^k)^{-1}}$$

By the mean value theorem, there exists $\boldsymbol{\xi}_h^k = (1-c)\boldsymbol{\theta}_h^\star + c\boldsymbol{\theta}_h^{k+1}$ for some $c \in (0,1)$ such that

$$\left\|\sum_{a' \in A_h^k} \left(\mathcal{P}_h\left(a'|s_h^k, A_h^k; \boldsymbol{\theta}_h^{k+1}\right) - \mathcal{P}_h\left(a'|s_h^k, A_h^k; \boldsymbol{\theta}_h^\star\right)\right) \phi(s_h^k, a')\right\|_{(\mathbf{H}_h^k)^{-1}}$$

$$= \left\|\sum_{a' \in A_h^k} \nabla \mathcal{P}_h\left(a'|s_h^k, A_h^k; \boldsymbol{\xi}_h^k\right)^\top \left(\boldsymbol{\theta}_h^{k+1} - \boldsymbol{\theta}_h^\star\right) \phi(s_h^k, a')\right\|_{(\mathbf{H}_h^k)^{-1}}$$

$$\leqslant \sum_{a' \in A_h^k} \left|\nabla \mathcal{P}_h\left(a'|s_h^k, A_h^k; \boldsymbol{\xi}_h^k\right)^\top \left(\boldsymbol{\theta}_h^{k+1} - \boldsymbol{\theta}_h^\star\right)\right| \left\|\phi(s_h^k, a')\right\|_{(\mathbf{H}_h^k)^{-1}}$$

$$\leqslant \sum_{a' \in A_h^k} \left|\mathcal{P}_h\left(a'|s_h^k, A_h^k; \boldsymbol{\xi}_h^k\right) \phi(s_h^k, a')^\top \left(\boldsymbol{\theta}_h^{k+1} - \boldsymbol{\theta}_h^\star\right)\right| \left\|\phi(s_h^k, a')\right\|_{(\mathbf{H}_h^k)^{-1}}$$

$$+ \sum_{a' \in A_h^k} \left|\mathcal{P}_h\left(a'|s_h^k, A_h^k; \boldsymbol{\xi}_h^k\right) \sum_{a'' \in A_h^k} \mathcal{P}_h\left(a''|s_h^k, A_h^k; \boldsymbol{\xi}_h^k\right)) \phi(s_h^k, a'')^\top \left(\boldsymbol{\theta}_h^{k+1} - \boldsymbol{\theta}_h^\star\right)\right| \left\|\phi(s_h^k, a')\right\|_{(\mathbf{H}_h^k)^{-1}}$$

$$\leqslant \alpha_h^{k+1} \sum_{a' \in A_h^k} \mathcal{P}_h\left(a'|s_h^k, A_h^k; \boldsymbol{\xi}_h^k\right) \left\|\phi(s_h^k, a')\right\|_{(\mathbf{H}_h^k)^{-1}}^2$$

$$+ \left(\alpha_h^{k+1} \sum_{a' \in A_h^k} \mathcal{P}_h\left(a'|s_h^k, A_h^k; \boldsymbol{\xi}_h^k\right) \left\|\phi(s_h^k, a')\right\|_{(\mathbf{H}_h^k)^{-1}}\right)^2$$

$$\leqslant 2\alpha_h^{k+1} \sum_{a' \in A_h^k} \mathcal{P}_h\left(a'|s_h^k, A_h^k; \boldsymbol{\xi}_h^k\right) \left\|\phi(s_h^k, a')\right\|_{(\mathbf{H}_h^k)^{-1}}^2 \leqslant 2\alpha_h^{k+1} \max_{a' \in A_h^k} \left\|\phi(s_h^k, a')\right\|_{(\mathbf{H}_h^k)^{-1}}^2, \tag{D.34}$$

where the third inequality by Hölder's inequality under the good event $\mathcal{E}^\theta$, and the second-to-last inequality follows from Jensen's inequality.

Similarly, we can also bound the second term of Equation D.33 by the mean value theorem. By the mean value theorem, there exists $\bar{\xi}_h^k = (1 - c')\boldsymbol{\theta}_h^\star + c'\boldsymbol{\theta}_h^{k+1}$ for some $c' \in (0, 1)$ such that

$$
\sum_{a \in A_h^k} \left( \mathcal{P}_h\left(a|s_h^k, A_h^k; \boldsymbol{\theta}_h^\star\right) - \mathcal{P}_h\left(a|s_h^k, A_h^k; \boldsymbol{\theta}_h^{k+1}\right) \right) \left\| \widetilde{\phi}(s_h^k, a) \right\|_{(\mathbf{H}_h^k)^{-1}}
$$

$$
= \sum_{a \in A_h^k} \nabla \mathcal{P}_h\left(a|s_h^k, A_h^k; \bar{\xi}_h^k\right)^\top \left(\boldsymbol{\theta}_h^\star - \boldsymbol{\theta}_h^{k+1}\right) \left\| \widetilde{\phi}(s_h^k, a) \right\|_{(\mathbf{H}_h^k)^{-1}}
$$

$$
= \sum_{a \in A_h^k} \mathcal{P}_h\left(a|s_h^k, A_h^k; \bar{\xi}_h^k\right) \phi(s_h^k, a)^\top \left(\boldsymbol{\theta}_h^\star - \boldsymbol{\theta}_h^{k+1}\right) \left\| \widetilde{\phi}(s_h^k, a) \right\|_{(\mathbf{H}_h^k)^{-1}}
$$

$$
- \sum_{a \in A_h^k} \mathcal{P}_h\left(a|s_h^k, A_h^k; \bar{\xi}_h^k\right) \left\| \widetilde{\phi}(s_h^k, a) \right\|_{(\mathbf{H}_h^k)^{-1}} \sum_{a' \in A_h^k} \mathcal{P}_h\left(a'|s_h^k, A_h^k; \bar{\xi}_h^k\right) \phi(s_h^k, a')^\top \left(\boldsymbol{\theta}_h^\star - \boldsymbol{\theta}_h^{k+1}\right)
$$

$$
\leqslant \alpha_h^{k+1} \max_{a \in A_h^k} \left\| \phi(s_h^k, a) \right\|_{(\mathbf{H}_h^k)^{-1}} \left\| \widetilde{\phi}(s_h^k, a) \right\|_{(\mathbf{H}_h^k)^{-1}}
$$

$$
+ \alpha_h^{k+1} \max_{a \in A_h^k} \left\| \widetilde{\phi}(s_h^k, a) \right\|_{(\mathbf{H}_h^k)^{-1}} \max_{a \in A_h^k} \left\| \phi(s_h^k, a) \right\|_{(\mathbf{H}_h^k)^{-1}}
$$

$$
\leqslant 2\alpha_h^{k+1} \max \left\{ \max_{a \in A_h^k} \left\| \widetilde{\phi}(s_h^k, a) \right\|_{(\mathbf{H}_h^k)^{-1}}^2, \max_{a \in A_h^k} \left\| \phi(s_h^k, a) \right\|_{(\mathbf{H}_h^k)^{-1}}^2 \right\}, \tag{D.35}
$$

where the last inequality holds by applying the AM-GM inequality to each term:

$$
\max_{a \in A_h^k} \left\| \phi(s_h^k, a) \right\|_{(\mathbf{H}_h^k)^{-1}} \left\| \widetilde{\phi}(s_h^k, a) \right\|_{(\mathbf{H}_h^k)^{-1}} + \max_{a \in A_h^k} \left\| \widetilde{\phi}(s_h^k, a) \right\|_{(\mathbf{H}_h^k)^{-1}} \max_{a \in A_h^k} \left\| \phi(s_h^k, a) \right\|_{(\mathbf{H}_h^k)^{-1}}
$$

$$
\leqslant \max_{a \in A_h^k} \frac{\left\| \phi(s_h^k, a) \right\|_{(\mathbf{H}_h^k)^{-1}}^2 + \left\| \widetilde{\phi}(s_h^k, a) \right\|_{(\mathbf{H}_h^k)^{-1}}^2}{2}
$$

$$
+ \frac{\left( \max_{a \in A_h^k} \left\| \widetilde{\phi}(s_h^k, a) \right\|_{(\mathbf{H}_h^k)^{-1}} \right)^2 + \left( \max_{a \in A_h^k} \left\| \phi(s_h^k, a) \right\|_{(\mathbf{H}_h^k)^{-1}} \right)^2}{2}
$$

$$
\leqslant 2 \max \left\{ \max_{a \in A_h^k} \left\| \widetilde{\phi}(s_h^k, a) \right\|_{(\mathbf{H}_h^k)^{-1}}^2, \max_{a \in A_h^k} \left\| \phi(s_h^k, a) \right\|_{(\mathbf{H}_h^k)^{-1}}^2 \right\}.
$$

Combining Equation D.32, Equation D.33, Equation D.34 and Equation D.35, we obtain that

$$
\nabla \tilde{R}(\boldsymbol{v}_h^\star(s_h^k))^\top \left( \boldsymbol{v}_h^k(s_h^k) - \boldsymbol{v}_h^\star(s_h^k) \right)
$$

$$
\leqslant 2\alpha_h^k \sum_{a \in A_h^k} \mathcal{P}_h\left(a|s_h^k, A_h^k; \boldsymbol{\theta}_h^{k+1}\right) \left\| \widetilde{\phi}(s_h^k, a) \right\|_{(\mathbf{H}_h^k)^{-1}}
$$

$$
+ 8\alpha_h^k \alpha_h^{k+1} \max \left\{ \max_{a \in A_h^k} \left\| \widetilde{\phi}(s_h^k, a) \right\|_{(\mathbf{H}_h^k)^{-1}}^2, \max_{a \in A_h^k} \left\| \phi(s_h^k, a) \right\|_{(\mathbf{H}_h^k)^{-1}}^2 \right\}. \tag{D.36}
$$

Now, we consider the second case to bound the term (A).

**Case (ii)** For $(k, h) \in \mathcal{K} \times [H]$ such that $v_h^k(s_h^k, a) = \phi(s_h^k, a)^\top \boldsymbol{\theta}_h^\star - 2\alpha_h^k \|\phi(s_h^k, a)\|_{(\mathbf{H}_h^k)^{-1}}$.

In this case, we know that $f_{h,J(k,h)}^k(s_h^k, a) < f_{h,J(k,h)}^k(s_h^k, a_0)$ for all $a \in \mathcal{I}\backslash\{a_0\}$. This implies that $|A_h^k| = 2$, since adding any item $a \in \mathcal{I}\backslash\{a_0\}$ to the set $\{a_0\}$ always decreases the expected value of $f_{h,J(k,h)}^k$. Furthermore, since we assume $\phi(s_h^k, a_0) = 0$ (which also implies $v_h^k(s_h^k, a_0) = 0$), and

denoting $A_h^k = \{a_0, \tilde{a}_h^k\}$, we have:

$$
\begin{aligned}
&\nabla \tilde{R}(\boldsymbol{v}_h^\star(s_h^k))^\top \left(\boldsymbol{v}_h^k(s_h^k) - \boldsymbol{v}_h^\star(s_h^k)\right) \\
&= 2\alpha_h^k \mathcal{P}_h\left(\tilde{a}_h^k|s_h^k, A_h^k; \boldsymbol{\theta}_h^\star\right) f_{h,J(k,h)}^k(s_h^k, \tilde{a}_h^k) \\
&\qquad \cdot \left(\|\phi(s_h^k, \tilde{a}_h^k)\|_{(\mathbf{H}_h^k)^{-1}} - \mathcal{P}_h\left(\tilde{a}_h^k|s_h^k, A_h^k; \boldsymbol{\theta}_h^\star\right) \left[\|\phi(s_h^k, \tilde{a}_h^k)\|_{(\mathbf{H}_h^k)^{-1}}\right]\right) \\
&= 2\alpha_h^k \mathcal{P}_h\left(\tilde{a}_h^k|s_h^k, A_h^k; \boldsymbol{\theta}_h^\star\right) \mathcal{P}_h\left(a_0|s_h^k, A_h^k; \boldsymbol{\theta}_h^\star\right) f_{h,J(k,h)}^k(s_h^k, \tilde{a}_h^k) \|\phi(s_h^k, \tilde{a}_h^k)\|_{(\mathbf{H}_h^k)^{-1}} \\
&\leqslant 2\alpha_h^k \mathcal{P}_h\left(\tilde{a}_h^k|s_h^k, A_h^k; \boldsymbol{\theta}_h^\star\right) \mathcal{P}_h\left(a_0|s_h^k, A_h^k; \boldsymbol{\theta}_h^\star\right) \|\phi(s_h^k, \tilde{a}_h^k)\|_{(\mathbf{H}_h^k)^{-1}},
\end{aligned}
\tag{D.37}
$$

where the last inequality is due to the fact that $f_{h,J(k,h)}^k \leqslant 1$. Now, we bound the term $\mathcal{P}_h\left(\tilde{a}_h^k|s_h^k, A_h^k; \boldsymbol{\theta}_h^\star\right) \mathcal{P}_h\left(a_0|s_h^k, A_h^k; \boldsymbol{\theta}_h^\star\right)$ by the mean value theorem. Recall that we can express the term as follows:

$$
\mathcal{P}_h\left(\tilde{a}_h^k|s_h^k, A_h^k; \boldsymbol{\theta}_h^\star\right) \mathcal{P}_h\left(a_0|s_h^k, A_h^k; \boldsymbol{\theta}_h^\star\right) = \frac{\exp\left(\phi(s_h^k, \tilde{a}_h^k)^\top \boldsymbol{\theta}_h^\star\right)}{\left(1 + \exp\left(\phi(s_h^k, \tilde{a}_h^k)^\top \boldsymbol{\theta}_h^\star\right)\right)^2}.
$$

Then, by the mean value theorem, there exists $\tilde{\boldsymbol{\xi}}_h^k = (1 - c')\boldsymbol{\theta}_h^\star + c'\boldsymbol{\theta}_h^{k+1}$ for some $c' \in (0,1)$ such that

$$
\begin{aligned}
&\mathcal{P}_h\left(\tilde{a}_h^k|s_h^k, A_h^k; \boldsymbol{\theta}_h^\star\right) \mathcal{P}_h\left(a_0|s_h^k, A_h^k; \boldsymbol{\theta}_h^\star\right) - \mathcal{P}_h\left(\tilde{a}_h^k|s_h^k, A_h^k; \boldsymbol{\theta}_h^{k+1}\right) \mathcal{P}_h\left(a_0|s_h^k, A_h^k; \boldsymbol{\theta}_h^{k+1}\right) \\
&= \frac{\exp\left(\phi(s_h^k, \tilde{a}_h^k)^\top \boldsymbol{\theta}_h^\star\right)}{\left(1 + \exp\left(\phi(s_h^k, \tilde{a}_h^k)^\top \boldsymbol{\theta}_h^\star\right)\right)^2} - \frac{\exp\left(\phi(s_h^k, \tilde{a}_h^k)^\top \boldsymbol{\theta}_h^{k+1}\right)}{\left(1 + \exp\left(\phi(s_h^k, \tilde{a}_h^k)^\top \boldsymbol{\theta}_h^{k+1}\right)\right)^2} \\
&= \frac{\exp\left(\phi(s_h^k, \tilde{a}_h^k)^\top \tilde{\boldsymbol{\xi}}_h^k\right)}{\left(1 + \exp\left(\phi(s_h^k, \tilde{a}_h^k)^\top \tilde{\boldsymbol{\xi}}_h^k\right)\right)^2} \left(1 - 2\frac{\exp\left(\phi(s_h^k, \tilde{a}_h^k)^\top \tilde{\boldsymbol{\xi}}_h^k\right)}{1 + \exp\left(\phi(s_h^k, \tilde{a}_h^k)^\top \tilde{\boldsymbol{\xi}}_h^k\right)}\right) \phi(s_h^k, \tilde{a}_h^k)^\top \left(\boldsymbol{\theta}_h^\star - \boldsymbol{\theta}_h^{k+1}\right) \\
&\leqslant \frac{\exp\left(\phi(s_h^k, \tilde{a}_h^k)^\top \tilde{\boldsymbol{\xi}}_h^k\right)}{\left(1 + \exp\left(\phi(s_h^k, \tilde{a}_h^k)^\top \tilde{\boldsymbol{\xi}}_h^k\right)\right)^2} \phi(s_h^k, \tilde{a}_h^k)^\top \left(\boldsymbol{\theta}_h^\star - \boldsymbol{\theta}_h^{k+1}\right) \\
&\leqslant \alpha_h^{k+1} \mathcal{P}_h\left(\tilde{a}_h^k|s_h^k, A_h^k; \tilde{\boldsymbol{\xi}}_h^k\right) \mathcal{P}_h\left(a_0|s_h^k, A_h^k; \tilde{\boldsymbol{\xi}}_h^k\right) \|\phi(s_h^k, \tilde{a}_h^k)\|_{(\mathbf{H}_h^k)^{-1}} \\
&\leqslant \alpha_h^{k+1} \max_{a \in A_h^k} \|\phi(s_h^k, a)\|_{(\mathbf{H}_h^k)^{-1}},
\end{aligned}
\tag{D.38}
$$

where the first inequality comes from the fact that $\left|1 - 2\frac{\exp\left(\phi(s_h^k, \tilde{a}_h^k)^\top \tilde{\boldsymbol{\xi}}_h^k\right)}{\left(1 + \exp\left(\phi(s_h^k, \tilde{a}_h^k)^\top \tilde{\boldsymbol{\xi}}_h^k\right)\right)^2}\right| \leqslant 1$, and the second inequality holds due to Hölder's inequality, conditioned on the good event $\mathcal{E}^\theta$. Plugging Equation D.38 into Equation D.37, we get

$$
\begin{aligned}
&\nabla \tilde{R}(\boldsymbol{v}_h^\star(s_h^k))^\top \left(\boldsymbol{v}_h^k(s_h^k) - \boldsymbol{v}_h^\star(s_h^k)\right) \\
&\leqslant 2\alpha_h^k \mathcal{P}_h\left(\tilde{a}_h^k|s_h^k, A_h^k; \boldsymbol{\theta}_h^{k+1}\right) \mathcal{P}_h\left(a_0|s_h^k, A_h^k; \boldsymbol{\theta}_h^{k+1}\right) \|\phi(s_h^k, \tilde{a}_h^k)\|_{(\mathbf{H}_h^k)^{-1}} \\
&\quad + 2\alpha_h^k \alpha_h^{k+1} \max_{a \in A_h^k} \|\phi(s_h^k, a)\|_{(\mathbf{H}_h^k)^{-1}} \\
&= 2\alpha_h^k \sum_{a \in A_h^k} \mathcal{P}_h\left(a|s_h^k, A_h^k; \boldsymbol{\theta}_h^{k+1}\right) \mathcal{P}_h\left(a_0|s_h^k, A_h^k; \boldsymbol{\theta}_h^{k+1}\right) \|\phi(s_h^k, a)\|_{(\mathbf{H}_h^k)^{-1}} \\
&\quad + 2\alpha_h^k \alpha_h^{k+1} \max_{a \in A_h^k} \|\phi(s_h^k, a)\|_{(\mathbf{H}_h^k)^{-1}},
\end{aligned}
\tag{D.39}
$$

where the equality holds due to the assumption that $\phi(s, a_0) = 0$.

Now, we are ready to bound the term (A). For any fixed $h \in [H]$, let $\mathcal{K}_{(i)}$ denote the set of episodes where **Case (i)** holds, and $\mathcal{K}_{(ii)}$ denote the set of episodes where **Case (ii)** holds. More formally, we

define:

$$\mathcal{K}_{(i)} = \left\{ k \in \mathcal{K} : v_h^k(s_h^k, a) = \phi(s_h^k, a)^\top \boldsymbol{\theta}_h^\star + 2\alpha_h^k \|\phi(s_h^k, a)\|_{(\mathbf{H}_h^k)^{-1}} \right\}$$

$$\mathcal{K}_{(ii)} = \left\{ k \in \mathcal{K} : v_h^k(s_h^k, a) = \phi(s_h^k, a)^\top \boldsymbol{\theta}_h^\star - 2\alpha_h^k \|\phi(s_h^k, a)\|_{(\mathbf{H}_h^k)^{-1}} \right\}.$$

Then, by combining Equation D.36 and Equation D.39, using the fact that $\alpha_h^1 \leqslant \cdots \leqslant \alpha_h^K$, and summing over $k \in \mathcal{K}$, we obtain that

$$\sum_{k \in \mathcal{K}} \nabla \tilde{R}(\boldsymbol{v}_h^\star(s_h^k))^\top \left( \boldsymbol{v}_h^k(s_h^k) - \boldsymbol{v}_h^\star(s_h^k) \right)$$

$$\leqslant 2\alpha_h^K \sum_{k \in \mathcal{K}_{(i)}} \sum_{a \in A_h^k} \mathcal{P}_h \left( a | s_h^k, A_h^k; \boldsymbol{\theta}_h^{k+1} \right) \left\| \widetilde{\phi}(s_h^k, a) \right\|_{(\mathbf{H}_h^k)^{-1}}$$

$$+ 2\alpha_h^K \sum_{k \in \mathcal{K}_{(ii)}} \sum_{a \in A_h^k} \mathcal{P}_h \left( a | s_h^k, A_h^k; \boldsymbol{\theta}_h^{k+1} \right) \mathcal{P}_h \left( a_0 | s_h^k, A_h^k; \boldsymbol{\theta}_h^{k+1} \right) \|\phi(s_h^k, a)\|_{(\mathbf{H}_h^k)^{-1}}$$

$$+ 10 \left( \alpha_h^K \right)^2 \sum_{k \in \mathcal{K}} \max \left\{ \max_{a \in A_h^k} \left\| \widetilde{\phi}(s_h^k, a) \right\|_{(\mathbf{H}_h^k)^{-1}}^2, \max_{a \in A_h^k} \|\phi(s_h^k, a)\|_{(\mathbf{H}_h^k)^{-1}}^2 \right\}$$

$$\leqslant 2\alpha_h^K \sqrt{|\mathcal{K}|} \cdot \left( \sum_{k=1}^K \sum_{a \in A_h^k} \mathcal{P}_h \left( a | s_h^k, A_h^k; \boldsymbol{\theta}_h^{k+1} \right) \left\| \widetilde{\phi}(s_h^k, a) \right\|_{(\mathbf{H}_h^k)^{-1}}^2 \right.$$

$$+ \sum_{k=1}^K \sum_{a \in A_h^k} \mathcal{P}_h \left( a | s_h^k, A_h^k; \boldsymbol{\theta}_h^{k+1} \right) \mathcal{P}_h \left( a_0 | s_h^k, A_h^k; \boldsymbol{\theta}_h^{k+1} \right) \|\phi(s_h^k, a)\|_{(\mathbf{H}_h^k)^{-1}}^2 \right)^{1/2}$$

$$+ 10 \left( \alpha_h^K \right)^2 \sum_{k=1}^K \max \left\{ \max_{a \in A_h^k} \left\| \widetilde{\phi}(s_h^k, a) \right\|_{(\mathbf{H}_h^k)^{-1}}^2, \max_{a \in A_h^k} \|\phi(s_h^k, a)\|_{(\mathbf{H}_h^k)^{-1}}^2 \right\}$$

$$= \mathcal{O} \left( d\sqrt{|\mathcal{K}|} \cdot \log K \log M + \frac{1}{\kappa} d^2 \left( \log K \log M \right)^2 \right), \tag{D.40}$$

where the second inequality follows from the Cauchy-Schwarz inequality, and the last equality holds by applying Lemma D.3 and substituting in the value of $\alpha_h^k$.

Now, we bound the term (B) in Equation D.30. Let $p_a(\bar{v}_h^k(s_h^k)) = \frac{\exp(\bar{v}_h^k(s_h^k, a))}{1 + \sum_{a'' \in A_h^k} \exp(\bar{v}_h^k(s_h^k, a''))}$.

$$\frac{1}{2} \sum_{k \in \mathcal{K}} \left( \boldsymbol{v}_h^k(s_h^k) - \boldsymbol{v}_h^\star(s_h^k) \right)^\top \nabla^2 \tilde{R}(\bar{\boldsymbol{v}}_h^k(s_h^k)) \left( \boldsymbol{v}_h^k(s_h^k) - \boldsymbol{v}_h^\star(s_h^k) \right)$$

$$= \frac{1}{2} \sum_{k \in \mathcal{K}} \sum_{a \in A_h^k} \sum_{a' \in A_h^k} \left( v_h^k(s_h^k, a) - \phi(s_h^k, a)^\top \boldsymbol{\theta}_h^\star \right) \frac{\partial^2 \tilde{R}(\bar{\boldsymbol{v}}_h^k(s_h^k))}{\partial a \partial a'} \left( v_h^k(s_h^k, a') - \phi(s_h^k, a')^\top \boldsymbol{\theta}_h^\star \right)$$

$$= \frac{1}{2} \sum_{k \in \mathcal{K}} \sum_{\substack{a \in A_h^k \\ }} \sum_{\substack{a' \in A_h^k \\ a' \neq a}} \left( v_h^k(s_h^k, a) - \phi(s_h^k, a)^\top \boldsymbol{\theta}_h^\star \right) \frac{\partial^2 \tilde{R}(\bar{\boldsymbol{v}}_h^k(s_h^k))}{\partial a \partial a'} \left( v_h^k(s_h^k, a') - \phi(s_h^k, a')^\top \boldsymbol{\theta}_h^\star \right)$$

$$+ \frac{1}{2} \sum_{k \in \mathcal{K}} \sum_{a \in A_h^k} \left( v_h^k(s_h^k, a) - \phi(s_h^k, a)^\top \boldsymbol{\theta}_h^\star \right)^2 \frac{\partial^2 \tilde{R}(\bar{\boldsymbol{v}}_h^k(s_h^k))}{\partial a \partial a}$$

$$\leqslant \sum_{k \in \mathcal{K}} \sum_{\substack{a \in A_h^k \\ }} \sum_{\substack{a' \in A_h^k \\ a' \neq a}} \left| v_h^k(s_h^k, a) - \phi(s_h^k, a)^\top \boldsymbol{\theta}_h^\star \right| p_a(\bar{\boldsymbol{v}}_h^k(s_h^k)) p_{a'}(\bar{\boldsymbol{v}}_h^k(s_h^k)) \left| v_h^k(s_h^k, a') - \phi(s_h^k, a')^\top \boldsymbol{\theta}_h^\star \right|$$

$$+ \frac{3}{2} \sum_{k \in \mathcal{K}} \sum_{a \in A_h^k} \left( v_h^k(s_h^k, a) - \phi(s_h^k, a)^\top \boldsymbol{\theta}_h^\star \right)^2 p_a(\bar{\boldsymbol{v}}_h^k(s_h^k)), \tag{D.41}$$

where the inequality holds by Lemma D.4 and $f^k_{j,J(k,h)} \leqslant 1$. To bound the first term in Equation D.41, by applying the AM-GM inequality, we get

$$\sum_{k \in \mathcal{K}} \sum_{a \in A^k_h} \sum_{\substack{a' \in A^k_h \\ a' \neq a}} \left| v^k_h(s^k_h, a) - \phi(s^k_h, a)^\top \boldsymbol{\theta}^\star_h \right| p_a(\bar{\boldsymbol{v}}^k_h(s^k_h)) p_{a'}(\bar{\boldsymbol{v}}^k_h(s^k_h)) \left| v^k_h(s^k_h, a') - \phi(s^k_h, a')^\top \boldsymbol{\theta}^\star_h \right|$$

$$\leqslant \sum_{k \in \mathcal{K}} \sum_{a \in A^k_h} \sum_{a' \in A^k_h} \left| v^k_h(s^k_h, a) - \phi(s^k_h, a)^\top \boldsymbol{\theta}^\star_h \right| p_a(\bar{\boldsymbol{v}}^k_h(s^k_h)) p_{a'}(\bar{\boldsymbol{v}}^k_h(s^k_h)) \left| v^k_h(s^k_h, a') - \phi(s^k_h, a')^\top \boldsymbol{\theta}^\star_h \right|$$

$$\leqslant \frac{1}{2} \sum_{k \in \mathcal{K}} \sum_{a \in A^k_h} \sum_{a' \in A^k_h} \left( v^k_h(s^k_h, a) - \phi(s^k_h, a)^\top \boldsymbol{\theta}^\star_h \right)^2 p_a(\bar{\boldsymbol{v}}^k_h(s^k_h)) p_{a'}(\bar{\boldsymbol{v}}^k_h(s^k_h))$$

$$+ \frac{1}{2} \sum_{k \in \mathcal{K}} \sum_{a \in A^k_h} \sum_{a' \in A^k_h} p_a(\bar{\boldsymbol{v}}^k_h(s^k_h)) p_{a'}(\bar{\boldsymbol{v}}^k_h(s^k_h)) \left( v^k_h(s^k_h, a') - \phi(s^k_h, a')^\top \boldsymbol{\theta}^\star_h \right)^2$$

$$= \sum_{k \in \mathcal{K}} \sum_{a \in A^k_h} \left( v^k_h(s^k_h, a) - \phi(s^k_h, a)^\top \boldsymbol{\theta}^\star_h \right)^2 p_a(\bar{\boldsymbol{v}}^k_h(s^k_h)). \tag{D.42}$$

Plugging Equation D.42 into Equation D.41, we have

$$\frac{1}{2} \sum_{k \in \mathcal{K}} \left( \boldsymbol{v}^k_h(s^k_h) - \boldsymbol{v}^\star_h(s^k_h) \right)^\top \nabla^2 \tilde{R}(\bar{\boldsymbol{v}}^k_h(s^k_h)) \left( \boldsymbol{v}^k_h(s^k_h) - \boldsymbol{v}^\star_h(s^k_h) \right)$$

$$\leqslant \frac{5}{2} \sum_{k \in \mathcal{K}} \sum_{a \in A^k_h} p_a(\bar{\boldsymbol{v}}^k_h(s^k_h)) \left( v^k_h(s^k_h, a) - \phi(s^k_h, a)^\top \boldsymbol{\theta}^\star_h \right)^2$$

$$\leqslant 10 \left( \alpha^K_h \right)^2 \sum_{k \in \mathcal{K}} \sum_{a \in A^k_h} p_a(\bar{\boldsymbol{v}}^k_h(s^k_h)) \| \phi(s^k_h, a) \|_{(\mathbf{H}^k_h)^{-1}}$$

$$\leqslant 10 \left( \alpha^K_h \right)^2 \sum_{k \in \mathcal{K}} \max_{a \in A^k_h} \| \phi(s^k_h, a) \|_{(\mathbf{H}^k_h)^{-1}} = \mathcal{O} \left( \frac{1}{\kappa} d^2 \left( \log K \log M \right)^2 \right), \tag{D.43}$$

where the second inequality holds since $\alpha^1_h \leqslant \cdots \leqslant \alpha^K_h$, and the last inequality holds by Lemma D.3.

Combining Equation D.40 and Equation D.43, we conclude the proof of Lemma D.7. □

### D.4 OPTIMISM

In this subsection, we prove the optimism of our value estimates $V^k_h$.

**Lemma D.8** (Point-wise monotonicity, Lemma 31 of Agarwal et al. 2023). *Suppose Algorithm 1 uses a consistent bonus oracle satisfying Definition 4. For any fixed $(k, h) \in [K] \times [H]$, conditioning on events $\mathcal{E}_{\leqslant k-1} \bigcap \left( \bigcap^H_{h'=h} \mathcal{E}^k_{h'} \right)$, for all $(s_h, a_h) \in \mathcal{S} \times \mathcal{I}$, we have*

1. $\overline{Q}^\star_h(s_h, a_h) \leqslant f^k_{h,1}(s_h, a_h)$;

2. $f^k_{h,-1}(s_h, a_h) \leqslant \overline{Q}^\star_h(s_h, a_h)$;

3. $f^\tau_{h,2}(s_h, a_h) \geqslant \max \left\{ \mathcal{T}_h V^k_{h+1,1}(s_h, a_h), f^k_{h,1}(s_h, a_h) \right\}, \quad \forall \tau \in [k]$.

**Lemma D.9** (Optimism). *Let $V^k_h$ be the realized optimistic value function defined in Equation D.18. Suppose Algorithm 1 uses a consistent bonus oracle satisfying Definition 4. On the even conditioning on the good event $\mathcal{E}^\theta \bigcap \mathcal{E}_{\leqslant K}$, for all $(k, h) \in [K] \times [H]$, we have*

$$V^k_h(s^k_h) \geqslant V^\star_h(s^k_h).$$

*Proof of Lemma D.9.* We denote $A_h^{k,\star} \in \arg\max_A \sum_{a \in A} \mathcal{P}_h(a|s_h^k, A)\overline{Q}_h^{\star}(s_h^k, a)$. If $A_h^k = A_{h,1}^k$, by the definition of the optimal value function $V_h^{\star}$, we have

$$
\begin{aligned}
V_h^{\star}(s_h^k) &= \max_{A \in \mathcal{A}} \sum_{a \in A} \mathcal{P}_h(a|s_h^k, A)\overline{Q}_h^{\star}(s_h^k, a) \\
&= \sum_{a \in A_h^{k,\star}} \mathcal{P}_h(a|s_h^k, A_h^{k,\star})\overline{Q}_h^{\star}(s_h^k, a) \\
&\leqslant \sum_{a \in A_h^{k,\star}} \mathcal{P}_h(a|s_h^k, A_h^{k,\star})f_{h,1}^k(s_h^k, a) \\
&\leqslant \sum_{a \in \tilde{A}_h^k} \widetilde{\mathcal{P}}_{h,1}^k(a|s_h^k, \tilde{A}_h^k)f_{h,1}^k(s_h^k, a) \\
&\leqslant \sum_{a \in A_{h,1}^k} \widetilde{\mathcal{P}}_{h,1}^k(a|s_h^k, A_{h,1}^k)f_{h,1}^k(s_h^k, a) \\
&= \max_{A \in \mathcal{A}} \sum_{a \in A} \widetilde{\mathcal{P}}_{h,1}^k(a|s_h^k, A)f_{h,1}^k(s_h^k, a) = V_h^k(s_h^k),
\end{aligned}
$$

where the first inequality holds by Lemma D.8, in the second inequality, we use the fact that, by Lemma D.2, there exists a subset $\tilde{A}_h^k \subseteq A_h^{k,\star}$ with $\tilde{A}_h^k \in \mathcal{A}$ such that the inequality holds, and the last inequality holds by the definition of $A_{h,1}^k$.

The case where $A_h^k = A_{h,2}^k$ can be proven using the same reasoning. $\qquad\square$

### D.5 VARIANCES

In this subsection, we present properties related to variances.

**Lemma D.10** (Upper bound of variance estimator, Lemma 34 of Agarwal et al. 2023)**.** *Let* $z_h^k = (s_h^k, a_h^k)$*. We denote* $\mathbb{E}_{\mathbb{P}}[\cdot|s_h^k, a_h^k] = \mathbb{E}_{s_{h+1} \sim \mathbb{P}_h(\cdot|s_h^k, a_h^k)}[\cdot|s_h^k, a_h^k]$ *and* $\mathbb{V}_{\mathbb{P}}[\cdot|s_h^k, a_h^k] = \mathbb{V}_{s_{h+1} \sim \mathbb{P}_h(\cdot|s_h^k, a_h^k)}[\cdot|s_h^k, a_h^k]$*, where the expectation in only taken over* $s_{h+1}$ *due to the model transition for shorthand. Suppose Algorithm 1 uses a consistent bonus oracle satisfying Definition 4. For any episode* $k \geqslant 2$ *conditioning on the good event* $\mathcal{E}_{\leqslant K}$*, the variance estimator* $\sigma_h^k$ *satisfies*

$$
\begin{aligned}
\left(\sigma_h^k\right)^2 &\leqslant \mathbb{V}\left[r_h + V_{h+1,1}^k(s_{h+1}) \mid z_h^k\right] + 4\left(f_{h,2}^k(z_h^k) - f_{h,-2}^k(z_h^k)\right) \\
&\quad + 4\min\left\{1, D_{\mathcal{F}_h}\left(z_h^k; \{z_h^\tau\}_{\tau=1}^{k-1}, \{\mathbf{1}^\tau\}_{\tau=1}^{k-1}\right) \cdot \left(2\sqrt{\left(\bar{\beta}_h^k\right)^2 + \rho} + 4L\sqrt{\left(\beta_{h,2}^k\right)^2 + \rho}\right)\right\}.
\end{aligned}
$$

**Lemma D.11** (Sum of variances, Corollary 50 of Agarwal et al. 2023)**.** *Let* $z_h^k = (s_h^k, a_h^k)$*. We denote* $\mathbb{E}_{\mathbb{P}}[\cdot|s_h^k, a_h^k] = \mathbb{E}_{s_{h+1} \sim \mathbb{P}_h(\cdot|s_h^k, a_h^k)}[\cdot|s_h^k, a_h^k]$ *and* $\mathbb{V}_{\mathbb{P}}[\cdot|s_h^k, a_h^k] = \mathbb{V}_{s_{h+1} \sim \mathbb{P}_h(\cdot|s_h^k, a_h^k)}[\cdot|s_h^k, a_h^k]$*, where the expectation in only taken over* $s_{h+1}$ *due to the model transition for shorthand. When* $L = \mathcal{O}(1)$*, with probability at least* $1 - \delta$*, we have*

$$
\begin{aligned}
&\sum_{k=1}^K \sum_{h=1}^H \mathbb{V}\left[r_h + V_{h+1,1}^k(s_{h+1})|z_h^k\right] \\
&\leqslant \mathcal{O}\left(H\sum_{k=1}^K \sum_{h=1}^H \left(f_{h,2}^k(z_h^k) - f_{h,-2}^k(z_h^k)\right) + K + KH^2\delta + H^2|\mathcal{K}_{oo}| + H^4 \log^2 \frac{KH}{\delta}\right).
\end{aligned}
$$

### D.6 APPROXIMATION ERROR OF OPTIMISTIC, OVERLY OPTIMISTIC (PESSIMISTIC) $\overline{Q}$-VALUES

In this section, we provide some inequalities for bounding the optimistic values, overly optimistic values, and overly pessimistic values sequence, which are useful for the proofs in Subsection D.7.

**Lemma D.12** (Approximation error of overly pessimistic $\overline{Q}$)**.** *Suppose Algorithm 1 uses a consistent bonus oracle satisfying Definition 4. Conditioning on the good event* $\mathcal{E}_{\leqslant K}$*, for any* $(k, h) \in [K] \times [H]$*,*

*it holds that*

$$(f_{h,-2}^k - \overline{Q}_h^{\pi_k})(s_h^k, a_h^k) \geqslant \sum_{h'=h+1}^{H} \sum_{a' \in A_{h'}^k} \left( \widetilde{\mathcal{P}}_{h',-2}^k(a'|s_{h'}^k, A_{h'}^k) - \mathcal{P}_{h'}(a'|s_{h'}^k, A_{h'}^k) \right) f_{h',-2}^k(s_{h'}^k, a')$$

$$- 2 \sum_{h'=h}^{H} b_{h',2}^k(s_{h'}^k, a_{h'}^k) + \sum_{h'=h+1}^{H} \zeta_{h',-2}^k + \sum_{h'=h+1}^{H} \dot{\zeta}_{h',-2}^k,$$

*where* $\zeta_{h,-2}^k := \mathbb{E}_{\mathbb{P}}\left[ (V_{h,-2}^k - V_h^{\pi_k})(s_h) \mid s_{h-1}^k, a_{h-1}^k \right] - (V_{h,-2}^k - V_h^{\pi_k})(s_h^k)$ *and* $\dot{\zeta}_{h,-2}^k :=$ $\mathbb{E}_{\mathcal{P}}\left[ \left( f_{h,-2}^k - \overline{Q}_h^{\pi_k} \right)(s_h^k, a_h) \mid s_h^k, A_h^k \right] - \left( f_{h,-2}^k - \overline{Q}_h^{\pi_k} \right)(s_h^k, a_h^k).$

*Proof of Lemma D.12.* Under the event $\mathcal{E}_{\leqslant K}$, we have

$$(f_{h,-2}^k - \overline{Q}_h^{\pi_k})(s_h^k, a_h^k)$$
$$= (f_{h,-2}^k - \mathcal{T}_h V_{h+1,-2}^k)(s_h^k, a_h^k) + (\mathcal{T}_h V_{h+1,-2}^k - \overline{Q}_h^{\pi_k})(s_h^k, a_h^k)$$
$$\geqslant -2b_{h,2}^k(s_h^k, a_h^k) + \mathbb{E}_{\mathbb{P}}\left[ (r_h - r_h) + (V_{h+1,-2}^k - V_{h+1}^{\pi_k})(s_{h+1}) | s_h^k, a_h^k \right]$$
$$= -2b_{h,2}^k(s_h^k, a_h^k) + (V_{h+1,-2}^k - V_{h+1}^{\pi_k})(s_{h+1}^k) + \zeta_{h+1,-2}^k$$
$$\geqslant -2b_{h,2}^k(s_h^k, a_h^k) + (Q_{h+1,-2}^k - Q_{h+1}^{\pi_k})(s_{h+1}^k, A_{h+1}^k) + \zeta_{h+1,-2}^k$$
$$= \sum_{a' \in A_{h+1}^k} \left( \widetilde{\mathcal{P}}_{h+1,-2}^k(a'|s_{h+1}^k, A_{h+1}^k) f_{h+1,-2}^k(s_{h+1}^k, a') - \mathcal{P}_{h+1}(a'|s_{h+1}^k, A_{h+1}^k) \overline{Q}_{h+1}^{\pi_k}(s_{h+1}^k, a') \right)$$
$$- 2b_{h,2}^k(s_h^k, a_h^k) + \zeta_{h+1,-2}^k$$
$$= \sum_{a' \in A_{h+1}^k} \left( \widetilde{\mathcal{P}}_{h+1,-2}^k(a'|s_{h+1}^k, A_{h+1}^k) - \mathcal{P}_{h+1}(a'|s_{h+1}^k, A_{h+1}^k) \right) f_{h+1,-2}^k(s_{h+1}^k, a')$$
$$+ \mathbb{E}_{\mathcal{P}}\left[ \left( f_{h+1,-2}^k - \overline{Q}_{h+1}^{\pi_k} \right)(s_{h+1}^k, a_{h+1}) | s_{h+1}^k, A_{h+1}^k \right] - 2b_{h,2}^k(s_h^k, a_h^k) + \zeta_{h+1,-2}^k$$
$$= \sum_{a' \in A_{h+1}^k} \left( \widetilde{\mathcal{P}}_{h+1,-2}^k(a'|s_{h+1}^k, A_{h+1}^k) - \mathcal{P}_{h+1}(a'|s_{h+1}^k, A_{h+1}^k) \right) f_{h+1,-2}^k(s_{h+1}^k, a')$$
$$+ \left( f_{h+1,-2}^k - \overline{Q}_{h+1}^{\pi_k} \right)(s_{h+1}^k, a_{h+1}^k) - 2b_{h,2}^k(s_h^k, a_h^k) + \zeta_{h+1,-2}^k + \dot{\zeta}_{h+1,-2}^k,$$

where the first inequality holds because $\mathcal{T}_h V_{h+1,-2}^k \in \mathcal{F}_{h,-2}^k$ under the event $\mathcal{E}_{\leqslant K}$ and definition of $b_{h,2}^k$, and the last inequality holds since $V_{h+1,-2}^k(s_{h+1}^k) \geqslant Q_{h+1,-2}^k(s_{h+1}^k, A_{h+1}^k)$.

Hence, by recursion we obtain that

$$(f_{h,-2}^k - \overline{Q}_h^{\pi_k})(s_h^k, a_h^k) \geqslant \sum_{h'=h+1}^{H} \sum_{a' \in A_{h'}^k} \left( \widetilde{\mathcal{P}}_{h',-2}^k(a'|s_{h'}^k, A_{h'}^k) - \mathcal{P}_{h'}(a'|s_{h'}^k, A_{h'}^k) \right) f_{h',-2}^k(s_{h'}^k, a')$$

$$- 2 \sum_{h'=h}^{H} b_{h',2}^k(s_{h'}^k, a_{h'}^k) + \sum_{h'=h+1}^{H} \zeta_{h',-2}^k + \sum_{h'=h+1}^{H} \dot{\zeta}_{h',-2}^k.$$

$\square$

**Lemma D.13** (Approximation error of overly optimistic $\overline{Q}$)**.** *Suppose Algorithm 1 uses a consistent bonus oracle satisfying Definition 4. Conditioning on the good event $\mathcal{E}_{\leqslant K}$, for any $k \in [K]$ and any $h \geqslant h_k$, it holds that*

$$(f_{h,2}^k - \overline{Q}_h^{\pi_k})(s_h^k, a_h^k) \leqslant \sum_{h'=h+1}^{H} \sum_{a' \in A_{h'}^k} \left( \widetilde{\mathcal{P}}_{h',2}^k(a'|s_{h'}^k, A_{h'}^k) - \mathcal{P}_{h'}(a'|s_{h'}^k, A_{h'}^k) \right) f_{h',2}^k(s_{h'}^k, a')$$

$$+ 2 \sum_{h'=h}^{H} b_{h',1}^k(s_{h'}^k, a_{h'}^k) + 2 \sum_{h'=h}^{H} b_{h',2}^k(s_{h'}^k, a_{h'}^k) + \sum_{h'=h+1}^{H} \zeta_{h',2}^k + \sum_{h'=h+1}^{H} \dot{\zeta}_{h',2}^k,$$

where $\zeta_{h,2}^k := \mathbb{E}_{\mathbb{P}}\left[(V_{h,2}^k - V_h^{\pi_k})(s_h) \mid s_{h-1}^k, a_{h-1}^k\right] - (V_{h,2}^k - V_h^{\pi_k})(s_h^k)$ and $\dot{\zeta}_{h,2}^k :=$
$\mathbb{E}_{\mathcal{P}}\left[\left(f_{h,2}^k - \overline{Q}_h^{\pi_k}\right)(s_h^k, a_h) \mid s_h^k, A_h^k\right] - \left(f_{h,2}^k - \overline{Q}_h^{\pi_k}\right)(s_h^k, a_h^k)$.

*Proof of Lemma D.13.* Under the event $\mathcal{E}_{\leqslant K}$, at $h \geqslant h_k$, we have

$$(f_{h,2}^k - \overline{Q}_h^{\pi_k})(s_h^k, a_h^k) = (f_{h,2}^k - \mathcal{T}_h V_{h+1,2}^k)(s_h^k, a_h^k) + (\mathcal{T}_h V_{h+1,2}^k - \overline{Q}_h^{\pi_k})(s_h^k, a_h^k)$$
$$\leqslant 2b_{h,1}^k(s_h^k, a_h^k) + 2b_{h,2}^k(s_h^k, a_h^k) + \mathbb{E}\left[(V_{h+1,2}^k - V_{h+1}^{\pi_k})(s_{h+1})|s_h^k, a_h^k\right]$$
$$= 2b_{h,1}^k(s_h^k, a_h^k) + 2b_{h,2}^k(s_h^k, a_h^k) + (V_{h+1,2}^k - V_{h+1}^{\pi_k})(s_{h+1}^k) + \zeta_{h+1,2}^k$$
$$= 2b_{h,1}^k(s_h^k, a_h^k) + 2b_{h,2}^k(s_h^k, a_h^k) + (Q_{h+1,2}^k - Q_{h+1}^{\pi_k})(s_{h+1}^k, A_{h+1}^k) + \zeta_{h+1,2}^k$$
$$= \sum_{a' \in A_{h+1}^k} \left(\widetilde{\mathcal{P}}_{h+1,2}^k(a'|s_{h+1}^k, A_{h+1}^k) f_{h+1,2}^k(s_{h+1}^k, a') - \mathcal{P}_{h+1}(a'|s_{h+1}^k, A_{h+1}^k)\overline{Q}_{h+1}^{\pi_k}(s_{h+1}^k, a')\right)$$
$$+ 2b_{h,1}^k(s_h^k, a_h^k) + 2b_{h,2}^k(s_h^k, a_h^k) + \zeta_{h+1,2}^k$$
$$= \sum_{a' \in A_{h+1}^k} \left(\widetilde{\mathcal{P}}_{h+1,2}^k(a'|s_{h+1}^k, A_{h+1}^k) - \mathcal{P}_{h+1}(a'|s_{h+1}^k, A_{h+1}^k)\right) f_{h+1,2}^k(s_{h+1}^k, a')$$
$$+ \mathbb{E}_{\mathcal{P}}\left[\left(f_{h+1,2}^k - \overline{Q}_{h+1}^{\pi_k}\right)(s_{h+1}^k, a_{h+1})|s_{h+1}^k, A_{h+1}^k\right] + 2b_{h,1}^k(s_h^k, a_h^k) + 2b_{h,2}^k(s_h^k, a_h^k) + \zeta_{h+1,2}^k$$
$$= \sum_{a' \in A_{h+1}^k} \left(\widetilde{\mathcal{P}}_{h+1,2}^k(a'|s_{h+1}^k, A_{h+1}^k) - \mathcal{P}_{h+1}(a'|s_{h+1}^k, A_{h+1}^k)\right) f_{h+1,2}^k(s_{h+1}^k, a')$$
$$+ \left(f_{h+1,2}^k - \overline{Q}_{h+1}^{\pi_k}\right)(s_{h+1}^k, a_{h+1}^k) + 2b_{h,1}^k(s_h^k, a_h^k) + 2b_{h,2}^k(s_h^k, a_h^k) + \zeta_{h+1,2}^k + \dot{\zeta}_{h+1,2}^k,$$

where the first inequality holds based on the assumption that $\mathcal{T}_h V_{h+1,2}^k \in \mathcal{F}_{h,2}^k$ and definition of $b_{h,2}^k$, and the third equality holds because for $h \geqslant h_k$, we know that $A_{h+1}^k \in \arg\max_A Q_{h+1}^k(s_{h+1}^k, A) = \arg\max_A \sum_{a \in A} \widetilde{\mathcal{P}}_{h+1,2}^k(a|s_{h+1}^k, A) f_{h+1,2}^k(s_{h+1}^k, a)$ by the data collection policy in Equation 10.

Therefore, by recursion we get

$$(f_{h,2}^k - \overline{Q}_h^{\pi_k})(s_h^k, a_h^k) \leqslant \sum_{h'=h+1}^{H} \sum_{a' \in A_{h'}^k} \left(\widetilde{\mathcal{P}}_{h',2}^k(a'|s_{h'}^k, A_{h'}^k) - \mathcal{P}_{h'}(a'|s_{h'}^k, A_{h'}^k)\right) f_{h',2}^k(s_{h'}^k, a')$$
$$+ 2\sum_{h'=h}^{H} b_{h',1}^k(s_{h'}^k, a_{h'}^k) + 2\sum_{h'=h}^{H} b_{h',2}^k(s_{h'}^k, a_{h'}^k) + \sum_{h'=h+1}^{H} \zeta_{h',2}^k + \sum_{h'=h+1}^{H} \dot{\zeta}_{h',2}^k.$$

$\square$

**Lemma D.14** (Approximation error of optimistic $\overline{Q}$)**.** *Suppose Algorithm 1 uses a consistent bonus oracle satisfying Definition 4. Conditioning on the good event $\mathcal{E}^\theta \bigcap \mathcal{E}_{\leqslant K}$, for any $k \in [K]$ and any $h \leqslant h_k$, we have*

$$(f_{h,1}^k - \overline{Q}_h^{\pi_k})(s_h^k, a_h^k)$$
$$\leqslant \sum_{h'=h+1}^{h_k-1} \sum_{a' \in A_{h'}^k} \left(\widetilde{\mathcal{P}}_{h',1}^k(a'|s_{h'}^k, A_{h'}^k) - \mathcal{P}_{h'}(a'|s_{h'}^k, A_{h'}^k)\right) f_{h',1}^k(s_{h'}^k, a')$$
$$+ \sum_{h'=h_k}^{H} \sum_{a' \in A_{h'}^k} \left(\widetilde{\mathcal{P}}_{h',2}^k(a'|s_{h'}^k, A_{h'}^k) - \mathcal{P}_{h'}(a'|s_{h'}^k, A_{h'}^k)\right) f_{h',2}^k(s_{h'}^k, a')$$
$$+ 2\sum_{h'=h}^{H} b_{h',1}^k(s_{h'}^k, a_{h'}^k) + 2\sum_{h'=h_k}^{H} b_{h',2}^k(s_{h'}^k, a_{h'}^k) + \sum_{h'=h+1}^{h_k-1} (\zeta_{h',1}^k + \dot{\zeta}_{h',1}^k) + \sum_{h'=h_k}^{H} (\zeta_{h',2}^k + \dot{\zeta}_{h',2}^k),$$

*where*

$$\zeta_{h,1}^k := \mathbb{E}_{\mathbb{P}}\left[(V_{h,1}^k - V_h^{\pi_k})(s_h) \mid s_{h-1}^k, a_{h-1}^k\right] - (V_{h,1}^k - V_h^{\pi_k})(s_h^k),$$

$$\dot{\zeta}_{h,1}^k := \mathbb{E}_{\mathcal{P}}\left[\left(f_{h,1}^k - \overline{Q}_h^{\pi_k}\right)(s_h^k, a_h) \mid s_h^k, A_h^k\right] - \left(f_{h,1}^k - \overline{Q}_h^{\pi_k}\right)(s_h^k, a_h^k),$$

$$\zeta_{h,2}^k := \mathbb{E}_{\mathbb{P}}\left[(V_{h,2}^k - V_h^{\pi_k})(s_h) \mid s_{h-1}^k, a_{h-1}^k\right] - (V_{h,2}^k - V_h^{\pi_k})(s_h^k),$$

$$\dot{\zeta}_{h,2}^k := \mathbb{E}_{\mathcal{P}}\left[\left(f_{h,2}^k - \overline{Q}_h^{\pi_k}\right)(s_h^k, a_h) \mid s_h^k, A_h^k\right] - \left(f_{h,2}^k - \overline{Q}_h^{\pi_k}\right)(s_h^k, a_h^k).$$

*Here, following the convention, we use the empty sum notation, i.e., $\sum_{i=a}^b x_i = 0$, when $b \leqslant a$.*

*Proof of Lemma D.14.* Under the event $\mathcal{E}_{\leqslant K}$, by Lemma D.8 it holds that $f_{h,1}^k(s,a) \leqslant f_{h,2}^k(s,a)$ for all $(s,a) \in \mathcal{S} \times \mathcal{I}$. Therefore, for $h = h_k$, by Lemma D.13, we get

$$(f_{h,1}^k - \overline{Q}_h^{\pi_k})(s_h^k, a_h^k) \leqslant (f_{h,2}^k - \overline{Q}_h^{\pi_k})(s_h^k, a_h^k)$$

$$\leqslant \sum_{h'=h+1}^H \sum_{a' \in A_{h'}^k} \left(\widetilde{\mathcal{P}}_{h',2}^k(a'|s_{h'}^k, A_{h'}^k) - \mathcal{P}_{h'}(a'|s_{h'}^k, A_{h'}^k)\right) f_{h',2}^k(s_{h'}^k, a')$$

$$+ 2\sum_{h'=h}^H b_{h',1}^k(s_{h'}^k, a_{h'}^k) + 2\sum_{h'=h}^H b_{h',2}^k(s_{h'}^k, a_{h'}^k) + \sum_{h'=h+1}^H \zeta_{h',2}^k + \sum_{h'=h+1}^H \dot{\zeta}_{h',2}^k. \tag{D.44}$$

For $h = h_k - 1$, we have

$$(f_{h,1}^k - \overline{Q}_h^{\pi_k})(s_h^k, a_h^k) = (f_{h,1}^k - \mathcal{T}_h V_{h+1,1}^k)(s_h^k, a_h^k) + (\mathcal{T}_h V_{h+1,1}^k - \overline{Q}_h^{\pi_k})(s_h^k, a_h^k)$$

$$\leqslant 2b_{h,1}^k(s_h^k, a_h^k) + \mathbb{E}\left[(V_{h+1,1}^k - V_h^{\pi_k})(s_{h+1})|s_h^k, a_h^k\right]$$

$$\leqslant 2b_{h,1}^k(s_h^k, a_h^k) + \mathbb{E}\left[(V_{h+1,2}^k - V_h^{\pi_k})(s_{h+1})|s_h^k, a_h^k\right]$$

$$= 2b_{h,1}^k(s_h^k, a_h^k) + (V_{h+1,2}^k - V_h^{\pi_k})(s_{h+1}^k) + \zeta_{h+1,2}^k$$

$$= 2b_{h,1}^k(s_h^k, a_h^k) + (Q_{h+1,2}^k - Q_h^{\pi_k})(s_{h+1}^k, A_{h+1}^k) + \zeta_{h+1,2}^k$$

$$\leqslant \sum_{a' \in A_{h+1}^k} \left(\widetilde{\mathcal{P}}_{h+1,2}^k(a'|s_{h+1}^k, A_{h+1}^k) f_{h+1,2}^k(s_{h+1}^k, a') - \mathcal{P}_{h+1}(a'|s_{h+1}^k, A_{h+1}^k)\overline{Q}_{h+1}^{\pi_k}(s_{h+1}^k, a')\right)$$

$$+ 2b_{h,1}^k(s_h^k, a_h^k) + \zeta_{h+1,2}^k$$

$$= \sum_{a' \in A_{h+1}^k} \left(\widetilde{\mathcal{P}}_{h+1,2}^k(a'|s_{h+1}^k, A_{h+1}^k) - \mathcal{P}_{h+1}(a'|s_{h+1}^k, A_{h+1}^k)\right) f_{h+1,2}^k(s_{h+1}^k, a')$$

$$+ \mathbb{E}_{\mathcal{P}}\left[\left(f_{h+1,2}^k - \overline{Q}_{h+1}^{\pi_k}\right)(s_{h+1}^k, a_{h+1})|s_{h+1}^k, A_{h+1}^k\right] + 2b_{h,1}^k(s_h^k, a_h^k) + \zeta_{h+1,2}^k$$

$$= \sum_{a' \in A_{h+1}^k} \left(\widetilde{\mathcal{P}}_{h+1,2}^k(a'|s_{h+1}^k, A_{h+1}^k) - \mathcal{P}_{h+1}(a'|s_{h+1}^k, A_{h+1}^k)\right) f_{h+1,2}^k(s_{h+1}^k, a')$$

$$+ \left(f_{h+1,2}^k - \overline{Q}_{h+1}^{\pi_k}\right)(s_{h+1}^k, a_{h+1}^k) + 2b_{h,1}^k(s_h^k, a_h^k) + \zeta_{h+1,2}^k + \dot{\zeta}_{h+1,2}^k, \tag{D.45}$$

where the first inequality holds based on the assumption that $\mathcal{T}_h V_{h+1,1}^k \in \mathcal{F}_{h,1}^k$ and definition of $b_{h,1}^k$, and the second inequality holds because for any $s_{h+1} \in \mathcal{S}$, we have

$$V_{h+1,1}^k(s_{h+1}) = \sum_{a' \in A_{h+1,1}} \widetilde{\mathcal{P}}_{h+1,1}^k(a'|s_{h+1}, A_{h+1,1}) f_{h+1,1}^k(s_{h+1}, a')$$

$$\leqslant \sum_{a' \in A_{h+1,1}} \widetilde{\mathcal{P}}_{h+1,1}^k(a'|s_{h+1}, A_{h+1,1}) f_{h+1,2}^k(s_{h+1}, a')$$

$$\leqslant \sum_{a' \in \tilde{A}_{h+1,1}} \widetilde{\mathcal{P}}_{h+1,2}^k(a'|s_{h+1}, \tilde{A}_{h+1,1}) f_{h+1,2}^k(s_{h+1}, a')$$

$$\leqslant \sum_{a' \in A_{h+1,2}^k} \widetilde{\mathcal{P}}_{h+1,2}^k(a'|s_{h+1}, A_{h+1,2}^k) f_{h+1,2}^k(s_{h+1}, a') = V_{h+1,2}^k(s_{h+1}),$$

where in the first equality, we denote $A_{h+1,1} \in \arg\max_A \sum_{a' \in A} \widetilde{\mathcal{P}}_{h+1,1}^k(a'|s_{h+1}, A) f_{h+1,1}^k(s_{h+1}, a')$, for the first inequality, we use the fact that $f_{h+1,1}^k(s,a) \leqslant f_{h+1,2}^k(s,a)$ (Lemma D.8), the second inequality holds for some $\tilde{A}_{h+1,1} \subseteq A_{h+1,1}$ (Lemma D.2), and the last inequality follows from the definition of $A_{h+1,2}^k$. Moreover, the third equality of Equation D.45 holds because for $h + 1 = h_k$, we know that $A_{h+1}^k = A_{h+1,2}^k$ by the data collection policy in Equation 10.

Therefore, by recursion, we have

$$
(f_{h,1}^k - \overline{Q}_h^{\pi_k})(s_h^k, a_h^k)
$$
$$
\leqslant \sum_{h'=h+1}^{H} \sum_{a' \in A_{h'}^k} \left( \widetilde{\mathcal{P}}_{h',2}^k(a'|s_{h'}^k, A_{h'}^k) - \mathcal{P}_{h'}(a'|s_{h'}^k, A_{h'}^k) \right) f_{h',2}^k(s_{h'}^k, a')
$$
$$
+ 2 \sum_{h'=h}^{H} b_{h',1}^k(s_{h'}^k, a_{h'}^k) + 2 \sum_{h'=h+1}^{H} b_{h',2}^k(s_{h'}^k, a_{h'}^k) + \sum_{h'=h+1}^{H} \zeta_{h',2}^k + \sum_{h'=h+1}^{H} \dot{\zeta}_{h',2}^k \quad \text{(D.46)}
$$

Finally, we consider the case where $h < h_k - 1$.

$$
(f_{h,1}^k - \overline{Q}_h^{\pi_k})(s_h^k, a_h^k) = (f_{h,1}^k - \mathcal{T}_h V_{h+1,1}^k)(s_h^k, a_h^k) + (\mathcal{T}_h V_{h+1,1}^k - \overline{Q}_h^{\pi_k})(s_h^k, a_h^k)
$$
$$
\leqslant 2 b_{h,1}^k(s_h^k, a_h^k) + \mathbb{E}\left[ (V_{h+1,1}^k - V_{h+1}^{\pi_k})(s_{h+1}) | s_h^k, a_h^k \right]
$$
$$
= 2 b_{h,1}^k(s_h^k, a_h^k) + (V_{h+1,1}^k - V_{h+1}^{\pi_k})(s_{h+1}^k) + \zeta_{h+1,1}^k
$$
$$
= 2 b_{h,1}^k(s_h^k, a_h^k) + (Q_{h+1,1}^k - Q_{h+1}^{\pi_k})(s_{h+1}^k, A_{h+1}^k) + \zeta_{h+1,1}^k
$$
$$
= \sum_{a' \in A_{h+1}^k} \left( \widetilde{\mathcal{P}}_{h+1,1}^k(a'|s_{h+1}^k, A_{h+1}^k) f_{h+1,1}^k(s_{h+1}^k, a') - \mathcal{P}_{h+1}(a'|s_{h+1}^k, A_{h+1}^k) \overline{Q}_{h+1}^{\pi_k}(s_{h+1}^k, a') \right)
$$
$$
+ 2 b_{h,1}^k(s_h^k, a_h^k) + \zeta_{h+1,1}^k
$$
$$
= \sum_{a' \in A_{h+1}^k} \left( \widetilde{\mathcal{P}}_{h+1,1}^k(a'|s_{h+1}^k, A_{h+1}^k) - \mathcal{P}_{h+1}(a'|s_{h+1}^k, A_{h+1}^k) \right) f_{h+1,1}^k(s_{h+1}^k, a')
$$
$$
+ \mathbb{E}_{\mathcal{P}}\left[ \left( f_{h+1,1}^k - \overline{Q}_{h+1}^{\pi_k} \right)(s_{h+1}^k, a_{h+1})|s_{h+1}^k, A_{h+1}^k \right] + 2 b_{h,1}^k(s_h^k, a_h^k) + \zeta_{h+1,1}^k
$$
$$
= \sum_{a' \in A_{h+1}^k} \left( \widetilde{\mathcal{P}}_{h+1,1}^k(a'|s_{h+1}^k, A_{h+1}^k) - \mathcal{P}_{h+1}(a'|s_{h+1}^k, A_{h+1}^k) \right) f_{h+1,1}^k(s_{h+1}^k, a')
$$
$$
+ \left( f_{h+1,1}^k - \overline{Q}_{h+1}^{\pi_k} \right)(s_{h+1}^k, a_{h+1}^k) + 2 b_{h,1}^k(s_h^k, a_h^k) + \zeta_{h+1,1}^k + \dot{\zeta}_{h+1,1}^k,
$$

where the first inequality holds based on the assumption that $\mathcal{T}_h V_{h+1,1}^k \in \mathcal{F}_{h,1}^k$ and definition of $b_{h,1}^k$ and the third equality holds because for $h + 1 < h_k$, we have $A_{h+1}^k = A_{h+1,1}^k$ by the data collection policy in Equation 10.

Hence, by recursion we have

$$
(f_{h,1}^k - \overline{Q}_h^{\pi_k})(s_h^k, a_h^k) \leqslant \sum_{h'=h+1}^{h_k-1} \sum_{a' \in A_{h'}^k} \left( \widetilde{\mathcal{P}}_{h',1}^k(a'|s_{h'}^k, A_{h'}^k) - \mathcal{P}_{h'}(a'|s_{h'}^k, A_{h'}^k) \right) f_{h',1}^k(s_{h'}^k, a')
$$
$$
+ \sum_{h'=h_k}^{H} \sum_{a' \in A_{h'}^k} \left( \widetilde{\mathcal{P}}_{h',2}^k(a'|s_{h'}^k, A_{h'}^k) - \mathcal{P}_{h'}(a'|s_{h'}^k, A_{h'}^k) \right) f_{h',2}^k(s_{h'}^k, a')
$$
$$
+ 2 \sum_{h'=h}^{H} b_{h',1}^k(s_{h'}^k, a_{h'}^k) + 2 \sum_{h'=h_k}^{H} b_{h',2}^k(s_{h'}^k, a_{h'}^k) + \sum_{h'=h+1}^{h_k-1} (\zeta_{h',1}^k + \dot{\zeta}_{h',1}^k)
$$
$$
+ \sum_{h'=h_k}^{H} (\zeta_{h',2}^k + \dot{\zeta}_{h',2}^k). \quad \text{(D.47)}
$$

Combining Equation D.44, Equation D.46, and Equation D.47, we conclude the proof. $\qquad\square$

### D.7 BOUNDS ON BONUSES AND $|\mathcal{K}_{oo}|$

In this subsection, we provide proofs for the bounds on the sum of bonuses (Lemma D.15, Lemma D.16, and Lemma D.17) as well as the bound on the size of $\mathcal{K}_{oo}$ (Lemma D.18).

**Lemma D.15** (Crude bound on $b_{h,1}^k$, Lemma 39 of Agarwal et al. 2023). *Let $z_h^k = (s_h^k, a_h^k)$. Given*

$$b_{h,1}^k(\cdot) \leqslant C \cdot \left( D_{\mathcal{F}_h}\left(\cdot; \{z_h^\tau\}_{\tau=1}^{k-1}, \{\bar{\sigma}_h^\tau\}_{\tau=1}^{k-1}\right) \cdot \sqrt{\left(\beta_{h,1}^k\right)^2 + \rho} + \epsilon_b \beta_{h,1}^k \right), \text{ when } \rho = 1, \nu \leqslant 1, \text{ it holds}$$

*that for any subset $\mathcal{K} \in [K]$, we have*

$$\sum_{k \in \mathcal{K}} \sum_{h=1}^H \min\left\{1 + L, b_{h,1}^k(z_h^k)\right\}$$

$$= \mathcal{O}\left(\sqrt{\log \frac{\mathcal{N}KH}{\nu\delta}} \cdot \left(\sqrt{\log \frac{\mathcal{N}\mathcal{N}_b KH}{\nu\delta}} \cdot H\sqrt{d_\nu|\mathcal{K}|} + \log \frac{\mathcal{N}\mathcal{N}_b KH}{\nu\delta} \cdot d_\nu H + |\mathcal{K}|H\epsilon_b\right)\right).$$

**Lemma D.16** (Crude bound on $b_{h,2}^k$, Lemma 38 of Agarwal et al. 2023). *Let $z_h^k = (s_h^k, a_h^k)$. Given*

$$b_{h,2}^k(\cdot) \leqslant C \cdot \left( D_{\mathcal{F}_h}\left(\cdot; \{z_h^\tau\}_{\tau=1}^{k-1}, \{\mathbf{1}^\tau\}_{\tau=1}^{k-1}\right) \cdot \sqrt{\left(\beta_{h,2}^k\right)^2 + \rho} + \epsilon_b \beta_{h,1}^k \right), \text{ when } \rho = 1, \nu \leqslant 1, \text{ it holds}$$

*that for any subset $\mathcal{K} \in [K]$, we have*

$$\sum_{k \in \mathcal{K}} \sum_{h=1}^H \min\left\{1 + L, b_{h,2}^k(z_h^k)\right\} = \mathcal{O}\left(\sqrt{\log \frac{\mathcal{N}\mathcal{N}_b KH}{\delta}} \cdot \left(H\sqrt{d_\nu|\mathcal{K}|} + d_\nu H + |\mathcal{K}|H\epsilon_b\right)\right).$$

**Lemma D.17** (Fine-grained bound on $b_{h,1}^k$). *Let $z_h^k = (s_h^k, a_h^k)$. Recall that the bonus oracle $\mathcal{B}$ outputs*

*a bonus function such that $b_{h,1}^k(\cdot) \leqslant C \cdot \left( D_{\mathcal{F}_h}\left(\cdot; \{z_h^\tau\}_{\tau=1}^{k-1}, \{\bar{\sigma}_h^\tau\}_{\tau=1}^{k-1}\right) \cdot \sqrt{\left(\beta_{h,1}^k\right)^2 + \rho} + \epsilon_b \beta_{h,1}^k \right)$.*

*When $\rho = 1, \nu = 1/\sqrt{KH}$, $\delta < (0, 1/7)$ and the event $\mathcal{E}_{\leqslant K}$ holds, with probability at least $1 - 7\delta$, we have*

$$\sum_{k=1}^K \sum_{h=1}^H \min\left\{1 + L, b_{h,1}^k(z_h^k)\right\}$$

$$= \mathcal{O}\left(\sqrt{d_\nu HK \cdot \log \frac{\mathcal{N}KH}{\delta}} + \frac{1}{\sqrt{\kappa}}dH^{7/2}\sqrt{d_\nu}\log K \log M \cdot \log \frac{\mathcal{N}\mathcal{N}_b KH}{\delta} \cdot \sqrt{\log \frac{\mathcal{N}KH}{\delta}}\right)$$

$$+ \mathcal{O}\left(d_\nu H^{7/2}\log \frac{\mathcal{N}KH}{\delta} \cdot \left(\log \frac{\mathcal{N}\mathcal{N}_b KH}{\delta}\right)^{3/2} + \sqrt{\log \frac{\mathcal{N}KH}{\delta}} \cdot \left(KH\epsilon_b + \sqrt{d_\nu KH^3\delta}\right)\right)$$

$$+ \mathcal{O}\left(\sqrt{\log \frac{\mathcal{N}KH}{\delta} \log \frac{\mathcal{N}\mathcal{N}_b KH}{\delta}} \cdot \sqrt{d_\nu H} \cdot \left(\sqrt{H^2 \sum_{k \in \mathcal{K}_o} u_k} + \sqrt{H^2|\mathcal{K}_{oo}|}\right)\right).$$

*Proof of Lemma D.17.* By the definition of the oracle $\mathcal{B}$ (Definition 4), we have

$$\sum_{k=1}^K \sum_{h=1}^H \min\left\{1 + L, b_{h,1}^k(z_h^k)\right\}$$

$$= \mathcal{O}\left(\sum_{k=1}^K \sum_{h=1}^H \min\left\{1, D_{\mathcal{F}_h}\left(z_h^k; \{z_h^\tau\}_{\tau=1}^{k-1}, \{\bar{\sigma}\}_{\tau=1}^{k-1}\right) \cdot \sqrt{\left(\beta_{h,1}^k\right)^2 + \rho}\right\} + KH\epsilon_b \cdot \max_{k,h} \beta_{h,1}^k\right)$$

$$= \mathcal{O}\left(\sqrt{\log \frac{\mathcal{N}KH}{\delta}} \cdot \sum_{k=1}^K \sum_{h=1}^H \min\left\{1, D_{\mathcal{F}_h}\left(z_h^k; \{z_h^\tau\}_{\tau=1}^{k-1}, \{\bar{\sigma}\}_{\tau=1}^{k-1}\right)\right\} + KH\epsilon_b\right), \qquad \text{(D.48)}$$

where the last equality holds by the definition of $\beta_{h,1}^k$.

Now, we bound the summation terms

$$\sum_{k=1}^{K}\sum_{h=1}^{H}\min\left\{1, D_{\mathcal{F}_h}\left(z_h^k; \{z_h^\tau\}_{\tau=1}^{k-1}, \{\bar\sigma\}_{\tau=1}^{k-1}\right)\right\}$$

$$=\sum_{k=1}^{K}\sum_{h=1}^{H}\min\left\{1, \bar\sigma_h^k\cdot\left(\bar\sigma_h^k\right)^{-1}D_{\mathcal{F}_h}\left(z_h^k; \{z_h^\tau\}_{\tau=1}^{k-1}, \{\bar\sigma\}_{\tau=1}^{k-1}\right)\right\}$$

by dividing into the following cases:

$$\mathcal{I}_1 = \left\{(k,h)\in[K]\times[H] : \left(\bar\sigma_h^k\right)^{-1}D_{\mathcal{F}_h}\left(z_h^k; \{z_h^\tau\}_{\tau=1}^{k-1}, \{\bar\sigma\}_{\tau=1}^{k-1}\right)\geq 1\right\},$$

$$\mathcal{I}_2 = \left\{(k,h)\in[K]\times[H] : \bar\sigma_h^k = \nu, (k,h)\neq\mathcal{I}_1\right\},$$

$$\mathcal{I}_3 = \left\{(k,h)\in[K]\times[H] : \bar\sigma_h^k = 2\left(\sqrt{o(\delta_h^k)}+\iota(\delta_h^k)\right)\cdot\sqrt{D_{\mathcal{F}_h}\left(z_h^k; \{z_h^\tau\}_{\tau=1}^{k-1}, \{\bar\sigma_h^\tau\}_{\tau=1}^{k-1}\right)},\right.$$
$$\left.(k,h)\neq\mathcal{I}_1\right\},$$

$$\mathcal{I}_4 = \left\{(k,h)\in[K]\times[H] : \bar\sigma_h^k = \sqrt{2}\iota(\delta_h^k)\sqrt{f_{h,2}^k(z_h^k) - f_{h,-2}^k(z_h^k)}, (k,h)\neq\mathcal{I}_1\right\}$$

$$\mathcal{I}_5 = \left\{(k,h)\in[K]\times[H] : \bar\sigma_h^k = \sigma_h^k, (k,h)\neq\mathcal{I}_1\right\},.$$

For the case of $\mathcal{I}_1$, we have

$$\sum_{(k,h)\in\mathcal{I}_1}\min\left\{1, \bar\sigma_h^k\cdot\left(\bar\sigma_h^k\right)^{-1}D_{\mathcal{F}_h}\left(z_h^k; \{z_h^\tau\}_{\tau=1}^{k-1}, \{\bar\sigma\}_{\tau=1}^{k-1}\right)\right\}$$

$$\leq\sum_{(k,h)\in\mathcal{I}_1}\left(\bar\sigma_h^k\right)^{-1}D_{\mathcal{F}_h}\left(z_h^k; \{z_h^\tau\}_{\tau=1}^{k-1}, \{\bar\sigma\}_{\tau=1}^{k-1}\right)\leq\sum_{h=1}^{H}\dim_{\nu,K}(\mathcal{F}_h) = d_\nu H. \quad\text{(D.49)}$$

For $\mathcal{I}_2$, we use the Cauchy-Schwarz inequality to get

$$\sum_{(k,h)\in\mathcal{I}_2}\min\left\{1, \bar\sigma_h^k\cdot\left(\bar\sigma_h^k\right)^{-1}D_{\mathcal{F}_h}\left(z_h^k; \{z_h^\tau\}_{\tau=1}^{k-1}, \{\bar\sigma\}_{\tau=1}^{k-1}\right)\right\}$$

$$\leq\sqrt{\nu^2 KH}\cdot\sqrt{\sum_{(k,h)\in\mathcal{I}_2}\left(\bar\sigma_h^k\right)^{-2}D_{\mathcal{F}_h}^2\left(z_h^k; \{z_h^\tau\}_{\tau=1}^{k-1}, \{\bar\sigma\}_{\tau=1}^{k-1}\right)}\leq\sqrt{\sum_{h=1}^{H}\dim_{\nu,K}(\mathcal{F}_h)} = \sqrt{d_\nu H}.$$

$$\text{(D.50)}$$

For $\mathcal{I}_3$, we have

$$\sum_{(k,h)\in\mathcal{I}_3}\min\left\{1, \bar\sigma_h^k\cdot\left(\bar\sigma_h^k\right)^{-1}D_{\mathcal{F}_h}\left(z_h^k; \{z_h^\tau\}_{\tau=1}^{k-1}, \{\bar\sigma\}_{\tau=1}^{k-1}\right)\right\}$$

$$\leq\sum_{(k,h)\in\mathcal{I}_3}\left(8o(\delta_h^k) + \iota^2(\delta_h^k)\right)\cdot\min\left\{1, \left(\bar\sigma_h^k\right)^{-2}D_{\mathcal{F}_h}^2\left(z_h^k; \{z_h^\tau\}_{\tau=1}^{k-1}, \{\bar\sigma\}_{\tau=1}^{k-1}\right)\right\}$$

$$=\mathcal{O}\left(\left(\sqrt{\log\frac{\mathcal{N}KH}{\delta}} + \log\frac{\mathcal{N}\mathcal{N}_b KH}{\delta}\right)\cdot\sum_{h=1}^{H}\dim_{\nu,K}(\mathcal{F}_h)\right)$$

$$=\mathcal{O}\left(\left(\sqrt{\log\frac{\mathcal{N}KH}{\delta}} + \log\frac{\mathcal{N}\mathcal{N}_b KH}{\delta}\right)d_\nu H\right), \quad\text{(D.51)}$$

where the inequality holds because, by dividing both sides of $\bar\sigma_h^k = 2\left(\sqrt{o(\delta_h^k)}+\iota(\delta_h^k)\right)\cdot$ $\sqrt{D_{\mathcal{F}_h}\left(z_h^k; \{z_h^\tau\}_{\tau=1}^{k-1}, \{\bar\sigma_h^\tau\}_{\tau=1}^{k-1}\right)}$ by $\sqrt{\bar\sigma_h^k}$ and rearranging terms, we get:

$$\bar\sigma_h^k \leq \left(8o(\delta_h^k) + \iota^2(\delta_h^k)\right)D_{\mathcal{F}_h}\left(z_h^k; \{z_h^\tau\}_{\tau=1}^{k-1}, \{\bar\sigma\}_{\tau=1}^{k-1}\right).$$

We also use the property that $\left(\bar\sigma_h^k\right)^{-1}D_{\mathcal{F}_h}\left(z_h^k; \{z_h^\tau\}_{\tau=1}^{k-1}, \{\bar\sigma\}_{\tau=1}^{k-1}\right)\leq 1$ for $(k,h)\in\mathcal{I}_3$, which follows directly from the definition of $\mathcal{I}_3$.

For $\mathcal{I}_4$, we have

$$\sum_{(k,h)\in\mathcal{I}_4} \min\left\{1, \bar{\sigma}_h^k \cdot \left(\bar{\sigma}_h^k\right)^{-1} D_{\mathcal{F}_h}\left(z_h^k; \{z_h^\tau\}_{\tau=1}^{k-1}, \{\bar{\sigma}\}_{\tau=1}^{k-1}\right)\right\}$$

$$\leqslant \sum_{(k,h)\in\mathcal{I}_4} \bar{\sigma}_h^k \cdot \left(\bar{\sigma}_h^k\right)^{-1} D_{\mathcal{F}_h}\left(z_h^k; \{z_h^\tau\}_{\tau=1}^{k-1}, \{\bar{\sigma}\}_{\tau=1}^{k-1}\right)$$

$$= \sum_{(k,h)\in\mathcal{I}_4} \sqrt{2}\iota(\delta_h^k)\sqrt{f_{h,2}^k(z_h^k) - f_{h,-2}^k(z_h^k)} \cdot \left(\bar{\sigma}_h^k\right)^{-1} D_{\mathcal{F}_h}\left(z_h^k; \{z_h^\tau\}_{\tau=1}^{k-1}, \{\bar{\sigma}\}_{\tau=1}^{k-1}\right)$$

$$\leqslant \mathcal{O}\left(\sqrt{\log\frac{\mathcal{N}\mathcal{N}_b KH}{\delta}}\sqrt{\sum_{k=1}^{K}\sum_{h=1}^{H} f_{h,2}^k(z_h^k) - f_{h,-2}^k(z_h^k)} \cdot \sqrt{d_\nu H}\right), \tag{D.52}$$

where the last inequality holds by the Cauchy-Schwarz inequality together with the definition of $\iota(\delta_h^k)$.

Lastly, restricting on $\mathcal{I}_5$, if the event $\mathcal{E}_{\leqslant K}$ holds, we have

$$\sum_{(k,h)\in\mathcal{I}_5} \min\left\{1, \bar{\sigma}_h^k \cdot \left(\bar{\sigma}_h^k\right)^{-1} D_{\mathcal{F}_h}\left(z_h^k; \{z_h^\tau\}_{\tau=1}^{k-1}, \{\bar{\sigma}\}_{\tau=1}^{k-1}\right)\right\}$$

$$\leqslant \sum_{(k,h)\in\mathcal{I}_5} \bar{\sigma}_h^k \cdot \left(\bar{\sigma}_h^k\right)^{-1} D_{\mathcal{F}_h}\left(z_h^k; \{z_h^\tau\}_{\tau=1}^{k-1}, \{\bar{\sigma}\}_{\tau=1}^{k-1}\right)$$

$$\leqslant \sqrt{\sum_{(k,h)\in\mathcal{I}_5} \left(\sigma_h^k\right)^2} \cdot \sqrt{\sum_{(k,h)\in\mathcal{I}_5} \left(\bar{\sigma}_h^k\right)^{-2} D_{\mathcal{F}_h}^2\left(z_h^k; \{z_h^\tau\}_{\tau=1}^{k-1}, \{\bar{\sigma}\}_{\tau=1}^{k-1}\right)}$$

$$\leqslant \mathcal{O}\left(\sqrt{\sum_{k,h} \mathbb{V}\left[r_h + V_{h+1,1}^k(s_{h+1}) \mid z_h^k\right] + \sum_{k,h}\left(f_{h,2}^k(z_h^k) - f_{h,-2}^k(z_h^k)\right)} \cdot \sqrt{d_\nu H}\right)$$

$$+ \mathcal{O}\left(\sqrt{\sum_{k,h}\min\left\{1, D_{\mathcal{F}_h}\left(z_h^k; \{z_h^\tau\}_{\tau=1}^{k-1}, \{\mathbf{1}^\tau\}_{\tau=1}^{k-1}\right)\right\}}\sqrt{\log\frac{\mathcal{N}\mathcal{N}_b KH}{\delta}} \cdot \sqrt{d_\nu H}\right), \tag{D.53}$$

where the second inequality holds by the Cauchy-Schwarz inequality, the last inequality holds by Lemma D.10 and the definition of Eluder dimension.

To further bound the first term on the right-hand side of Equation D.53, we apply Lemma D.11. Therefore, with probability at least $1 - \delta$, we have

$$\sum_{(k,h)\in\mathcal{I}_5} \min\left\{1, \bar{\sigma}_h^k \cdot \left(\bar{\sigma}_h^k\right)^{-1} D_{\mathcal{F}_h}\left(z_h^k; \{z_h^\tau\}_{\tau=1}^{k-1}, \{\bar{\sigma}\}_{\tau=1}^{k-1}\right)\right\}$$

$$\leqslant \mathcal{O}\left(\sqrt{H\sum_{k,h}\left(f_{h,2}^k(z_h^k) - f_{h,-2}^k(z_h^k)\right) + K + KH^2\delta + H^2|\mathcal{K}_{\text{oo}}| + H^4\log^2\frac{KH}{\delta}} \cdot \sqrt{d_\nu H}\right)$$

$$+ \mathcal{O}\left(\sqrt{\sum_{k,h}\left(f_{h,2}^k(z_h^k) - f_{h,-2}^k(z_h^k)\right)} \cdot \sqrt{d_\nu H}\right) + \mathcal{O}\left(\sqrt{\left(d_\nu H + H\sqrt{d_\nu K}\right)\sqrt{\log\frac{\mathcal{N}\mathcal{N}_b KH}{\delta}}} \cdot \sqrt{d_\nu H}\right)$$

$$\leqslant \mathcal{O}\left(\sqrt{K + KH^2\delta + H^2|\mathcal{K}_{\text{oo}}| + H^4\log^2\frac{KH}{\delta}} \cdot \sqrt{d_\nu H} + \sqrt{H\sum_{k,h}\left(f_{h,2}^k(z_h^k) - f_{h,-2}^k(z_h^k)\right)} \cdot \sqrt{d_\nu H}\right)$$

$$+ \mathcal{O}\left(d_\nu H^{1.5}\sqrt{\log\frac{\mathcal{N}\mathcal{N}_b KH}{\delta}}\right), \tag{D.54}$$

where the first inequality holds by the fact that

$$\sum_{k,h} \min\left\{1, D_{\mathcal{F}_h}\left(z_h^k; \{z_h^\tau\}_{\tau=1}^{k-1}, \{\mathbf{1}^\tau\}_{\tau=1}^{k-1}\right)\right\} \leq d_\nu H$$

$$\sum_{k,h} \min\left\{1, D_{\mathcal{F}_h}\left(z_h^k; \{z_h^\tau\}_{\tau=1}^{k-1}, \{\mathbf{1}^\tau\}_{\tau=1}^{k-1}\right)\right\} \leq \sqrt{KH}\sqrt{\sum_{k,h} D_{\mathcal{F}_h}^2\left(z_h^k; \{z_h^\tau\}_{\tau=1}^{k-1}, \{\mathbf{1}^\tau\}_{\tau=1}^{k-1}\right)} \leq H\sqrt{d_\nu K},$$

and the last inequality holds by the AM-GM inequality such that

$$H\sqrt{K \cdot d_\nu \log\frac{\mathcal{N}\mathcal{N}_b KH}{\delta}} \leq K + H^2 d_\nu \log\frac{\mathcal{N}\mathcal{N}_b KH}{\delta}.$$

Combining Equation D.49, D.50, D.51, D.52, and D.54, we get

$$\sum_{k=1}^K \sum_{h=1}^H \min\left\{1, \bar{\sigma}_h^k \cdot \left(\bar{\sigma}_h^k\right)^{-1} D_{\mathcal{F}_h}\left(z_h^k; \{z_h^\tau\}_{\tau=1}^{k-1}, \{\bar{\sigma}\}_{\tau=1}^{k-1}\right)\right\}$$

$$\leq \mathcal{O}\left(\sqrt{\log\frac{\mathcal{N}\mathcal{N}_b KH}{\delta}} \cdot \sqrt{d_\nu H} \cdot \sqrt{H\sum_{k=1}^K \sum_{h=1}^H f_{h,2}^k(z_h^k) - f_{h,-2}^k(z_h^k)}\right)$$

$$+ \mathcal{O}\left(\sqrt{K + KH^2\delta + H^2|\mathcal{K}_{oo}| + H^4\log^2\frac{KH}{\delta}} \cdot \sqrt{d_\nu H} + d_\nu H^{1.5}\sqrt{\log\frac{\mathcal{N}\mathcal{N}_b KH}{\delta}}\right). \quad \text{(D.55)}$$

Now, we bound the term $\sum_{k,h}\left(f_{h,2}^k(z_h^k) - f_{h,-2}^k(z_h^k)\right)$. For $k \in \mathcal{K}_{oo}$, we have

$$\sum_{k\in\mathcal{K}_{oo}} \sum_{h=1}^H \left(f_{h,2}^k(z_h^k) - f_{h,-2}^k(z_h^k)\right) = \mathcal{O}(|\mathcal{K}_{oo}|H).$$

Otherwise, for episodes $k \in \mathcal{K}_o$, we know that it holds true that $f_{h,1}^k(z_h^k) \geq f_{h,2}^k(z_h^k) - u_k$ by Equation 10. Therefore, under the event $\mathcal{E}_{\leq K}$, we have

$$\sum_{k\in\mathcal{K}_o} \sum_{h=1}^H \left(f_{h,2}^k(z_h^k) - f_{h,-2}^k(z_h^k)\right) \leq \sum_{k\in\mathcal{K}_o} \sum_{h=1}^H \left(f_{h,1}^k(z_h^k) - f_{h,-2}^k(z_h^k)\right) + H\sum_{k\in\mathcal{K}_o} u_k$$

$$= \sum_{k\in\mathcal{K}_o} \sum_{h=1}^H \left((f_{h,1}^k - \overline{Q}_h^{\pi_k})(z_h^k) + (\overline{Q}_h^{\pi_k} - f_{h,-2}^k)(z_h^k)\right) + H\sum_{k\in\mathcal{K}_o} u_k$$

$$\leq \sum_{k\in\mathcal{K}_o} \sum_{h=1}^H \sum_{h'=h+1}^H \sum_{a'\in A_{h'}^k} \left(\widetilde{\mathcal{P}}_{h',1}^k(a'|s_{h'}^k, A_{h'}^k) - \mathcal{P}_{h'}(a'|s_{h'}^k, A_{h'}^k)\right) f_{h',1}^k(s_{h'}^k, a')$$

$$- \sum_{k\in\mathcal{K}_o} \sum_{h=1}^H \sum_{h'=h+1}^H \sum_{a'\in A_{h'}^k} \left(\widetilde{\mathcal{P}}_{h',-2}^k(a'|s_{h'}^k, A_{h'}^k) - \mathcal{P}_{h'}(a'|s_{h'}^k, A_{h'}^k)\right) f_{h',-2}^k(s_{h'}^k, a')$$

$$+ 2\sum_{k\in\mathcal{K}_o} \sum_{h=1}^H \min\left\{1 + L, \sum_{h'=h}^H b_{h',1}^k(s_{h'}^k, a_{h'}^k)\right\} + 2\sum_{k\in\mathcal{K}_o} \sum_{h=1}^H \min\left\{1 + L, \sum_{h'=h}^H b_{h',2}^k(s_{h'}^k, a_{h'}^k)\right\}$$

$$+ \underbrace{\sum_{k\in\mathcal{K}_o} \sum_{h=1}^H \sum_{h'=h+1}^H \left(\zeta_{h',1}^k + \dot{\zeta}_{h',1}^k - \zeta_{h',-2}^k - \dot{\zeta}_{h',-2}^k\right)}_{\text{martingale difference sequences (MDSs)}} + H\sum_{k\in\mathcal{K}_o} u_k, \quad \text{(D.56)}$$

where the second inequality holds by Lemma D.12 and D.14.

To further the right-hand side of Equation D.56, we apply Lemma D.6 (which holds with probability at least $1 - 2\delta$) to the first and the second terms, Lemma D.15 to the third term, Lemma D.16 to the

forth term, and we bound the fifth term using the Azuma-Hoeffding inequality (which holds with probability at least $1 - 4\delta$). As a result, absorbing the low-order terms, we obtain that

$$\sum_{k=1}^{K} \sum_{h=1}^{H} \left( f_{h,2}^{k}(z_h^k) - f_{h,-2}^{k}(z_h^k) \right)$$

$$\leqslant \mathcal{O}\left( \frac{1}{\sqrt{\kappa}} d H^2 \sqrt{K} \cdot \log K \log M \right)$$

$$+ \mathcal{O}\left( \sqrt{\log \frac{\mathcal{N} K H}{\delta}} \cdot \left( \sqrt{\log \frac{\mathcal{N} \mathcal{N}_b K H}{\delta}} \cdot H^2 \sqrt{d_\nu K} + \log \frac{\mathcal{N} \mathcal{N}_b K H}{\delta} \cdot d_\nu H^2 \right) \right)$$

$$+ \mathcal{O}\left( \sqrt{\log \frac{\mathcal{N} \mathcal{N}_b K H}{\delta}} \cdot K H^2 \epsilon_b + H \sqrt{K H \log \frac{K H}{\delta}} + |\mathcal{K}_{\mathrm{oo}}| H + H \sum_{k \in \mathcal{K}_\mathrm{o}} u_k \right), \qquad \text{(D.57)}$$

where the last inequality holds by the AM-GM inequality.

Plugging Equation D.57 to Equation D.55, we have

$$\sum_{k=1}^{K} \sum_{h=1}^{H} \min \left\{ 1, \bar{\sigma}_h^k \cdot \left( \bar{\sigma}_h^k \right)^{-1} D_{\mathcal{F}_h}\left( z_h^k; \{z_h^\tau\}_{\tau=1}^{k-1}, \{\bar{\sigma}\}_{\tau=1}^{k-1} \right) \right\}$$

$$\leqslant \mathcal{O}\left( \sqrt{\log \frac{\mathcal{N} \mathcal{N}_b K H}{\delta}} \cdot \sqrt{d_\nu H} \cdot \left( \sqrt{\frac{1}{\sqrt{\kappa}} d H^3 \log K \log M \cdot \sqrt{K}} + H^2 \sum_{k \in \mathcal{K}_\mathrm{o}} u_k \right) \right)$$

$$+ \mathcal{O}\left( \left( \log \frac{\mathcal{N} \mathcal{N}_b K H}{\delta} \right)^{3/4} \cdot \sqrt{d_\nu H} \cdot \sqrt{\sqrt{\log \frac{\mathcal{N} K H}{\delta}} \cdot \left( H^3 \sqrt{d_\nu K} + \sqrt{\log \frac{\mathcal{N} \mathcal{N}_b K H}{\delta}} \cdot d_\nu H^3 \right)} \right)$$

$$+ \mathcal{O}\left( \sqrt{\log \frac{\mathcal{N} \mathcal{N}_b K H}{\delta}} \cdot \sqrt{d_\nu H} \cdot \sqrt{\sqrt{\log \frac{\mathcal{N} \mathcal{N}_b K H}{\delta}} \cdot K H^3 \epsilon_b + H^2 \sqrt{K H \log \frac{K H}{\delta}} + |\mathcal{K}_{\mathrm{oo}}| H^2} \right)$$

$$+ \mathcal{O}\left( \sqrt{K + K H^2 \delta + H^4 \log^2 \frac{K H}{\delta}} \cdot \sqrt{d_\nu H} + d_\nu H^{1.5} \sqrt{\log \frac{\mathcal{N} \mathcal{N}_b K H}{\delta}} \right)$$

$$\leqslant \mathcal{O}\left( \sqrt{\log \frac{\mathcal{N} \mathcal{N}_b K H}{\delta}} \cdot \sqrt{d_\nu H} \cdot \sqrt{\frac{K}{\log \frac{\mathcal{N} \mathcal{N}_b K H}{\delta}} + \frac{1}{\kappa} d^2 H^6 (\log K \log M)^2 \cdot \log \frac{\mathcal{N} \mathcal{N}_b K H}{\delta}} \right)$$

$$+ \mathcal{O}\left( \sqrt{\log \frac{\mathcal{N} \mathcal{N}_b K H}{\delta}} \cdot \sqrt{d_\nu H} \cdot \sqrt{d_\nu H^6 \log \frac{\mathcal{N} K H}{\delta} \left( \log \frac{\mathcal{N} \mathcal{N}_b K H}{\delta} \right)^2 + H^2 \sum_{k \in \mathcal{K}_\mathrm{o}} u_k} \right)$$

$$+ \mathcal{O}\left( \sqrt{\log \frac{\mathcal{N} \mathcal{N}_b K H}{\delta}} \cdot \sqrt{d_\nu H} \cdot \sqrt{H^2 |\mathcal{K}_{\mathrm{oo}}|} + K H \epsilon_b + \sqrt{d_\nu H} \cdot \sqrt{K H^2 \delta} \right)$$

$$= \mathcal{O}\left( \sqrt{d_\nu H K} + \frac{1}{\sqrt{\kappa}} d H^{7/2} \sqrt{d_\nu} \log K \log M \cdot \log \frac{\mathcal{N} \mathcal{N}_b K H}{\delta} \right)$$

$$+ \mathcal{O}\left( d_\nu H^{7/2} \sqrt{\log \frac{\mathcal{N} K H}{\delta}} \cdot \left( \log \frac{\mathcal{N} \mathcal{N}_b K H}{\delta} \right)^{3/2} \right)$$

$$+ \mathcal{O}\left( \sqrt{\log \frac{\mathcal{N} \mathcal{N}_b K H}{\delta}} \cdot \sqrt{d_\nu H} \cdot \left( \sqrt{H^2 \sum_{k \in \mathcal{K}_\mathrm{o}} u_k} + \sqrt{H^2 |\mathcal{K}_{\mathrm{oo}}|} \right) + K H \epsilon_b + \sqrt{d_\nu K H^3 \delta} \right),$$

$$\text{(D.58)}$$

where the second inequality holds by applying the AM-GM inequality and absorbing the lower-order terms.

Finally, plugging Equation D.58 to Equation D.48, we derive that

$$\sum_{k=1}^{K}\sum_{h=1}^{H}\min\left\{1+L, b_{h,1}^{k}(z_{h}^{k})\right\}$$

$$\leqslant \mathcal{O}\left(\sqrt{d_{\nu}HK \cdot \log\frac{\mathcal{N}KH}{\delta}} + \frac{1}{\sqrt{\kappa}}dH^{7/2}\sqrt{d_{\nu}}\log K\log M \cdot \log\frac{\mathcal{N}\mathcal{N}_{b}KH}{\delta} \cdot \sqrt{\log\frac{\mathcal{N}KH}{\delta}}\right)$$

$$+ \mathcal{O}\left(d_{\nu}H^{7/2}\log\frac{\mathcal{N}KH}{\delta} \cdot \left(\log\frac{\mathcal{N}\mathcal{N}_{b}KH}{\delta}\right)^{3/2} + \sqrt{\log\frac{\mathcal{N}KH}{\delta}} \cdot \left(KH\epsilon_{b} + \sqrt{d_{\nu}KH^{3}\delta}\right)\right)$$

$$+ \mathcal{O}\left(\sqrt{\log\frac{\mathcal{N}KH}{\delta}\log\frac{\mathcal{N}\mathcal{N}_{b}KH}{\delta}} \cdot \sqrt{d_{\nu}H} \cdot \left(\sqrt{H^{2}\sum_{k\in\mathcal{K}_{o}}u_{k}} + \sqrt{H^{2}|\mathcal{K}_{oo}|}\right)\right).$$

This concludes the proof of Lemma D.17. $\qquad\square$

**Lemma D.18** (Bounding size of $\mathcal{K}_{oo}$). *Suppose $\nu \leqslant 1$ and we set*

$$u_{k} \geqslant C \cdot \left(\frac{\sqrt{\log\frac{\mathcal{N}KH}{\nu\delta}} \cdot \left(\log\frac{\mathcal{N}\mathcal{N}_{b}KH}{\nu\delta} \cdot H^{5/2}\sqrt{d_{\nu}} + \sqrt{k}H\epsilon_{b}\right)}{\sqrt{k}} + \frac{dH^{5/2}\log K\log M\sqrt{\log\frac{\mathcal{N}\mathcal{N}_{b}KH}{\nu\delta}}}{\sqrt{k}}\right),$$

*for some large enough constant $0 < C < \infty$, when the event $\mathcal{E}_{\leqslant K}$ holds true, then with probability at least $1 - 2\delta$, it holds that*

$$|\mathcal{K}_{oo}| \leqslant \mathcal{O}\left(\frac{K}{H^{3}\log\frac{\mathcal{N}\mathcal{N}_{b}KH}{\nu\delta}} + \frac{d_{\nu}}{H^{3}} + \frac{d^{4}(\log K\log M)^{4}}{\kappa^{2}d_{\nu}H^{3} \cdot \log\frac{\mathcal{N}KH}{\nu\delta}\left(\log\frac{\mathcal{N}\mathcal{N}_{b}KH}{\nu\delta}\right)^{2}}\right).$$

*Proof of Lemma D.18.* By the definition of $h_{k}$, for each $k \in \mathcal{K}_{oo}$, we have $f_{h_{k},2}^{k}(s_{h_{k}}^{k}, a_{h_{k}}^{k}) \geqslant f_{h_{k},1}^{k}(s_{h_{k}}^{k}, a_{h_{k}}^{k}) + u_{k}$, which implies that

$$\sum_{k\in\mathcal{K}_{oo}}(f_{h_{k},2}^{k} - f_{h_{k},1}^{k})(s_{h_{k}}^{k}, a_{h_{k}}^{k}) \geqslant \sum_{k\in\mathcal{K}_{oo}}u_{k}$$

$$\geqslant C \cdot \left(\sqrt{\log\frac{\mathcal{N}KH}{\nu\delta}} \cdot \left(\log\frac{\mathcal{N}\mathcal{N}_{b}KH}{\nu\delta} \cdot H^{5/2}\sqrt{d_{\nu}} \cdot \frac{|\mathcal{K}_{oo}|}{\sqrt{K}} + |\mathcal{K}_{oo}|H\epsilon_{b}\right)\right.$$

$$\left. + dH^{5/2}\log K\log M\sqrt{\log\frac{\mathcal{N}\mathcal{N}_{b}KH}{\nu\delta}} \cdot \frac{|\mathcal{K}_{oo}|}{\sqrt{K}}\right). \tag{D.59}$$

Furthermore, under the event $\mathcal{E}_{\leqslant K}$, by Lemma D.8, it holds that $f_{h,1}^k(s_h, a_h) \geqslant \overline{Q}_h^\star(s_h, a_h) \geqslant \overline{Q}_h^{\pi_k}(s_h, a_h)$ for all $(s_h, a_h) \in \mathcal{S} \times \mathcal{I}$. Thus, we get

$$\sum_{k \in \mathcal{K}_{\text{oo}}} (f_{h_k,2}^k - f_{h_k,1}^k)(s_{h_k}^k, a_{h_k}^k) \leqslant \sum_{k \in \mathcal{K}_{\text{oo}}} (f_{h_k,2}^k - \overline{Q}_{h_k}^{\pi_k})(s_{h_k}^k, a_{h_k}^k)$$

$$\leqslant \sum_{k \in \mathcal{K}_{\text{oo}}} \sum_{h'=h_k+1}^H \sum_{a' \in A_{h',2}^k} \left( \widetilde{\mathcal{P}}_{h',2}^k(a'|s_{h'}^k, A_{h',2}^k) - \mathcal{P}_{h'}(a'|s_{h'}^k, A_{h',2}^k) \right) f_{h',2}^k(s_{h'}^k, a')$$

$$+ 2 \sum_{k \in \mathcal{K}_{\text{oo}}} \sum_{h'=h_k}^H \min\left\{1 + L, b_{h',1}^k(s_{h'}^k, a_{h'}^k)\right\} + 2 \sum_{k \in \mathcal{K}_{\text{oo}}} \sum_{h'=h_k}^H \min\left\{1 + L, b_{h',2}^k(s_{h'}^k, a_{h'}^k)\right\}$$

$$+ \sum_{k \in \mathcal{K}_{\text{oo}}} \sum_{h'=h_k+1}^H \left( \zeta_{h',2}^k + \dot{\zeta}_{h',2}^k \right)$$

$$\leqslant \mathcal{O}\left( dH\sqrt{|\mathcal{K}_{\text{oo}}|} \cdot \log K \log M + \frac{1}{\kappa} d^2 H \left(\log K \log M\right)^2 + \sqrt{|\mathcal{K}_{\text{oo}}| H \log \frac{KH}{\delta}} \right)$$

$$+ \mathcal{O}\left( \sqrt{\log \frac{\mathcal{N}KH}{\nu\delta}} \cdot \left( \sqrt{\log \frac{\mathcal{N}\mathcal{N}_b KH}{\nu\delta}} \cdot H\sqrt{d_\nu |\mathcal{K}_{\text{oo}}|} + \log \frac{\mathcal{N}\mathcal{N}_b KH}{\nu\delta} \cdot d_\nu H + |\mathcal{K}_{\text{oo}}| H \epsilon_b \right) \right),$$

$$\text{(D.60)}$$

where in the first inequality, we note that $A_{h'}^k = A_{h',2}^k$ for $h' \geqslant h_k$, the second inequality follows from D.13. And for the third inequality, we apply Lemma D.7 to the first term, Lemma D.15 to the second term, Lemma D.16 to the third term. Finally, we bound the last term using the Azuma-Hoeffding inequality with probability at least $1 - 2\delta$.

Thus, in order for the two inequalities D.59 and D.60 to hold simultaneously, the following condition must be satisfied:

$$|\mathcal{K}_{\text{oo}}| \leqslant \mathcal{O}\Bigg( \max\Bigg\{ \frac{K}{H^3 \log \frac{\mathcal{N}\mathcal{N}_b KH}{\nu\delta}},$$

$$\frac{\left( d_\nu \sqrt{\log \frac{\mathcal{N}KH}{\nu\delta}} \log \frac{\mathcal{N}\mathcal{N}_b KH}{\nu\delta} + \frac{1}{\kappa} d^2 (\log K \log M)^2 \right) \cdot \sqrt{K}}{H^{3/2}\sqrt{\log \frac{\mathcal{N}\mathcal{N}_b KH}{\nu\delta}} \left( \sqrt{d_\nu \log \frac{\mathcal{N}KH}{\nu\delta} \log \frac{\mathcal{N}\mathcal{N}_b KH}{\nu\delta}} + d \log K \log M \right)} \Bigg\} \Bigg).$$

Using the AM-GM inequality, we can further bound the second term inside the max operation.

$$\mathcal{O}\left( \frac{\left( d_\nu \sqrt{\log \frac{\mathcal{N}KH}{\nu\delta}} \log \frac{\mathcal{N}\mathcal{N}_b KH}{\nu\delta} + \frac{1}{\kappa} d^2 (\log K \log M)^2 \right) \cdot \sqrt{K}}{H^{3/2}\sqrt{\log \frac{\mathcal{N}\mathcal{N}_b KH}{\nu\delta}} \left( \sqrt{d_\nu \log \frac{\mathcal{N}KH}{\nu\delta} \log \frac{\mathcal{N}\mathcal{N}_b KH}{\nu\delta}} + d \log K \log M \right)} \right)$$

$$\leqslant \mathcal{O}\left( \frac{K}{H^3 \log \frac{\mathcal{N}\mathcal{N}_b KH}{\nu\delta}} + \left( \frac{d_\nu \sqrt{\log \frac{\mathcal{N}KH}{\nu\delta}} \log \frac{\mathcal{N}\mathcal{N}_b KH}{\nu\delta} + \frac{1}{\kappa} d^2 (\log K \log M)^2}{H^{3/2}\sqrt{d_\nu \log \frac{\mathcal{N}KH}{\nu\delta} \log \frac{\mathcal{N}\mathcal{N}_b KH}{\nu\delta}}} \right)^2 \right)$$

$$\leqslant \mathcal{O}\left( \frac{K}{H^3 \log \frac{\mathcal{N}\mathcal{N}_b KH}{\nu\delta}} + \left( \sqrt{\frac{d_\nu}{H^3}} + \frac{d^2 (\log K \log M)^2}{\kappa \cdot H^{3/2}\sqrt{d_\nu \log \frac{\mathcal{N}KH}{\nu\delta} \log \frac{\mathcal{N}\mathcal{N}_b KH}{\nu\delta}}} \right)^2 \right)$$

$$\leqslant \mathcal{O}\left( \frac{K}{H^3 \log \frac{\mathcal{N}\mathcal{N}_b KH}{\nu\delta}} + \frac{d_\nu}{H^3} + \frac{d^4 (\log K \log M)^4}{\kappa^2 d_\nu H^3 \cdot \log \frac{\mathcal{N}KH}{\nu\delta} \left(\log \frac{\mathcal{N}\mathcal{N}_b KH}{\nu\delta}\right)^2} \right),$$

where the second inequality also follow from the AM-GM inequality and the last inequality holds due to the fact that $(a + b)^2 \leqslant 2a^2 + 2b^2$ for any $a, b \in \mathbb{R}^+$. This concludes the proof. $\qquad\square$

## D.8 Proof of Theorem 1

Now, we are ready to provide the proof of Theorem 1.

*Proof of Theorem 1.* When the event $\mathcal{E}^\theta \bigcap \mathcal{E}_{\leqslant K}$ happens (with probability at least $1 - 2\delta$), we can bound the regret as follows:

$$\textbf{Regret}(\mathcal{M}, K) = \sum_{k=1}^K (V_1^\star - V_1^{\pi_k})(s_1^k) \leqslant \sum_{k=1}^K (V_1^k - V_1^{\pi_k})(s_1^k) = \sum_{k=1}^K (Q_1^k - Q_1^{\pi_k})(s_1^k, A_1^k)$$

$$\leqslant \mathcal{O}(1) + \sum_{k=2}^K (Q_1^k - Q_1^{\pi_k})(s_1^k, A_1^k)$$

$$= \mathcal{O}(1) + \sum_{k\in\mathcal{K}_o\backslash\{1\}} (Q_1^k - Q_1^{\pi_k})(s_1^k, A_1^k) + \sum_{k\in\mathcal{K}_{oo}\backslash\{1\}} (Q_1^k - Q_1^{\pi_k})(s_1^k, A_1^k),$$

$$\text{(D.61)}$$

where the first inequality holds by Lemma D.9.

For $k \in \mathcal{K}_o\backslash\{1\}$, recall that $Q_h^k(s, A_h^k) = \sum_{a_1\in A_h^k} \widetilde{\mathcal{P}}_{h,1}^k(a_1|s, A_h^k) f_{h,1}^k(s_h^k, a_1)$ for all $h \in [H]$, as defined in Equation D.18. Therefore, we have

$$\left(Q_1^k - Q_1^{\pi_k}\right)(s_1^k, A_1^k)$$

$$= \sum_{a_1\in A_1^k} \widetilde{\mathcal{P}}_{1,1}^k(a_1|s_1^k, A_1^k) f_{1,1}^k(s_h^k, a_1) - \sum_{a_1\in A_1^k} \mathcal{P}_1(a_1|s_1^k, A_1^k) \overline{Q}_1^{\pi_k}(s_1^k, a_1)$$

$$= \sum_{a_1\in A_1^k} \left(\widetilde{\mathcal{P}}_{1,1}^k(a_1|s_1^k, A_1^k) - \mathcal{P}_1(a_1|s_1^k, A_1^k)\right) f_{1,1}^k(s_h^k, a_1)$$

$$+ \sum_{a_1\in A_1^k} \mathcal{P}_1(a_1|s_1^k, A_1^k) \left(f_{1,1}^k - \overline{Q}_1^{\pi_k}\right)(s_1^k, a_1)$$

$$= \sum_{a_1\in A_1^k} \left(\widetilde{\mathcal{P}}_{1,1}^k(a_1|s_1^k, A_1^k) - \mathcal{P}_1(a_1|s_1^k, A_1^k)\right) f_{1,1}^k(s_h^k, a_1) + \left(f_{1,1}^k - \overline{Q}_1^{\pi_k}\right)(s_1^k, a_1)$$

$$+ \mathbb{E}_{\mathcal{P}}\left[\left(f_{1,1}^k - \overline{Q}_1^{\pi_k}\right)(s_1^k, a_1) \mid s_1^k, A_1^k\right] - \left(f_{1,1}^k - \overline{Q}_1^{\pi_k}\right)(s_1^k, a).$$

Then, by applying Lemma D.14 with $h_k = H + 1$, we have

$$\left(Q_1^k - Q_1^{\pi_k}\right)(s_1^k, A_1^k) \leqslant \sum_{h=1}^H \sum_{a_h\in A_h^k} \left(\widetilde{\mathcal{P}}_{h,1}^k(a_h|s_h^k, A_h^k) - \mathcal{P}_h(a_h|s_h^k, A_h^k)\right) f_{h,1}^k(s_h^k, a_h)$$

$$+ 2\sum_{h=1}^H b_{h,1}^k(s_h^k, a_h^k) + \sum_{h=1}^H \dot{\zeta}_{h,1}^k + \sum_{h=2}^H \zeta_{h,1}^k, \qquad \text{(D.62)}$$

where $\zeta_{h,1}^k = \mathbb{E}_{\mathbb{P}}\left[(V_{h,1}^k - V_h^{\pi_k})(s_h) \mid s_{h-1}^k, a_{h-1}^k\right] - (V_{h,1}^k - V_h^{\pi_k})(s_h^k)$ and $\dot{\zeta}_{h,1}^k = \mathbb{E}_{\mathcal{P}}\left[\left(f_{h,1}^k - \overline{Q}_h^{\pi_k}\right)(s_h^k, a_h) \mid s_h^k, A_h^k\right] - \left(f_{h,1}^k - \overline{Q}_h^{\pi_k}\right)(s_h^k, a_h^k)$.

Now, we consider the case where $k \in \mathcal{K}_{oo}\backslash\{1\}$. In this cases, note that $h_k \in [H]$. Similar to the above analysis, by Lemma D.14, we get

$$\left(Q_1^k - Q_1^{\pi_k}\right)(s_1^k, A_1^k)$$

$$\leqslant \sum_{h=1}^{h_k-1} \sum_{a_h\in A_h^k} \left(\widetilde{\mathcal{P}}_{h,1}^k(a_h|s_h^k, A_h^k) - \mathcal{P}_h(a_h|s_h^k, A_h^k)\right) f_{h,1}^k(s_h^k, a_h)$$

$$+ \sum_{h=h_k}^H \sum_{a_h\in A_h^k} \left(\widetilde{\mathcal{P}}_{h,2}^k(a_h|s_h^k, A_h^k) - \mathcal{P}_h(a_h|s_h^k, A_h^k)\right) f_{h,2}^k(s_h^k, a_h)$$

$$+ 2\sum_{h=1}^H b_{h,1}^k(s_h^k, a_h^k) + 2\sum_{h=h_k}^H b_{h,2}^k(s_h^k, a_h^k) + \sum_{h=1}^{h_k-1} \dot{\zeta}_{h,1}^k + \sum_{h=2}^{h_k-1} \zeta_{h,1}^k + \sum_{h=h_k}^H \dot{\zeta}_{h,2}^k + \sum_{h=h_k}^H \zeta_{h,2}^k,$$

$$\text{(D.63)}$$

where $\zeta_{h,2}^k = \mathbb{E}_{\mathbb{P}}\left[(V_{h,2}^k - V_h^{\pi_k})(s_h) \mid s_{h-1}^k, a_{h-1}^k\right] - (V_{h,2}^k - V_h^{\pi_k})(s_h^k)$ and $\dot{\zeta}_{h,2}^k = \mathbb{E}_{\mathcal{P}}\left[\left(f_{h,2}^k - \overline{Q}_h^{\pi_k}\right)(s_h^k, a_h) \mid s_h^k, A_h^k\right] - \left(f_{h,2}^k - \overline{Q}_h^{\pi_k}\right)(s_h^k, a_h^k)$.

Plugging Equation D.62 and Equation D.63 into Equation D.61, and denoting $J(k,h) : [K] \times [H] \to \{1,2\}$ as the one-to-one function that maps from $[K] \times [H]$ to the index set $\{1,2\}$ such that $A_h^k = A_{h,J(k,h)}^k \in \arg\max_{A \in \mathcal{A}} \sum_{a \in A} \widetilde{\mathcal{P}}_{h,J(k,h)}^k(a|s_h^k, A) f_{h,J(k,h)}^k(s_h^k, a)$, we obtain that

$\textbf{Regret}(\mathcal{M}, K)$

$$\leqslant \mathcal{O}(1) + \sum_{k=2}^{K} \sum_{h=1}^{H} \sum_{a_h \in A_h^k} \left(\widetilde{\mathcal{P}}_{h,J(k,h)}^k(a_h|s_h^k, A_h^k) - \mathcal{P}_h(a_h|s_h^k, A_h^k)\right) f_{h,J(k,h)}^k(s_h^k, a_h)$$

$$+ 2\sum_{k=2}^{K} \sum_{h=1}^{H} \min\left\{1 + L, b_{h,1}^k(s_h^k, a_h^k)\right\} + 2\sum_{k \in \mathcal{K}_{oo}} \sum_{h=h_k}^{H} \min\left\{1 + L, b_{h,2}^k(s_h^k, a_h^k)\right\}$$

$$+ \sum_{k=2}^{K} \sum_{h=1}^{h_k-1} \dot{\zeta}_{h,1}^k + \sum_{k=2}^{K} \sum_{h=2}^{h_k-1} \zeta_{h,1}^k + \sum_{k=2}^{K} \sum_{h=h_k}^{H} \dot{\zeta}_{h,2}^k + \sum_{k=2}^{K} \sum_{h=h_k}^{H} \zeta_{h,2}^k.$$

Now, by applying the results from Lemma D.7 to bound the first term (which holds with probability at least $1 - \delta$), Lemma D.17 for the second term (which holds with probability at least $1 - 7\delta$), Lemma D.16 for the third term, and applying the Azuma-Hoeffding inequality to the remaining terms (which holds with probability at least $1 - 4\delta$), we get

$\textbf{Regret}(\mathcal{M}, K)$

$$\leqslant \mathcal{O}\left(dH\sqrt{K} \cdot \log K \log M + \frac{1}{\kappa} d^2 H (\log K \log M)^2\right)$$

$$+ \mathcal{O}\left(\sqrt{d_\nu H K \cdot \log \frac{\mathcal{N}KH}{\delta}} + \frac{1}{\sqrt{\kappa}} dH^{7/2} \sqrt{d_\nu} \log K \log M \cdot \log \frac{\mathcal{N}\mathcal{N}_b KH}{\delta} \cdot \sqrt{\log \frac{\mathcal{N}KH}{\delta}}\right)$$

$$+ \mathcal{O}\left(d_\nu H^{7/2} \log \frac{\mathcal{N}KH}{\delta} \cdot \left(\log \frac{\mathcal{N}\mathcal{N}_b KH}{\delta}\right)^{3/2} + \sqrt{\log \frac{\mathcal{N}KH}{\delta}} \cdot \left(KH\epsilon_b + \sqrt{d_\nu KH^3\delta}\right)\right)$$

$$+ \mathcal{O}\left(\sqrt{\log \frac{\mathcal{N}KH}{\delta} \log \frac{\mathcal{N}\mathcal{N}_b KH}{\delta}} \cdot \sqrt{d_\nu H} \cdot \left(\sqrt{H^2 \sum_{k \in \mathcal{K}_o} u_k} + \sqrt{H^2 |\mathcal{K}_{oo}|}\right)\right)$$

$$+ \mathcal{O}\left(\sqrt{\log \frac{\mathcal{N}\mathcal{N}_b KH}{\delta}} \cdot \left(H\sqrt{d_\nu |\mathcal{K}_{oo}|} + |\mathcal{K}_{oo}|H\epsilon_b\right)\right)$$

$$\leqslant \mathcal{O}\left(dH\sqrt{K} \cdot \log K \log M + \frac{1}{\kappa} d^2 H (\log K \log M)^2\right)$$

$$+ \mathcal{O}\left(\sqrt{d_\nu H K \cdot \log \frac{\mathcal{N}KH}{\delta}} + \frac{1}{\sqrt{\kappa}} dH^{7/2} \sqrt{d_\nu} \log K \log M \cdot \log \frac{\mathcal{N}\mathcal{N}_b KH}{\delta} \cdot \sqrt{\log \frac{\mathcal{N}KH}{\delta}}\right)$$

$$+ \mathcal{O}\left(d_\nu H^{7/2} \log \frac{\mathcal{N}KH}{\delta} \cdot \left(\log \frac{\mathcal{N}\mathcal{N}_b KH}{\delta}\right)^{3/2} + \sqrt{\log \frac{\mathcal{N}\mathcal{N}_b KH}{\delta}} \cdot \left(KH\epsilon_b + \sqrt{d_\nu KH^3\delta}\right)\right)$$

$$+ \mathcal{O}\left(\sqrt{\log \frac{\mathcal{N}KH}{\delta} \log \frac{\mathcal{N}\mathcal{N}_b KH}{\delta}} \cdot \sqrt{d_\nu H} \cdot \sqrt{H^2 \sum_{k \in \mathcal{K}_o} u_k}\right), \tag{D.64}$$

where the second inequality holds by Lemma D.18 with probability at least $1 - 2\delta$, and use the fact that $|\mathcal{K}_{oo}|H\epsilon_b \leqslant KH\epsilon_b$. Now, we apply the AM-GM inequality to the term

$\mathcal{O}\left(\frac{1}{\sqrt{\kappa}} dH^{7/2}\sqrt{d_\nu}\log K \log M \cdot \log \frac{\mathcal{N}\mathcal{N}_b KH}{\delta} \cdot \sqrt{\log \frac{\mathcal{N}KH}{\delta}}\right)$, thus we get

$$\mathcal{O}\left(\frac{1}{\sqrt{\kappa}} dH^{7/2}\sqrt{d_\nu}\log K \log M \cdot \log \frac{\mathcal{N}\mathcal{N}_b KH}{\delta} \cdot \sqrt{\log \frac{\mathcal{N}KH}{\delta}}\right)$$

$$\leqslant \mathcal{O}\left(\frac{1}{\kappa} d^2 H^2 \left(\log K \log M\right)^2 + d_\nu H^5 \left(\log \frac{\mathcal{N}\mathcal{N}_b KH}{\delta}\right)^2 \cdot \log \frac{\mathcal{N}KH}{\delta}\right). \tag{D.65}$$

Furthermore, by substituting the chosen values of $u_k$ and applying the AM-GM inequality, we get

$$\mathcal{O}\left(\sqrt{\log \frac{\mathcal{N}KH}{\delta}\log\frac{\mathcal{N}\mathcal{N}_b KH}{\delta}} \cdot \sqrt{d_\nu H} \cdot \sqrt{H^2 \sum_{k\in\mathcal{K}_o} u_k}\right)$$

$$= \mathcal{O}\left(\sqrt{d_\nu HK \cdot \log \frac{\mathcal{N}KH}{\delta}} + d^2 H^2 \left(\log K \log M\right)^2 + d_\nu H^5 \log \frac{\mathcal{N}KH}{\delta}\cdot\left(\log\frac{\mathcal{N}\mathcal{N}_b KH}{\delta}\right)^2\right)$$

$$+ \mathcal{O}\left(\sqrt{\log \frac{\mathcal{N}KH}{\delta}}\cdot KH\epsilon_b\right). \tag{D.66}$$

Then, by plugging Equation D.65 and Equation D.66 into Equation D.64, and setting $\delta < \frac{1}{H^2+14}$, we derive that

$$\mathbf{Regret}(\mathcal{M}, K)$$

$$= \mathcal{O}\left(dH\sqrt{K}\cdot\log K \log M + \sqrt{d_\nu HK \cdot \log \frac{\mathcal{N}KH}{\delta}} + \frac{1}{\kappa} d^2 H^2 \left(\log K \log M\right)^2\right)$$

$$+ \mathcal{O}\left(d_\nu H^5 \log \frac{\mathcal{N}KH}{\delta}\cdot\left(\log\frac{\mathcal{N}\mathcal{N}_b KH}{\delta}\right)^2 + \sqrt{\log \frac{\mathcal{N}\mathcal{N}_b KH}{\delta}}\cdot KH\epsilon_b\right).$$

We conclude the proof of Theorem 1. $\qquad\qquad\square$

# E    PROOF OF THEOREM 2

In this section, we introduce several properties of linear function class. For linear MDPs, let $\mathcal{F}_h^{\mathrm{lin}}(\epsilon_c)$ be an $\epsilon_c$-cover of $\mathcal{F}_h^{\mathrm{lin}}$ under the $\ell_\infty$ norm, so that

$$\log |\mathcal{F}_h^{\mathrm{lin}}(\epsilon_c)| = \mathcal{O}\left(d^{\mathrm{lin}}\log\frac{2\sqrt{d^{\mathrm{lin}}}}{\epsilon_c}\right) = \tilde{\mathcal{O}}\left(d^{\mathrm{lin}}\right). \tag{E.1}$$

Then, the definition of generalized Eluder dimension for the linear function class $\mathcal{F}_h^{\mathrm{lin}}$ can be expressed as:

**Lemma E.1** (Lemma 3 of Agarwal et al. 2023). *For the class $\mathcal{F}_h^{lin}$ defined in Equation 1, letting $\mathcal{F}_h^{lin}(\epsilon_c)$ be the $\epsilon_c$-cover of $\mathcal{F}_h^{lin}$ for some $\epsilon_c > 0$, we have*

$$\dim_{\nu,K}(\mathcal{F}_h^{lin}(\epsilon_c)) \leqslant \dim_{\nu,K}(\mathcal{F}_h^{lin}) = \mathcal{O}\left(d^{lin}\log\left(1+\frac{K}{\nu^2\rho}\right)\right) = \tilde{\mathcal{O}}(d^{lin}).$$

The bonus oracle for linear MDPs can be easily instantiated using the standard elliptical bonus, and, as demonstrated in the next lemma, satisfies all the required properties for a bonus oracle.

**Lemma E.2** (Bonus oracle $\mathcal{B}$ for linear MDPs, Lemma 7 of Agarwal et al. 2023). *Given $K, H \in \mathbb{Z}_+$, suppose all $\beta_h^k \leqslant \beta$ and $\beta_h^k$ is non-decreasing in $k \in [K]$ for each $h \in [H]$. For any $k \geqslant 1, h \in [H]$, variances $\{\bar{\sigma}_h^\tau\}_{\tau=1}^h$ satisfying $\bar{\sigma}_h^\tau \geqslant \nu$ for some $\nu > 0$, dataset $\mathcal{D}_h^{k-1} = \{\psi(s_h^\tau, a_h^\tau), a_h^\tau, r_h^\tau, \psi(s_{h+1}^\tau, a_{h+1}^\tau)\}_{\tau=1}^{k-1}$, function class $\mathcal{F}_h^k$ and $\hat{f}_h^k \in \mathcal{F}_h^k$ defined via weighted regression in Equation 4, and parameters $\rho, \epsilon_c > 0$, let $\mathcal{B}(\{\bar{\sigma}_h^\tau\}_{\tau=1}^h, \mathcal{D}_h^{k-1}, \mathcal{F}_h^k, \hat{f}_h^k, \beta_h^k, \rho, \epsilon_c) =$*

$\|\psi(s,a)\|_{\left(\Sigma_h^k\right)^{-1}}\sqrt{\left(\beta_h^k\right)^2+\rho}$, *where* $\Sigma_h^k = \frac{\rho}{16d}I+\sum_{\tau=1}^{k-1}\frac{1}{(\hat{\sigma}_h^\tau)^2}\psi(s_h^\tau,a_h^\tau)\psi(s_h^\tau,a_h^\tau)^\top$. *For any choice of covering radius* $\epsilon_c \leqslant \nu\sqrt{\rho/8K}$, *the oracle satisfies all the properties of Definition 4 with*

$$\log\mathcal{N}_b = \log|\mathcal{W}| = \mathcal{O}\left((d^{lin})^2\log\left(1+d^{lin}\sqrt{d^{lin}}\beta/(\rho\epsilon_c^2)\right)\right) = \tilde{\mathcal{O}}\left((d^{lin})^2\right).$$

**Theorem E.1** (Regret upper bound of `MNL-VQL` for linear MDPs, proof in Section E). *Under the same conditions with Theorem 1, suppose that the underlying MDP has linear transition probabilities and rewards, so that the function class for linear MDPs, $\mathcal{F}_h^{lin}$, satisfies Assumption 2. Let $\mathcal{F}_h^{lin}(\epsilon_c)$ be an $\epsilon_c$-cover of $\mathcal{F}_h^{lin}$ under $\ell_\infty$ norm. We set $\rho = 1$, $u_k = \tilde{\Theta}\left((d^{lin})^3H^{5/2}+d(d^{lin})^{3/2}H^{5/2})/\sqrt{K}\right)$, $\nu = \sqrt{1/HK}$, $\epsilon_b = \epsilon_c \leqslant 1/(8HK)$ and $\delta < 1/(H^2+14)$. Then, with probability at least $1-\delta$, the cumulative regret of `MNL-VQL`, with bonus oracle defined in Lemma E.2, is upper-bounded by*

$$\textbf{Regret}\left(\mathcal{M}_{\boldsymbol{\theta}^\star,\boldsymbol{\mu}^\star,\mathbf{w}^\star}, K\right) = \tilde{\mathcal{O}}\bigg(\underbrace{dH\sqrt{K}+\frac{1}{\kappa}d^2H^2}_{\text{regret from MNL model}}+\underbrace{d^{lin}\sqrt{HK}+(d^{lin})^6H^5}_{\text{regret from linear MDPs}}\bigg).$$

*Proof of Theorem 2.* We apply the above results to linear MDPs $\mathcal{M}_{\boldsymbol{\theta}^\star,\boldsymbol{\mu}^\star,\mathbf{w}^\star}$ with function class $\mathcal{F}_h^{lin}(\epsilon_c)$, $h \in [H]$, and bonus oracle $\mathcal{B}$. From Equation E.1, we know that $\mathcal{N} = \tilde{\mathcal{O}}(d^{lin})$. Additionally, Lemma E.1 shows that $d_\nu = \tilde{\mathcal{O}}(d^{lin})$. Therefore, by combining these results with Theorem 1 and Lemma E.2, we can establish the upper regret bounds for linear MDPs.

$$\textbf{Regret}\left(\mathcal{M}_{\boldsymbol{\theta}^\star,\boldsymbol{\mu}^\star,\mathbf{w}^\star}, K\right) = \tilde{\mathcal{O}}\bigg(dH\sqrt{K}+d^{lin}\sqrt{HK}+\frac{1}{\kappa}d^2H^2+(d^{lin})^6H^5\bigg),$$

where we set $\rho = 1$, $u_k = \tilde{\Theta}\left((d^{lin})^3H^{5/2}+d(d^{lin})^{3/2}H^{5/2}\right)/\sqrt{K}$, $\nu = \sqrt{1/HK}$, $\epsilon_b = \epsilon_c \leqslant 1/(8HK)$ and $\delta < 1/(H^2+14)$. $\square$

# F    PROOF OF THEOREM 3

In this section, we provide a regret lower bound for linear MDPs with preference model. We construct a hard instance $\mathcal{M}(\mathcal{S}, \mathcal{I}, \mathcal{A}, M, \{\mathcal{P}_h\}_{h=1}^H, \{\mathbb{P}_h\}_{h=1}^H, \{r_{h=1}^H\}, H)$, illustrated as in Figure E.1. This instance is based on an $H+1$-layered structure, where each layer is a variation of the hard-to-learn MDPs introduced in Zhou et al. (2021b).

Without loss of generality, we assume that $d^{lin} \geqslant 6$ and that $d^{lin}-5$ is divisible by 2.[3] Let $i \in [H+2]$ represent the layer index. For each layer $i \in [H+2]$, there are $H-i+3$ states, denoted as $x_i^{(i)}, \ldots, x_{H+2}^{(i)}$, where $x_{H+2}^{(i)}$ is the absorbing state. Furthermore, there is a *global* absorbing state $x_0$, which can only be reached at any state and horizon through the user's choice of the outside option $\mathbf{a}_0$ (not choosing any item in the assortment). Thus, there are $(H+1)(H+2)/2+1$ states in total in the set of states $\mathcal{S}$. There are $2^{(d^{lin}-5)/2}+1$ items, so the item set is $\mathcal{I} = \{-1,1\}^{(d^{lin}-5)/2} \cup \{\mathbf{a}_0\}$. The set of candidate assortments follows the definition in Section 3, i.e., $\mathcal{A} = \{A \subseteq \mathcal{I} : \mathbf{a}_0 \in A, 1 \leqslant |A\backslash\{\mathbf{a}_0\}| \leqslant M\}$.

## F.1    CONSTRUCTION OF LINEAR TRANSITIONS AND REWARDS

At each episode $k \in [K]$, the agent starts from the fixed initial state $x_1^{(1)}$. We define $\mathbf{a}_h^\star$ as an item such that $\mathbf{a}_h^\star \in \arg\max_{\mathbf{a}\in\mathcal{I}\backslash\{\mathbf{a}_0\}}\langle\boldsymbol{\mu}_h,\mathbf{a}\rangle$, where $\boldsymbol{\mu}_h \in \{-\Delta,\Delta\}^{(d^{lin}-5)/2}$ with $\Delta = \sqrt{\delta/K}/(4\sqrt{2})$ and $\delta = 1/H$.

If the state is $x_h^{(i)}$ with $i \in [H+1]$ and $h \in [i, H+1]$, and the user chooses the item $\mathbf{a}_h^\star$, the agent remains in the same layer $i$ and receives a reward of $\gamma^{i-1}/H$, where $\gamma = \frac{H}{1+H}$. The next state will be either $x_{h+1\wedge H+1}^{(i)}$ or $x_{H+2}^{(i)}$, with probabilities $1-(\delta+\langle\boldsymbol{\mu}_h,\mathbf{a}\rangle)$ and $\delta+\langle\boldsymbol{\mu}_h,\mathbf{a}\rangle$, respectively. If the user chooses an item $\mathbf{a} \neq \mathbf{a}_0, \mathbf{a}_h^\star$ in the state $x_h^{(i)}$ with $i \in [H+1]$ and $h \in [i, H+1]$, the agent obtains

---

[3] If $d^{lin}-5$ is not divisible by 2, we can set $d^{lin} \leftarrow d^{lin}+1$ by adding zero padding.

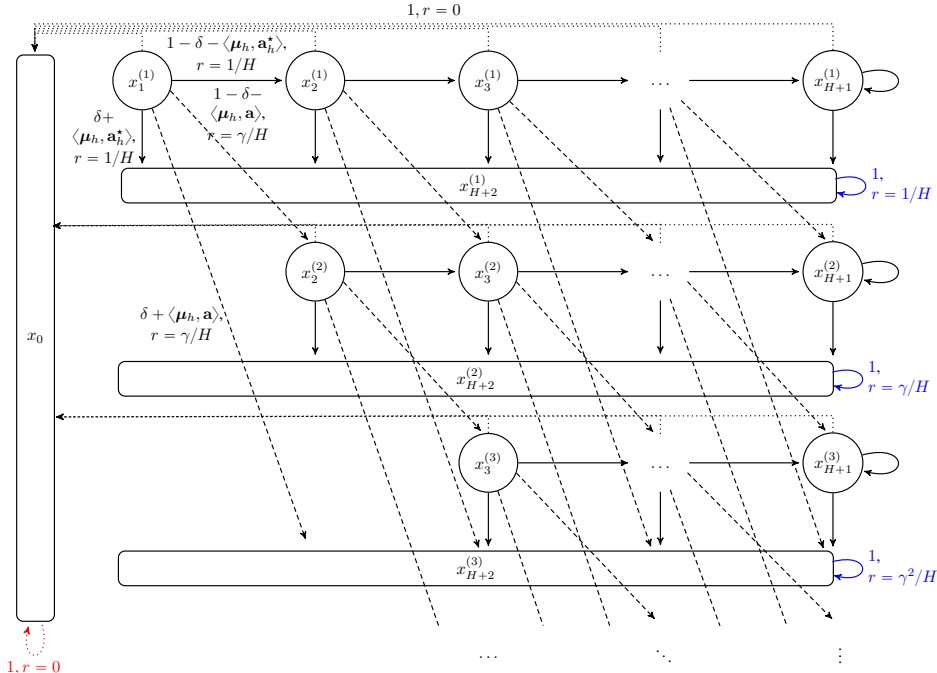

Figure E.1: Inhomogeneous, hard-to-learn linear MDPs with MNL preference model. The solid line indicates the transition caused by the user choosing the item $\mathbf{a}_h^\star$ (with a reward of $r_h = \gamma^{i-1}/H$), the dashed line shows the transition caused by the user choosing any item $\mathbf{a} \neq \mathbf{a}_h^\star, \mathbf{a}_0$ (with a reward of $r_h = \gamma^i/H$), and the dotted line represents the transition caused by the user choosing the outside option $\mathbf{a}_0$ (with a reward of $r_h = 0$). The blue solid line indicates a transition from the absorbing state back to itself, caused by the user choosing any item (with a reward of $r_h = \gamma^{i-1}/H$), and the red dotted line indicates a transition from the *global* absorbing state back to itself, caused by the user choosing any item (with a reward of $r_h = 0$).

a reward of $\gamma^i/H$ and transitions to $x_{h+1 \wedge H+1}^{(i+1)}$ or $x_{H+2}^{(i+2 \wedge H+2)}$, with probabilities $1 - (\delta + \langle \boldsymbol{\mu}_h, \mathbf{a} \rangle)$ and $\delta + \langle \boldsymbol{\mu}_h, \mathbf{a} \rangle$, respectively. If the user does not choose any item, i.e., chooses the outside option $\mathbf{a}_0$, in the state $x_h^{(i)}$ with $i \in [H+1]$ and $h \in [i, H+1]$, the agent will deterministically transition to the global absorbing state $x_0$ and receive no reward.

If the agent is in any of the absorbing states–$x_{H+2}^{(i)}$ for $i \in [H+2]$–the agent will remain in the same state and receive a reward of $\gamma^{i-1}/H$, regardless of which item (including the outside option) the user chooses.

Formally, we construct transition probabilities $\mathbb{P}_h(s'|s, \mathbf{a}) = \langle \psi(s, \mathbf{a}), \boldsymbol{\mu}_h^\star(s') \rangle$, with

$$
\psi(s, \mathbf{a}) = \begin{cases}
(\alpha, \beta \mathbf{a}^\top, 0, \mathbf{0}, 0, 0, \frac{\gamma^{i-1}}{\sqrt{2}})^\top, & s = x_h^{(i)}, \mathbf{a} = \mathbf{a}_h^\star, i \in [H+1], h \in [i, H+1]; \\
(0, \mathbf{0}, \alpha, \beta \mathbf{a}^\top, 0, 0, \frac{\gamma^i}{\sqrt{2}})^\top, & s = x_h^{(i)}, \mathbf{a} \neq \mathbf{a}_h^\star, \mathbf{a}_0, i \in [H+1], h \in [i, H+1]; \\
(0, \mathbf{0}^\top, 0, \mathbf{0}^\top, 0, 1, 0)^\top, & s = x_h^{(i)}, \mathbf{a} = \mathbf{a}_0, i \in [H+1], h \in [i, H+1]; \\
(0, \mathbf{0}^\top, 0, \mathbf{0}^\top, \frac{1}{\sqrt{2}}, 0, \frac{\gamma^{i-1}}{\sqrt{2}})^\top, & s = x_{H+2}^{(i)}, i \in [H+2],
\end{cases}
\tag{F.1}
$$

and

$$
\boldsymbol{\mu}_h^\star(s') = \begin{cases}
(\frac{1-\delta}{\alpha}, -\frac{\boldsymbol{\mu}_h^\top}{\beta}, 0, \mathbf{0}, 0, 0, 0)^\top, & s' = x_{h+1 \wedge H+1}^{(i)}; \\
(\frac{\delta}{\alpha}, \frac{\boldsymbol{\mu}_h^\top}{\beta}, 0, \mathbf{0}, \sqrt{2}, 0, 0)^\top, & s' = x_{H+2}^{(i)}; \\
(0, \mathbf{0}, \frac{1-\delta}{\alpha}, -\frac{\boldsymbol{\mu}_h^\top}{\beta}, 0, 0, 0)^\top, & s' = x_{h+1}^{(i+1)}; \\
(0, \mathbf{0}, \frac{\delta}{\alpha}, \frac{\boldsymbol{\mu}_h^\top}{\beta}, 0, 0, 0)^\top, & s' = x_{H+2}^{(i+2 \wedge H+2)}; \\
(0, \mathbf{0}^\top, 0, \mathbf{0}, 0, 1, 0)^\top, & s' = x_0; \\
(0, \mathbf{0}^\top, 0, \mathbf{0}, 0, 0, 0)^\top, & \text{otherwise,}
\end{cases}
\tag{F.2}
$$

where we denote $\mathbf{0} \in \mathbb{R}^{(d^{\text{lin}}-5)/2}$ as the zero vector of dimension $(d^{\text{lin}} - 5)/2$, and set $\gamma = \frac{H}{H+1}$ as the discount factor for transitioning to the next layer. Additionally, we choose $\delta = 1/H$, $\boldsymbol{\mu}_h \in \{-\Delta, \Delta\}^{(d^{\text{lin}}-5)/2}$ with $\Delta = \sqrt{\delta/K}/(4\sqrt{2})$, $\alpha = \sqrt{1/(2 + \Delta \cdot (d^{\text{lin}} - 5))}$, and $\beta = \sqrt{\Delta/(2 + \Delta \cdot (d^{\text{lin}} - 5))}$.

And the parameter vectors for the linear rewards $r_h(s, \mathbf{a}) = \langle \psi(s, \mathbf{a}), \mathbf{w}_h^\star \rangle$ are as follows:

$$
\mathbf{w}_h^\star = (0, \mathbf{0}^\top, 0, \mathbf{0}, 0, 0, \sqrt{2}/H)^\top,
$$

which ensures that the reward function satisfies:

$$
r_h(s, \mathbf{a}) = \begin{cases}
\gamma^{i-1}/H, & s = x_h^{(i)}, \mathbf{a} = \mathbf{a}_h^\star, i \in [H+1], h \in [i, H+1]; \\
\gamma^i/H, & s = x_h^{(i)}, \mathbf{a} \neq \mathbf{a}_h^\star, \mathbf{a}_0, i \in [H+1], h \in [i, H+1]; \\
0, & s = x_h^{(i)}, \mathbf{a} = \mathbf{a}_0, i \in [H+1], h \in [i, H+1]; \\
\gamma^{i-1}/H, & s = x_{H+2}^{(i)}, i \in [H+2],
\end{cases}
$$

where $0 < \gamma \leqslant \frac{H}{H+1}$ is the discount factor for transitioning to the next layer.

This parameter setting satisfies the boundedness assumption of linear MDPs (refer Definition 2). First, we show that $\|\psi(s, \mathbf{a})\|_2 \leqslant 1$:

$$
\|\psi(s, \mathbf{a})\|_2^2 \leqslant \alpha^2 + \frac{d^{\text{lin}} - 5}{2}\beta^2 + \frac{1}{2} = 1, \qquad \text{(the first and second cases of Equation F.1),}
$$

$$
\|\psi(s, \mathbf{a})\|_2^2 = 1, \qquad \text{(the third case of Equation F.1),}
$$

$$
\|\psi(s, \mathbf{a})\|_2^2 \leqslant \frac{1}{2} + \frac{1}{2} = 1, \qquad \text{(the fourth case of Equation F.1),}
$$

Moreover, since $d^{\text{lin}} \geqslant 6$ and $K \geqslant 13(d^{\text{lin}} - 5)^2/H$, we ensure that $\max\{\|\sum_{s \in S} \boldsymbol{\mu}_h(s)\|_2, \|\mathbf{w}_h^\star\|_2\} \leqslant \sqrt{d^{\text{lin}}}$:

$$
\left\| \sum_{s \in S} |\boldsymbol{\mu}_h(s)| \right\|_2^2 = \frac{2(1-\delta)^2 + 2\delta^2}{\alpha^2} + \frac{\|\boldsymbol{\mu}_h\|_2}{\beta^2} + 3
$$
$$
\leqslant 2\left(2 + \Delta \cdot (d^{\text{lin}} - 5)\right) + 2\Delta \cdot (d^{\text{lin}} - 5)\left(2 + \Delta \cdot (d^{\text{lin}} - 5)\right)
$$
$$
\leqslant \left(2 + 2\Delta \cdot (d^{\text{lin}} - 5)\right)^2 \leqslant d^{\text{lin}},
$$
$$
\text{and} \quad \|\mathbf{w}_h^\star\|_2^2 \leqslant \frac{2}{H^2} \leqslant d^{\text{lin}}.
$$

## F.2 CONSTRUCTION OF MNL PREFERENCE MODEL

Inspired by the lower bound proposed in Lee & Oh (2024), we construct an adversarial setting for the MNL preference model.

We assume that $d \geqslant 2$ and that $d - 1$ is divisible by 4 (without loss of generality). Let $\epsilon \in \left(0, \frac{1}{(d-1)\sqrt{d-1}}\right)$ be a small positive parameter. Throughout the proof, we set $\epsilon = \sqrt{\frac{d-1}{144C \cdot K} \cdot \frac{(H+1)^2}{H}}$, for some $C > 0$. For every subset $W \subseteq [d-1]$, we define the corresponding parameter $\boldsymbol{\theta}_W \in \mathbb{R}^{d-1}$ as $[\boldsymbol{\theta}_W]_j = \epsilon$ for all $j \in W$, and $[\boldsymbol{\theta}_W]_j = 0$ for all $j \notin W$.

Next, for any $h \in [H]$, we define the parameter set as:

$$\boldsymbol{\theta}_h^\star \in \Theta := \{(\boldsymbol{\theta}_W^\top, -\log H)^\top : W \in \mathcal{W}_{(d-1)/4}\}$$
$$= \{(\boldsymbol{\theta}_W^\top, -\log H)^\top : W \subseteq [d-1], |W| = (d-1)/4\},$$

where $\mathcal{W}_k$ denotes the class of all subsets of $[d-1]$ of size $k$.

The feature vector $\phi(s, \mathbf{a})$ is invariant across the state $s$. For each $U \in \mathcal{W}_{(d-1)/4}$, we define vectors $z_U \in \mathbb{R}^{d-1}$ as follows:

$$[z_U]_j = 1/\sqrt{d-1} \quad \text{for } j \in U; \quad [z_U]_j = 0 \quad \text{for } j \notin U.$$

Let $\mathcal{Z} := \{z_U : U \in \mathcal{W}_{(d-1)/4}\}$. We define the function $Z : \mathcal{I} \to \mathcal{Z}$, so that $Z(\mathbf{a}) \in \mathcal{Z}$. Then, the feature vector $\phi(s, \mathbf{a})$ is constructed as follows:

$$\phi(s, \mathbf{a}) = \begin{cases} (Z(\mathbf{a})^\top, 0)^\top, & \mathbf{a} \neq \mathbf{a}_0; \\ (\mathbf{0}, 1)^\top, & \mathbf{a} = \mathbf{a}_0, \end{cases}$$

where $\mathbf{0} \in \mathbb{R}^{d-1}$. For all $V \in \mathcal{V}_{d/4}$ and $(s, \mathbf{a}) \in \mathcal{S} \times \mathcal{I}$, it can be verified that $\boldsymbol{\theta}_V$ and $\phi(s, \mathbf{a})$ satisfy the boundedness in Assumption 1 as follows:

$$\|\phi(s, \mathbf{a})\|_2 \leqslant \sqrt{(d-1) \cdot 1/(d-1)} = 1,$$
$$\|\boldsymbol{\theta}_h^\star\|_2 \leqslant \sqrt{(d-1)\epsilon^2 + (-\log H)^2} \leqslant \sqrt{2} \log H =: B.$$

Let $\mathbf{a}_h^\star$ (defined in the previous subsection) also have the maximum utility, i.e., $\mathbf{a}_h^\star \in \arg\max_{a \in \mathcal{I} \setminus \{\mathbf{a}_0\}} \langle \boldsymbol{\theta}_h^\star, \phi(s, \mathbf{a}) \rangle$ (note that $\phi(\cdot, \cdot)$ is identical for all $s \in \mathcal{S}$).

### F.3 PROOF OF THEOREM 3

A good policy is one that quickly reaches the state $x_{H+2}^{(i)}$ while remaining in the lower layers (i.e., with lower $i$). Recall that the item $\mathbf{a}_h^\star$ has the highest utility and, therefore, the highest choice probability. It also has the best chance of quickly reaching the state $x_{H+2}^{(i)}$ while staying within the same layer. In other words, a good policy encourages the user to frequently select the item $\mathbf{a}_h^\star \in \arg\max_{a \in \mathcal{I} \setminus \{\mathbf{a}_0\}} \langle \boldsymbol{\mu}_h, \mathbf{a} \rangle = \arg\max_{a \in \mathcal{I} \setminus \{\mathbf{a}_0\}} \langle \boldsymbol{\theta}_h^\star, \phi(s, \mathbf{a}) \rangle$. Note that $\mathbf{a}_h^\star$ is unique due to the way the action space and transition probabilities are constructed.

*Proof of Theorem 3.* Fix $\boldsymbol{\theta}$ and $\boldsymbol{\mu}$ so that we can omit the parameter dependency of $\mathbb{P}$ and $\mathcal{P}$ throughout the proof. Based on the construction of the hard instance $\mathcal{M}$ discussed in the previous subsections, the following lemma shows that the optimal assortment at horizon $h \in [H]$ is $\{\mathbf{a}_0, \mathbf{a}_h^\star\}$.

**Lemma F.1.** *For any $h \in [H]$, we have $A_h^\star = \{\mathbf{a}_0, \mathbf{a}_h^\star\}$.*

Furthermore, we can bound the expected value of $\overline{Q}^\star$ for any assortment as follows:

**Lemma F.2.** *For any $(A, i, h) \in \mathcal{A} \times [H] \times [H]$, let $\tilde{\mathbf{a}}_h^{(i)} \in \arg\max_{\mathbf{a} \in A \setminus \{\mathbf{a}_0\}} \phi(x_h^{(i)}, \mathbf{a})^\top \boldsymbol{\theta}_h^\star$, $\tilde{A}_h^{(i)} = \{\tilde{\mathbf{a}}_h^{(i)}, \mathbf{a}_0\}$, and $\bar{\mathbf{a}}_h^{(i)} \in \arg\max_{\mathbf{a} \in A \setminus \{\mathbf{a}_0\}} \overline{Q}_h^\pi(x_h^{(i)}, \mathbf{a})$. For any $\mathbf{a}' \neq \mathbf{a}_0$, we define*

$$\widetilde{Q}_h^\pi(x_h^{(i)}, \mathbf{a}_h^\star, \mathbf{a}')$$
$$:= \begin{cases} \frac{\gamma^{i-1}}{H} + \mathbb{P}_h(x_{h+1}^{(i)}|x_h^{(i)}, \mathbf{a}_h^\star)V_{h+1}^\pi(x_{h+1}^{(i)}) + \mathbb{P}_h(x_{H+2}^{(i)}|x_h^{(i)}, \mathbf{a}')\frac{(H-h)\gamma^{i-1}}{H}, & \mathbf{a}' = \mathbf{a}_h^\star, \\ \frac{\gamma^{i-1}}{H} + \mathbb{P}_h(x_{h+1}^{(i)}|x_h^{(i)}, \mathbf{a}_h^\star)V_{h+1}^\pi(x_{h+1}^{(i)}) + \mathbb{P}_h(x_{H+2}^{(i+2)}|x_h^{(i)}, \mathbf{a}')\frac{(H-h)\gamma^{i-1}}{H}, & \mathbf{a}' \neq \mathbf{a}_h^\star. \end{cases}$$

*Then, for any policy $\pi$, if $K \geqslant 4(d^{lin} - 5)^2 H(H+1)^2$, we have*

$$\sum_{\mathbf{a} \in A} \mathcal{P}_h(\mathbf{a}|x_h^{(i)}, A)\overline{Q}_h^\pi(x_h^{(i)}, \mathbf{a}) \leqslant \mathcal{P}_h(\tilde{\mathbf{a}}_h^{(i)}|x_h^{(i)}, \tilde{A}_h^{(i)})\widetilde{Q}_h^\pi(x_h^{(i)}, \mathbf{a}_h^\star, \bar{\mathbf{a}}_h^{(i)}).$$

Now, we are ready to provide the proof of Theorem 3.

For any $h \in [H]$ and any $A_h \in \mathcal{A}$, let $\tilde{\mathbf{a}}_h \in \arg\max_{\mathbf{a} \in A_h \setminus \{\mathbf{a}_0\}} \phi(x_h^{(i)}, \mathbf{a})^\top \boldsymbol{\theta}_h^\star$. We also denote $\tilde{A}_h = \{\tilde{\mathbf{a}}_h, \mathbf{a}_0\}$ and $\bar{\mathbf{a}}_h \in \arg\max_{\mathbf{a} \in A \setminus \{\mathbf{a}_0\}} \overline{Q}_h^\pi(x_h, \mathbf{a})$. Recall that the index $i$ can be omitted for $\tilde{\mathbf{a}}_h$

because the transition and choice probabilities are identical across all $x_h^{(1)}, \ldots x_h^{(H)}$ given $\mathbf{a} \in \mathcal{I}$. The change in layer $i$ only affects the scaling of rewards, and consequently the $\overline{Q}$-values, but the item that maximizes $\overline{Q}(x_h^{(i)}, \mathbf{a})$ remains the same across layers.

By applying Lemma F.2, the value of policy $\pi$ in state $x_1^{(1)}$ can be bounded as follows:

$$V_1^\pi(x_1^{(1)}) = \sum_{\mathbf{a} \in A_1} \mathcal{P}_1(\mathbf{a}|x_1^{(1)}, A_1) \overline{Q}_1^\pi(x_1^{(1)}, \mathbf{a}) \leqslant \mathcal{P}_1(\tilde{\mathbf{a}}_1|x_1^{(1)}, \tilde{A}_1) \widetilde{Q}_h^\pi(x_1^{(1)}, \mathbf{a}_1^\star, \bar{\mathbf{a}}_1). \qquad \text{(F.3)}$$

Moreover, according to Lemma F.1, the optimal assortment for horizon $h \in [H]$ is $A_h^\star = \{\mathbf{a}_0, \mathbf{a}_h^\star\}$. Thus, the optimal value function in state $x_1^{(1)}$ can be written as follows:

$$V_1^\star(x_1^{(1)}) = \sum_{\mathbf{a} \in A_1^\star} \mathcal{P}_1(\mathbf{a}|x_1^{(1)}, A_1^\star) \overline{Q}_1^\star(x_1^{(1)}, \mathbf{a}) = \mathcal{P}_1(\mathbf{a}_1^\star|x_1^{(1)}, A_1^\star) \overline{Q}_1^\star(x_1^{(1)}, \mathbf{a}_1^\star),$$

where the last equality holds because $\overline{Q}_h^\star(x_h^{(i)}, \mathbf{a}_0) = 0$. We denote $s_{H+2}$ can be either $x_{H+2}^{(1)}$ or $x_{H+2}^{(3)}$, depending on whether the item (for transition) is $\mathbf{a}_h^\star$ or any $\mathbf{a} \neq \mathbf{a}_h^\star, \mathbf{a}_0$. Then, we have

$$\begin{aligned}
&(V_1^\star - V_1^\pi)(x_1^{(1)}) \\
&\geqslant \mathcal{P}_1(\mathbf{a}_1^\star|x_1^{(1)}, A_1^\star) \overline{Q}_1^\star(x_1^{(1)}, \mathbf{a}_1^\star) - \mathcal{P}_1(\tilde{\mathbf{a}}_1|x_1^{(1)}, \tilde{A}_1) \widetilde{Q}_h^\pi(x_1^{(1)}, \mathbf{a}_1^\star, \bar{\mathbf{a}}_1) \\
&= \left( \mathcal{P}_1(\mathbf{a}_1^\star|x_1^{(1)}, A_1^\star) - \mathcal{P}_1(\tilde{\mathbf{a}}_1|x_1^{(1)}, \tilde{A}_1) \right) \overline{Q}_1^\star(x_1^{(1)}, \mathbf{a}_1^\star) \\
&\quad + \mathcal{P}_1(\tilde{\mathbf{a}}_1|x_1^{(1)}, \tilde{A}_1) \left( \overline{Q}_1^\star(x_1^{(1)}, \mathbf{a}_1^\star) - \widetilde{Q}_h^\pi(x_1^{(1)}, \mathbf{a}_1^\star, \bar{\mathbf{a}}_1) \right) \\
&= \left( \mathcal{P}_1(\mathbf{a}_1^\star|x_1^{(1)}, A_1^\star) - \mathcal{P}_1(\tilde{\mathbf{a}}_1|x_1^{(1)}, \tilde{A}_1) \right) \overline{Q}_1^\star(x_1^{(1)}, \mathbf{a}_1^\star) \\
&\quad + \mathcal{P}_1(\tilde{\mathbf{a}}_1|x_1^{(1)}, \tilde{A}_1) \left( \frac{1}{H} + \mathbb{P}_1(x_2^{(1)}|x_1^{(1)}, \mathbf{a}_1^\star) V_2^\star(x_2^{(1)}) + \mathbb{P}_1(x_{H+2}^{(1)}|x_1^{(1)}, \mathbf{a}_1^\star) \frac{(H-1)}{H} \right. \\
&\qquad \left. - \left( \frac{1}{H} + \mathbb{P}_1(x_2^{(1)}|x_1^{(1)}, \mathbf{a}_1^\star) V_2^\pi(x_2^{(1)}) + \mathbb{P}_1(s_{H+2}|x_1^{(1)}, \bar{\mathbf{a}}_1) \frac{(H-1)}{H} \right) \right) \\
&= \left( \mathcal{P}_1(\mathbf{a}_1^\star|x_1^{(1)}, A_1^\star) - \mathcal{P}_1(\tilde{\mathbf{a}}_1|x_1^{(1)}, \tilde{A}_1) \right) \overline{Q}_1^\star(x_1^{(1)}, \mathbf{a}_1^\star) \\
&\quad + \mathcal{P}_1(\tilde{\mathbf{a}}_1|x_1^{(1)}, \tilde{A}_1) \mathbb{P}_1(x_2^{(1)}|x_1^{(1)}, \mathbf{a}_1^\star)(V_2^\star - V_2^\pi)(x_2^{(1)}) \\
&\quad + \mathcal{P}_1(\tilde{\mathbf{a}}_1|x_1^{(1)}, \tilde{A}_1) \left( \mathbb{P}_1(x_{H+2}^{(1)}|x_1^{(1)}, \mathbf{a}_1^\star) - \mathbb{P}_1(s_{H+2}|x_1^{(1)}, \bar{\mathbf{a}}_1) \right) \frac{(H-1)}{H}, \qquad \text{(F.4)}
\end{aligned}$$

where the first inequality holds by Equation F.3. Note that, by construction, for any $h \in [H]$, we have

$$\overline{Q}_h^\star(x_h^{(1)}, \mathbf{a}_h^\star) = \frac{H - h + 1}{H},$$

$$\mathcal{P}_h(\tilde{\mathbf{a}}_h|x_h^{(1)}, \tilde{A}_h) \geqslant \frac{1}{1/H + 1} = \frac{H}{1 + H},$$

$$\mathbb{P}_h(x_{h+1}^{(1)}|x_h^{(1)}, \mathbf{a}_h^\star) = 1 - \delta - (d^{\text{lin}} - 5)\Delta,$$

$$\mathbb{P}_h(x_{H+2}^{(1)}|x_h^{(1)}, \mathbf{a}_h^\star) - \mathbb{P}_h(s_{H+2}|x_h^{(1)}, \bar{\mathbf{a}}_h) = (d^{\text{lin}} - 5)\Delta - \langle \boldsymbol{\mu}_h, \bar{\mathbf{a}}_h \rangle. \qquad \text{(F.5)}$$

Hence, by plugging Equation F.5 into Equation F.4 and applying recursion, we get

$$(V_1^\star - V_1^\pi)(x_1^{(1)})$$

$$\geqslant \sum_{h=1}^{H} \left( \mathcal{P}_h(\mathbf{a}_h^\star | x_h^{(1)}, A_h^\star) - \mathcal{P}_h(\tilde{\mathbf{a}}_h | x_h^{(1)}, \tilde{A}_h) \right) \frac{H - h + 1}{H} \cdot \left( \frac{H}{H+1} \right)^{h-1}$$

$$\cdot \left( (1 - \delta - (d^{\text{lin}} - 5)\Delta) \right)^{h-1}$$

$$+ \sum_{h=1}^{H} \left( (d^{\text{lin}} - 5)\Delta - \langle \boldsymbol{\mu}_h, \bar{\mathbf{a}}_h \rangle \right) \frac{H - h}{H} \cdot \left( \frac{H}{H+1} \right)^{h} \cdot \left( (1 - \delta - (d^{\text{lin}} - 5)\Delta) \right)^{h-1}$$

$$- \sum_{h=1}^{H} \frac{(H - h)}{H\sqrt{K}} \cdot \left( \frac{H}{H+1} \right)^{h} \cdot \left( (1 - \delta - (d^{\text{lin}} - 5)\Delta) \right)^{h-1}. \tag{F.6}$$

Furthermore, since $H \geqslant 3$ and $3(d^{\text{lin}} - 5)\Delta \leqslant \delta = 1/H$, we have

$$\left( \frac{H}{H+1} \right)^{h} \geqslant \left( \frac{H}{H+1} \right)^{H+1} \geqslant \frac{3}{10},$$

$$\left( (1 - \delta - (d^{\text{lin}} - 5)\Delta) \right)^{h-1} \geqslant \left( 1 - \frac{4\delta}{3} \right)^{H} \geqslant \frac{1}{3}. \tag{F.7}$$

Therefore, by substituting Equation F.7 into Equation F.6, and considering the terms where $h \geqslant H/2$, we obtain

$$(V_1^\star - V_1^\pi)(x_1^{(1)})$$

$$\geqslant \frac{1}{20} \sum_{h=1}^{H/2} \left( \mathcal{P}_h(\mathbf{a}_h^\star | x_h^{(1)}, A_h^\star) - \mathcal{P}_h(\tilde{\mathbf{a}}_h | x_h^{(1)}, \tilde{A}_h) \right) + \frac{1}{20} \sum_{h=1}^{H/2} \left( (d^{\text{lin}} - 5)\Delta - \langle \boldsymbol{\mu}_h, \bar{\mathbf{a}}_h \rangle \right)$$

$$= \frac{1}{20} \underbrace{\sum_{h=1}^{H/2} \left( \mathcal{P}_h(\mathbf{a}_h^\star | x_h^{(1)}, A_h^\star) - \mathcal{P}_h(\tilde{\mathbf{a}}_h | x_h^{(1)}, \tilde{A}_h) \right)}_{\text{MNL bandit regret}} + \frac{1}{20} \sum_{h=1}^{H/2} \underbrace{\left( \max_{\mathbf{a} \in \mathcal{I}} \langle \boldsymbol{\mu}_h, \mathbf{a} \rangle - \langle \boldsymbol{\mu}_h, \bar{\mathbf{a}}_h \rangle \right)}_{\text{linear bandit regret}}. \tag{F.8}$$

On the right-hand side of Equation F.8, the first term corresponds to an MNL bandit problem. Recall that $|A_h^\star| = |\tilde{A}_h| = 2$ and, by construction, we have

$$\mathcal{P}_h(\mathbf{a}_h^\star | x_h^{(1)}, A_h^\star) = \frac{\exp\left( \phi(x_h^{(1)}, \mathbf{a}_h^\star)^\top \boldsymbol{\theta}_h^\star \right)}{1/H + \exp\left( \phi(x_h^{(1)}, \mathbf{a}_h^\star)^\top \boldsymbol{\theta}_h^\star \right)},$$

$$\mathcal{P}_h(\tilde{\mathbf{a}}_h | x_h^{(1)}, \tilde{A}_h) = \frac{\exp\left( \phi(x_h^{(1)}, \tilde{\mathbf{a}}_h)^\top \boldsymbol{\theta}_h^\star \right)}{1/H + \exp\left( \phi(x_h^{(1)}, \tilde{\mathbf{a}}_h)^\top \boldsymbol{\theta}_h^\star \right)}.$$

Hence, this corresponds to an MNL bandit problem with a maximum assortment size of $M = 2$, where the *attraction parameter for the outside option* (the constant in the denominator) is $1/H$.

Furthermore, the second term on the right-hand side of Equation F.8 represents a linear bandit problem. To sum up, the learning problem is not harder than minimizing the regret on $\Omega(H/2)$ MNL and linear bandit problems.

To bound each term of Equation F.8, we introduce the following propositions:

**Proposition F.1** (Regret lower bound of MNL bandits, Lee & Oh 2024)**.** *Let $v_0$ denote the attraction parameter for the outisde option. Let $d$ be divisible by $4$. Suppose $K \geqslant C \cdot d^4 M/(M - 1)$ for some constant $C > 0$. Then, in the uniform reward setting (where rewards are identical) with the reward for the outside option being zero, for any policy and the MNL preference model parameterized by $\boldsymbol{\theta}$, there exists a worst-case problem instance such that the worst-case expected regret is lower bounded as follows:*

$$\sup_{\boldsymbol{\theta}} \mathbb{E}_{\boldsymbol{\theta}} \left[ \mathbf{MNLBanditRegret}(\boldsymbol{\theta}, K) \right] = \Omega \left( \frac{\sqrt{v_0(M - 1)}}{v_0 + M - 1} \cdot d\sqrt{K} \right).$$

**Proposition F.2** (Lemma C.8 in Zhou et al. 2021a). *Fix $0 < \delta < 1/3$. Consider the linear bandit problem parameterized with a vector $\boldsymbol{\mu} \in \{-\Delta, \Delta\}^d$ and action set $\mathcal{I} = \{-1, 1\}^d$. And the reward distribution for taking action $a \in \mathcal{I}$ is a Bernoulli distribution denoted as $B(\delta + \langle \boldsymbol{\mu}, \mathbf{a} \rangle)$. Let $K$ be the number of time steps playing this bandit problem. Assume $K \geqslant d^2/(2\delta)$ and $\Delta = \sqrt{\delta/K}/(4\sqrt{2})$. Then, for any bandit algorithm $\mathcal{B}$, there exists $\boldsymbol{\mu}$ such that the expected pseudo-regret of $\mathcal{B}$ over $K$ steps is lower bounded as follows:*

$$\mathbb{E}_{\boldsymbol{\mu}}[\mathbf{LinearBanditRegret}(\boldsymbol{\mu}, K)] \geqslant \frac{d\sqrt{K\delta}}{8\sqrt{2}}.$$

*where the expectation is with respect to the reward distribution that depends on $\mu$.*

Now, by using Proposition F.1 and F.2, we can bound the regret as follows:

$$\sup_{\boldsymbol{\theta}, \boldsymbol{\mu}, \mathbf{w}} \mathbb{E}_{\boldsymbol{\theta}, \boldsymbol{\mu}, \mathbf{w}} \left[ \mathbf{Regret}\left( \mathcal{M}_{\boldsymbol{\theta}, \boldsymbol{\mu}, \mathbf{w}}, K \right) \right]$$

$$\geqslant \frac{1}{20} \sum_{h=1}^{H/2} \sup_{\boldsymbol{\theta}} \mathbb{E}_{\boldsymbol{\theta}} \left[ \sum_{k=1}^{K} \left( \mathcal{P}_h(\mathbf{a}_h^\star | x_h^{(1)}, A_h^\star) - \mathcal{P}_h(\tilde{\mathbf{a}}_h | x_h^{(1)}, \tilde{A}_h) \right) \right]$$

$$+ \frac{1}{20} \sum_{h=1}^{H/2} \sup_{\boldsymbol{\mu}} \mathbb{E}_{\boldsymbol{\mu}} \left[ \sum_{k=1}^{K} \left( \max_{\mathbf{a} \in \mathcal{I}} \langle \boldsymbol{\mu}_h, \mathbf{a} \rangle - \langle \boldsymbol{\mu}_h, \bar{\mathbf{a}}_h \rangle \right) \right]$$

$$= \Omega \left( d\sqrt{HK} + d^{\mathrm{lin}}\sqrt{HK} \right),$$

where, in the last equality, we use $v_0 = 1/H$, $M = 2$, and $\delta = 1/H$. This concludes the proof of Theorem 3. $\qquad\square$

### F.4 PROOF OF LEMMAS FOR THEOREM 3

#### F.4.1 PROOF OF LEMMA F.1

*Proof of Lemma F.1.* For any $i \in [H]$, we can write the optimal $\overline{Q}$-value in state $x_h^{(i)}$ at horizon $h \in [H]$ as follows:

$$\overline{Q}_h^\star(x_h^{(i)}, \mathbf{a})$$
$$= \begin{cases} \frac{\gamma^{i-1}}{H} + \mathbb{P}_h(x_{h+1}^{(i)} | x_h^{(i)}, \mathbf{a}) V_{h+1}^\star(x_{h+1}^{(i)}) + \mathbb{P}_h(x_{H+2}^{(i)} | x_h^{(i)}, \mathbf{a}) \frac{(H-h)\gamma^{i-1}}{H}, & \mathbf{a} = \mathbf{a}_h^\star; \\ \frac{\gamma^i}{H} + \mathbb{P}_h(x_{h+1}^{(i+1)} | x_h^{(i)}, \mathbf{a}) V_{h+1}^\star(x_{h+1}^{(i+1)}) + \mathbb{P}_h(x_{H+2}^{(i+2)} | x_h^{(i)}, \mathbf{a}) \frac{(H-h)\gamma^{i+1}}{H}, & \mathbf{a} = \mathbf{a}_h^\star, \mathbf{a}_0; \\ 0, & \mathbf{a} = \mathbf{a}_0. \end{cases}$$

First, we show that for any $(i, h) \in [H] \times [H]$, we have

$$\overline{Q}_h^\star(x_h^{(i)}, \mathbf{a}) \geqslant \sum_{\mathbf{a}' \in A_h^\star} \mathcal{P}_h(\mathbf{a}' | x_h^{(i)}, A_h^\star) \overline{Q}_h^\star(x_h^{(i)}, \mathbf{a}'), \quad \forall \mathbf{a} \in A_h^\star \backslash \{\mathbf{a}_0\}. \tag{F.9}$$

We prove this by contradiction. Suppose there exists $\mathbf{a} \in A_h^\star$ such that $\overline{Q}_h^\star(x_h^{(i)}, \mathbf{a}) < \sum_{\mathbf{a}' \in A_h^\star} \mathcal{P}_h(\mathbf{a}' | x_h^{(i)}, A_h^\star) \overline{Q}_h^\star(x_h^{(i)}, \mathbf{a}')$. In that case, removing the item $\mathbf{a}$ from the assortment $A_h^\star$ results in a higher expected value of $\overline{Q}_h^\star$. This contradicts the optimality of $A_h^\star$. Therefore, Equation F.9 must hold.

By the definition of $\overline{Q}_h^\star(x_h^{(i)}, \mathbf{a})$, for any $\mathbf{a} \in \mathcal{I} \backslash \{\mathbf{a}_h^\star\}$, we have

$$\overline{Q}_h^\star(x_h^{(i)}, \mathbf{a}) \leqslant \frac{\gamma^{i-1}}{H} + \mathbb{P}_h(x_{h+1}^{(i+1)} | x_h^{(i)}, \mathbf{a}) V_{h+1}^\star(x_{h+1}^{(i)}) + \mathbb{P}_h(x_{H+2}^{(i+2)} | x_h^{(i)}, \mathbf{a}) \frac{(H-h)\gamma^{i-1}}{H}$$

$$\leqslant \frac{\gamma^{i-1}}{H} + \mathbb{P}_h(x_{h+1}^{(i)} | x_h^{(i)}, \mathbf{a}_h^\star) V_{h+1}^\star(x_{h+1}^{(i)}) + \mathbb{P}_h(x_{H+2}^{(i)} | x_h^{(i)}, \mathbf{a}_h^\star) \frac{(H-h)\gamma^{i-1}}{H}$$

$$= \overline{Q}_h^\star(x_h^{(i)}, \mathbf{a}_h^\star),$$

where the first inequality holds since $V^\star_{h+1}(x^{(i+1)}_{h+1}) \leqslant V^\star_{h+1}(x^{(i)}_{h+1})$, and the second inequality holds due to the fact that $V^\star_{h+1}(x^{(i)}_{h+1}) \leqslant \frac{(H-h)\gamma^{i-1}}{H}$ and $\mathbb{P}_h(x^{(i+2)}_{H+2}|x^{(i)}_h, \mathbf{a}) \leqslant \mathbb{P}_h(x^{(i)}_{H+2}|x^{(i)}_h, \mathbf{a}^\star_h)$.

Since $\overline{Q}^\star_h(x^{(i)}_h, \mathbf{a}^\star_h)$ has the highest value among all items, the optimal assortment $A^\star_h$ should include $\mathbf{a}^\star_h$. Thus, we have $\mathbf{a}^\star_h, \mathbf{a}_0 \in A^\star_h$. In other words, when $A^\star_h = \{\mathbf{a}^\star_h, \mathbf{a}_0\}$, the condition in Equation F.9 is satisfied. Thus, we begin with $A^\star_h = \{\mathbf{a}^\star_h, \mathbf{a}_0\}$ and check if there exist an item $\mathbf{a} \neq \mathbf{a}^\star_h, \mathbf{a}_0$ that can increase the expected value of $\overline{Q}^\star_h$. To this end, for $A^\star_h = \{\mathbf{a}^\star_h, \mathbf{a}_0\}$, we get

$$\sum_{\mathbf{a}' \in A^\star_h} \mathcal{P}_h(\mathbf{a}'|x^{(i)}_h, A^\star_h)\overline{Q}^\star_h(x^{(i)}_h, \mathbf{a}') = \mathcal{P}_h(\mathbf{a}^\star_h|x^{(i)}_h, A^\star_h)\overline{Q}^\star_h(x^{(i)}_h, \mathbf{a}^\star_h) \geqslant \gamma \overline{Q}^\star_h(x^{(i)}_h, \mathbf{a}^\star_h), \quad \text{(F.10)}$$

where the equality holds since $\overline{Q}^\star_h(x^{(i)}_h, a_0) = 0$, and the inequality holds by the definition of $\gamma$:

$$\gamma = \frac{H}{1+H} = \frac{1}{1/H + 1} \leqslant \min_{h \in [H]} \min_{s \in \mathcal{S}} \min_{A \in \mathcal{A}} \min_{\mathbf{a} \in A \setminus \{a_0\}} \frac{\exp\left(\phi(s, \mathbf{a})^\top \boldsymbol{\theta}^\star_h\right)}{1/H + \exp\left(\phi(s, \mathbf{a})^\top \boldsymbol{\theta}^\star_h\right)}$$

$$\leqslant \frac{\exp\left(\phi(x^{(i)}_h, \mathbf{a}^\star_h)^\top \boldsymbol{\theta}^\star_h\right)}{1/H + \exp\left(\phi(x^{(i)}_h, \mathbf{a}^\star_h)^\top \boldsymbol{\theta}^\star_h\right)} = \mathcal{P}_h(\mathbf{a}^\star_h|x^{(i)}_h, A^\star_h). \quad \text{(F.11)}$$

Here, we rely on the fact that the sigmoid function is monotonically increasing to establish the inequalities.

On the other hand, for any item $\mathbf{a} \neq \mathbf{a}^\star_h, \mathbf{a}_0$, we have

$$\overline{Q}^\star_h(x^{(i)}_h, \mathbf{a}) = \frac{\gamma^i}{H} + \mathbb{P}_h(x^{(i+1)}_{h+1}|x^{(i)}_h, \mathbf{a})V^\star_{h+1}(x^{(i+1)}_{h+1}) + \mathbb{P}_h(x^{(i+2)}_{H+2}|x^{(i)}_h, \mathbf{a})\frac{(H-h)\gamma^{i+1}}{H}$$

$$\leqslant \frac{\gamma^i}{H} + \mathbb{P}_h(x^{(i+1)}_{h+1}|x^{(i)}_h, \mathbf{a})V^\star_{h+1}(x^{(i+1)}_{h+1}) + \mathbb{P}_h(x^{(i+2)}_{H+2}|x^{(i)}_h, \mathbf{a})\frac{(H-h)\gamma^i}{H}$$

$$\leqslant \frac{\gamma^i}{H} + \mathbb{P}_h(x^{(i)}_{h+1}|x^{(i)}_h, \mathbf{a}^\star_h)V^\star_{h+1}(x^{(i+1)}_{h+1}) + \mathbb{P}_h(x^{(i)}_{H+2}|x^{(i)}_h, \mathbf{a}^\star_h)\frac{(H-h)\gamma^i}{H}$$

$$= \gamma \cdot \left(\frac{\gamma^{i-1}}{H} + \mathbb{P}_h(x^{(i)}_{h+1}|x^{(i)}_h, \mathbf{a}^\star_h)V^\star_{h+1}(x^{(i)}_{h+1}) + \mathbb{P}_h(x^{(i)}_{H+2}|x^{(i)}_h, \mathbf{a}^\star_h)\frac{(H-h)\gamma^{i-1}}{H}\right)$$

$$= \gamma \overline{Q}^\star_h(x^{(i)}_h, \mathbf{a}^\star_h), \quad \text{(F.12)}$$

where the second inequality holds because $V^\star_{h+1}(x^{(i+1)}_{h+1}) \leqslant \frac{(H-h)\gamma^i}{H}$, $\mathbb{P}_h(x^{(i+2)}_{H+2}|x^{(i)}_h, \mathbf{a}) \leqslant \mathbb{P}_h(x^{(i)}_{H+2}|x^{(i)}_h, \mathbf{a}^\star_h)$, and the second equality follows from the fact that $\gamma V^\star_{h+1}(x^{(i)}_{h+1}) = V^\star_{h+1}(x^{(i+1)}_{h+1})$ by construction.

Combining Equation F.10 and Equation F.12, when $A^\star_h = \{\mathbf{a}^\star_h, \mathbf{a}_0\}$, for any item $\mathbf{a} \neq \mathbf{a}^\star_h, \mathbf{a}_0$, we get

$$\sum_{\mathbf{a}' \in A^\star_h} \mathcal{P}_h(\mathbf{a}'|x^{(i)}_h, A^\star_h)\overline{Q}^\star_h(x^{(i)}_h, \mathbf{a}') \geqslant \overline{Q}^\star_h(x^{(i)}_h, \mathbf{a}).$$

Since $\overline{Q}^\star_h(x^{(i)}_h, \mathbf{a})$ for $\mathbf{a} \neq \mathbf{a}^\star_h, \mathbf{a}_0$ is not greater than the expected value of $\overline{Q}^\star_h$ for $A^\star_h$, adding any item $\mathbf{a} \neq \mathbf{a}^\star_h, \mathbf{a}_0$ to $A^\star_h$ does not increase the expected value of $\overline{Q}^\star_h$. This confirms the optimality of $A^\star_h$. □

### F.4.2 PROOF OF LEMMA F.2

*Proof of Lemma F.2.* For any $i \in [H]$, we can write the $\overline{Q}$-value for the policy $\pi$ in state $x^{(i)}_h$ at horizon $h \in [H]$ as follows:

$$\overline{Q}^\pi_h(x^{(i)}_h, \mathbf{a})$$
$$= \begin{cases} \frac{\gamma^{i-1}}{H} + \mathbb{P}_h(x^{(i)}_{h+1}|x^{(i)}_h, \mathbf{a})V^\pi_{h+1}(x^{(i)}_{h+1}) + \mathbb{P}_h(x^{(i)}_{H+2}|x^{(i)}_h, \mathbf{a})\frac{(H-h)\gamma^{i-1}}{H}, & \mathbf{a} = \mathbf{a}^\star_h; \\ \frac{\gamma^i}{H} + \mathbb{P}_h(x^{(i+1)}_{h+1}|x^{(i)}_h, \mathbf{a})V^\pi_{h+1}(x^{(i+1)}_{h+1}) + \mathbb{P}_h(x^{(i+2)}_{H+2}|x^{(i)}_h, \mathbf{a})\frac{(H-h)\gamma^{i+1}}{H}, & \mathbf{a} = \mathbf{a}^\star_h, \mathbf{a}_0; \\ 0, & \mathbf{a} = \mathbf{a}_0. \end{cases}$$

We provide a proof by considering the following cases:

**Case (i)** $\mathbf{a}_h^\star \in A$.

Recall that, by Equation F.11, we have

$$\gamma = \frac{H}{1+H} \leqslant \frac{\exp\left(\phi(x_h^{(i)}, \mathbf{a}_h^\star)^\top \boldsymbol{\theta}_h^\star\right)}{1/H + \exp\left(\phi(x_h^{(i)}, \mathbf{a}_h^\star)^\top \boldsymbol{\theta}_h^\star\right)}. \tag{F.13}$$

By multiplying $\widetilde{Q}_h^\pi(x_h^{(i)}, \mathbf{a}_h^\star, \mathbf{a}')$ on both sides of Equation F.13, we get

$$\gamma \cdot \widetilde{Q}_h^\pi(x_h^{(i)}, \mathbf{a}_h^\star, \mathbf{a}') \leqslant \frac{\exp\left(\phi(x_h^{(i)}, \mathbf{a}_h^\star)^\top \boldsymbol{\theta}_h^\star\right) \widetilde{Q}_h^\pi(x_h^{(i)}, \mathbf{a}_h^\star, \mathbf{a}')}{1/H + \exp\left(\phi(x_h^{(i)}, \mathbf{a}_h^\star)^\top \boldsymbol{\theta}_h^\star\right)}$$

$$\Leftrightarrow \left(\sum_{\mathbf{a}\in A\backslash\{\mathbf{a}_h^\star, \mathbf{a}_0\}} \exp\left(\phi(x_h^{(i)}, \mathbf{a})^\top \boldsymbol{\theta}_h^\star\right)\right) \gamma \cdot \widetilde{Q}_h^\pi(x_h^{(i)}, \mathbf{a}_h^\star, \mathbf{a}') \left(1/H + \exp\left(\phi(x_h^{(i)}, \mathbf{a}_h^\star)^\top \boldsymbol{\theta}_h^\star\right)\right)$$

$$\leqslant \left(\sum_{\mathbf{a}\in A\backslash\{\mathbf{a}_h^\star, \mathbf{a}_0\}} \exp\left(\phi(x_h^{(i)}, \mathbf{a})^\top \boldsymbol{\theta}_h^\star\right)\right) \exp\left(\phi(x_h^{(i)}, \mathbf{a}_h^\star)^\top \boldsymbol{\theta}_h^\star\right) \widetilde{Q}_h^\pi(x_h^{(i)}, \mathbf{a}_h^\star, \mathbf{a}')$$

$$\Leftrightarrow \left(\exp\left(\phi(x_h^{(i)}, \mathbf{a}_h^\star)^\top \boldsymbol{\theta}_h^\star\right) + \sum_{\mathbf{a}\in A\backslash\{\mathbf{a}_h^\star, \mathbf{a}_0\}} \gamma \cdot \exp\left(\phi(x_h^{(i)}, \mathbf{a})^\top \boldsymbol{\theta}_h^\star\right)\right) \widetilde{Q}_h^\pi(x_h^{(i)}, \mathbf{a}_h^\star, \mathbf{a}')$$

$$\cdot \left(1/H + \exp\left(\phi(x_h^{(i)}, \mathbf{a}_h^\star)^\top \boldsymbol{\theta}_h^\star\right)\right)$$

$$\leqslant \left(1/H + \exp\left(\phi(x_h^{(i)}, \mathbf{a}_h^\star)^\top \boldsymbol{\theta}_h^\star\right) + \sum_{\mathbf{a}\in A\backslash\{\mathbf{a}_h^\star, \mathbf{a}_0\}} \exp\left(\phi(x_h^{(i)}, \mathbf{a})^\top \boldsymbol{\theta}_h^\star\right)\right) \exp\left(\phi(x_h^{(i)}, \mathbf{a}_h^\star)^\top \boldsymbol{\theta}_h^\star\right)$$

$$\cdot \widetilde{Q}_h^\pi(x_h^{(i)}, \mathbf{a}_h^\star, \mathbf{a}')$$

$$\Leftrightarrow \frac{\exp\left(\phi(x_h^{(i)}, \mathbf{a}_h^\star)^\top \boldsymbol{\theta}_h^\star\right) \widetilde{Q}_h^\pi(x_h^{(i)}, \mathbf{a}_h^\star, \mathbf{a}') + \sum_{\mathbf{a}\in A\backslash\{\mathbf{a}_h^\star, \mathbf{a}_0\}} \exp\left(\phi(x_h^{(i)}, \mathbf{a})^\top \boldsymbol{\theta}_h^\star\right) \gamma \widetilde{Q}_h^\pi(x_h^{(i)}, \mathbf{a}_h^\star, \mathbf{a}')}{1/H + \sum_{\mathbf{a}\in A\backslash\{\mathbf{a}_0\}} \exp\left(\phi(x_h^{(i)}, \mathbf{a})^\top \boldsymbol{\theta}_h^\star\right)}$$

$$\leqslant \frac{\exp\left(\phi(x_h^{(i)}, \mathbf{a}_h^\star)^\top \boldsymbol{\theta}_h^\star\right) \widetilde{Q}_h^\pi(x_h^{(i)}, \mathbf{a}_h^\star, \mathbf{a}')}{1/H + \exp\left(\phi(x_h^{(i)}, \mathbf{a}_h^\star)^\top \boldsymbol{\theta}_h^\star\right)}. \tag{F.14}$$

On the other hand, by the definition of $\widetilde{Q}_h^\pi(x_h^{(i)}, \mathbf{a}_h^\star, \mathbf{a}')$, for any $\mathbf{a}' \neq \mathbf{a}_h^\star, \mathbf{a}_0$, we have

$$\gamma \widetilde{Q}_h^\pi(x_h^{(i)}, \mathbf{a}_h^\star, \mathbf{a}')$$

$$= \gamma \cdot \left(\frac{\gamma^{i-1}}{H} + \mathbb{P}_h(x_{h+1}^{(i)}|x_h^{(i)}, \mathbf{a}_h^\star)V_{h+1}^\pi(x_{h+1}^{(i)}) + \mathbb{P}_h(x_{H+2}^{(i+2)}|x_h^{(i)}, \mathbf{a}')\frac{(H-h)\gamma^{i-1}}{H}\right)$$

$$= \frac{\gamma^i}{H} + \mathbb{P}_h(x_{h+1}^{(i)}|x_h^{(i)}, \mathbf{a}_h^\star)V_{h+1}^\pi(x_{h+1}^{(i+1)}) + \mathbb{P}_h(x_{H+2}^{(i+2)}|x_h^{(i)}, \mathbf{a}')\frac{(H-h)\gamma^i}{H}$$

$$\geqslant \frac{\gamma^i}{H} + \mathbb{P}_h(x_{h+1}^{(i)}|x_h^{(i)}, \mathbf{a}')V_{h+1}^\pi(x_{h+1}^{(i+1)}) + \mathbb{P}_h(x_{H+2}^{(i+2)}|x_h^{(i)}, \mathbf{a}')\frac{(H-h)\gamma^{i+1}}{H}$$

$$= \overline{Q}_h^\pi(x_h^{(i)}, \mathbf{a}'), \tag{F.15}$$

where the second equality holds since $\gamma V_{h+1}^{\pi}(x_{h+1}^{(i)}) = V_{h+1}^{\pi}(x_{h+1}^{(i+1)})$, and the inequality holds because, for $K \geqslant 4(d^{\mathrm{lin}} - 5)^2 H(H+1)^2$, the following inequality holds:

$$\mathbb{P}_h(x_{h+1}^{(i)}|x_h^{(i)}, \mathbf{a}')V_{h+1}^{\pi}(x_{h+1}^{(i+1)}) + \mathbb{P}_h(x_{H+2}^{(i+2)}|x_h^{(i)}, \mathbf{a}')\frac{(H-h)\gamma^{i+1}}{H}$$

$$\leqslant \mathbb{P}_h(x_{h+1}^{(i)}|x_h^{(i)}, \mathbf{a}_h^{\star})V_{h+1}^{\pi}(x_{h+1}^{(i+1)}) + \mathbb{P}_h(x_{H+2}^{(i+2)}|x_h^{(i)}, \mathbf{a}')\frac{(H-h)\gamma^{i}}{H}$$

$$\Leftrightarrow \left( \mathbb{P}_h(x_{h+1}^{(i)}|x_h^{(i)}, \mathbf{a}') - \mathbb{P}_h(x_{h+1}^{(i)}|x_h^{(i)}, \mathbf{a}_h^{\star}) \right) V_{h+1}^{\pi} \leqslant \mathbb{P}_h(x_{H+2}^{(i+2)}|x_h^{(i)}, \mathbf{a}')\frac{(H-h)}{H} \left( \gamma^i - \gamma^{i+1} \right).$$

Specifically, if the upper bound of the left-hand side is less than or equal to the lower bound of the right-hand side, the inequality holds. To demonstrate this, we have:

$$\left( \mathbb{P}_h(x_{h+1}^{(i)}|x_h^{(i)}, \mathbf{a}') - \mathbb{P}_h(x_{h+1}^{(i)}|x_h^{(i)}, \mathbf{a}_h^{\star}) \right) V_{h+1}^{\pi} \leqslant 2(d^{\mathrm{lin}} - 5)\Delta \cdot \frac{(H-h)}{H}, \qquad \text{(F.16)}$$

and, since $\gamma^i = \left( \frac{H}{H+1} \right)^i \geqslant \left( \frac{H}{H+1} \right)^{H+1} \geqslant \frac{3}{10}$, we get

$$\mathbb{P}_h(x_{H+2}^{(i+2)}|x_h^{(i)}, \mathbf{a}')\frac{(H-h)}{H} \left( \gamma^i - \gamma^{i+1} \right) \geqslant \left( \delta - (d^{\mathrm{lin}} - 5)\Delta \right) \frac{(H-h)}{H} \gamma^i (1 - \gamma)$$

$$\geqslant \left( \frac{1}{H} - (d^{\mathrm{lin}} - 5)\Delta \right) \frac{(H-h)}{H} \cdot \frac{3}{10} \cdot \frac{1}{H+1}. \quad \text{(F.17)}$$

Combining Equation F.16 and Equation F.17, and rearranging the terms, we get

$$(d^{\mathrm{lin}} - 5)\Delta \cdot \left( 2 + \frac{3}{10(H+1)} \right) \leqslant \frac{3}{10H(H+1)},$$

which holds when $K \geqslant 4(d^{\mathrm{lin}} - 5)^2 H(H+1)^2$. This explains how the inequality in Equation F.15 is satisfied.

Let $\bar{\mathbf{a}}_h^{(i)} \in \arg\max_{\mathbf{a} \in A_h \setminus \{\mathbf{a}_0\}} \overline{Q}_h^{\pi}(x_h^{(i)}, \mathbf{a}) = \mathbf{a}_h^{\star}$. Note that $\bar{\mathbf{a}}_h^{(i)}$ is unique due to the way the action space and transition probabilities are constructed. Then, by combining Equation F.14 and Equation F.15, and using the fact that $\overline{Q}_h^{\pi}(x_h^{(i)}, \mathbf{a}_0) = 0$, we obtain that

$$\sum_{\mathbf{a} \in A} \mathcal{P}_h(\mathbf{a}|x_h^{(i)}, A)\overline{Q}_h^{\pi}(x_h^{(i)}, \mathbf{a}) \leqslant \sum_{\mathbf{a} \in A} \mathcal{P}_h(\mathbf{a}|x_h^{(i)}, A)\overline{Q}_h^{\pi}(x_h^{(i)}, \bar{\mathbf{a}}_h^{(i)})$$

$$\leqslant \frac{\exp \left( \phi(x_h^{(i)}, \mathbf{a}_h^{\star})^{\top}\boldsymbol{\theta}_h^{\star} \right) \widetilde{Q}_h^{\pi}(x_h^{(i)}, \mathbf{a}_h^{\star}, \bar{\mathbf{a}}_h^{(i)})}{1/H + \exp \left( \phi(x_h^{(i)}, \mathbf{a}_h^{\star})^{\top}\boldsymbol{\theta}_h^{\star} \right)}$$

$$= \frac{\max_{\mathbf{a} \in A \setminus \{\mathbf{a}_0\}} \exp \left( \phi(x_h^{(i)}, \mathbf{a})^{\top}\boldsymbol{\theta}_h^{\star} \right) \widetilde{Q}_h^{\pi}(x_h^{(i)}, \mathbf{a}_h^{\star}, \bar{\mathbf{a}}_h^{(i)})}{1/H + \max_{\mathbf{a} \in A \setminus \{\mathbf{a}_0\}} \exp \left( \phi(x_h^{(i)}, \mathbf{a})^{\top}\boldsymbol{\theta}_h^{\star} \right)},$$

where the first inequality holds since $\bar{\mathbf{a}}_h^{(i)}$ is the action that maximizes the $\overline{Q}$-value. The second inequality follows from Equation F.14 and Equation F.15, and from the fact that $\overline{Q}_h^{\pi}(x_h^{(i)}, \bar{\mathbf{a}}_h^{(i)}) = \overline{Q}_h^{\pi}(x_h^{(i)}, \mathbf{a}_h^{\star}) = \widetilde{Q}_h^{\pi}(x_h^{(i)}, \mathbf{a}_h^{\star}, \bar{\mathbf{a}}_h^{(i)})$. Finally, the last equality holds by the definition of $\mathbf{a}_h^{\star}$.

**Case (ii)** $\mathbf{a}_h^{\star} \notin A$.

Again, by Equation F.11, for any $A \in \mathcal{A}$, we have

$$\gamma \leqslant \min_{\mathbf{a} \in A \setminus \{a_0\}} \frac{\exp \left( \phi(x_h^{(i)}, \mathbf{a})^{\top}\boldsymbol{\theta}_h^{\star} \right)}{1/H + \exp \left( \phi(x_h^{(i)}, \mathbf{a})^{\top}\boldsymbol{\theta}_h^{\star} \right)}$$

$$\leqslant \frac{\max_{\mathbf{a} \in A \setminus \{a_0\}} \exp \left( \phi(x_h^{(i)}, \mathbf{a})^{\top}\boldsymbol{\theta}_h^{\star} \right)}{1/H + \max_{\mathbf{a} \in A \setminus \{a_0\}} \exp \left( \phi(x_h^{(i)}, \mathbf{a})^{\top}\boldsymbol{\theta}_h^{\star} \right)}$$

$$\leqslant \frac{\max_{\mathbf{a} \in A \setminus \{a_0\}} \exp \left( \phi(x_h^{(i)}, \mathbf{a})^{\top}\boldsymbol{\theta}_h^{\star} \right)}{1/H + \max_{\mathbf{a} \in A \setminus \{a_0\}} \exp \left( \phi(x_h^{(i)}, \mathbf{a})^{\top}\boldsymbol{\theta}_h^{\star} \right)} \cdot \frac{1/H + \sum_{\mathbf{a} \in A \setminus \{\mathbf{a}_0\}} \exp \left( \phi(x_h^{(i)}, \mathbf{a})^{\top}\boldsymbol{\theta}_h^{\star} \right)}{\sum_{\mathbf{a} \in A \setminus \{\mathbf{a}_0\}} \exp \left( \phi(x_h^{(i)}, \mathbf{a})^{\top}\boldsymbol{\theta}_h^{\star} \right)},$$

where the second inequality holds since the sigmoid function is a monotonically increasing function.

We denote $\bar{\mathbf{a}}_h^{(i)} \in \arg\max_{\mathbf{a} \in A_h \setminus \{\mathbf{a}_0\}} \overline{Q}_h^\pi(x_h^{(i)}, \mathbf{a})$. Then, multiplying $\widetilde{Q}_h^\pi(x_h^{(i)}, \mathbf{a}_h^\star, \bar{\mathbf{a}}_h^{(i)})$ (note that $\bar{\mathbf{a}}_h^{(i)} \neq \mathbf{a}_h^\star, \mathbf{a}_0$) on both sides and rearranging terms, we get

$$\frac{\sum_{\mathbf{a} \in A \setminus \{\mathbf{a}_0\}} \exp\left(\phi(x_h^{(i)}, \mathbf{a})^\top \boldsymbol{\theta}_h^\star\right) \gamma \cdot \widetilde{Q}_h^\pi(x_h^{(i)}, \mathbf{a}_h^\star, \bar{\mathbf{a}}_h^{(i)})}{1/H + \sum_{\mathbf{a} \in A \setminus \{\mathbf{a}_0\}} \exp\left(\phi(x_h^{(i)}, \mathbf{a})^\top \boldsymbol{\theta}_h^\star\right)}$$

$$\leqslant \frac{\max_{\mathbf{a} \in A \setminus \{a_0\}} \exp\left(\phi(x_h^{(i)}, \mathbf{a})^\top \boldsymbol{\theta}_h^\star\right) \widetilde{Q}_h^\pi(x_h^{(i)}, \mathbf{a}_h^\star, \bar{\mathbf{a}}_h^{(i)})}{1/H + \max_{\mathbf{a} \in A \setminus \{a_0\}} \exp\left(\phi(x_h^{(i)}, \mathbf{a})^\top \boldsymbol{\theta}_h^\star\right)}.$$

Recall that for any $\mathbf{a}' \neq \mathbf{a}_h^\star, \mathbf{a}_0$, we have $\gamma \cdot \widetilde{Q}_h^\pi(x_h^{(i)}, \mathbf{a}_h^\star, \mathbf{a}') \geqslant \overline{Q}_h^\pi(x_h^{(i)}, \mathbf{a}')$ by Equation F.15. Thus, we get

$$\sum_{\mathbf{a} \in A} \mathcal{P}_h(\mathbf{a}|x_h^{(i)}, A) \overline{Q}_h^\pi(x_h^{(i)}, \mathbf{a}) \leqslant \sum_{\mathbf{a} \in A} \mathcal{P}_h(\mathbf{a}|x_h^{(i)}, A) \overline{Q}_h^\pi(x_h^{(i)}, \bar{\mathbf{a}}_h^{(i)})$$

$$\leqslant \sum_{\mathbf{a} \in A} \mathcal{P}_h(\mathbf{a}|x_h^{(i)}, A) \gamma \cdot \widetilde{Q}_h^\pi(x_h^{(i)}, \mathbf{a}_h^\star, \bar{\mathbf{a}}_h^{(i)})$$

$$\leqslant \frac{\max_{\mathbf{a} \in A \setminus \{a_0\}} \exp\left(\phi(x_h^{(i)}, \mathbf{a})^\top \boldsymbol{\theta}_h^\star\right) \widetilde{Q}_h^\pi(x_h^{(i)}, \mathbf{a}_h^\star, \bar{\mathbf{a}}_h^{(i)})}{1/H + \max_{\mathbf{a} \in A \setminus \{a_0\}} \exp\left(\phi(x_h^{(i)}, \mathbf{a})^\top \boldsymbol{\theta}_h^\star\right)}.$$

This concludes the proof of Lemma F.2. □

# G  NUMERICAL EXPERIMENTS

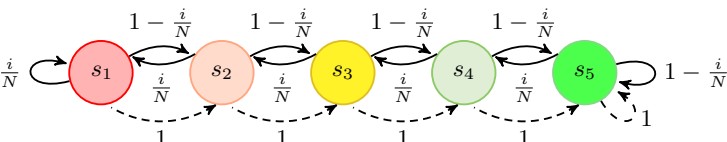

Figure G.1: The "online shopping with budget" environment with $|\mathcal{S}| = 5$. Each state represents the user's budget level of $1, 2, 3, 4,$ or $5$. The solid line indicates the transition when the user purchases an actual item $a_i$ (with a reward of $(i/100N + j/|\mathcal{S}|)/H$), and the dashed line shows the transition when the user does not purchase any item (with a reward of 0). The initial state is $s_3$.

In this section, we empirically evaluate the performance of our algorithm, MNL-VQL, in linear MDPs. We consider an *online shopping with budget* (refer Figure G.1) environment under linear MDPs and an MNL user preference model. We denote the set of states as $\mathcal{S} = \{s_1, \ldots, s_{|\mathcal{S}|}\}$ and the set of items as $\mathcal{I} = \{a_1, \ldots, a_N, a_0\}$ ($a_0$ denotes the outside option). Each state $s_j \in \mathcal{S}$ corresponds to a *user's budget level*, where a larger index $j$ indicates a higher budget (e.g., $s_{|\mathcal{S}|}$ represents the state with the largest budget). The initial state is set to the medium budget state $s_{\lceil |\mathcal{S}|/2 \rceil}$. Furthermore, we let the transition probabilities $\mathbb{P}_h$, rewards $r_h$, and preference model $\mathcal{P}_h$ be the same for all $h \in [H]$, and thus we omit the subscript $h$.

At state $s_j$, the agent offers an assortment $A \in \mathcal{A}$ with a maximum size of $M$. The user then either purchases an item $a_i \in A$ or or opts not to buy anything, represented by the outside option $a_0 \in A$. Then, the reward is defined as follows:

- If the user purchases an item $a_i \in A$, the reward is: $r(s_j, a_i) = \left(\frac{i}{100N} + \frac{j}{|\mathcal{S}|}\right)/H$.

- If the user does not buy anything ($a_0$), the reward is: $r(s_j, a_0) = 0$.

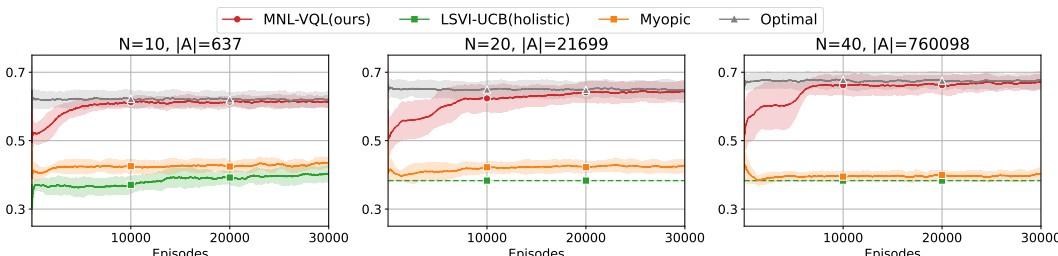

Figure G.2: Episodic returns over 10 independent runs. The dotted lines represent estimated (virtual) episodic returns for cases that could not be run due to excessively long runtimes.

|  | Myoptic | LSVI-UCB | MNL-VQL (ours) |
|---|---|---|---|
| $N = 10, |\mathcal{A}| = 637$ | 0.089 s | 0.136 s | 0.463 s |
| $N = 20, |\mathcal{A}| = 21,699$ | 0.097 s | 4.861 s | 0.526 s |
| $N = 40, |\mathcal{A}| = 760,098$ | 0.113 s | 453.641 s | 0.620 s |

Table G.1: Average runtime (seconds) per episode.

The reward can be regarded as the *user's rating* of the purchased item. It is reasonable to assume that, at higher budget states, users tend to be more generous in their ratings, leading to higher ratings (rewards). And the transition probability is defined as follows:

- If the user purchases an item $a_i \in A$, the transition probability is:

$$\mathbb{P}(s_{\min(j+1,|\mathcal{S}|)}|s_j, a_i) = 1 - \frac{i}{N}, \quad \text{and } \mathbb{P}(s_{\max(j-1,0)}|s_j, a_i) = \frac{i}{N}.$$

- If the user does not buy anything ($a_0$), the transition probability is:

$$\mathbb{P}(s_{\min(j+1,|\mathcal{S}|)}|s_j, a_0) = 1$$

If the user does purchase an item, the budget level decreases with a certain probability that depends on the chosen item. Conversely, if the user does not purchase any item ($a_0$), the budget level increases deterministically.

We construct the feature map $\psi(s, a)$ (for linear MDPs) using SVD. Specifically, the transition kernel $\mathbb{P}(\cdot|\cdot, \cdot) \in \mathbb{R}^{|\mathcal{S}||\mathcal{I}| \times |\mathcal{S}|}$ has at most $|\mathcal{S}|$ singular values, and the reward vector $r(\cdot, \cdot) \in \mathbb{R}^{|\mathcal{S}||\mathcal{I}|}$ has one singular value. Consequently, the feature map $\psi(s, a) \in \mathbb{R}^{d_{lin}}$ lies in a space of dimension $|\mathcal{S}| + 1$, i.e., $d_{lin} = |\mathcal{S}| + 1$.

For MNL preference model, the true parameter $\boldsymbol{\theta}^\star \in \mathbb{R}^d$, and the feature $\phi(s, a) \in \mathbb{R}^d$ (for MNL preference model) are randomly sampled from a $d$-dimensional uniform distribution in each instance.

We set $K = 30000, H = 5, M = 6, |\mathcal{S}| = 5, d = 5$ (feature dimension for MNL preference model), $d^{lin} = 6$ (feature dimension for linear MDP), $N \in \{10, 20, 40\}$ (the number of items), and $|\mathcal{A}| = \sum_{m'=1}^{M-1} \binom{N}{m} \in \{637, 21699, 760098\}$ (the number of assortments). Moreover, for simplicity, we set $\bar{\sigma}_h^k = 1$ in our algorithm. As a result, we use unweighted regression to estimate the $\overline{Q}$-values.

We compare our algorithm with two baselines: `Myopic` and `LSVI-UCB` (Jin et al., 2020). `Myopic` is a variant of `OFU-MNL+` (Lee & Oh, 2024) adapted for *unknown* rewards. It is a myopic algorithm that selects assortments based only on immediate rewards, ignoring state transitions. `LSVI-UCB` (Jin et al., 2020) treats each assortment as a single, atomic (holistic) action, requiring enumeration of all possible assortments. To demonstrate the effectiveness of our approach, we also include the performance of the optimal policy (`Optimal`) to highlight that our algorithm is converging toward optimality. We run the algorithms on 10 independent instances and report the episodic return across all episodes.

Figure G.2 demonstrates that our algorithm significantly outperforms other baseline algorithms. And Table G.1 shows that our algorithm remains robust even as the total number of assortments $|\mathcal{A}|$ increases. Although the runtime of `Myopic` is approximately 5.3 times faster than ours, its

performance is substantially worse, converging to a suboptimal solution. This underscores a key limitation of the myopic strategy—it can completely fail in certain environments, highlighting the importance of accounting for long-term outcomes. Additionally, the runtime of LSVI-UCB increases exponentially as $N$ grows, because it requires enumerating all possible assortments. Due to the extremely slow runtime of LSVI-UCB, we did not include its performance results for $N = 20$ and $N = 40$. Instead, for these cases, we used dotted lines to represent the average episodic return observed for $N = 10$. Even for the smaller case of $N = 10$, LSVI-UCB demonstrated the worst performance. Based on this observation, we suspect that its performance is unlikely to improve as $N$ increases.

