# OpenReview forum: "Combinatorial Reinforcement Learning with Preference Feedback"
_ICLR.cc/2025/Conference — Submitted to ICLR 2025_

### Official Review · Reviewer_5zZf · 2024-10-30

**Soundness:** 3
**Presentation:** 2
**Contribution:** 2
**Rating:** 5
**Confidence:** 3

**Summary:**

This paper considers combinatorial reinforcement learning with preference feedback, where a learning agent sequentially offers an action—an assortment of multiple items—to a user, whose preference feedback follows a multinomial logit (MNL) model. This framework can model real-world scenarios, particularly those involving long-term user engagement, such as in recommender systems and online advertising. However, this framework faces two main challenges: (1) the unknown value of each item, unlike traditional MNL bandits (which only account for single-step preference feedback), and (2) computational complexity due to the combinatorial action space. In this paper, the authors assume a contextual MNL preference model, where mean utilities are linear, and the value of each item is approximated using general function approximation. The authors propose an algorithm, MNL-VQL, that addresses these challenges, making it both computationally and statistically efficient. As a special case, for linear MDPs (with the MNL preference model), the authors establish a regret lower bound and show that MNL-VQL achieves near-optimal regret.

**Strengths:**

1.	The authors design a computationally efficient algorithm, called MNL-VQL, and provide a regret upper bound for the parameterized contextual MNL preference model and general function approximation for item values. This is the first theoretical regret guarantee in combinatorial RL with preference feedback.
2.	For the special case of linear MDPs, the authors establish a regret lower bound, and algorithm MNL-VQL achieves a regret upper bound which nearly matches this lower bound.

**Weaknesses:**

1.	The writing of this paper needs improvement. This paper is too dense.
2.	There is no experiment provided. It is hard to validate the effectiveness and implementability of the proposed algorithm, and the provided theoretical results.
3.	The authors should highlight more on the computational challenge brought by the combinatorial action space, and novel techniques used to overcome this computational challenge, using concise words to convey the intuition behind, perhaps in the introduction.
4.	This work looks like a combination between combinatorial RL with preference feedback and existing RL works with function approximation. The authors should elaborate more on technical novelty, e.g., provide a paragraph to summarize it.

**Questions:**

Please see the weaknesses above.

---

> ### Author Response · Authors · 2024-11-19
> **Official Comment by Authors [1/4]**
>
> We sincerely appreciate the time and effort you have dedicated to reviewing our paper, and we thank you for your valuable feedback. As we begin our discussion, we kindly ask for your patience with our detailed and lengthy responses, as we have a lot to share to fully address your concerns. The paper has been thoroughly updated to reflect all of your feedback.
>
> ---
> ### **Presentation**
>
> In response to the reviewers' feedback, we have revised the paper to better emphasize our contributions and technical novelties. The major changes are highlighted in red. The main revisions include:
>
> - Providing a clearer explanation of our key challenges, main contributions, and technical novelties in the Introduction (Section 1).
> - Adding an overview of the algorithm and improving the connections between its components in Section 4.
> - Including numerical experiments in Section 5 and Appendix G.
>
> ---
> ### **Empirical results**
>
> Following the reviewer’s suggestion, we conducted experiments in the linear MDPs.
>
> We consider an *online shopping with budget* (see Figure G.1) under linear MDPs and an MNL user preference model. To simplify the presentation, we represent the set of states as $\mathcal{S} = \\{s_1, \dots, s_{|\mathcal{S}|} \\}$ and  the set of items as $\mathcal{I} = \\{1, \dots, N , 0 \\}$ ($0$ denotes the outside option).  Each state $s_j \in \mathcal{S}$ corresponds to a user's budget level, where a larger index $j$  indicates a higher budget (e.g., $s_{|\mathcal{S}|}$ represents the state with the largest budget). The initial state is set to the medium budget state $s_{\lceil  |\mathcal{S}|/2 \rceil }$. Furthermore, we let the transition probabilities, rewards, and preference model be the same for all $h \in [H]$, and thus we omit the subscript $h$.
>
> At state $s_j$, the agent offers an assortment $A \in \mathcal{A}$ with the maximum size of $M$.  The user then purchases an item $i \in A$. Then, the reward  is defined as follows:
>
> - If the chosen item $i$ is not the outside option, the reward is $r(s_j,i) = (\frac{i}{100N} + \frac{j}{|\mathcal{S}|})/H$.
> - If the chosen item $i$ is the outside option, the reward is $r(s_j,i) = 0$.
>
> The reward can be regarded as the *user's rating* of the purchased item. It is reasonable to assume that, at higher budget states, users tend to be more generous in their ratings, leading to higher ratings (rewards). And the transition probability is defined as follows:
>
> - If the chosen item $i$ is not the outside option, the transition probability is:
>     $\mathbb{P}(s_{\min(j+1, |\mathcal{S}|) } |s_j,i) = 1 - \frac{i}{N}$ and $\mathbb{P}( s_{\max(j-1, 0)} |s_j,i) = \frac{i}{N}.$
> - If the chosen item $i$ is the outside option, the transition probability is $\mathbb{P}(s_{\min(j+1, |\mathcal{S}|) } |s_j,i) = 1$.
>
> If the user does purchase an item, the budget level decreases with a certain probability that depends on the chosen item. Conversely, if the user does not purchase any item ($i=0$), the budget level increases deterministically.
>
> We construct the feature map $\psi(s,i)$ (for linear MDPs) using SVD. Specifically, the transition kernel $\mathbb{P}(\cdot | \cdot, \cdot) \in \mathbb{R}^{|\mathcal{S}||\mathcal{I}| \times |\mathcal{S}|}$ has at most $|\mathcal{S}|$ singular values, and the reward vector $r( \cdot, \cdot) \in \mathbb{R}^{|\mathcal{S}||\mathcal{I}|}$ has one singular value. Consequently, the feature map $\psi(s,i) \in \mathbb{R}^{d_{lin}}$ lies in a space of dimension $|\mathcal{S}| + 1$, i.e., $d_{lin} = |\mathcal{S}| + 1$.
>
>
> For MNL preference model, the true parameter $\theta^\star \in \mathbb{R}^d$, and the feature  $\phi(s,a) \in \mathbb{R}^d$ (for MNL preference model) are randomly sampled from a $d$-dimensional uniform distribution in each instance.
>
>
> We set $K = 30000, H=5, M=6,  |\mathcal{S}|=5, d = 5$ (feature dimension for MNL preference model), $d^{lin} = 6$ (feature dimension for linear MDP), $N \in \\{10, 20, 40\\}$ (the number of items), and $|\mathcal{A}| = \sum_{m=1}^{M-1}$$N \choose m$ $\in \\{ 637, 21699, 760098\\}$ (the number of assortments). Moreover, for simplicity, we set $\bar{\sigma}^k_h = 1$ in our algorithm.
>
> We compare our algorithm with two baselines: *Myopic* and *LSVI-UCB*.
>
> - *Myopic* is a variant of OFU-MNL+ (Lee & Oh, 2024) adapted for **unknown** rewards. It is a short-sighted algorithm that selects assortments based only on immediate rewards, ignoring state transitions.
> - *LSVI-UCB* (Jin et al., 2020) treats each assortment as a single, atomic (holistic) action, requiring enumeration of all possible assortments.

---

> ### Author Response · Authors · 2024-11-19
> **Official Comment by Authors [2/4]**
>
> To demonstrate the effectiveness of our approach, we also include the performance of the optimal policy (*Optimal*) to highlight that our algorithm is converging toward optimality. We run the algorithms on 10 independent instances and report the average total return and runtime across all episodes.
>
>
> | Return (Runtime) | Optimal  | Myopic               | LSVI-UCB(holistic)            | MNL-VQL(ours)       |
> |------------------------------|----------|-----------------------|-------------------------------|---------------------|
> | $N = 10, \|\mathcal{A}\| = 637$          | 18,632.7  | 12,754.8 (2,663.5s)     | 11,495.5 (4,088.8s)             | **18,097.4** (13,904.9s)  |
> | $N = 20, \|\mathcal{A}\| = 21,699$        | 19,500.5  | 12,588.2 (2,906.1s)     | still running (1.7 days estimated) | **18,586.3** (15,780.7s)  |
> | $N = 40, \|\mathcal{A}\| = 760,098$       | 20,283.2  | 11,826.3 (3,398.5s)     | still running (157.5 days estimated) | **19487.5** (18,605.2s)  |
>
>
> As shown in the table above, our algorithm **significantly outperforms** the baseline algorithms. We have also included experimental results in the revised paper (Appendix G).
>
> Although the runtime of *Myopic* is approximately 5.3 times faster than ours, its performance is substantially worse, converging to a suboptimal solution (see Figure G.2). This underscores a key limitation of the myopic strategy—it can completely fail in certain environments, highlighting the importance of accounting for long-term outcomes.
> Additionally, the runtime of *LSVI-UCB* increases exponentially as $N$ grows, because it requires enumerating all possible assortments. Due to the extremely slow runtime of *LSVI-UCB*, we omitted its performance results for $N=20$ and $N=40$. However, even for smaller $N$ ($N=10$), *LSVI-UCB* shows the worst performance. Based on these results, we hypothesize that its performance would not improve for larger values of $N$.
>
> ---
> ### **Computational challenge due to combinatorial action space**
> If by "computational challenge brought by the combinatorial action space" the reviewer meant (or implied) that there is possible exponential computation complexity often caused by combinatorial action space, we clearly state that our method does NOT incur
> an exponential computational cost at all for combinatorial actions.
> Our method does not naively enumerate all possible assortments (see Remark 4).
>
> In Remark 4, we clarified that the combinatorial optimization can be avoided by transforming the problem into a **linear programming** formulation. For further details, please refer to Appendix C.
>
> We strongly believe that this aspect should NOT be considered as a weakness of our work.
> Additionally, as per your suggestion, we have explicitly addressed how to avoid exponential computational complexity in the Introduction of the revised version.
>
> ---
>
> ### **Contributions & Technical novelties**
>
> If the reviewer's characterization of our work were to be a possibly simple "combination" of combinatorial RL with preference feedback and existing RL works with function approximation, we would be happy to provide more details. To ensure clarity, we pose the following questions:
>
> - Has there been any prior work on combinatorial RL with function approximation (with regret guarantees)? **No**.
> - Has there been any prior work on RL with MNL choice feedback on actions (with regret guarantees)? **No**.
>
> Our contribution goes well beyond a mere combination of these areas. Each of these components—combinatorial RL, preference feedback, and function approximation—is non-trivial in its own right. Combining them rigorously under a single framework requires careful and sophisticated theoretical development. Furthermore, this combination introduces unique challenges that have not been addressed in previous methods.
>
> - **Unknown item values**, as the rewards and transition probabilities are unknown, unlike in MNL bandits.
> - **Limitations of using optimistic utilities alone**, as they are insufficient to guarantee optimism, unlike in MNL bandits.
> - **Technical challenges in proving optimism** when item values are unknown.
> - The **absence of regret lower bounds**, even for linear MDPs.
>
> We emphasize that this work provides the **first theoretical guarantee for combinatorial RL with preference feedback**, to our best knowledge.
> Moreover, we establish the **first regret lower bound for combinatorial linear MDPs with preference feedback**. This lower bound establishes a fundamental theoretical baseline and highlights the intrinsic difficulty of the problem. Additionally, our analysis of the regret is novel and tailored specifically to address the interplay between combinatorial actions, preference feedback, and linear function approximation.

---

> ### Author Response · Authors · 2024-11-19
> **Official Comment by Authors [3/4]**
>
> The followings are our main technical novelties:
>
> **1) Proof of optimism (Lemma D.9):**
> - Unlike MNL bandits, which assume the revenue parameters are known, we do **NOT have access to the true item-level Q functions**. As a result, we must select an assortment based on estimated values.
> - From an algorithmic perspective, we begin by computing the optimistic item-level Q-functions. However, unlike in MNL bandits, where item values are known and the outside option always has a value of zero, simply using optimistic utilities (for MNL preference model) is not enough to ensure optimism in our setting. This is because the optimistic Q-values for the outside option can be large, requiring us to adapt our strategy accordingly. To address this, we carefully alternate between **optimistic and pessimistic utilities** based on the estimated item-level Q-functions (Equation 8).
>
>
> - Furthermore, in our analysis, proving optimism requires demonstrating that the value
> $$
> \sum\_{a \in A^{k,\star}\_h} \mathcal{P}\_h (a | s^k_h, A^{k,\star}\_h) f^k\_{h,1}(s^k\_h,a)
> $$ increases when using optimistic and pessimistic utilities (for MNL preference model), rather than the true utilities (using $\tilde{\mathcal{P}}$ instead of $\mathcal{P}$).  However, we cannot directly prove that
> $$
> \sum\_{a \in A^{k,\star}\_h} \mathcal{P}\_h (a | s^k_h, A^{k,\star}\_h) f^k\_{h,1}(s^k_h,a) \leq \sum_{a \in A^{k,\star}\_h} \tilde{\mathcal{P}}\_h (a | s^k_h, A^{k,\star}\_h) f^k\_{h,1}(s^k_h,a)
> $$   for the same optimal assortment $A^{k,\star}\_h$.  This is because the item-level Q-value estimates for the outside option $f^k_{h,j}(s, a_0)$  can be larger than those for other items. Unlike MNL bandits, where the revenue parameter for the outside option is always zero, our setting allows for a positive value for the item-level Q functions (not choosing any item can lead to larger rewards later).
>
> - To overcome this, we instead **prove the inequality for a specific subset** $\tilde{A}^{k}\_h \subseteq A^{k,\star}\_h$. Thus, we obtain:
> $$
> \sum\_{a \in A^{k,\star}_h} \mathcal{P}\_h (a | s^k_h, A^{k,\star}\_h) f^k\_{h,1}(s^k_h,a) \leq \sum\_{a \in \tilde{A}^{k}\_h} \tilde{\mathcal{P}}\_h (a | s^k_h, \tilde{A}^{k}\_h) f^k\_{h,1}(s^k_h,a) \quad \text{(Lemma D.2)}
> $$   This is sufficient to prove the optimism (Lemma D.9). This analysis is novel and has not been addressed in previous works. These represent the key technical challenges in proving optimism.
>
>
>
> **2) First lower bound in combinatorial RL with preference feedback under linear MDPs:**
> - In Theorem 3, we establish, to the best of our knowledge, the **first regret lower bound for combinatorial RL with preference feedback under linear MDPs**. To prove this, we construct a novel and complex **multi-layered MDP** (see Figure F.1) with carefully defined features and parameters. We believe that both the construction of the MDP and the proof structure represent a **non-trivial extension of previous works**, such as Zhou et al. (2021b). Moreover, the result is particularly powerful, as there is only a  $\sqrt{H}$ gap in the first regret term when compared to the upper bound (Theorem 2)
>
>
>
>
> Other (relatively minor) technical novelties are as follows:
>
> **1) Careful analysis due to unknown item-level Q-values $\bar{Q}^\star_h$:**
> - Additional technical difficulties arise from the unknown $\bar{Q}^\star_h$. To prove several lemmas (Lemmas D.2, D.5, and D.7), we need to consider different cases based on the value of $f^k_{h,j}$ (optimistic estimates of $\bar{Q}^\star_h$).
>
>
> **2) $1/\sqrt{\kappa}$-improved bound for MNL bandits (Lemma D.6):**
> - The crude MNL bandit regret in Lemma D.6 improves upon the one proposed by Oh & Iyengar (2021) by a factor of  $1/\sqrt{\kappa}$, which can be exponentially large. We achieve this by using the Hessian matrix $\mathbf{H}^k_h$,  in contrast to the approach used in Oh & Iyengar (2021). This enhancement allows us to eliminate the $\frac{1}{\kappa^2}$ term from the lower-order term in the regret upper bound, leaving only a $\frac{1}{\kappa}$ factor.  Lemma D.6 is used to carefully bound the sum of $b^k_{h,1}$ (Lemma D.17).
> - Another interesting point is that $\tilde{\mathcal{P}}$ can be constructed using either optimistic or pessimistic utilities. This distinguishes our work from Oh & Iyengar (2021), which only proves the regret using optimistic utilities.
>
>
>
> **3) Choice of the exploration threshold $u_k$:**
> - To bound the size of the overly optimistic episodes $|\mathcal{K}_{oo}|$ (Lemma D.18), we must carefully choose the exploration threshold $u_k$. Since we consider the MNL preference model, the value of $u_k$ needs to be different from that used in Agrawal et al. (2023).

---

> ### Author Response · Authors · 2024-11-19
> **Official Comment by Authors [4/4]**
>
> In conclusion, we believe that our work not only addresses a previously unexplored problem setting but also makes significant theoretical contributions by establishing regret guarantees and deriving a regret lower bound in this complex framework (under the additional assumption of linear MDPs). These achievements are far from being a simple combination and represent a substantial advancement in the field.

---

> > ### Author Response · Authors · 2024-11-25
> > **Dear Reviewer 5zZf**
> >
> > Dear Reviewer 5zZf,
> >
> > As we approach the conclusion of the discussion period, we want to ensure that all your questions and comments have been comprehensively addressed. However, if you have any remaining questions or need further clarification, please don’t hesitate to let us know. Otherwise, we kindly ask you to take a moment to re-evaluate our work.

---

> > > ### Comment · Reviewer_5zZf · 2024-11-25
> > > **Thank you for your response**
> > >
> > > Thank you for your response and the added experiments!
> > >
> > > I know this work provides the first theoretical analysis for combinatorial RL with preference feedback, and avoids naive enumeration on combinatorial action space (but this is standard and not surprising given many prior works in the literature of combinatorial bandits). I think my main concern is still on how large the technical challenge and novelty are in combining combinatorial RL and preference feedback. In other words, why can't one simply combine the existing techniques in these two literatures to solve this combined problem.
> > >
> > > I tend to maintain my score now, and will participate in the discussion with other reviewers and AC.

---

> ### Author Response · Authors · 2024-11-28
> **Clarification of technical novelties [1/2]**
>
> We are happy to elaborate further on our technical contributions. Before diving into the discussion, we want to emphasize that **our lower bound—which may have been overlooked by most of reviewers—is highly innovative in both technical and creative aspects**. The instance design and proof techniques we employ are exceptionally novel.
>
> To address your comments, we have outlined our key technical challenges and compared them point-by-point with existing methods. Additionally, we demonstrate why simply combining existing methods fails entirely.
>
> ### **1. vs MNL bandits**
>
> Our framework generalizes MNL bandits in several significant ways by incorporating: 1) **state transitions** (accounting for long-term values), 2) **unknown rewards (and transitions)**, and 3) **non-zero rewards for the outside option**. These extensions introduce **unique challenges** that do not occur in traditional MNL bandits.
>
> To start, let us simplify the discussion by fix $h \in [H]$ in our framework. It facilitates a clearer comparison with MNL bandit frameworks in terms of technical novelty. We can then express the $Q$-function, abbreviating the index $h$, as follows:
> $$
> Q(s,A) = \sum_{a \in A} \mathcal{P}(a | s, A; \theta^\star) \bar{Q}(s,a),
> $$
> where the item-level $Q$-function $\bar{Q}$ and the preference model $\mathcal{P}$ are **unknown**. Here, $\bar{Q}$ is assumed to be a general function, while  $\mathcal{P}$ follows the MNL model.
>
> In contrast, the revenue function in MNL bandits is defined as:
> $$
> R(A) = \sum_{a \in A} \mathcal{P}(a | A ; \theta^\star) r(a),
> $$
> where $r$ is **known** and the reward for the outside option $r(a_o)$ is zero. In our framework, $r$ (and $\bar{Q}$) is unknown and $r(s,a_o)$ can be non-zero (and even very large). These differences lead to several unique challenges:
>
> - **Algorithmic Challenges**
>     1. **Estimating unknown $\bar{Q}$**
>      In our framework, we estimate $\bar{Q}$ optimistically using a UCB-style approach, inspired by methods in RL with general function approximation (Agarwal et al., 2023).
>
>     2. **Optimism**
>     - In MNL bandits, optimism is straightforward because rewards are known, and $r(a_o)=0$. Increasing the utilities of items in the  assortment $A^\star \setminus \{0\}$ ($A^\star$ is the optimal assortment) ensures optimism. Specifically, using the UCB parameter $\tilde{\theta}\_{t,i} = \arg\max\_{\theta \in \mathcal{C}_t} x_i^\top \theta$, the following holds (see Lemma 4 in Oh and Iyengar., 2021):
>      $$
>      R(A^\star)
>      = \sum\_{a \in A^\star} \mathcal{P}(a | A^\star ; \theta^\star) r(a)
>      \leq \sum\_{a \in A^\star} \mathcal{P}(a | A^\star ; \tilde{\theta}_t) r(a)
>      \leq \sum\_{a \in A_t} \mathcal{P}(a | A_t ; \tilde{\theta}_t) r(a),
>     $$
>
>     - In contrast, in our framework, because $\bar{Q}$ is **unknown**, we rely on the optimistic estimates $f$. Since we allow $f(s,a_o) > 0$ (i.e., non-zero optimistic estimates for the outside option), naively using a UCB (optimistic) parameter (for the MNL model) does not always guarantee an increase in $Q$-values. This issue arises because if the optimistic estimate $f(s,a_o)$ is larger than the optimistic estimate $f(s,a)$ for other items in the optimal assortment $A^\star$, i.e., $a \in A^\star \setminus \{0\}$, increasing the probability of selecting non-outside options reduces the estimated $Q$-value.
>     - To address this, we carefully alternate between **optimistic and pessimistic utilities** (for the MNL model) depending on the value of $f(s,a)$. This nuanced approach to utility selection represents the key algorithmic novelty of our framework.
>
>
> - **Analytical Challenges**
>     1. **Optimism proof**
>      Even if we properly use optimistic and pessimistic utilities, novel analysis is required to prove optimism. Recall that in MNL bandits, we have:
>     $$
>      R(A^\star)
>      = \sum\_{a \in A^\star} \mathcal{P}(a | A^\star ; \theta^\star) r(a)
>      \leq \sum\_{a \in A^\star} \mathcal{P}(a | A^\star ; \tilde{\theta}\_t) r(a).
>     $$
>     However, in our case, we cannot obtain this result for the same $A^\star$. Instead, we **prove the inequality for a specific subset** $\tilde{A} \subseteq A^{\star}$. Thus, we obtain:
> $$
> \sum_{a \in A^{\star}} \mathcal{P} (a | s, A^{\star}; \theta^\star) f(s,a) \leq \sum_{a \in \tilde{A}} \mathcal{P} (a | s, \tilde{A}; \tilde{\theta}\_k) f(s,a) \quad \text{(Lemma D.2)},
> $$
> where $\tilde{\theta}$ can be optimistic or pessimistic depending on $f$.
> Then, choosing the assortment $A_k = \arg\max_A \sum_{a \in A} \ \mathcal{P} (a | s, A; \tilde{\theta}\_k) f(s,a)$  is sufficient to prove optimism (Lemma D.9). This analysis is novel and has not been addressed in previous works. These represent the key technical challenges in proving optimism.
>
>     2. **Careful analysis due to unknown $\bar{Q}$  (Lemmas D.2, D.5, and D.7)**
>     Since we do not know the rewards, we need to consider different cases based on the value of $f$ when proving other lemmas related to MNL bandits.

---

> ### Author Response · Authors · 2024-11-28
> **Clarification of technical novelties [2/2]**
>
> 3. **$1/\sqrt{\kappa}$-improved bound for MNL bandits (Lemma D.6)**
>     - The crude MNL bandit regret in Lemma D.6 improves upon the one proposed by Oh & Iyengar (2021) by a factor of  $1/\sqrt{\kappa}$, which can be exponentially large. This improvement eliminates the $\frac{1}{\kappa^2}$ term in the lower-order our final regret bound, leaving only $\frac{1}{\kappa}$ factor.  Lemma D.6 is used to carefully bound the sum of $b^k_{h,1}$ (Lemma D.17).
>     - Additionally, $\tilde{\mathcal{P}}$ can be constructed using either optimistic or pessimistic utilities, unlike Oh & Iyengar (2021), which relies solely on optimistic utilities.
>
>
> ### **2. General function approximation (Agarwal et al., 2023)**
>
> Since the rewards and transitions are unknown, we approximate item-level $Q$-values using a general function approximation approach, inspired by Agarwal et al. (2023). However, our framework incorporates a preference model, which significantly influences many of the key lemmas (e.g., Lemmas D.12, D.13, D.14, D.17, and D.18).
>
> 1. **Estimation error**
>      The overall estimation error is a combination of two sources: 1) errors from the MNL preference model, and 2) errors in estimating item-level $Q$-values.
>
> 3. **Choice of the exploration threshold $u_k$**
>     To bound the number of overly optimistic episodes ($|\mathcal{K}_{oo}|$, as shown in Lemma D.18), the exploration threshold $u_k$ must be chosen carefully. Because our framework incorporates the MNL preference model, the value of $u_k$ depends on the estimation error associated with the preference model.
>
>
> ### **3. vs Combinatorial bandit**
>
> Our framework is fundamentally different from combinatorial bandits in its modeling assumptions, making a direct comparison of technical novelty meaningless. Below, we highlight how the two frameworks differ:
>
> - **Model**: In combinatorial bandits, the goal is to maximize the expected reward $R(A_t)$, where $A_t$ is an assortment (or super arm). Unlike our framework, combinatorial bandits typically do not incorporate a user preference model. Instead, the expected reward is treated solely as a function of the assortment $A_t$.
>
> - **Assumptions**: Combinatorial bandits commonly rely on the monotone reward function assumption [1,2]. In contrast, our framework does not need this assumption. For instance, in our case, even if the utility for any $a \in A$ increases, the $Q$-function does not necessarily increase because the reward $r(s,a)$ for that item could be very low.
>
> - **Optimization**: Combinatorial bandits often use *approximation oracles*[1,2] to select the assortment, while our framework computes the optimal assortment exactly.
>
>
> [1] Chen, Wei, Yajun Wang, and Yang Yuan. "Combinatorial multi-armed bandit: General framework and applications." International conference on machine learning. PMLR, 2013.
>
> [2] Chen, Wei, et al. "Combinatorial multi-armed bandit with general reward functions." Advances in Neural Information Processing Systems 29 (2016).
>
> ### **4. vs PbRL**
>
> Our framework is also fundamentally different from PbRL. In PbRL, the agent does not learn from explicit numerical rewards but instead relies on preferences as feedback. Specifically, the user is presented with two (or sometimes more) items and selects the preferred one. The preference model is typically defined as:
> $$
>     \mathcal{P}(a_1|s, A=\{a_1,a_2\}) = \frac{\exp(r(s,a_1)) }{\exp(r(s,a_1)) + \exp(r(s,a_2))}.
> $$
> This model inherently assumes that the item with the larger reward is more preferable [3,4,5].  In contrast, our framework allows the incurred reward to deviate from the preference utility, meaning the reward does not need to be proportional to the preference.
>
> Additionally, our framework differs from PbRL in its objective. While PbRL typically aims to offer a single item based on preferences [3, 4, 5], our framework focuses on offering multiple items—a combinatorial (base) action—at each timestep.
>
> [3] Saha, Aadirupa, and Aditya Gopalan. "Combinatorial bandits with relative feedback." Advances in Neural Information Processing Systems 32 (2019).
>
> [4] Zhu, Banghua, Michael Jordan, and Jiantao Jiao. "Principled reinforcement learning with human feedback from pairwise or k-wise comparisons." International Conference on Machine Learning. PMLR, 2023.
>
> [5] Saha, Aadirupa, Aldo Pacchiano, and Jonathan Lee. "Dueling rl: Reinforcement learning with trajectory preferences." International Conference on Artificial Intelligence and Statistics. PMLR, 2023.
>
>
> In summary, our results cannot be achieved by simply combining existing methods. If we naively combine techniques from MNL bandits and general function approximation, the **optimism fails entirely**, along with other key technical lemmas, making it impossible to establish regret upper bounds.
> Furthermore, comparing our framework with general combinatorial RL (without the MNL model) or PbRL is not meaningful, as the modeling assumptions or algorithmic objectives are fundamentally different.

---

> ### Author Response · Authors · 2024-12-02
> **Dear Reviewer 5zZf**
>
> Dear Reviewer 5zZf,
>
> As the extended discussion period comes to an end, we want to ensure that our previous responses have fully addressed your concerns regarding our contributions.
>
> We believe our work is significant as it is the **first theoretical study in combinatorial RL with a preference feedback framework.** Developing new frameworks based on existing ones is highly impactful because it creates new research directions. Our framework, in particular, broadens the scope of RL research, with applications beyond recommender systems and online shopping, extending to fields like online dialogue systems (e.g., ChatGPT).
>
> For instance, in a system like ChatGPT, users interact with the model through multiple actions: accepting an answer, regenerating it, or asking follow-up questions. These interactions fit naturally into our combinatorial RL with preference feedback framework. Unlike existing works that focus solely on and learn from pairwise feedback (PbRL), our framework enables models to learn from ongoing multi-turn conversations, opening exciting possibilities for future advancements in AI dialogue systems.
>
> Additionally, the **technical challenges addressed in our paper are non-trivial**, including:
>
> 1) Proving optimism through a novel algorithm and supporting lemmas (Lemma D.2, D.9),
> 2) Establishing the first lower bound for linear MDPs with preference feedback (Theorem 3), and
> 3) Deriving tighter bounds for general function approximation (with MNL preference model) through several new lemmas (Lemmas D.12, D.13, D.14, D.17, and D.18).
>
> If there are no further questions or clarification needed, we kindly request you to reconsider our score. Thank you for your time and thoughtful evaluation.

---

### Official Review · Reviewer_aS1z · 2024-11-03

**Soundness:** 3
**Presentation:** 3
**Contribution:** 3
**Rating:** 6
**Confidence:** 3

**Summary:**

This paper studies the combinatorial RL with preference feedback, where the agent aims to select an assortment of items for users and get preference feedback to improve long-term user engagement in online recommender systems. To tackle the uncertainty in both the user choice model and the state transitions, the authors propose to use the contextual MNL preference model and RL with function approximation to model the problem. Then they propose a computationally and statistically efficient algorithm MNL-VQL, with rigorous theoretical guarantee.

**Strengths:**

1. This paper frames a new combinatorial RL problem with preference feedback. Based on the decomposition model in SlateQ (Ie et al, 2019), the authors explicitly use contextual MNL preference assumption to model the user’s single item choice after the agent recommends a combinatorial action (i.e., an assortment), which is the key structural property to maintain the traceability of the problem. For the state transition and reward modeling, the authors use RL with general function approximation to guarantee the generalizability of the model.
2. The authors naturally combine the MNL bandit leaning for the user’s MNL choice model and the model-free VOQL to learn the item-level Q-functions. To obtain optimistic choice probabilities, they propose a new method to judiciously assign UCB or LCB for each selected item in the assortment considering the relative value of outside option $a_0$.  Then they also propose a provably efficient method to compute the policy based on the optimistic assortment Q-values.
3. The authors also provide theoretical guarantees for the regret upper bounds as well as discussions for the tightness of the regret bounds considering several degenerate cases, such as contextual MNL bandit with $H=1$ and RL with linear function approximation

**Weaknesses:**

1. Although quite natural and well-motivated, the problem setting heavily relies on three previous works: the SlateQ (Ie et al, 2019) as the basic slate-based combinatorial RL model, the contextual MNL choice model for user behavior Oh & Iyengar (2019), and model-free RL with non-linear function approximation in Agarwal et al. 2023. The algorithm is also a combination of Lee & Oh (2024) for MNL model learning and VOQL in Lee & Oh (2024). The main contribution beyond the above combination is how to adjust the UCB/LCB for MNL model learning as well as an efficient combinatorial optimization subroutine. In general, the paper does not provide extremely novel algorithmic components/insights, but only essential parts to make the combination work.
2. The theoretical results are not validated by any empirical results, which may not give readers enough evidence that structural assumptions will hold for real-world application scenarios and confidence that the proposed algorithm would work well in practice.

**Questions:**

1. Please comment on whether I am correct about the contribution and novelty of the paper. One interesting direction to improve the novelty of the paper is to discuss a new setting where more than one item can be adopted by the user, such as the DCM bandit choice model (Katariya et al., 2016), which is more interesting and practical when users may purchase several items recommended by the system.
2. For the empirical evaluation, one valuable comparison, that I would like to see, is to use the similar simulation setting of Section 5 of SlateQ (Ie et al, 2019) to demonstrate the algorithm's performance, which is a simulated environment that mimics real-world recommender systems. For baseline algorithms, the authors could use SARSA and Q-learning with different combinatorial optimization methods (e.g., greedy, top-K as in Ie et al, 2019) as well as the holistic optimization that uses the non-decomposed Q-learning method that treats each assortment atomically as a single action. The authors are also encouraged to compare with other possible baselines as well as the real world data as in Section 6 in SlateQ (Ie et al, 2019).

---

> ### Author Response · Authors · 2024-11-19
> **Official Comment by Authors [1/3]**
>
> We sincerely appreciate the time and effort you put into reviewing our paper. Your comments and suggestions have been invaluable in strengthening our work. We have revised the paper to address all your concerns.
>
> ---
>
> ### **Main contributions and technical novelties**
>
> If the reviewer views our work as a possibly simple "combination" of existing frameworks, we would be glad to elaborate further. To facilitate understanding, we propose the following questions:
>
> - Has there been any prior work on combinatorial RL with function approximation (with regret guarantees)? **No**.
> - Has there been any prior work on RL with MNL choice feedback on actions (with regret guarantees)? **No**.
>
> Our contribution goes well beyond a mere combination of these areas. Each of these components—combinatorial RL (Ie et al., 2019), MNL preference feedback (Oh & Iyengar.,  2019, Lee & Oh., 2024), and function approximation (Agarwal et al., 2023)—is non-trivial in its own right. Combining them rigorously under a single framework requires careful and sophisticated theoretical development. Furthermore, this combination introduces unique challenges that have not been addressed in previous methods.
>
> - **Unknown item values**, as the rewards and transition probabilities are unknown, unlike in MNL bandits.
> - **Limitations of using optimistic utilities alone**, as they are insufficient to guarantee optimism, unlike in MNL bandits.
> - **Technical challenges in proving optimism** when item values are unknown.
> - The **absence of regret lower bounds**, even for linear MDPs.
>
> We emphasize that this work provides the **first theoretical guarantee for combinatorial RL with preference feedback**, to our best knowledge.
> Moreover, we establish the **first regret lower bound for combinatorial linear MDPs with preference feedback**. This lower bound establishes a fundamental theoretical baseline and highlights the intrinsic difficulty of the problem. Additionally, our analysis of the regret is novel and tailored specifically to address the interplay between combinatorial actions, preference feedback, and linear function approximation.
>
> The followings are our main technical novelties:
>
> **1) Proof of optimism (Lemma D.9):**
> - Unlike MNL bandits, which assume the revenue parameters are known, we do **NOT have access to the true item-level Q functions**. As a result, we must select an assortment based on estimated values.
> - From an algorithmic perspective, we begin by computing the optimistic item-level Q-functions. However, unlike in MNL bandits, where item values are known and the outside option always has a value of zero, simply using optimistic utilities (for MNL preference model) is not enough to ensure optimism in our setting. This is because the optimistic Q-values for the outside option can be large, requiring us to adapt our strategy accordingly. To address this, we carefully alternate between **optimistic and pessimistic utilities** based on the estimated item-level Q-functions (Equation 8).
>
>
> - Furthermore, in our analysis, proving optimism requires demonstrating that the value
> $$
> \sum\_{a \in A^{k,\star}\_h} \mathcal{P}\_h (a | s^k_h, A^{k,\star}\_h) f^k\_{h,1}(s^k\_h,a)
> $$ increases when using optimistic and pessimistic utilities (for MNL preference model), rather than the true utilities (using $\tilde{\mathcal{P}}$ instead of $\mathcal{P}$).  However, we cannot directly prove that
> $$
> \sum\_{a \in A^{k,\star}\_h} \mathcal{P}\_h (a | s^k_h, A^{k,\star}\_h) f^k\_{h,1}(s^k_h,a) \leq \sum_{a \in A^{k,\star}\_h} \tilde{\mathcal{P}}\_h (a | s^k_h, A^{k,\star}\_h) f^k\_{h,1}(s^k_h,a)
> $$   for the same optimal assortment $A^{k,\star}\_h$.  This is because the item-level Q-value estimates for the outside option $f^k_{h,j}(s, a_0)$  can be larger than those for other items. Unlike MNL bandits, where the revenue parameter for the outside option is always zero, our setting allows for a positive value for the item-level Q functions (not choosing any item can lead to larger rewards later).
>
> - To overcome this, we instead **prove the inequality for a specific subset** $\tilde{A}^{k}\_h \subseteq A^{k,\star}\_h$. Thus, we obtain:
> $$
> \sum\_{a \in A^{k,\star}_h} \mathcal{P}\_h (a | s^k_h, A^{k,\star}\_h) f^k\_{h,1}(s^k_h,a) \leq \sum\_{a \in \tilde{A}^{k}\_h} \tilde{\mathcal{P}}\_h (a | s^k_h, \tilde{A}^{k}\_h) f^k\_{h,1}(s^k_h,a) \quad \text{(Lemma D.2)}
> $$   This is sufficient to prove the optimism (Lemma D.9). This analysis is novel and has not been addressed in previous works. These represent the key technical challenges in proving optimism.

---

> ### Author Response · Authors · 2024-11-19
> **Official Comment by Authors [2/3]**
>
> **2) First lower bound in combinatorial RL with preference feedback under linear MDPs:**
> - In Theorem 3, we establish, to the best of our knowledge, the **first regret lower bound for combinatorial RL with preference feedback under linear MDPs**. To prove this, we construct a novel and complex **multi-layered MDP** (see Figure F.1) with carefully defined features and parameters. We believe that both the construction of the MDP and the proof structure represent a **non-trivial extension of previous works**, such as Zhou et al. (2021b). Moreover, the result is particularly powerful, as there is only a  $\sqrt{H}$ gap in the first regret term when compared to the upper bound (Theorem 2)
>
> Other (relatively minor) technical novelties are as follows:
>
> **1) Careful analysis due to unknown item-level Q-values $\bar{Q}^\star_h$:**
> - Additional technical difficulties arise from the unknown $\bar{Q}^\star_h$. To prove several lemmas (Lemmas D.2, D.5, and D.7), we need to consider different cases based on the value of $f^k_{h,j}$ (optimistic estimates of $\bar{Q}^\star_h$).
>
>
> **2) $1/\sqrt{\kappa}$-improved bound for MNL bandits (Lemma D.6):**
> - The crude MNL bandit regret in Lemma D.6 improves upon the one proposed by Oh & Iyengar (2021) by a factor of  $1/\sqrt{\kappa}$, which can be exponentially large. We achieve this by using the Hessian matrix $\mathbf{H}^k_h$,  in contrast to the approach used in Oh & Iyengar (2021). This enhancement allows us to eliminate the $\frac{1}{\kappa^2}$ term from the lower-order term in the regret upper bound, leaving only a $\frac{1}{\kappa}$ factor.  Lemma D.6 is used to carefully bound the sum of $b^k_{h,1}$ (Lemma D.17).
> - Another interesting point is that $\tilde{\mathcal{P}}$ can be constructed using either optimistic or pessimistic utilities. This distinguishes our work from Oh & Iyengar (2021), which only proves the regret using optimistic utilities.
>
>
>
> **3) Choice of the exploration threshold $u_k$:**
> - To bound the size of the overly optimistic episodes $|\mathcal{K}_{oo}|$ (Lemma D.18), we must carefully choose the exploration threshold $u_k$. Since we consider the MNL preference model, the value of $u_k$ needs to be different from that used in Agrawal et al. (2023).
>
>
> In conclusion, we believe our work tackles a previously unexplored problem setting while making significant theoretical contributions. Specifically, we establish regret guarantees and derive a regret lower bound within this complex framework (with the additional assumption of linear MDPs). These accomplishments go well beyond a simple combination of existing ideas and represent a meaningful advancement in the field.
>
> ---
>
> ### **Empirical results**
>
> Following the reviewer’s suggestion, we conducted experiments in the linear MDP setting.
>
> We consider an *online shopping with budget* (see Figure G.1) under linear MDPs and an MNL user preference model. To simplify the presentation, we represent the set of states as $\mathcal{S} = \\{s_1, \dots, s_{|\mathcal{S}|} \\}$ and  the set of items as $\mathcal{I} = \\{1, \dots, N , 0 \\}$ ($0$ denotes the outside option).  Each state $s_j \in \mathcal{S}$ corresponds to a user's budget level, where a larger index $j$  indicates a higher budget (e.g., $s_{|\mathcal{S}|}$ represents the state with the largest budget). The initial state is set to the medium budget state $s_{\lceil  |\mathcal{S}|/2 \rceil }$. Furthermore, we let the transition probabilities, rewards, and preference model be the same for all $h \in [H]$, and thus we omit the subscript $h$.
>
> At state $s_j$, the agent offers an assortment $A \in \mathcal{A}$ with the maximum size of $M$.  The user then purchases an item $i \in A$. Then, the reward  is defined as follows:
>
> - If the chosen item $i$ is not the outside option, the reward is $r(s_j,i) = (\frac{i}{100N} + \frac{j}{|\mathcal{S}|})/H$.
> - If the chosen item $i$ is the outside option, the reward is $r(s_j,i) = 0$.
>
> The reward can be regarded as the *user's rating* of the purchased item. It is reasonable to assume that, at higher budget states, users tend to be more generous in their ratings, leading to higher ratings (rewards). And the transition probability is defined as follows:
>
> - If the chosen item $i$ is not the outside option, the transition probability is:
>     $\mathbb{P}(s_{\min(j+1, |\mathcal{S}|) } |s_j,i) = 1 - \frac{i}{N}$ and $\mathbb{P}( s_{\max(j-1, 0)} |s_j,i) = \frac{i}{N}.$
> - If the chosen item $i$ is the outside option, the transition probability is $\mathbb{P}(s_{\min(j+1, |\mathcal{S}|) } |s_j,i) = 1$.
>
> If the user does purchase an item, the budget level decreases with a certain probability that depends on the chosen item. Conversely, if the user does not purchase any item ($i=0$), the budget level increases deterministically.

---

> ### Author Response · Authors · 2024-11-19
> **Official Comment by Authors [3/3]**
>
> We construct the feature map $\psi(s,i)$ (for linear MDPs) using SVD. Specifically, the transition kernel $\mathbb{P}(\cdot | \cdot, \cdot) \in \mathbb{R}^{|\mathcal{S}||\mathcal{I}| \times |\mathcal{S}|}$ has at most $|\mathcal{S}|$ singular values, and the reward vector $r( \cdot, \cdot) \in \mathbb{R}^{|\mathcal{S}||\mathcal{I}|}$ has one singular value. Consequently, the feature map $\psi(s,i) \in \mathbb{R}^{d_{lin}}$ lies in a space of dimension $|\mathcal{S}| + 1$, i.e., $d_{lin} = |\mathcal{S}| + 1$.
>
> For MNL preference model, the true parameter $\theta^\star \in \mathbb{R}^d$, and the feature  $\phi(s,a) \in \mathbb{R}^d$ (for MNL preference model) are randomly sampled from a $d$-dimensional uniform distribution in each instance.
>
>
> We set $K = 30000, H=5, M=6,  |\mathcal{S}|=5, d = 5$ (feature dimension for MNL preference model), $d^{lin} = 6$ (feature dimension for linear MDP), $N \in \\{10, 20, 40\\}$ (the number of items), and $|\mathcal{A}| = \sum_{m=1}^{M-1}$$N \choose m$ $\in \\{ 637, 21699, 760098\\}$ (the number of assortments). Moreover, for simplicity, we set $\bar{\sigma}^k_h = 1$ in our algorithm.
>
> We compare our algorithm with two baselines: *Myopic* and *LSVI-UCB*.
>
> - *Myopic* is a variant of OFU-MNL+ (Lee & Oh, 2024) adapted for **unknown** rewards. It is a short-sighted algorithm that selects assortments based only on immediate rewards, ignoring state transitions.
> - *LSVI-UCB* (Jin et al., 2020) treats each assortment as a single, atomic (holistic) action, requiring enumeration of all possible assortments.
>
> To demonstrate the effectiveness of our approach, we also include the performance of the optimal policy (*Optimal*) to highlight that our algorithm is converging toward optimality. We run the algorithms on 10 independent instances and report the average total return and runtime across all episodes.
>
>
> | Return (Runtime) | Optimal  | Myopic               | LSVI-UCB(holistic)            | MNL-VQL(ours)       |
> |------------------------------|----------|-----------------------|-------------------------------|---------------------|
> | $N = 10, \|\mathcal{A}\| = 637$          | 18,632.7  | 12,754.8 (2,663.5s)     | 11,495.5 (4,088.8s)             | **18,097.4** (13,904.9s)  |
> | $N = 20, \|\mathcal{A}\| = 21,699$        | 19,500.5  | 12,588.2 (2,906.1s)     | still running (1.7 days estimated) | **18,586.3** (15,780.7s)  |
> | $N = 40, \|\mathcal{A}\| = 760,098$       | 20,283.2  | 11,826.3 (3,398.5s)     | still running (157.5 days estimated) | **19487.5** (18,605.2s)  |
>
>
> As shown in the table above, our algorithm **significantly outperforms** the baseline algorithms. We have also included experimental results in the revised paper (Appendix G).
>
> Although the runtime of *Myopic* is approximately 5.3 times faster than ours, its performance is substantially worse, converging to a suboptimal solution (see Figure G.2). This underscores a key limitation of the myopic strategy—it can completely fail in certain environments, highlighting the importance of accounting for long-term outcomes.
> Additionally, the runtime of *LSVI-UCB* increases exponentially as $N$ grows, because it requires enumerating all possible assortments. Due to the extremely slow runtime of *LSVI-UCB*, we omitted its performance results for $N=20$ and $N=40$. However, even for smaller $N$ ($N=10$), *LSVI-UCB* shows the worst performance. Based on these results, we hypothesize that its performance would not improve for larger values of $N$.
>
> ---
> ### **Extension to the multiple-choice setting**
>
> As you suggested in Question 1, we agree that developing a new framework where a user can adopt multiple items while considering state transitions is a very intriguing idea. We believe that extending our framework to include such a multiple-choice setting would be both meaningful and valuable as a direction for future research. Thank you for suggesting this nice idea!

---

> > ### Comment · Reviewer_aS1z · 2024-11-27
> >
> > The reviewer appreciates the authors’ detailed rebuttal and acknowledges the additional experiments and clarification of the main contributions and technical novelties, which are well-presented and helpful. I am still positive about the current work and keep my score unchanged.

---

> > > ### Author Response · Authors · 2024-11-27
> > >
> > > Thank you so much for your continued support of our work! We’re delighted to know that we successfully addressed your concerns and clarified our main contributions and technical novelties. Your constructive feedback has been incredibly valuable in improving our paper, and we deeply appreciate your input. Thank you again!

---

### Official Review · Reviewer_Qxxw · 2024-11-04

**Soundness:** 3
**Presentation:** 2
**Contribution:** 2
**Rating:** 6
**Confidence:** 2

**Summary:**

The paper tackles Combinatorial Reinforcement Learning (RL), which refers to as a class of RL problems where the action space is combinatorial, meaning that the agent selects a combination or subset of base actions from a set of possible base actions. The paper proposes a computationally efficient algorithm and provides theoretical regret guarantee.

**Strengths:**

* The paper tries to integrate the concept of multinomial logit user feedback into reinforcement learning, which could model the practical cases in applications like online retail and advertising, etc.

**Weaknesses:**

* In Line 48, "However, these studies take a myopic approach, optimizing for immediate rewards without considering the long-term impact on user behavior." -> I cannot agree with this. I believe that any approaches that balance exploration-exploitation take into account the long-term impact, including bandit algorithms.
* The regret definition seems strange to me. The definition in Line 192 seems like the regret is only measured on state $s_1$.
* The organization of the paper is not very easy to follow.

**Questions:**

* I am confused with the role of function class mentioned in Assumption 2. Personally I don't regard such formulation as standard in RL literature, rather, it seems more like a way to simplify the function approximation, in the sense that the problem is formulated more in a bandit setting side.
* Another confusion of mine is regarding the state space. It is not explicitly defined in the paper whether the state space is finite or not. If it's finite, then by estimating value function of each state action pair, I am more interested in the sample complexity of the proposed algorithm.

---

> ### Author Response · Authors · 2024-11-19
>
> Thank you for your valuable feedback and the time and effort you invested in reviewing our paper. We have revised the paper to incorporate all your comments and will address your questions below.
>
> ---
> ### **RL vs bandit algorithms (Line 48 in the original version, Line 50 in the revised version)**
> The goal of bandit algorithms is to maximize the immediate reward from a single decision, while RL algorithms aim to maximize the cumulative reward over the entire horizon (until the end of the episode). This fundamental difference leads to distinctly different behaviors. Let us clarify this with an example illustrated in Figure A.1:
>
> In Figure A.1, a user might choose a *junk* item that provides a high immediate reward but causes a transition to a low-satisfaction state, where all subsequent rewards are uniformly reduced. Repeatedly recommending such *junk* items can lead to user fatigue, resulting in lower satisfaction or loyalty to the system and ultimately a lower cumulative reward.
>
> In this scenario, RL algorithms take the user's state into account to decide which item will maximize the sum of future rewards. In contrast, bandit algorithms prioritize items with immediate high rewards (e.g., *junk* items) and continue recommending them. As a result, **bandit algorithms can completely fail to maximize the cumulative reward**, whereas RL algorithms excel at doing so by considering long-term outcomes.
>
> ---
> ### **Initial state in the regret definition (Line 192 in the original version, Line 194 in the revised version)**
>
> The regret is measured at state $s^k_1$, **NOT** $s_1$. The initial state $s^k_1$ is chosen arbitrarily (Line 159 in the original version, Line 161 in the revised version). It appears that you have misread the text.
>
> ---
> ### **Presentation**
> To clearly convey our main contributions and technical novelties, we have revised the paper. The major changes are highlighted in red. The key revisions are as follows:
>
> - Providing a clearer explanation of our key challenges, main contributions, and technical novelties in the Introduction (Section 1).
> - Adding an overview of the algorithm and improving the connections between its components in Section 4.
> - Including numerical experiments in Section 5 and Appendix G.
>
> ---
> ### **Questions**
>
> #### **Q1. I am confused with the role of function class mentioned in Assumption 2. Personally I don't regard such formulation as standard in RL literature, rather, it seems more like a way to simplify the function approximation, in the sense that the problem is formulated more in a bandit setting side.**
>
> First, we are unclear why you describe Assumption 2 as being "formulated more on the bandit setting side." The **completeness assumption is a concept specific to RL** (and is commonly used in RL with general function approximation)  and does not apply to the bandit setting.
>
> Furthermore, as outlined in Remark 2, the second moment completeness assumption we employ is stronger than those used in prior works such as (Wang et al., 2021; Jin et al., 2021). However:
>
> - It naturally **holds for both tabular [1] and linear MDPs [2]**.
> - It is the same as the assumptions in Agarwal et al. (2023) and Zhao et al. (2023).
>
> To the best of our knowledge, no existing theoretical results address combinatorial RL with preference feedback, even in simpler settings like tabular or linear MDPs. Therefore, **we strongly believe that concerns about the use of general function approximation (e.g., the second moment completeness assumption in a general function class) should NOT be viewed as a limitation of our paper.** Instead, addressing general function approximation—a more complex and general case—**further strengthens** the contributions of our work.
>
> For reference, we could remove the second moment completeness assumption. However, this would result in slightly looser bounds, as the analysis would then rely solely on unweighted regression.
>
>
> [1] Mohammad Gheshlaghi Azar, Ian Osband, and R´emi Munos. Minimax regret bounds for reinforcement learning. In Proceedings of the 34th International Conference on Machine Learning-Volume 70, pages 263–272. JMLR. org, 2017.
>
> [2] He, Jiafan, et al. "Nearly minimax optimal reinforcement learning for linear markov decision processes." International Conference on Machine Learning. PMLR, 2023.
>
> #### **Q2. Another confusion of mine is regarding the state space. It is not explicitly defined in the paper whether the state space is finite or not. If it's finite, then by estimating value function of each state action pair, I am more interested in the sample complexity of the proposed algorithm.**
>
> We allow the state space to be infinite, and we will clarify this point in the revised version of our paper.

---

### Official Review · Reviewer_BLXB · 2024-11-04

**Soundness:** 3
**Presentation:** 2
**Contribution:** 3
**Rating:** 5
**Confidence:** 3

**Summary:**

This paper studies combinatorial reinforcement learning with preference feedback problems in which a learner offers a set of actions (referred to as assortment) to a user, and he prefers one of the actions, where his preference follows a multinomial logit (MNL) model. The goal is to learn a policy for selecting an assortment from a combinatorial action space in each step of every episode that minimizes the total regret, i.e., the sum of the difference between the state value of the optimal policy for a given initial state and state value of the policy for given initial state.

The authors proposed an algorithm named MNL-VQL (MNL Preference Model with Variance-weighted Item-level Q-Learning) for this problem with a sub-linear regret guarantee, even achieving near-optimal regret bounds for linear Markov Decision Processes (MDPs).

**Strengths:**

**The following are the strengths of the paper:**
1. This paper extends the existing MNL bandit setting to the combinatorial RL setting. This problem is motivated by many real-world applications like online recommendation and advertising, where long-term user engagement is modeled using state transitions and preference feedback over shown assortments.

2. The authors propose the MNL-VQL algorithm for combinatorial reinforcement learning with preference feedback problems. They have shown that their algorithm enjoys sub-linear regret bounds guarantees, showing near-optimal performance for linear MDPs.

**Weaknesses:**

**The following are the weaknesses of the paper:**
1. It is unclear how the proposed algorithm is computationally efficient as it has to search all possible assortments (that can be exponentially large) to find the assortment to users (as defined in Eq. (9)). Is there a way to solve the optimization problem in Eq. (9) efficiently, avoiding the exponential cost of enumerating all assortments?

2. The proposed algorithm has many hyperparameters, and it is unclear how one can choose them in practice. Furthermore, without any experiment results (or ablations study), it is unclear how the proposed algorithm will behave in practice. Are there any guidelines for selecting the key hyperparameters, or is it possible to have a small experiment demonstrating the algorithm's sensitivity to different choices of hyperparameters?

3. This paper considers time-inhomogeneous MDPs, but it is unclear how the non-stationarity (in transition kernel and reward function) across episodes is dealt with. Can you point where in algorithm we can see explicit treatment of these time-varying dynamics?

4. The rewards can not be sparse (Line 249) because rewards in each step are needed to get estimates of the item-level Q-functions (Line 281 and Line 287), also needed for estimating the second moment (Line 294).

**Questions:**

Please address the weaknesses of the paper. I have a few more questions/comments:
1. How does one deal with the problem of changing horizon length (e.g., horizon length cannot be fixed in real-life applications like online recommendations and can be different for different users)?

2. What is the value of generalized Eluder dimension for different function classes, e.g., linear, neural network, and kernel regression with different kernels? Is it always finite, irrespective of function class?

3. Why is the outside option $a_0$ considered when selecting the assortment, as it is already part of every assortment? Without this, pessimistic utility estimates (Eq. (7)) are not needed. Right?

**Minor comment:**
1. It is clear enough how the pessimistic estimates ($j = -2$) of item-level Q values are used in Eq. (10). My understanding is that it is part of the **otherwise** case.

2. I find it hard to follow the paper from page 5 to page 9 (except page 6). The authors can make this part of the paper more readable by clearly connecting different parts of the algorithm.

I am open to changing my score based on the authors' responses.

**Details Of Ethics Concerns:**

Since this work is a theoretical paper, I do not find any ethical concerns.

---

> ### Author Response · Authors · 2024-11-19
> **Official Comment by Authors [1/5]**
>
> Thank you for taking the time to review our paper and provide feedback. To begin with, we kindly ask for your patience with our detailed responses. We have revised the paper to fully address your concerns.
>
> ---
> ### **Computational efficiency in Eq. (9)**
> We respectfully clarify the misunderstaning in the reviewer's comment. We do **NOT** need to enumerate all possible assortments. As clearly explained in Remark 4 (immediately following Equation 9), the optimization problem presented in Equation 9 can be reformulated as a linear programming (LP) problem. This reformulation enables efficient solving, with a computational complexity that is polynomial in the number of items. For more details, please refer to Appendix C. There is NO "exponential cost of enumerating all assortments."
>
> ---
>
> ### **Hyperparameters & General function approximation**
>
> In practice, a simple way to implement our algorithm is to use unweighted regression ($\bar{\sigma}^k_h = 1$) instead of the weighted regression described in Equation (4). In this case, the **only** parameters to specify are the confidence radius $\alpha^k_h$ (for MNL) and $\beta^k_h$ (for function approximation). The parameter $\alpha^k_h$ can be computed explicitly in MNL bandits, and $\beta^k_h$ can also be determined in closed form for both tabular and linear MDPs.
>
> In the case of general function approximation with unweighted regression ($\bar{\sigma}^k_h = 1$), the main requirement is the computation of the bonus term $b^k_h (s,a)$. While this term may not always be explicitly computable in practice (e.g., when using neural networks), this limitation reflects a fundamental challenge inherent in using general function approximation—one that is common across the RL community. It is important to emphasize that the difficulty of conducting experiments with general function approximation is **NOT** unique to our work. Instead, it is a **common challenge** faced by all algorithms relying on general function approximation. To the best of our knowledge, no existing RL algorithms have conducted experiments under general function approximation.
>
> For reference, we would like to justify why unweighted regression can be used in practice. The weighted regression is typically employed to achieve a tighter regret bound, but it requires specifying additional parameters. To the best of our knowledge, there are no empirical results evaluating the use of unweighted regression in RL theory, even in tabular settings [1], linear MDPs [2], linear mixture MDPs [3], or general function approximation (Agarwal et al., 2023). Thus, we believe that simply setting $\bar{\sigma}^k_h = 1$ is sufficient in practice.
>
> We strongly believe that **concerns regarding challenges such as the difficulty of specifying parameters when using general function approximation should not be viewed as a weakness of our work.** To the best of our knowledge, there are no existing theoretical results for combinatorial RL with preference feedback, even in simpler cases like tabular or linear MDPs. By tackling general function approximation—a much broader and more complex framework than tabular or linear MDPs—we ONLY strengthen the significance and contributions of our work. While we could have chosen to focus solely on results for linear MDPs, we deliberately chose to address the more general and challenging setting instead.
>
> [1] Mohammad Gheshlaghi Azar, Ian Osband, and R´emi Munos. Minimax regret bounds for reinforcement learning. In Proceedings of the 34th International Conference on Machine Learning-Volume 70, pages 263–272. JMLR. org, 2017.
>
> [2] He, Jiafan, et al. "Nearly minimax optimal reinforcement learning for linear markov decision processes." International Conference on Machine Learning. PMLR, 2023.
>
> [3] Zhou, Dongruo, Quanquan Gu, and Csaba Szepesvari. "Nearly minimax optimal reinforcement learning for linear mixture markov decision processes." Conference on Learning Theory. PMLR, 2021.
>
> ---
>
> ### **Empirical results**
>
> Following the reviewer’s suggestion, we conducted experiments in the linear MDP setting. In linear MDPs, all parameters can be explicitly calculated (as previously explained), **so there is no need to specify any additional parameters.**
>
> We consider an *online shopping with budget* (see Figure G.1) under linear MDPs and an MNL user preference model. To simplify the presentation, we represent the set of states as $\mathcal{S} = \\{s_1, \dots, s_{|\mathcal{S}|} \\}$ and  the set of items as $\mathcal{I} = \\{1, \dots, N , 0 \\}$ ($0$ denotes the outside option).  Each state $s_j \in \mathcal{S}$ corresponds to a user's budget level, where a larger index $j$  indicates a higher budget (e.g., $s_{|\mathcal{S}|}$ represents the state with the largest budget). The initial state is set to the medium budget state $s_{\lceil  |\mathcal{S}|/2 \rceil }$. Furthermore, we let the transition probabilities, rewards, and preference model be the same for all $h \in [H]$, and thus we omit the subscript $h$.

---

> ### Author Response · Authors · 2024-11-19
> **Official Comment by Authors [2/5]**
>
> At state $s_j$, the agent offers an assortment $A \in \mathcal{A}$ with the maximum size of $M$.  The user then purchases an item $i \in A$. Then, the reward  is defined as follows:
>
> - If the chosen item $i$ is not the outside option, the reward is $r(s_j,i) = (\frac{i}{100N} + \frac{j}{|\mathcal{S}|})/H$.
> - If the chosen item $i$ is the outside option, the reward is $r(s_j,i) = 0$.
>
> The reward can be regarded as the *user's rating* of the purchased item. It is reasonable to assume that, at higher budget states, users tend to be more generous in their ratings, leading to higher ratings (rewards). And the transition probability is defined as follows:
>
> - If the chosen item $i$ is not the outside option, the transition probability is:
>     $\mathbb{P}(s_{\min(j+1, |\mathcal{S}|) } |s_j,i) = 1 - \frac{i}{N}$ and $\mathbb{P}( s_{\max(j-1, 0)} |s_j,i) = \frac{i}{N}.$
> - If the chosen item $i$ is the outside option, the transition probability is $\mathbb{P}(s_{\min(j+1, |\mathcal{S}|) } |s_j,i) = 1$.
>
> If the user does purchase an item, the budget level decreases with a certain probability that depends on the chosen item. Conversely, if the user does not purchase any item ($i=0$), the budget level increases deterministically.
>
> We construct the feature map $\psi(s,i)$ (for linear MDPs) using SVD. Specifically, the transition kernel $\mathbb{P}(\cdot | \cdot, \cdot) \in \mathbb{R}^{|\mathcal{S}||\mathcal{I}| \times |\mathcal{S}|}$ has at most $|\mathcal{S}|$ singular values, and the reward vector $r( \cdot, \cdot) \in \mathbb{R}^{|\mathcal{S}||\mathcal{I}|}$ has one singular value. Consequently, the feature map $\psi(s,i) \in \mathbb{R}^{d_{lin}}$ lies in a space of dimension $|\mathcal{S}| + 1$, i.e., $d_{lin} = |\mathcal{S}| + 1$.
>
>
> For MNL preference model, the true parameter $\theta^\star \in \mathbb{R}^d$, and the feature  $\phi(s,a) \in \mathbb{R}^d$ (for MNL preference model) are randomly sampled from a $d$-dimensional uniform distribution in each instance.
>
>
> We set $K = 30000, H=5, M=6,  |\mathcal{S}|=5, d = 5$ (feature dimension for MNL preference model), $d^{lin} = 6$ (feature dimension for linear MDP), $N \in \\{10, 20, 40\\}$ (the number of items), and $|\mathcal{A}| = \sum_{m=1}^{M-1}$$N \choose m$ $\in \\{ 637, 21699, 760098\\}$ (the number of assortments). Moreover, for simplicity, we set $\bar{\sigma}^k_h = 1$ in our algorithm.
>
> We compare our algorithm with two baselines: *Myopic* and *LSVI-UCB*.
>
> - *Myopic* is a variant of OFU-MNL+ (Lee & Oh, 2024) adapted for **unknown** rewards. It is a short-sighted algorithm that selects assortments based only on immediate rewards, ignoring state transitions.
> - *LSVI-UCB* (Jin et al., 2020) treats each assortment as a single, atomic (holistic) action, requiring enumeration of all possible assortments.
>
> To demonstrate the effectiveness of our approach, we also include the performance of the optimal policy (*Optimal*) to highlight that our algorithm is converging toward optimality. We run the algorithms on 10 independent instances and report the average total return and runtime across all episodes.
>
>
> | Return (Runtime) | Optimal  | Myopic               | LSVI-UCB(holistic)            | MNL-VQL(ours)       |
> |------------------------------|----------|-----------------------|-------------------------------|---------------------|
> | $N = 10, \|\mathcal{A}\| = 637$          | 18,632.7  | 12,754.8 (2,663.5s)     | 11,495.5 (4,088.8s)             | **18,097.4** (13,904.9s)  |
> | $N = 20, \|\mathcal{A}\| = 21,699$        | 19,500.5  | 12,588.2 (2,906.1s)     | still running (1.7 days estimated) | **18,586.3** (15,780.7s)  |
> | $N = 40, \|\mathcal{A}\| = 760,098$       | 20,283.2  | 11,826.3 (3,398.5s)     | still running (157.5 days estimated) | **19487.5** (18,605.2s)  |
>
>
> As shown in the table above, our algorithm **significantly outperforms** the baseline algorithms. We have also included experimental results in the revised paper (Appendix G).
>
> Although the runtime of *Myopic* is approximately 5.3 times faster than ours, its performance is substantially worse, converging to a suboptimal solution (see Figure G.2). This underscores a key limitation of the myopic strategy—it can completely fail in certain environments, highlighting the importance of accounting for long-term outcomes.
> Additionally, the runtime of *LSVI-UCB* increases exponentially as $N$ grows, because it requires enumerating all possible assortments. Due to the extremely slow runtime of *LSVI-UCB*, we omitted its performance results for $N=20$ and $N=40$. However, even for smaller $N$ ($N=10$), *LSVI-UCB* shows the worst performance. Based on these results, we hypothesize that its performance would not improve for larger values of $N$.

---

> ### Author Response · Authors · 2024-11-19
> **Official Comment by Authors [3/5]**
>
> ### **Time-inhomogeneous MDPs**
>
> In model-free RL (our setting) [1, 2, 3, 4], which approximates the $Q_h$ functions (in our case, $\bar{Q}\_h$) without explicitly learning the underlying MDP such as the transition kernel or reward function, **estimating $Q_h$ is sufficient to handle time-inhomogeneous dynamics**. Specifically, recall the definition: $Q_h(s,a) = r_h(s,a) + \sum_{s'}\mathbb{P}\_h(s'|s,a) V\_{h+1}(s')$. This equation shows that $Q_h$ inherently captures the time-inhomogeneous nature of both the reward function and the transition kernel.  Therefore, even if the transition kernel and reward function vary over time, directly learning $Q_h$ is sufficient to address these variations.
>
> For your reference, it is very common and widespread in the RL literature [1, 2, 3, 4, 5] to consider time-inhomogeneous MDPs.
>
> [1] Chi Jin, Zhuoran Yang, Zhaoran Wang, and Michael I Jordan. Provably efficient reinforcement learning with linear function approximation. In Conference on Learning Theory, pp. 2137–2143. PMLR, 2020
>
>  [2] Ishfaq, Haque, et al. "Randomized exploration in reinforcement learning with general value function approximation." International Conference on Machine Learning. PMLR, 2021.
>
>  [3] Dylan J Foster, Sham M Kakade, Jian Qian, and Alexander Rakhlin. The statistical complexity of interactive decision making. arXiv preprint arXiv:2112.13487, 2021.
>
>  [4] Alekh Agarwal, Yujia Jin, and Tong Zhang. Vo q l: Towards optimal regret in model-free rl with nonlinear function approximation. In The Thirty Sixth Annual Conference on Learning Theory, pp. 987–1063. PMLR, 2023.
>
>  [5] Zhou, Dongruo, Quanquan Gu, and Csaba Szepesvari. "Nearly minimax optimal reinforcement learning for linear mixture markov decision processes." Conference on Learning Theory. PMLR, 2021.
>
> ---
>
> ### **Meaning of "sparse" reward**
>
> To clarify, by "sparse" rewards, we mean that **the total sum of rewards is bounded by 1**, not that most rewards are zero. Our primary focus is on a scenario where rewards are small in magnitude. Since the term "sparsity" might confuse readers, we have decided to remove it from Line 249 in the revised version of our paper.
>
> For reference, if the rewards are genuinely "sparse" (i.e., most rewards are zero, which is not the case in our study), you can handle this by assigning a reward of zero in Lines 281, 287, and 294. In other words, "sparse" rewards do not mean that these rewards should be excluded from the regression. Instead, it simply indicates that most rewards are zero, so including a zero reward as the regression target is sufficient.
>
> ---
> ### **Questions**
>
> #### **Q1. How does one deal with the problem of changing horizon length (e.g., horizon length cannot be fixed in real-life applications like online recommendations and can be different for different users)?**
>
> Let an absorbing state exist, representing the scenario where the user leaves the system. If a user exits the system before reaching the end of the horizon $H$, this situation is treated as a transition to the absorbing state (or "user-exited state"). Once in the absorbing state, the agent remains there for the rest of the episode. This approach ensures that our problem setting aligns with real-life applications, where a user may leave before the end of the horizon.
>
> #### **Q2. What is the value of generalized Eluder dimension for different function classes, e.g., linear, neural network, and kernel regression with different kernels? Is it always finite, irrespective of function class?**
>
>  In [1, 2], it was shown that the Eluder dimension (Russo & Van Roy, 2013) is small in several cases, such as generalized linear models and linear quadratic regulators (LQR). However, as demonstrated in [3], the Eluder dimension can be exponentially large even for a single ReLU neuron, suggesting significant challenges when applying it to neural network settings. Since the generalized Eluder dimension is upper bounded by the standard Eluder dimension up to logarithmic terms (Theorem 4.6 of Zhao et al., 2023), it too can be exponentially large for neural networks.
>
> That being said, as we previously emphasized, the limitations of general function approximation are not specific to our work and therefore should NOT be regarded as a weakness of our paper.
>
> [1] Ruosong Wang, Ruslan Salakhutdinov, and Lin F Yang. Provably efficient reinforcement learning with general value function approximation. arXiv preprint arXiv:2005.10804, 2020.
> [2] Jin, Chi, Qinghua Liu, and Sobhan Miryoosefi. "Bellman eluder dimension: New rich classes of rl problems, and sample-efficient algorithms." Advances in neural information processing systems 34 (2021): 13406-13418.
> [3] Dong, Kefan, Jiaqi Yang, and Tengyu Ma. "Provable model-based nonlinear bandit and reinforcement learning: Shelve optimism, embrace virtual curvature." Advances in neural information processing systems 34 (2021): 26168-26182.

---

> ### Author Response · Authors · 2024-11-19
> **Official Comment by Authors [4/5]**
>
> #### **Q3-1. Why is the outside option $a_0$ considered when selecting the assortment, as it is already part of every assortment?**
>
>  Including the outside option $a_0$ is both **natural and practical**, as it reflects real-world scenarios where users are not obligated to choose an item. In most cases, users have the freedom to opt out, making the inclusion of $a_0$ essential for accurately modeling user behavior. Excluding it would assume that users must always select an item, which is rarely the case. Moreover, the inclusion of the outside option is also common in MNL bandits (Oh & Iyengar, 2019; Faury et al., 2020; Abeille et al., 2021; Faury et al., 2022; Oh & Iyengar, 2021;Perivier & Goyal, 2022; Agrawal et al., 2023; Lee & Oh, 2024).
>
> #### **Q3-2. Without this, pessimistic utility estimates (Eq. (7)) are not needed. Right?**
>
> To answer your second question, our response is "Yes." However, in such a case, **the problem simplifies to a standard RL problem where only one item is selected at a time.** Given the item-level Q functions $\bar{Q}^\star_h(s,a)$, the optimal policy is to select the **single** item (not including the outside option) with the highest $\bar{Q}^\star_h(s,a)$.
> Note that if multiple items share the maximum value of  $\bar{Q}^\star_h(s,a)$, selecting any one of them or all of them as the assortment both constitute optimal policies.
> To formalize this, let $A^\star_h$ be the set of items with the maximum value of $\bar{Q}^\star_h(s,a)$.
> Then, we have $V_h^\star(s) =  \sum_{a'}\mathcal{P}_h(a'|s,A^\star_h)\bar{Q}^\star_h(s,a') = \bar{Q}^\star_h(s,a)$ for any $a \in A^\star_h$.
> This shows that any $a \in A^\star_h$ maximizes $\bar{Q}^\star_h(s,a)$, meaning that selecting the single item with the highest $\bar{Q}^\star_h(s,a)$ is one of the optimal policies.
>
> Since one of the optimal policies involves selecting only a single item, the task simplifies to selecting a single item during each training episode, which makes the problem less interesting.
>
> ---
> #### **Minor-1. It is clear enough how the pessimistic estimates ($j=-2$) of item-level Q values are used in Eq. (10). My understanding is that it is part of the otherwise case.**
>
>  In Equation 10, the assortment for $j=-2$ is not utilized. Only the assortments for $j=1,2$ are considered, as they are necessary to ensure optimism (as established in Lemma D.9). For reference, the value estimates for $j=-2$ are calculated to estimate the variance (see Line 15 in Algorithm 1).
>
> #### **Minor-2. I find it hard to follow the paper from page 5 to page 9 (except page 6). The authors can make this part of the paper more readable by clearly connecting different parts of the algorithm.**
>
> In response to the reviewers' feedback, we have revised the paper to more effectively highlight our contributions and technical novelties. The major changes are highlighted in red, and the key revisions are as follows:
>
> - Providing a clearer explanation of our key challenges, main contributions, and technical novelties in Introduction (Section 1).
> - Adding an overview of the algorithm and improving the connections between its components in Section 4.
> - Including numerical experiments in Section 5 and Appendix G.
>
> ---
>
> ### **Additional comment on our contributions**
>
> We believe we have thoroughly addressed all your concerns. To further clarify, we would like to highlight our main contributions.
>
> To begin, consider the following questions:
>
> - Has there been any prior work on combinatorial RL with function approximation (with regret guarantees)? **No**.
> - Has there been any prior work on RL with MNL choice feedback on actions (with regret guarantees)? **No**.
>
> Our contributions encompass both of these frameworks and extend far beyond a simple combination of these areas. Each component—combinatorial RL, preference feedback, and function approximation—is inherently challenging. Integrating these into a unified framework requires careful and sophisticated theoretical analysis. Moreover, this integration introduces unique challenges that prior methods have not addressed, such as:
>
>
> - **Unknown item values**, as the rewards and transition probabilities are unknown, unlike in MNL bandits.
> - **Limitations of using optimistic utilities alone**, as they are insufficient to guarantee optimism, unlike in MNL bandits.
> - **Technical challenges in proving optimism** when item values are unknown.
> - The **absence of regret lower bounds**, even for linear MDPs.
>
> We emphasize that, to the best of our knowledge, this work provides the **first theoretical guarantee for combinatorial RL with preference feedback**. Furthermore, we establish the **first regret lower bound for combinatorial linear MDPs with preference feedback**, which serves as a fundamental baseline and underscores the intrinsic difficulty of the problem. Our regret analysis is novel, carefully crafted to address the interplay between combinatorial actions, preference feedback, and linear function approximation.

---

> > ### Author Response · Authors · 2024-11-19
> > **Official Comment by Authors [5/5]**
> >
> > In summary, our work not only tackles a previously unexplored problem setting but also makes significant theoretical contributions. By providing regret guarantees and deriving a regret lower bound (in combinatorial linear MDPs with preference feedback), we move well beyond a simple combination of existing ideas, marking a substantial advancement in the field.

---

> > > ### Author Response · Authors · 2024-11-25
> > > **Dear Reviewer BLXB**
> > >
> > > Dear Reviewer BLXB,
> > >
> > > As we near the end of the discussion period, we want to make sure all your questions and comments have been fully addressed. We believe that the points you had mentioned are adequately addressed. However,
> > > if you have any remaining questions or need further clarification, please let us know. Otherwise, we kindly ask you to take a moment to re-evaluate our work.

---

> ### Author Response · Authors · 2024-12-02
> **Dear Reviewer BLXB,**
>
> Dear Reviewer BLXB,
>
> As our extended discussion period approaches its conclusion, we would like to ensure that our previous responses have fully addressed all your questions and concerns. We have made every effort to provide thorough and thoughtful answers and to demonstrate the practicality of our approach through clear experimental evidence.
>
> We strongly believe that **our work merits acceptance due to its significance as the first theoretical study in combinatorial RL with a preference feedback framework.** Introducing a new framework based on existing ones is impactful, as it opens up new research directions. Our framework, in particular, broadens RL research to include applications beyond traditional areas like recommender systems and online shopping, extending to fields such as online dialogue systems (e.g., ChatGPT).
>
> For instance, in a system like ChatGPT, users interact with the model through multiple actions: accepting an answer, regenerating it, or asking follow-up questions. These interactions fit naturally into our combinatorial RL with preference feedback framework. Unlike existing works that focus solely on and learn from pairwise feedback (PbRL), our framework allows models to learn from ongoing multi-turn conversations, opening up exciting opportunities for advancements in AI dialogue systems.
>
> Additionally, the **technical challenges addressed in our paper are substantial** and include:
>
> 1. Proving optimism through a novel algorithm and supporting lemmas (Lemma D.2, D.9),
> 2. Establishing the first lower bound for linear MDPs with preference feedback (Theorem 3), and
> 3. Deriving tighter bounds for general function approximation (with MNL preference model) through several new lemmas (Lemmas D.12, D.13, D.14, D.17, and D.18).
>
> If you have no further questions or require no additional clarification, we kindly and respectfully ask you to consider revisiting our score. Thank you for your time and thoughtful evaluation.

---

> > ### Comment · Reviewer_BLXB · 2024-12-02
> >
> > Dear Authors,
> >
> > \
> > Thank you for your detailed response. Please clarify the following:
> > 1. The MDP considered in the paper is time-inhomogeneous across time steps of an episode but time-homogeneous across episodes. Right?
> >
> > 2. Why can each assortment not be the same as base items (there will be only one assortment), i.e., $\mathcal{A} = \mathcal{I}$ for each step? I did not find any constraint (except $M$) given in the paper for which only a subset of base items should be selected. Even when there is a constraint that makes it impossible to select all base items, the problem is a subset selection problem, which is combinatorial in nature. However, the way base items are added to the assortment is simply a sum of the weighted values of base items (Eq. (9) where only the denominator will change with the change in assortment), which is much simpler (hence can be solved using LP) than solving a combinatorial problem to select the best assortment. As an analogy, the assortment selection problem in Eq. (9) becomes equivalent to the fractional knapsack, which is much easier to solve than a combinatorial problem like a 0-1 knapsack.
> >
> > For now, I will maintain my updated score until the Reviewers/AC discussions.

---

> ### Author Response · Authors · 2024-12-02
> **Thank you!**
>
> We sincerely appreciate you raising the score and for your continued thoughtful participation in the discussion. We are pleased to provide answers to your additional questions below:
>
> > Q1. The MDP considered in the paper is time-inhomogeneous across time steps of an episode but time-homogeneous across episodes. Right?
>
> Yes, you are exactly correct. We consider *inhomogeneous MDPs*, where the transition dynamics and rewards can vary across the horizon within a single episode (time-inhomogeneous within an episode). This is distinct from *non-stationary MDPs*, where the transition dynamics and rewards change across different episodes.
>
>
>
> > Q2-1. Why can each assortment not be the same as base items (there will be only one assortment) for each step? I did not find any constraint (except $M$) given in the paper for which only a subset of base items should be selected.
>
> We allow the assortment to include all base items, meaning $M = |\mathcal{I}|$. In this case, it is possible to select the assortment $A_t = \mathcal{I}$, where every base item is included. However, please note that in practical applications, it is more common to have $M \ll |\mathcal{I}|$. For example, in video recommendation systems like YouTube and Netflix, there are millions of available items (video clips), but only a few can be displayed as recommendations to a user.
>
>
>
> > Q2-2. Even when there is a constraint that makes it impossible to select all base items, the problem is a subset selection problem, which is combinatorial in nature. However, the way base items are added to the assortment is simply a sum of the weighted values of base items (Eq. (9) where only the denominator will change with the change in assortment), which is much simpler (hence can be solved using LP) than solving a combinatorial problem to select the best assortment. As an analogy, the assortment selection problem in Eq. (9) becomes equivalent to the fractional knapsack, which is much easier to solve than a combinatorial problem like a 0-1 knapsack.
>
> First, to clarify again, there is NO constraint that prevents the selection of all base items. It is entirely possible to include every base item in the assortment.
>
> Second, Equation (9) is NOT equivalent to the fractional knapsack problem; rather, it aligns more closely with the 0-1 knapsack problem. This distinction stems from the discrete nature of assortment decisions: each item must be either fully included in the assortment or completely excluded. Unlike the fractional knapsack problem, which allows items to be partially included, assortment optimization requires a binary decision for each item, making it fundamentally a discrete optimization problem.
>
> Third, as discussed in Remark 4 and Appendix C, Equation (9) can be converted into a linear program (LP). Initially, Equation (9) represents a fractional mixed-integer program (MIP). According to Chen & Hausman (2000), the binary indicator in the MIP can be relaxed, transforming the problem into a fractional linear program. Then, using the Charnes-Cooper transformation method, this fractional linear program can be reformulated as an LP. Consequently, the optimization problem in Equation (9) can be solved in polynomial time.
>
> ---
> We hope our answers fully address your remaining concerns. If you have any further questions, please do not hesitate to reach out. If everything is clear, we would be truly grateful if you could kindly consider voting to accept our paper.

---

### Author Response · Authors · 2024-11-25
**Summary of Comments**

We sincerely appreciate the time and effort you devoted to reviewing our work and offering insightful feedback.
In response to the reviewers' requests, please note that **we updated our paper to clearly emphasize our contributions and include the requested experiments.**

With the discussion period coming to a close, we would like to summarize the key points we have addressed.

---
### **1. Contribution & Technical Novelties**

We want to emphasize that, to our best knowledge, this is the **first theoretical work on combinatorial RL with preference feedback**, addressing a broad range of real-world applications, including recommender systems, online retail, and many more. Moreover, we establish the **first regret lower bound in linear MDPs (with preference feedback)**, proving the near optimality of our proposed algorithm.


Our work is **not just a combination of prior research at all.** Rather, rigorously integrating key elements, combinatorial RL, preference feedback, and (general) function approximation, into a unified framework requires very involved analysis and addresses several unique challenges that had not been addressed previously, including:

1. **Unknown Item Values**: Since the rewards and transition probabilities are unknown, unlike in MNL bandits, the item values need to be estimated. We use general function approximation to perform this estimation.

2. **The Need for a New Exploration Strategy**: Optimistic utilities alone are insufficient to guarantee optimism in our framework, unlike in MNL bandits. To address this, we employ a novel strategy that alternates between optimistic and pessimistic utilities based on the estimated item values (Equation 8).

3. **Technical Challenges in Proving Optimism**: The new exploration strategy we employ prevents us from directly applying the optimism analysis used in the MNL bandit literature (Oh & Iyengar., 2019, and Lee & Oh ., 2024).

4. **Technical Challenges Due to Unknown Item Values**: We require more sophisticated analyses in several lemmas (e.g., Lemmas D.2, D.5, and D.7) because the true item values are unknown.

5. **The Absence of Regret Lower Bounds**: Even for linear MDPs (with preference feedback), there are no existing regret lower bounds. To address this, we construct a complex multi-layered MDP and employ novel analytical techniques.

These challenges highlight the depth and novelty of our work, going far beyond a straightforward combination of existing ideas, which are evidently shown in the analysis.


### **2) Experiment**
 We have included the experiments in the revised version as per the reviewers' requests.


### **3) Presentation**
The paper's dense presentation primarily stems from our use of general function approximation and the complex notations necessary for weighted regression. These elements are essential for achieving a tight regret bound, and the notations largely follow those introduced by Agarwal et al., 2023. Consequently, the heavy notation should NOT be considered a weakness of our work. This part is rather improvable upon acceptance.

---
We believe that our results represent a significant advancement in RL theory and are substantial enough to warrant acceptance.

---

### Public Comment · ~Joongkyu_Lee1 · 2025-02-17
**Notification of Revision Update**

We have posted a revised version of our manuscript on arXiv (http://arxiv.org/abs/2502.10158). In this update, we have addressed all concerns raised during the ICLR 2025 discussion period.

We encourage readers to refer to the updated version.

---
### **Key Revisions and Improvements**:

- Inclusion of **experimental results** in the main paper (Section 6)

- Clearer articulation of our **technical novelties** in the Introduction (Section 1)

- Stronger emphasis on how **our work goes beyond a simple combination of existing methods** (Section 5.2)

- **Improved regret upper bound** from $\tilde{\mathcal{O}} \big( d H \sqrt{ K} + \sqrt{\operatorname{dim}(\mathcal{F}) KH \log |\mathcal{F}|  } \big)$ to $\tilde{\mathcal{O}} \big( d \sqrt{H K} + \sqrt{\operatorname{dim}(\mathcal{F}) KH \log |\mathcal{F}| }  \big)$ (Theorem 5.1)

- Achieving nearly **minimax optimality in linear MDPs** with preference feedback (Theorem 5.2)

- Enhanced overall presentation for improved clarity of our contributions

---

### Meta-Review · Area_Chair_Md1t · 2024-12-23

**Metareview:**

This papers studies combinatorial RL with preference feedback where the RL agent selects a subset of actions to users and receives the preference feedback (choice). To tackle the uncertainty in both the user choice model and the state transitions, the authors propose to use the contextual MNL preference model and RL with function approximation.  The authors presents MNL-VQL, a computationally and statistically efficient algorithm with theoretical guarantees. The reviewers appreciate the following strengths:
- Existing MNL bandit setting to the combinatorial RL setting, which is practically motivated.
- Offering the first theoretical regret guarantee in combinatorial RL with preference feedback.

The reviewers mainly have three shared concerns:
- Lack of experiments. The authors added numerical experiments during rebuttal period to resolve the concern.
- The technical challenge and novelty are not clear. The result heavily relies on existing works on combinatorial RL, RL/bandits under choice model, and RL with function approximation. The reviewers are confused about the technical novelty compared to simply combining existing works.
- Organization and presentation issues.

Other issues were also discussed such as choosing hyperparameters, computation efficiency, etc, and IMO they are fully addressed by author response. After discussion period, two reviewers remain the rating of "marginally below the acceptance threshold". I agree with the reviewers and recommend rejection. The paper would be stronger by addressing the shared concerns in next version.

**Additional Comments On Reviewer Discussion:**

- Lack of experiments. The authors added numerical experiments during rebuttal period and resolved the concern.
- The technical challenge and novelty is not clear. The result heavily relies on existing works on combinatorial RL, RL/bandits under choice model, and RL with function approximation. The authors provide clarifications to address this concern. The explanation is accepted by Reviewer aS1z but did not fully convince Reviewer 5zZf. I agree that this is a major concern.
- Organization and presentation issues. This is not a major concern in my recommendation but should be improved in future version.

Other issues are considered minor and fully addressed.

---

### Decision · Program_Chairs · 2025-01-22

Reject